# Distinct microglial transcriptomic signatures within the hippocampus

**Sana Chintamen**[1ᵒ], **Pallavi Gaur**[2ᵒ], **Nicole Vo**[2], **Elizabeth M. Bradshaw**[2], **Vilas Menon**[2], **Steven G. Kernie**[1,2] *

1 Department of Pediatrics, Columbia University College of Physicians and Surgeons, New York, New York, United States of America, 2 Department of Neurology, Columbia University College of Physicians and Surgeons, New York, New York, United States of America

ᵒ These authors contributed equally to this work.
* sk3516@cumc.columbia.edu

**Data Availability Statement:** All relevant data are within the manuscript and its Supporting information files.

**Funding:** This research was supported by National Institutes of Health/National Institute of

## Abstract

Microglia, the resident immune cells of the brain, are crucial in the development of the nervous system. Recent evidence demonstrates that microglia modulate adult hippocampal neurogenesis by inhibiting cell proliferation of neural precursors and survival both *in vitro* and *in vivo*, thus maintaining a balance between cell division and cell death in the neural stem cell pool. There are increasing reports suggesting these microglia found in neurogenic niches differ from their counterparts in non-neurogenic areas. Here, we present evidence that hippocampal microglia exhibit transcriptomic heterogeneity, with some cells expressing genes associated with neurogenesis. By comprehensively profiling myeloid lineage cells in the hippocampus using single cell RNA-sequencing, we have uncovered a small, yet distinct population of microglia which exhibit depletion in genes associated with homeostatic microglia and enrichment of genes associated with phagocytosis. Intriguingly, this population also expresses a gene signature with substantial overlap with previously characterized phenotypes, including disease associated microglia (DAM), a particularly unique and compelling microglial state.

## Introduction

The hippocampus is important for memory consolidation as well as declarative and spatial memory and learning [1–3]. Hippocampal function is also known to be affected early or more severely in a variety of neurodegenerative and psychiatric diseases. These include Alzheimer's disease (AD), epilepsy, and major depressive disorder, which all are known to exhibit alterations in immune activity and each manifest hallmark traits of inflammation [4–7]. Subsets of immune cells show proclivity towards disease progression in both the rodent and human brain [8–10]. Thus, characterizing various immune subsets in the hippocampus is crucial for uncovering mechanisms of disease development and progression.

Under resting conditions, the immune compartment of the central nervous system (CNS) is comprised of myeloid lineage cells, microglia and other macrophages which contain distinct transcriptomic and phenotypic properties [11, 12]. The latter of the two are typically found in the meninges, perivascular regions, and choroid plexus [13]. Collectively, these non-microglial

Neurological Disorders and Stroke Grants R01-NS-095803 (SGK) and the Paul Allen Foundation (SGK), and National Institute of Aging Grant R01-AG-066831 (VM). This research was funded in part through the National Institute of Health and National Cancer Institute (NIH/NCI) Cancer Center Support Grant P30CA013696 and used the Genomics and High Throughput Screening Shared Resource. Research reported in this publication was performed in the CCTI Flow Cytometry Core, supported in part by the Office of the Director, National Institutes of Health under awards S10OD020056. The content is solely the responsibility of the authors and does not necessarily represent the official views of the National Institutes of Health. Images were collected and/or image processing and analysis for this work was performed in the Confocal and Specialized Microscopy Shared Resource of the Herbert Irving Comprehensive Cancer Center at Columbia University, supported by NIH grant #P30 CA013696 (National Cancer Institute). The funders had no role in study design, data collection and analysis, decision to publish, or preparation of the manuscript.

**Competing interests:** The authors have declared that no competing interests exist.

macrophages are termed CNS-associated macrophages (CAMs) or Border Associated Macrophages (BAMs) [11, 12]. However, as microglia are the primary macrophage in the brain parenchyma and found to be actively interacting with neurons, they are more widely studied and characterized compared to macrophages in non-parenchymal tissue, particularly in the context of neurodevelopmental processes. During early postnatal development, the brain is highly plastic and microglia exhibit a great degree of heterogeneity. In contrast, previous studies in the adult rodent brain have shown limited heterogeneity, corresponding to a time point when the brain is less plastic [10, 14, 15]. However, since neurogenic niches undergo life-long development, immune cells show phenotypic differences that correlate with a specialized need to support these regions [16–18]. Increasing evidence demonstrates that microglia actively regulate adult hippocampal neurogenesis [19, 20], and in fact, immune input has been shown to alter neurogenesis during injury, stroke, and aging [21]. Importantly, adult hippocampal neurogenesis (AHN) is key in certain forms of spatial memory and learning, memory consolidation, and recovery from injury [22]. Deficits in AHN in the murine and human brain have been found in a host of neurodegenerative diseases such as depression, Alzheimer's Disease, and age-associated cognitive deficits [23–26]. This suggests that attenuating reductions in neurogenesis may prevent the cognitive decline associated with aging or neurodegeneration [27].

Bulk sequencing experiments show subtle differences in various genes between subregions in the hippocampus [16]. Single cell transcriptomic profiling of cells in the dentate gyrus has demonstrated that immune cells minimally express common microglia markers and more highly express some genes associated with microglial activation [28]. This necessitates a direct comparison between various populations within the hippocampus at the single-cell level to provide relative information on how immune cells are specialized to support the neurogenic niche. In this study, we leverage transcriptomic profiles from myeloid lineage cells in the hippocampus at the level of single cells to resolve heterogeneity previously obscured in bulk sequencing/profiling.

Our experimental paradigm profiles over 18,000 cells from twelve murine hippocampi to resolve heterogeneity in the myeloid landscape of the adult hippocampus. In doing so, we have a substantially higher number and resolution of hippocampal myeloid cells than previously reported [14, 16, 28]. Consequently, we uncovered rare populations that reside in the hippocampus and have previously not been identified. Here, we identify a unique subset or population of cells that correspond to myeloid cells in the subgranular zone, which shape, regulate and/or support the pool of hippocampal neural progenitor cells. By examining these cells within the myeloid cell pool in the hippocampus, not only are we able to make a direct comparison to other subsets of microglia, but we also can examine other populations that may influence the neurogenic niche, even when not in direct contact with stem/progenitor cells in this region. This novel and comprehensive transcriptomic study, with single cell resolution, uniquely highlights genes involved in immune activation and neuronal development and support that provides insight into how the neurogenic niche is regulated in development and disease.

## Materials and methods

### Animals

All experimental procedures were in accordance with the Guide for the Care and Use of Laboratory Animals of the National Institutes of Health and approved by the Institutional Animal Care and Use Committee at Columbia. Experimental animals were humanely housed and cared for under the supervision of the Institute of Comparative Medicine at Columbia University. For generation of the dual reporter mice, Cx3Cr1CreERT2+/+ (Jackson stock no. 021160)

males were bred with Rosa26-loxp-stop-tdtomato+/+ females, resulting in progeny (F1) that were heterozygous for both the Cre recombinase and the flox-stop tdTomato reporter (Jackson stock 007914). Mice from F1 were crossed mice from F2 that were homozygous for the CreERT2 and tdTomato were selected as breeders. Finally, these mice were crossed with Nestin-GFP mice developed by us (Jackson stock no. 02967), resulting in progeny (F3) that were heterozygous for each of the 3 alleles of interest, the Cx3Cr1CreERT2, Rosa26-tdtomato, and TK-Nestin-eGFP. See S1 Fig [29].

## Microglial isolation

Seven-week old dual reporter mice were injected with Tamoxifen (100mg/kg) intraperitoneally once a day for four consecutive days. Each sequencing sample replicate comprised of four bilateral hippocampi from two female and two male, eight week old mice. We had a total of three replicates for single cell RNA-sequencing, resulting in cells analzyed from twelve bilateral hippocampi. Our data set consists of cells from hippocampi originating from a total of six male mice and six female mice. Mice were perfused with approximately 25–30 mL of ice-cold sterile PBS (Corning Cellgro REF 21-040-CV) under general isoflurane anesthesia to minimize pain and suffering. Each brain was extracted whole and placed in 5 mL of homogenization buffer (see buffers list) at 4° while other mice were being perfused. After all brains were extracted, hippocampi were dissected on a sterile petri dish placed atop a cold metal platform on top of ice brick to ensure brains remain cold throughout. Each brain was hemisected along the midline using a sterile scalpel. Using curved forceps with sharpened ends, bilateral hippocampi were dissected from each hemisphere of each mouse and bilateral hippocampi from each of the four mice were pooled together in a 2 mL dounce with 1 ml of sample buffer (see buffer list). Cortical tissue was also dissected to be used for setting up sample gates during FACS.

Following homogenization, we adapted the isolation protocol from Bohlen et. al. 2018 [30]. In brief, cell suspensions were filtered by passing through a 70um filter. Samples were transferred to 2 mL eppendorf tubes coated with 10% sterile filtered FBS in PBS (to prevent cell adhesion on tubes) and centrifuged. Pellets were suspended in 1.8 mL myelin removal buffer. Myelin removal beads were briefly vortexed. 200 μL of myelin removal beads were added to each sample and incubated over ice for 15 minutes with gentle flicking every 5 minutes to mix settled beads. The reaction was stopped after incubation period by diluting with 2 mL of myelin buffer per sample. Samples were transferred to 2 mL Eppendorf tubes and centrifuged. Pellets were resuspended in MACS buffer (1ml buffer/pellet). After LS columns were washed twice with flow through discarded, cell suspension was applied to columns (1 tube/LS column). LS columns were washed to elute remaining cells adhering to columns. Flow through containing demyelinated cells were transferred to 2 mL eppendorf tubes and centrifuged. Pellets were resuspended in 1 mL Sterile PBS and incubated with 1 μl Live/Dead Violet per sample for 5 minutes covered from light over ice. Samples were centrifuged, resuspended in flow buffer (containing RNAsin and DNase), and transferred to 5 mL polypropylene tubes for FACS.

## Flow cytometry

Samples were sorted on BD Influx at the Columbia Center for Translational Immunology Flow Cytometry core. Between 113,00–132,000 cells were retrieved per sort sample. Samples were of high viability and yield. Gates were established as illustrated in Fig 1B. Cells were first selected by size and granularity (FSC and SSC, respectively). Next cells were gated to exclude doublets. Subsequently, cells were gated for viability and finally gated for td-Tomato

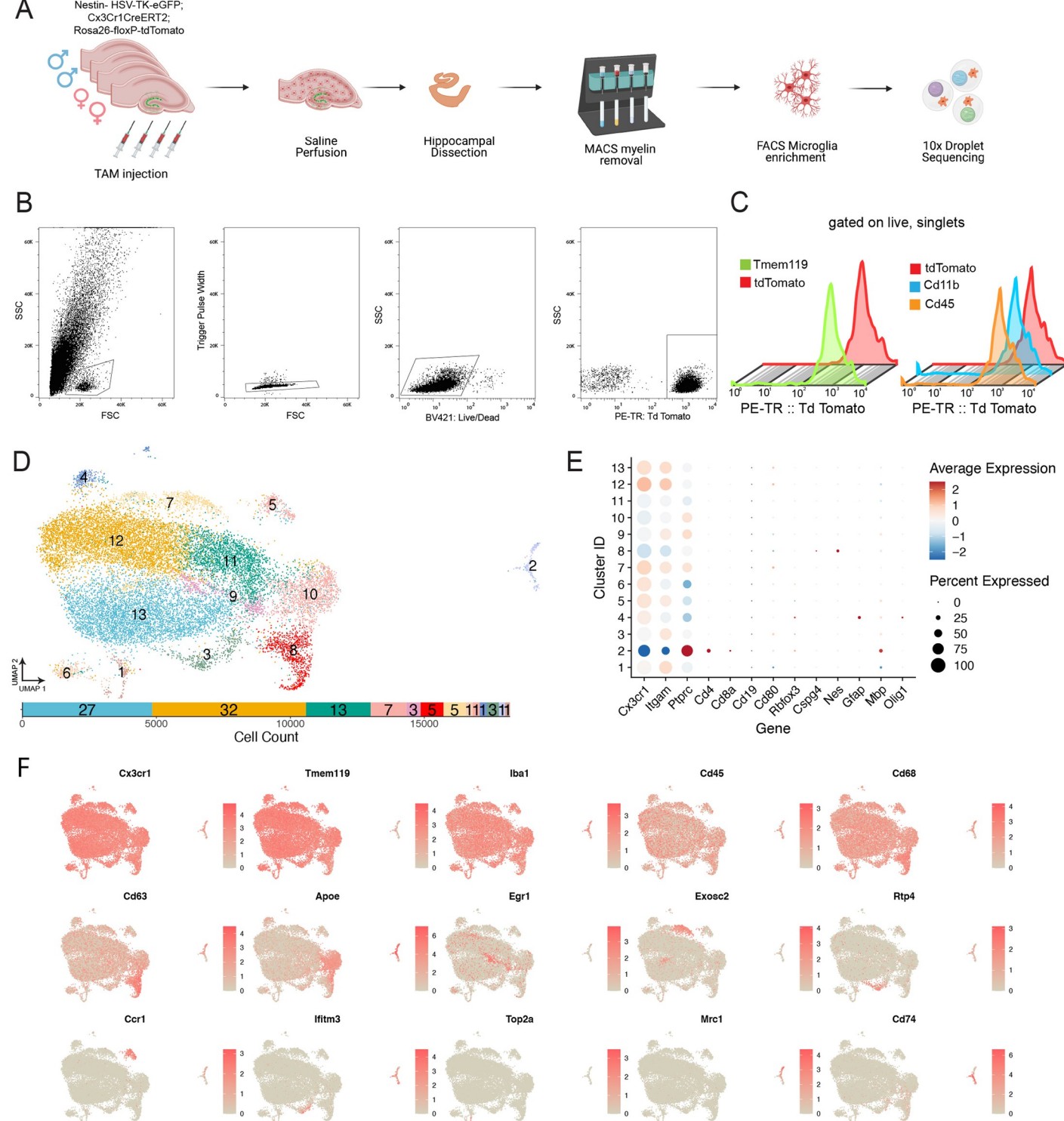

**Fig 1. Transcriptomic Heterogeneity of myeloid population in the hippocampus.** (A) Schematic illustrating experimental workflow and double reporter mouse model used in single cell sequencing experiments. (B) Fluorescent Activated Cell Sorting (FACS) gating scheme to isolate tdTomato+ cells. (C) Flow cytometry analysis of cells expressing Tdtomato with Tmem119 antibody(left) or CD11b/CD45 (right). (D) Top: UMAP Plot showing dimensionality reduction, colored by Seurat clusters. Each dot represents a cell. Bottom: Bar Plot showing distribution of cells across clusters to show cell count in each cluster (right to left starting with cluster 1). Percentage of cells in each cluster labeled within bar. (E) Dot Plot displaying expression of marker genes for cell types found in the brain across clusters. Each row represents a cluster while each column represents the level of expression of a selected gene marker. The fraction of cells in a given cluster is represented by the size of

the dot. The color of the dot represents the average expression of the cells within a given cluster. (F) UMAP projection displaying scaled expression of previously identified gene markers to investigate heterogeneity in hippocampal myeloid cells.

expression. TdTomato+ cells were validated using cd11b (1:100 BD biosciences 557396) and cd45 (1:50 BD Biosciences 59864) and Tmem119 (1:100 abcam ab225495). Antibodies serving as isotype controls against both FITC (BD 400607) and APC (BD 402205) were used to determine nonspecific binding of antibodies.

## Sequencing

**Single-cell 3′ library construction and library pooling.** After samples were sorted, suspensions were spun down and resuspended at a final concentration of 1000 cells /μL with between 90,000–13,000 cells. Cell viability was confirmed with trypan blue after resuspension. Between roughly 30,000–40,000 cells were loaded onto 10x Chromium Controller. Libraries were constructed as per manufacturer's (10x Genomics) instructions. Samples were sequenced at the Columbia Sulzberger genomics core. Chemistry Single Cell 3' v3. Cell ranger v 3.0.2 for Run 1, 3.1 for Run 2 and 3. Live cell suspensions were loaded onto GEM droplets. Pooled 3'-end libraries were sequenced on NovaSeq 6000.

## Analysis

**Single-cell RNA-seq preprocessing and alignment.** Read alignment to reference transcriptome mm10 was performed using Cell Ranger v 3.0. At least 94% of reads were mapped to genome. Cells were filtered using default parameters for UMI counts. Counts matrices were generated and further preprocessed as outlined below.

**Normalization and integration.** Preprocessing of the transcriptomic data was performed using CellRanger version 3.0.2 for the first sample and version 3.1.0 for the subsequent two samples using default parameters. Count matrices were then exported to Seurat version 4.0.0 to create a list of Seurat objects for each sample. To attain this, the raw gene-UMI matrix from each sample was converted to corresponding Seurat object in R 4.0.3 using Read10X function from Seurat package and a list of all Seurat objects was created. To avoid confounding effects due to low quality cells, we used a standard criterion of excluding cells with higher than 20% of total UMIs from mitochondrial genes, less than 1,000 features (nFeature_RNA), and less than 2500 transcripts (nCount_RNA) before proceeding further for normalization and integration (S1 Table).

To remove the potential influence of technical effects in the analyses, we normalized the raw data using function SCTransform separately for each dataset within the Seurat pipeline; this uses Pearson residuals to harmonize the data instead of regular log-normalized expression values. We regressed out percentage of mitochondrial genes expressed this metric from influencing clustering results.

Next, by running PrepSCTIntegration function that calculates all Pearson residuals, we proceeded to identify anchors and integrate the datasets using FindIntegrationAnchors and IntegrateData functions respectively. The anchor as well as integration dimension was set to 30, using 3000 features total.

**Dimensionality reduction and clustering.** After count normalization and integration, we used the RunPCA function in Seurat for Principal Component Analysis and proceeded with 20 principal components for further clustering and visualization; this value was selected based on the elbow plot cutoff (S3 Fig) for significant components.

The function FindClusters from the Seurat package was used for *K*-nearest neighbor clustering with a resolution parameter of 0.5. For the visualization of integration and clustering onto a 2D space, we used uniform manifold approximation and projection (UMAP) by implementing as implemented in the Seurat functions RunUMAP, DimPlot, and UMAPPlot.

**Identification of cluster markers via differential expression analysis.**   We used Seurat's FindAllMarkers function, which implements the non-parametric Wilcoxon rank sum test, to inspect differentially expressed genes by comparing a single cluster of interest with all others. We only tested the genes that were observed to be positively (up) regulated in minimum fraction of at least 70% cells in the cluster of interested showed at least ~1.5(logfc = 0.5) fold change between cluster of interest and all other groups, with an FDR-adjusted p-value <0.01.

**Gene enrichment analysis of differentially expressed genes.**   We used TopGO to find enriched Gene Ontology (GO) terms for high variance genes obtained in differential expression analysis. We ran TopGO with the Kolmogorov-Smirnov test and considered terms with a statistical significance of <0.05 (adjusted p-value).

**Comparison of novel cluster 8 and homeostatic clusters (12 and 13) with Keren-Shaul et al. disease-associated microglia data set.**   We compared the transcriptomic profiles of SGZ (cluster 8) and homeostatic clusters (cluster 12 and 13) to microglia subtypes signatures found in Keren-Shaul et. al.'s study (GEO: GSE98969). The primary goal of the integration was to compare the unique signature found in SGZ-enriched cluster with the disease-associated microglia (DAM) signature identified by the group.

We used Harmony to integrate our data with the Keren-Shaul et. al. data by passing a merged Seurat object consisting of all selected microglial cells from both datasets and following the standard pipeline through PCA. We then ran the *RunHarmony* function on the normalized data, where we ran 10 rounds of iteration with default values of theta and lambda to achieve the corrected harmony coordinates. We used the first 10 dimensions from the Harmony integration to generate the UMAP and run nearest neighbor analyses, using a resolution of 0.7 with the harmony embeddings rather than PCs. We investigated the expression of genes known to be downregulated in DAM profile to map the cluster associated with unique signature found in SGZ-enriched cluster in the integrated dataset.

**Data and code availability.**   Single cell sequencing analysis was done primarily using Seurat version 4.0.0. Scripts and code are available on GitHub at https://github.com/sanachintamen/HC_myeloid. Raw BAM files and Cell Ranger processed gene expression matrices are available in the NCBI GEO databank with accession number GSE182289.

## Brain sectioning and immunohistochemistry

Mice were perfused with PBS followed by 4% paraformaldehyde. Brains were dissected and post-fixed over night at 4°C. Free floating sections were cut with 50 μm thickness on Leica 1000S vibrating blade microtome. Sections were permeabilized in 0.3% PBST followed by 1 hour incubation with 5% Normal Donkey Serum at RT. Samples were then incubated overnight in primary antibody at the following concentrations 1:500 Iba1 (Wako 019–19741), 1:100 Cd68 (Biorad MCA1957), 1:200 Cd9 (BioLegend 124802) at 4°C. The next day, sections were washed thrice with PBST for 5 minutes at room temperature. Afterwards, they were incubated with secondary antibody staining solution containing secondary antibodies at a final concentration of 1:200 (Jackson immunoresearch). Sections were incubated in secondary antibody solution for two hours at room temperature after which they were washed thrice in PBST and twice in PBS. Sections were then mounted using LifeTechnologies Prolong mounting medium containing DAPI or NucBlue.

### Imaging and analysis

Images for Sholl Analysis were obtained using a Laser Scan confocal microscope (TCS SP8, Leica). Cells for reconstruction were obtained from a single section originating from one eight-old week male mouse. Z-stack images were acquired at 0.5μm intervals. Images were projected across z-planes for representative images. Cells were traced in a semi-automated manner in 3D using Neurolucida. Dendritic complexity was measured using intersections at 10 μm radii were used to determine microglial process ramification. These intersections were summed for each cell to obtain the total number of intersections and compared using an unpaired, Type 2 t-test. For confocal imaging of sections from dual reporter mice, Nikon Ti Eclipse inverted with Yokogawa CSU-X1 spinning disk was used to minimize photobleaching. Images were obtained using 25x water of 40x air objectives. Images were stitched together using NIS-Elements. Epifluorescent images were acquired with a Zeiss microscope (Axio Imager M2, Zeiss) equipped with a Hamamatsu camera (Orca-R2, Hamamatsu).

## Results

### Single cell sequencing of myeloid cells in the hippocampus

The brain immune compartment consists primarily of innate immune cells which include microglia (the predominant cell type) as well as other CNS-associated macrophages. To dissect cellular interactions between immune cells and neural progenitor/precursor cells in the neurogenic niche, we utilized a double reporter mouse model expressing eGFP under the control of the Nestin promoter and tdTomato conditionally active in cells expressing the fractalkine receptor (CX3CR1) and CreERT2 [31] (S2 and S3 Figs). These double reporter mice were used for both transcriptomic and histology experiments (Fig 1A). We isolated tdTomato+ myeloid lineage cells for single cell RNA-sequencing (Fig 1B). We validated the reporter line using flow cytometry analysis using antibodies common for microglia and showed near complete overlap with Cd11b$^{hi}$/Cd45$^{lo}$ cells as well as Tmem119$^{+}$ cells (Fig 1C and S1 Fig).

Single cell RNA-sequencing experiments were conducted to determine which cells compose the myeloid landscape of the hippocampus. For each experiment, four hippocampi of tdTomato positive mice (two male, two female) were pooled and enriched for tdTomato-positive myeloid lineage cells, which were then sequenced with three technical triplicates, yielding cells from the hippocampi of twelve mice in total.

The pre-processing of the data yielded 20,376 combined cells from all samples. To filter out cells with low quality, we excluded cells whose UMI counts were fewer than 2,500; comprised of higher than 20 percent of mitochondrial genes; and reflected less than 1,000 unique features (S1 Table and S2A Fig). We checked the percentage of genes encoding ribosomal protein which were consistently acceptable throughout each sample (S2A Fig). The remaining 18,198 cells were included in downstream normalization, analysis, and integration (S2 Fig). We applied Canonical Component Analysis (CCA)-based integration on our integrated dataset to account for batch effects and found that cells from different batches mixed together across all major cell types after applying CCA. Subsequently, we conducted principal component analysis for dimensionality reduction with the number of principal components set using an elbow plot (S2 Fig) in order to cluster myeloid cells based on transcriptome profiles; this approach (see Methods) yielded 14 clusters. However, since two of these clusters yielded virtually no differences in differential gene expression analysis, we merged them which then totaled 13 clusters. These are illustrated in Fig 1D and we refer to these clusters as such for downstream analysis. As further quality control checks, we applied cell cycle scoring to our dataset to determine whether cell proliferation status greatly influenced clustering (S2 Fig). It should be noted

that this scoring method has been developed on human genes as a reference for cell cycle status. This may be a caveat as we converted these lists to their mouse homologs and performed this analysis. As microglia do not rapidly turnover in the homeostatic brain, the majority of cells are not dividing and hence not in S-phase with the exception of cells from cluster 1 [32]. However, we chose to not regress out the cell cycle scores as microglial proliferation itself marks immune activation and the number of proliferating cells at baseline levels may serve as a useful reference to compare with disease and injury model systems. We also used Double-tFinder to check for potential doublets (S2C Fig). Predicted doublets with this method were not restricted to any one cluster, suggesting that their removal would not impact clustering results in a significant manner.

Cells from all clusters expressed common marker genes such as *Cx3cr1*, *Cd11b* (*Itgam*), and *Cd45* (*Ptprc*) that are known to be enriched in myeloid lineage cells such microglia and other macrophages (Fig 1E). We next plotted relative transcript levels of marker genes found broadly in myeloid lineage cells (*Iba1* aka *Aif1*), monocytes (*Ccr2*), and those specific to microglia (*Tmem119, P2ry12, and Hexb*; S3C Fig). Some clusters had small fractions of cells with negligible levels of marker genes from other immune cells such as T cells (*Cd4*, *Cd8a*), B cells (*Cd19*, *Cd80*), neurons (*Rbfox* aka NeuN), NG2 glia (*Cspg4*), neuronal stem cells (*Nes*, *Gfap*), astrocytes (*Gfap*), and oligodendrocytes (*Mbp* and *Olig1*) but in large part these transcripts are not detected at appreciable levels or in most cells (Fig 1E). These may reflect material from adjacent cells in close association with isolated microglia or alternatively transient expression of these genes as is the case in newly formed microglia with *Nestin* expression [33]. In addition, we examined in further detail marker genes to resolve types of myeloid lineage cells. We noted relatively consistent levels of microglial specific genes such as *Tmem119*, *P2ry12*, and *Hexb* (S3 and S4 Figs) across most clusters with the exception of two clusters which we describe in greater detail below.

We next determined whether there were sex-specific differences in the transcriptome profiles of cells originating from male versus female samples. To do this, we plotted *Xist* expression, a gene expressed specifically by the inactivated X chromosome in female cells (S5 Fig). We found that cluster 14 is the only cluster primarily comprising female cells and that *Xist* expression separates cluster 13 from clusters 11 and 12 (S4 Fig). Based on the expression of known genes, we designated these three clusters as homeostatic clusters. We then tested whether there were significant differences in gene expression related to immune function in these putative homeostatic clusters (Table 1). We found that none of the differentially expressed genes reflect significant changes in immune function between cells of male and female origin in these homeostatic populations.

To assess whether these clusters truly distinguish cells based on biological differences, we next performed differential gene expression analysis between clusters (Table 1 and S4 Fig). Due to the high enrichment of classical microglia-specific genes- *Tmem119*, *P2ry12*, *Selplg*- and low numbers of differentially expressed genes, we concluded that clusters 11,12, and 13, which represent about 72 percent of cells in the data set, reflect a homeostatic gene expression profile [34–36]. We also inspected some key genes previously identified to be implicated in immune dynamics (Fig 1F) and then delved deeper into different transcriptomic clusters to determine to what extent these clusters reflect distinct populations [10, 12, 15, 34].

Clusters 4,5, and 6 did not meet the threshold for differential gene expression for gene set enrichment analysis. These clusters failed to show enrichment of at least ten nuclear genes. Cluster 4 shows high expression of mitochondrial genes. While these cells passed the threshold during processing, they may be indicative of cells that are lower quality. Cluster 5 was observed to express high levels of *Ccr1* and upregulate *Tmem176a*. These genes are found in border macrophages transitioning to a microglia-like state [15]. However, in our dataset these cells

**Table 1. Differences in gene expression related to immune function in these putative homeostatic clusters.**

| | p_val | avg_log2FC | pct.1 | pct.2 | p_val_adj | cluster | gene |
|---|---|---|---|---|---|---|---|
| **Stmn1** | 1.405487958053E-206 | 3.07876097378747 | 0.832 | 0.131 | 4.36446175614199E-202 | 1 | Stmn1 |
| **Hmgb2** | 8.60450362450788E-99 | 2.39413825583662 | 0.8 | 0.231 | 2.67195651051843E-94 | 1 | Hmgb2 |
| **Ptma** | 1.51259994061616E-54 | 1.07018770963932 | 0.995 | 0.898 | 4.69707659559536E-50 | 1 | Ptma |
| **Dek** | 2.84321008688058E-51 | 1.36450422401494 | 0.805 | 0.381 | 8.82902028279026E-47 | 1 | Dek |
| **Tuba1b** | 2.5211811959346E-44 | 1.27690413462073 | 0.935 | 0.659 | 7.82902396773573E-40 | 1 | Tuba1b |
| **Tubb5** | 5.49430629964004E-44 | 1.45491436717803 | 0.919 | 0.674 | 1.70614693522722E-39 | 1 | Tubb5 |
| **Slbp** | 2.9995279292458E-43 | 1.13787083025338 | 0.703 | 0.289 | 9.31443407868699E-39 | 1 | Slbp |
| **H2afz** | 1.67617696783389E-39 | 1.59162937247635 | 0.924 | 0.735 | 5.20503233821458E-35 | 1 | H2afz |
| **Ran** | 2.3103161005717E-30 | 0.932352110222592 | 0.822 | 0.488 | 7.17422458710531E-26 | 1 | Ran |
| **Ppia** | 9.49734094653465E-30 | 0.677329773054638 | 1 | 0.978 | 2.9492092841274E-25 | 1 | Ppia |
| **Malat1** | 2.71503645925915E-26 | -0.507907024790841 | 1 | 0.999 | 8.43100271693745E-22 | 1 | Malat1 |
| **Fcgr3** | 1.47579356997005E-24 | -0.647677838303455 | 0.914 | 0.977 | 4.58278177282798E-20 | 1 | Fcgr3 |
| **Srsf7** | 1.12323083988607E-23 | 0.627731663609143 | 0.751 | 0.37 | 3.48796872709823E-19 | 1 | Srsf7 |
| **Hsp90aa1** | 9.03067923283368E-22 | 0.966906173083516 | 0.789 | 0.484 | 2.80429682217184E-17 | 1 | Hsp90aa1 |
| **Gapdh** | 1.96374437187526E-21 | 0.663395978278371 | 0.968 | 0.83 | 6.09801539798425E-17 | 1 | Gapdh |
| **Rps27** | 2.3721557743964E-20 | -0.538296874004709 | 0.984 | 0.99 | 7.36625532623313E-16 | 1 | Rps27 |
| **Hmgb1** | 7.5834208795323E-20 | 0.696452323022391 | 0.935 | 0.805 | 2.35487968572117E-15 | 1 | Hmgb1 |
| **Ybx1** | 1.43698687875737E-19 | 0.587052762794741 | 0.908 | 0.656 | 4.46227535460526E-15 | 1 | Ybx1 |
| **C1qb** | 2.1130376789021E-19 | -0.317571579786213 | 1 | 0.999 | 6.56161590429468E-15 | 1 | C1qb |
| **Ctss** | 3.87896330866326E-19 | -0.371111585194315 | 1 | 0.999 | 1.2045344762392E-14 | 1 | Ctss |
| **Rbm3** | 5.02442857182374E-19 | 0.720059055393812 | 0.822 | 0.583 | 1.56023580440842E-14 | 1 | Rbm3 |
| **Hspa8** | 8.46099287966658E-19 | 0.533963930241726 | 0.973 | 0.887 | 2.62739211892286E-14 | 1 | Hspa8 |
| **Tyrobp** | 4.32360751945819E-17 | -0.386412663083304 | 1 | 0.999 | 1.34260984301735E-12 | 1 | Tyrobp |
| **Hnrnpd** | 4.97460993026326E-17 | 0.510467402474567 | 0.778 | 0.43 | 1.54476562164465E-12 | 1 | Hnrnpd |
| **Cd300c2** | 5.59711452660734E-17 | -0.578958189118655 | 0.822 | 0.902 | 1.73807197394738E-12 | 1 | Cd300c2 |
| **Selenop** | 1.04478660247113E-15 | -0.534049653650265 | 0.973 | 0.984 | 3.24437583665359E-11 | 1 | Selenop |
| **Trem2** | 2.98433161948128E-15 | -0.348133915407529 | 0.995 | 0.997 | 9.26724497797523E-11 | 1 | Trem2 |
| **Hint1** | 3.40336260533027E-15 | 0.56227473209876 | 0.811 | 0.575 | 1.05684618983321E-10 | 1 | Hint1 |
| **Itm2b** | 5.42970254893917E-15 | -0.302825472442435 | 1 | 1 | 1.68608553252208E-10 | 1 | Itm2b |
| **Cst3** | 8.34803314181557E-15 | -0.273078787394303 | 1 | 1 | 2.59231473152799E-10 | 1 | Cst3 |
| **Set** | 6.87585763141454E-14 | 0.521393732728281 | 0.805 | 0.54 | 2.13516007028316E-09 | 1 | Set |
| **Sumo2** | 4.24089041328231E-13 | 0.478605323350229 | 0.849 | 0.669 | 1.31692370003655E-08 | 1 | Sumo2 |
| **Hnrnpf** | 4.98591132037963E-13 | 0.50611696743529 | 0.914 | 0.755 | 1.54827504231749E-08 | 1 | Hnrnpf |
| **Srsf3** | 1.17389744598453E-12 | 0.465040084255532 | 0.914 | 0.73 | 3.64530373901577E-08 | 1 | Srsf3 |
| **Tra2b** | 2.73248436302316E-12 | 0.464800630833684 | 0.854 | 0.571 | 8.48518369249582E-08 | 1 | Tra2b |
| **Fkbp2** | 2.82828767615021E-12 | 0.472862391580013 | 0.811 | 0.583 | 8.78268172074924E-08 | 1 | Fkbp2 |
| **Fau** | 3.80617373744277E-12 | -0.318549156826642 | 1 | 0.999 | 1.1819311306881E-07 | 1 | Fau |
| **Hmgn1** | 4.65068976610108E-12 | 0.530992653386055 | 0.746 | 0.495 | 1.44417869306737E-07 | 1 | Hmgn1 |
| **Mafb** | 4.66549584194598E-12 | -0.49449101126898 | 0.951 | 0.969 | 1.44877642379949E-07 | 1 | Mafb |
| **Ctsl** | 9.42729264986922E-12 | -0.348261968358648 | 0.989 | 0.985 | 2.92745718656389E-07 | 1 | Ctsl |
| **Srsf2** | 1.02575931740359E-11 | 0.438969505739476 | 0.881 | 0.684 | 3.18529040833336E-07 | 1 | Srsf2 |
| **Rpl10** | 1.43163108807182E-11 | -0.361700259002224 | 0.984 | 0.986 | 4.44564401778943E-07 | 1 | Rpl10 |
| **Fcrls** | 2.90994564327388E-11 | 0.311076370276798 | 0.984 | 0.981 | 9.03625420605838E-07 | 1 | Fcrls |
| **Rps21** | 4.26841268914367E-11 | -0.339062273869301 | 0.978 | 0.993 | 1.32547019235978E-06 | 1 | Rps21 |
| **Tnfaip8l2** | 5.36380944232295E-11 | -0.53814686726608 | 0.703 | 0.782 | 1.66562374612454E-06 | 1 | Tnfaip8l2 |
| **Fcer1g** | 6.37945635802906E-11 | -0.263936659146291 | 1 | 0.998 | 1.98101258285877E-06 | 1 | Fcer1g |
| **Tmed3** | 1.0754260347785E-10 | 0.466938395193858 | 0.822 | 0.586 | 3.33952046579768E-06 | 1 | Tmed3 |

*(Continued)*

**Table 1.** (Continued)

| | p_val | avg_log2FC | pct.1 | pct.2 | p_val_adj | cluster | gene |
|---|---|---|---|---|---|---|---|
| **Psma1** | 1.66119232220977E-10 | 0.446075860887791 | 0.795 | 0.559 | 5.158500518158E-06 | 1 | Psma1 |
| **Arpc5l** | 1.75191797056051E-10 | 0.39732277338557 | 0.719 | 0.447 | 5.44023087398157E-06 | 1 | Arpc5l |
| **Rdx** | 2.13276878714269E-10 | 0.437170143436664 | 0.735 | 0.471 | 6.62288691471421E-06 | 1 | Rdx |
| **Rpl34** | 2.37232782293395E-10 | -0.363219690411769 | 0.962 | 0.98 | 7.36678958855679E-06 | 1 | Rpl34 |
| **Rpl18a** | 3.06354794751935E-10 | -0.34323886858764 | 1 | 0.99 | 9.51323544143184E-06 | 1 | Rpl18a |
| **Hpgds** | 3.36135732673178E-10 | -0.438932432518431 | 0.854 | 0.894 | 1.04380229067002E-05 | 1 | Hpgds |
| **Rgs10** | 4.47597935361456E-10 | -0.307172198260009 | 0.995 | 0.991 | 1.38992586867793E-05 | 1 | Rgs10 |
| **Snrpb** | 4.67843328377099E-10 | 0.413190868562448 | 0.784 | 0.532 | 1.4527938876094E-05 | 1 | Snrpb |
| **Arhgap45** | 4.70613034812082E-10 | -0.452800221852119 | 0.827 | 0.863 | 1.46139465700196E-05 | 1 | Arhgap45 |
| **Rhob** | 5.52858229337787E-10 | -0.400284166352431 | 0.968 | 0.985 | 1.71679065956263E-05 | 1 | Rhob |
| **Mat2a** | 6.26598559094021E-10 | 0.530173555260338 | 0.773 | 0.539 | 1.94577650555466E-05 | 1 | Mat2a |
| **Cd14** | 7.67210172997373E-10 | -0.511463111549875 | 0.697 | 0.766 | 2.38241775020874E-05 | 1 | Cd14 |
| **Ubc** | 7.69507150300919E-10 | -0.326626404355029 | 0.978 | 0.982 | 2.38955055382944E-05 | 1 | Ubc |
| **Atp5j** | 1.75650670044555E-09 | 0.417626525618481 | 0.816 | 0.589 | 5.45448025689356E-05 | 1 | Atp5j |
| **Rps28** | 1.76539118301755E-09 | -0.360637976228019 | 0.935 | 0.96 | 5.48206924062439E-05 | 1 | Rps28 |
| **Cyth4** | 2.10874307651391E-09 | -0.351714215247656 | 0.951 | 0.952 | 6.54827987549865E-05 | 1 | Cyth4 |
| **Arpp19** | 2.38518165511063E-09 | 0.38091381431864 | 0.741 | 0.488 | 7.40670459361504E-05 | 1 | Arpp19 |
| **Serbp1** | 2.55101978251909E-09 | 0.38784972715402 | 0.838 | 0.641 | 7.92168173065652E-05 | 1 | Serbp1 |
| **Fus** | 2.66784790070403E-09 | 0.390071870812651 | 0.914 | 0.771 | 8.28446808605624E-05 | 1 | Fus |
| **Rpl27a** | 2.70291125013243E-09 | -0.280378564108046 | 1 | 0.994 | 8.39335030503622E-05 | 1 | Rpl27a |
| **Rps9** | 2.80398096428665E-09 | -0.299546709086227 | 0.989 | 0.992 | 8.70720208839932E-05 | 1 | Rps9 |
| **Unc93b1** | 5.48196964472685E-09 | -0.326759267704769 | 0.984 | 0.971 | 0.000170231603377703 | 1 | Unc93b1 |
| **Hsp90ab1** | 5.48759669595159E-09 | 0.379697208168531 | 0.962 | 0.842 | 0.000170406340199385 | 1 | Hsp90ab1 |
| **Sat1** | 7.00238163299055E-09 | -0.466706580894777 | 0.816 | 0.825 | 0.000217444956849256 | 1 | Sat1 |
| **Sdf2l1** | 7.71873440094701E-09 | 0.428214183100575 | 0.751 | 0.529 | 0.000239689859352608 | 1 | Sdf2l1 |
| **Ncl** | 1.02643612176332E-08 | 0.408798932373841 | 0.757 | 0.539 | 0.000318739208891163 | 1 | Ncl |
| **Ctsh** | 1.3463319994624E-08 | -0.365810852573241 | 0.957 | 0.959 | 0.000418076475793058 | 1 | Ctsh |
| **Luc7l3** | 1.63859177020499E-08 | 0.374340806920861 | 0.735 | 0.509 | 0.000508831902401754 | 1 | Luc7l3 |
| **Gpr34** | 1.75013116124881E-08 | -0.309543649893288 | 0.995 | 0.994 | 0.000543468229502592 | 1 | Gpr34 |
| **Hnrnpu** | 1.86697190265684E-08 | 0.444684316604392 | 0.773 | 0.575 | 0.000579750784932027 | 1 | Hnrnpu |
| **Npm1** | 1.93181103203264E-08 | 0.357196757849688 | 0.859 | 0.676 | 0.000599885279777096 | 1 | Npm1 |
| **Siglech** | 2.02177609332656E-08 | -0.314329101316197 | 0.984 | 0.985 | 0.000627822130260696 | 1 | Siglech |
| **Ly6e** | 2.13663690455209E-08 | 0.321078844882404 | 1 | 0.95 | 0.000663489857970561 | 1 | Ly6e |
| **Rpl37** | 3.12019157428408E-08 | -0.297504305854286 | 0.995 | 0.99 | 0.000968913089562434 | 1 | Rpl37 |
| **Rps11** | 3.35645910112025E-08 | -0.287174007621304 | 0.995 | 0.991 | 0.00104228124467087 | 1 | Rps11 |
| **Krtcap2** | 3.42358140248358E-08 | 0.356697568476341 | 0.773 | 0.559 | 0.00106312473291322 | 1 | Krtcap2 |
| **Vsir** | 3.5670013407074E-08 | -0.292221943492612 | 0.995 | 0.974 | 0.00110766092632987 | 1 | Vsir |
| **Rps14** | 4.88294740494431E-08 | -0.330743683722173 | 0.973 | 0.972 | 0.00151630165765736 | 1 | Rps14 |
| **Atp5g2** | 5.19240667853224E-08 | 0.342104746319781 | 0.865 | 0.715 | 0.00161239804588462 | 1 | Atp5g2 |
| **Tmem86a** | 5.93103286608248E-08 | -0.438237955943034 | 0.751 | 0.802 | 0.00184176363590459 | 1 | Tmem86a |
| **Hnrnpa3** | 8.03083969054611E-08 | 0.35880449845111 | 0.811 | 0.61 | 0.00249381664910528 | 1 | Hnrnpa3 |
| **Rpl30** | 8.92590016704123E-08 | -0.277157973870885 | 0.995 | 0.993 | 0.00277175977887131 | 1 | Rpl30 |
| **Ddx39b** | 1.094308618947E-07 | 0.312806546724517 | 0.724 | 0.472 | 0.00339815655441611 | 1 | Ddx39b |
| **Ucp2** | 1.558370009489E-07 | -0.478146626722825 | 0.676 | 0.715 | 0.0048392063904662 | 1 | Ucp2 |
| **Eef1a1** | 1.86689142780994E-07 | -0.255407835345328 | 1 | 0.999 | 0.0057972579507782 | 1 | Eef1a1 |
| **Npc2** | 2.03204908357552E-07 | -0.347704687966907 | 0.886 | 0.907 | 0.00631012201922706 | 1 | Npc2 |
| **Oxct1** | 2.26751989989881E-07 | 0.339510758811187 | 0.719 | 0.488 | 0.00704132954515576 | 1 | Oxct1 |

*(Continued)*

**Table 1.** (Continued)

| | p_val | avg_log2FC | pct.1 | pct.2 | p_val_adj | cluster | gene |
|---|---|---|---|---|---|---|---|
| **Asah1** | 2.72265402997823E-07 | -0.332856862920483 | 0.881 | 0.872 | 0.0084546575592914 | 1 | Asah1 |
| **Tmco1** | 2.82169670483339E-07 | 0.3623435952691 | 0.827 | 0.679 | 0.00876221477751911 | 1 | Tmco1 |
| **Manf** | 3.49670468087565E-07 | 0.311832302426752 | 0.881 | 0.694 | 0.0108583170455231 | 1 | Manf |
| **Rsrp1** | 5.00440099833167E-07 | -0.306751601607958 | 0.984 | 0.967 | 0.0155401664201193 | 1 | Rsrp1 |
| **Rtn3** | 5.01172309036219E-07 | 0.356832850856833 | 0.843 | 0.64 | 0.0155629037125017 | 1 | Rtn3 |
| **Pnn** | 5.21385357248612E-07 | 0.370181361947366 | 0.8 | 0.612 | 0.0161905794986412 | 1 | Pnn |
| **Ctsb** | 6.1023082567593E-07 | -0.359563025071504 | 0.984 | 0.981 | 0.0189494978297147 | 1 | Ctsb |
| **Arl6ip1** | 7.01630549834455E-07 | 0.750850023988365 | 0.935 | 0.846 | 0.0217877334640093 | 1 | Arl6ip1 |
| **H3f3a** | 8.38953437624406E-07 | 0.295467828959084 | 0.957 | 0.908 | 0.0260520210985507 | 1 | H3f3a |
| **Tecr** | 8.42991360865655E-07 | 0.303451132843531 | 0.784 | 0.572 | 0.0261774107289612 | 1 | Tecr |
| **Rps4x** | 1.103021160002E-06 | -0.287811032418148 | 1 | 0.984 | 0.034252116081542 | 1 | Rps4x |
| **Gnas** | 1.30870944451629E-06 | 0.315752460718663 | 0.93 | 0.803 | 0.0406393543805645 | 1 | Gnas |
| **Pf4** | 0 | 4.37853701752422 | 0.732 | 0.004 | 0 | 2 | Pf4 |
| **Mrc1** | 0 | 3.5577447697175 | 0.798 | 0.013 | 0 | 2 | Mrc1 |
| **Ms4a7** | 0 | 3.40638520934347 | 0.789 | 0.001 | 0 | 2 | Ms4a7 |
| **Ifi27l2a** | 0 | 3.24285935674792 | 0.781 | 0.066 | 0 | 2 | Ifi27l2a |
| **Ifitm3** | 0 | 3.23400977150058 | 0.772 | 0.038 | 0 | 2 | Ifitm3 |
| **Dab2** | 0 | 3.00672179121753 | 0.789 | 0.065 | 0 | 2 | Dab2 |
| **Ifitm2** | 0 | 2.49528029256268 | 0.789 | 0.008 | 0 | 2 | Ifitm2 |
| **Cybb** | 0 | 2.30254592729466 | 0.75 | 0.005 | 0 | 2 | Cybb |
| **Tgfbi** | 4.47252324827468E-265 | 1.77732641493551 | 0.741 | 0.084 | 1.38885264428673E-260 | 2 | Tgfbi |
| **Clec2d** | 7.46815160856889E-159 | 1.58661840196453 | 0.719 | 0.126 | 2.3190851190089E-154 | 2 | Clec2d |
| **Lyz2** | 6.02324004023014E-158 | 4.77109415482144 | 0.939 | 0.363 | 1.87039672969267E-153 | 2 | Lyz2 |
| **Ms4a6c** | 9.68378544392257E-146 | 2.22060451823559 | 0.89 | 0.278 | 3.00710589390128E-141 | 2 | Ms4a6c |
| **Hexb** | 3.2821132815041E-137 | -2.14183056509067 | 0.908 | 1 | 1.01919463730547E-132 | 2 | Hexb |
| **Cst3.1** | 1.44533002050378E-126 | -1.79844146468655 | 1 | 1 | 4.4881833126704E-122 | 2 | Cst3 |
| **Cd81** | 1.38644013247434E-119 | -1.76461710714318 | 0.763 | 1 | 4.30531254337256E-115 | 2 | Cd81 |
| **Selplg** | 7.73281113121192E-119 | -2.10983603366066 | 0.711 | 0.999 | 2.40126984057524E-114 | 2 | Selplg |
| **Ctsd** | 1.35285054753601E-115 | -1.84984214818543 | 0.917 | 1 | 4.20100680526356E-111 | 2 | Ctsd |
| **Slfn2** | 2.30893285361598E-112 | 1.53688870812899 | 0.768 | 0.191 | 7.1699291903337E-108 | 2 | Slfn2 |
| **Apoe** | 4.25915344390955E-109 | 5.30931722027963 | 0.912 | 0.536 | 1.32259491893723E-104 | 2 | Apoe |
| **Tmem119** | 1.08702043580267E-108 | -2.05633221631248 | 0.504 | 0.995 | 3.37552455929804E-104 | 2 | Tmem119 |
| **Sparc** | 1.10731990003302E-108 | -2.18920919186698 | 0.395 | 0.999 | 3.43856048557254E-104 | 2 | Sparc |
| **Gpr34.1** | 1.17601731559925E-108 | -1.9206505045212 | 0.697 | 0.997 | 3.65188657013036E-104 | 2 | Gpr34 |
| **P2ry12** | 5.99649932345756E-107 | -2.05673790980279 | 0.592 | 0.998 | 1.86209293491328E-102 | 2 | P2ry12 |
| **Emp3** | 3.58773656012687E-104 | 1.44030125325655 | 0.741 | 0.184 | 1.1140998340162E-99 | 2 | Emp3 |
| **Lgmn** | 1.18029779044412E-103 | -1.27122497369596 | 0.899 | 1 | 3.66517872866611E-99 | 2 | Lgmn |
| **Basp1** | 1.49916241944672E-97 | -1.56657531259554 | 0.711 | 0.992 | 4.6553490611079E-93 | 2 | Basp1 |
| **Lpcat2** | 1.56060019007985E-97 | -1.58297183978522 | 0.702 | 0.984 | 4.84613177025495E-93 | 2 | Lpcat2 |
| **Cd9** | 2.81122177457997E-97 | -1.95986925080908 | 0.5 | 0.977 | 8.72968697660318E-93 | 2 | Cd9 |
| **Siglech.1** | 6.9924039237018E-96 | -1.75210623490392 | 0.539 | 0.991 | 2.17135119042712E-91 | 2 | Siglech |
| **Trem2.1** | 3.64304257704326E-94 | -1.38189806163724 | 0.833 | 0.999 | 1.13127401144924E-89 | 2 | Trem2 |
| **Fth1** | 5.98699638709904E-92 | 1.38644395257302 | 1 | 0.996 | 1.85914198808586E-87 | 2 | Fth1 |
| **Olfml3** | 6.06554782323319E-91 | -1.88338859679 | 0.539 | 0.983 | 1.8835345655486E-86 | 2 | Olfml3 |
| **H2-D1** | 2.91813651021523E-90 | 1.98566369064107 | 0.982 | 0.73 | 9.06168930517134E-86 | 2 | H2-D1 |
| **Ftl1** | 2.37952852709334E-85 | 1.42217462241371 | 0.996 | 0.991 | 7.38914993518295E-81 | 2 | Ftl1 |
| **P2ry13** | 8.4235960171982E-82 | -1.58759443807461 | 0.553 | 0.948 | 2.61577927122056E-77 | 2 | P2ry13 |

*(Continued)*

**Table 1.** (Continued)

| | p_val | avg_log2FC | pct.1 | pct.2 | p_val_adj | cluster | gene |
|---|---|---|---|---|---|---|---|
| **Rps29** | 1.64269073202869E-81 | 0.89813665302241 | 1 | 1 | 5.10104753016868E-77 | 2 | Rps29 |
| **Rpl38** | 9.60280179882922E-81 | 1.30848779604445 | 1 | 0.948 | 2.98195804259044E-76 | 2 | Rpl38 |
| **Bst2** | 3.48009518058583E-80 | 1.93397290584532 | 0.89 | 0.454 | 1.08067395642732E-75 | 2 | Bst2 |
| **Tgfbr1** | 5.40223262672918E-80 | -1.44059083748452 | 0.68 | 0.983 | 1.67755529757821E-75 | 2 | Tgfbr1 |
| **Ecscr** | 7.18100739610371E-80 | -1.80915350358947 | 0.294 | 0.872 | 2.22991822671208E-75 | 2 | Ecscr |
| **F11r** | 9.60117541097144E-80 | -1.48570202289293 | 0.601 | 0.952 | 2.98145300036896E-75 | 2 | F11r |
| **Vsir.1** | 5.57946969933843E-77 | -1.34683059790175 | 0.702 | 0.978 | 1.73259272573556E-72 | 2 | Vsir |
| **Tpt1** | 2.74581544793537E-76 | 1.02463312939067 | 1 | 0.999 | 8.5265807104737E-72 | 2 | Tpt1 |
| **Rps28.1** | 3.53577718174795E-74 | 1.16956321559522 | 1 | 0.96 | 1.09796488824819E-69 | 2 | Rps28 |
| **H2-K1** | 6.17140763481889E-74 | 1.92678722793731 | 0.965 | 0.721 | 1.91640721284031E-69 | 2 | H2-K1 |
| **Plxdc2** | 2.80633430416144E-73 | -1.51331744882972 | 0.439 | 0.932 | 8.71450991471253E-69 | 2 | Plxdc2 |
| **Rhob.1** | 2.93193482936845E-72 | -1.28479026105139 | 0.724 | 0.988 | 9.10453722563784E-68 | 2 | Rhob |
| **Ptgs1** | 7.2002608414574E-72 | -1.51227437491087 | 0.474 | 0.918 | 2.23589699909777E-67 | 2 | Ptgs1 |
| **Rpl23.1** | 3.23109700045626E-71 | 1.08637702647503 | 1 | 0.987 | 1.00335255155168E-66 | 2 | Rpl23 |
| **Ldhb** | 9.05798562013014E-71 | -1.63829853914837 | 0.338 | 0.868 | 2.81277627461901E-66 | 2 | Ldhb |
| **Rpl37a** | 1.43310945448441E-70 | 1.01422226398218 | 1 | 0.99 | 4.45023478901044E-66 | 2 | Rpl37a |
| **Arhgap5** | 6.60636293556999E-70 | -1.6067265971305 | 0.425 | 0.9 | 2.05147388238255E-65 | 2 | Arhgap5 |
| **Rpl36** | 1.19721664514944E-69 | 1.14988474740412 | 0.991 | 0.94 | 3.71771684818255E-65 | 2 | Rpl36 |
| **Anxa3** | 1.81590409647394E-69 | -1.34705219627591 | 0.64 | 0.925 | 5.63892699078053E-65 | 2 | Anxa3 |
| **Ctsl.1** | 2.01461651198014E-69 | -1.2536159141747 | 0.737 | 0.988 | 6.25598865465194E-65 | 2 | Ctsl |
| **Itgb5** | 4.74246415531213E-68 | -1.18989921233838 | 0.741 | 0.973 | 1.47267739414908E-63 | 2 | Itgb5 |
| **Rpl35a** | 1.01960511083218E-67 | 0.878857222054372 | 1 | 0.996 | 3.16617975066717E-63 | 2 | Rpl35a |
| **Cx3cr1** | 1.62671077483876E-67 | -1.17100557971701 | 0.807 | 0.997 | 5.05142496910679E-63 | 2 | Cx3cr1 |
| **Serpine2** | 1.58976242454601E-66 | -1.6481148706064 | 0.272 | 0.81 | 4.93668925694274E-62 | 2 | Serpine2 |
| **Golm1** | 2.82521076190506E-66 | -1.52798439541282 | 0.386 | 0.87 | 8.77312697894377E-62 | 2 | Golm1 |
| **Bin1** | 2.90672846628869E-66 | -1.20513616006694 | 0.746 | 0.958 | 9.02626390636626E-62 | 2 | Bin1 |
| **Marcks** | 6.03288513195467E-66 | -1.01387761035684 | 0.895 | 0.998 | 1.87339182002588E-61 | 2 | Marcks |
| **Mafb.1** | 3.57243353116995E-65 | -1.32719830606566 | 0.645 | 0.973 | 1.1093477844342E-60 | 2 | Mafb |
| **Rps14.1** | 5.04552057570057E-64 | 1.04504392772565 | 1 | 0.972 | 1.5667855043723E-59 | 2 | Rps14 |
| **Csf1r** | 1.39554589011637E-63 | -0.842085678092108 | 0.947 | 0.999 | 4.33358865257837E-59 | 2 | Csf1r |
| **Fau.1** | 5.96253375808519E-63 | 0.908726323565812 | 1 | 0.998 | 1.85154560789819E-58 | 2 | Fau |
| **Epb41l2** | 1.13055315978884E-62 | -1.39707650837606 | 0.561 | 0.92 | 3.51070672709228E-58 | 2 | Epb41l2 |
| **Blvrb** | 1.13472513015967E-62 | 1.21667315947308 | 0.724 | 0.272 | 3.52366194668482E-58 | 2 | Blvrb |
| **Ctss.1** | 2.4625640409105E-62 | -0.807902515258847 | 0.974 | 0.999 | 7.64700011623938E-58 | 2 | Ctss |
| **Rpl41** | 3.25881180885314E-62 | 0.958201245201018 | 1 | 0.99 | 1.01195883100317E-57 | 2 | Rpl41 |
| **Frmd4a** | 4.00387893533487E-61 | -1.34974482658083 | 0.509 | 0.895 | 1.24332452578954E-56 | 2 | Frmd4a |
| **Rps24** | 8.77557802691455E-61 | 1.05993848607014 | 1 | 0.994 | 2.72508024469777E-56 | 2 | Rps24 |
| **Syngr1** | 2.61156370496386E-60 | -1.49672836820486 | 0.311 | 0.817 | 8.10968877302427E-56 | 2 | Syngr1 |
| **Rps16** | 9.88784117838514E-58 | 1.06824195585608 | 1 | 0.976 | 3.07047132112394E-53 | 2 | Rps16 |
| **Slc2a5** | 1.04127416006617E-57 | -1.59635854274501 | 0.276 | 0.751 | 3.23346864925347E-53 | 2 | Slc2a5 |
| **Rgs10.1** | 4.68493353484417E-57 | -0.894680861429364 | 0.908 | 0.992 | 1.45481241057516E-52 | 2 | Rgs10 |
| **Adgrg1** | 6.60135897920662E-57 | -1.50416282898137 | 0.25 | 0.764 | 2.04992000381303E-52 | 2 | Adgrg1 |
| **Apbb1ip** | 1.26901082167329E-56 | -1.20756930256106 | 0.667 | 0.913 | 3.94065930454206E-52 | 2 | Apbb1ip |
| **Sgk1** | 1.61830969338922E-56 | -1.6611609749098 | 0.399 | 0.823 | 5.02533709088155E-52 | 2 | Sgk1 |
| **Laptm5** | 1.04408337195052E-55 | -0.755142974938095 | 1 | 0.998 | 3.24219209491795E-51 | 2 | Laptm5 |
| **mt-Co1** | 1.21565705128143E-55 | 0.861842071564895 | 1 | 0.995 | 3.77497984134423E-51 | 2 | mt-Co1 |
| **Qk** | 2.16292981198365E-55 | -1.15060464424042 | 0.732 | 0.95 | 6.71654594515284E-51 | 2 | Qk |

*(Continued)*

**Table 1.** (Continued)

| | p_val | avg_log2FC | pct.1 | pct.2 | p_val_adj | cluster | gene |
|---|---|---|---|---|---|---|---|
| Lair1 | 5.87645385493111E-55 | -1.12265963056934 | 0.654 | 0.949 | 1.82481521557176E-50 | 2 | Lair1 |
| Rpl34.1 | 7.51178606524087E-55 | 0.851997036602266 | 1 | 0.98 | 2.33263492683925E-50 | 2 | Rpl34 |
| Ly86 | 7.96859894336671E-54 | -0.8430398632039 | 0.939 | 0.998 | 2.47448902988366E-49 | 2 | Ly86 |
| Ctsc | 9.62976248405467E-54 | 1.31457238506344 | 0.93 | 0.793 | 2.9903301441735E-49 | 2 | Ctsc |
| Rps20 | 5.13338900201088E-53 | 0.935431181690971 | 0.996 | 0.975 | 1.59407128679444E-48 | 2 | Rps20 |
| Scoc | 2.23458401200199E-52 | -1.23605155298513 | 0.522 | 0.879 | 6.93905373246979E-48 | 2 | Scoc |
| Mtdh | 3.82516583174803E-52 | -1.02669205808061 | 0.803 | 0.951 | 1.18782874573271E-47 | 2 | Mtdh |
| Rpl32 | 5.24332040362482E-52 | 1.07124573322813 | 0.996 | 0.976 | 1.62820828493762E-47 | 2 | Rpl32 |
| Cd37.1 | 6.94897702491066E-52 | -1.15320572072195 | 0.61 | 0.912 | 2.15786583554551E-47 | 2 | Cd37 |
| Abi3 | 1.40466419235075E-50 | -1.28755434239719 | 0.487 | 0.813 | 4.36190371650678E-46 | 2 | Abi3 |
| Rpsa.1 | 1.61464824119565E-50 | 1.29934946888645 | 0.996 | 0.927 | 5.01396718338487E-46 | 2 | Rpsa |
| Rps27.1 | 2.69830768249901E-49 | 0.918246575738408 | 1 | 0.99 | 8.37905484646417E-45 | 2 | Rps27 |
| Ywhah | 6.04398170133699E-49 | -1.10395660944086 | 0.645 | 0.891 | 1.87683763771618E-44 | 2 | Ywhah |
| Rps5 | 8.32478323478892E-49 | 1.17874358688984 | 0.982 | 0.935 | 2.585094937899E-44 | 2 | Rps5 |
| Rps21.1 | 1.33097135745923E-48 | 0.780835835863661 | 1 | 0.993 | 4.13306535631816E-44 | 2 | Rps21 |
| Smap2.1 | 3.31030089662752E-48 | -1.11136371948752 | 0.596 | 0.885 | 1.02794773742974E-43 | 2 | Smap2 |
| Sirpa | 2.78083792154E-47 | -0.962211967519336 | 0.803 | 0.953 | 8.63533599775816E-43 | 2 | Sirpa |
| Tspo | 5.72334987447643E-47 | 1.10649210379914 | 0.702 | 0.285 | 1.77727183652117E-42 | 2 | Tspo |
| Abhd12 | 6.52282304221931E-47 | -0.94860583957851 | 0.811 | 0.956 | 2.02553223930036E-42 | 2 | Abhd12 |
| Rps8 | 9.6621982205978E-47 | 0.818544937418142 | 1 | 0.988 | 3.00040241344223E-42 | 2 | Rps8 |
| Rps11.1 | 1.7077466735659E-46 | 0.943239073532013 | 1 | 0.991 | 5.30306574542417E-42 | 2 | Rps11 |
| Srgap2 | 4.05268128991621E-46 | -1.07099346742274 | 0.645 | 0.892 | 1.25847912095768E-41 | 2 | Srgap2 |
| Ckb | 6.89654116612596E-46 | -0.998274422757237 | 0.741 | 0.947 | 2.14158292831709E-41 | 2 | Ckb |
| Rps18 | 1.31827227921182E-45 | 0.997279685321421 | 0.996 | 0.947 | 4.09363090863646E-41 | 2 | Rps18 |
| Ms4a6b | 9.16833993975502E-45 | 1.27792244054769 | 0.846 | 0.531 | 2.84704460149213E-40 | 2 | Ms4a6b |
| Crybb1.1 | 2.08730665091023E-44 | -1.47503900453205 | 0.346 | 0.751 | 6.48171334307153E-40 | 2 | Crybb1 |
| Rpl9 | 3.19767960741829E-44 | 0.818950457928426 | 0.991 | 0.971 | 9.92975448491602E-40 | 2 | Rpl9 |
| Rps19 | 1.42626373898353E-43 | 1.0256620630084 | 0.996 | 0.955 | 4.42897678866557E-39 | 2 | Rps19 |
| Cmtm6.1 | 5.2967192036309E-43 | -1.06640093420613 | 0.57 | 0.846 | 1.6447902143035E-38 | 2 | Cmtm6 |
| Rpl35 | 9.36879505784493E-43 | 0.888952922927077 | 0.987 | 0.879 | 2.90929192931259E-38 | 2 | Rpl35 |
| Rpl19 | 1.05040006003692E-42 | 0.878038009827109 | 1 | 0.979 | 3.26180730643264E-38 | 2 | Rpl19 |
| Susd3 | 3.00815492263626E-42 | -1.14012521134304 | 0.469 | 0.793 | 9.34122348126239E-38 | 2 | Susd3 |
| Ifngr1 | 3.48575387480282E-42 | -0.765868368922916 | 0.877 | 0.986 | 1.08243115074252E-37 | 2 | Ifngr1 |
| Rplp2 | 1.1310891111252E-41 | 0.782773604849853 | 1 | 0.97 | 3.51237101677709E-37 | 2 | Rplp2 |
| Lst1 | 4.63101893743306E-41 | 1.1576677823991 | 0.86 | 0.524 | 1.43807031064109E-36 | 2 | Lst1 |
| Bin2 | 5.72091079395355E-41 | -1.11533586630113 | 0.5 | 0.789 | 1.7765144288464E-36 | 2 | Bin2 |
| Rps2 | 9.37280871500094E-41 | 0.950537762372105 | 0.987 | 0.93 | 2.91053829026924E-36 | 2 | Rps2 |
| Rpl26 | 1.00539157764906E-40 | 0.795670949450634 | 1 | 0.978 | 3.12204246607362E-36 | 2 | Rpl26 |
| Rpl37.1 | 1.19094531662208E-40 | 0.780985551079742 | 0.996 | 0.99 | 3.69824249170654E-36 | 2 | Rpl37 |
| Rps13 | 1.73666732047638E-40 | 0.924497538292701 | 1 | 0.973 | 5.39287303027529E-36 | 2 | Rps13 |
| Itgam | 2.74450358186367E-40 | -1.11131849358202 | 0.509 | 0.799 | 8.52250697276126E-36 | 2 | Itgam |
| Rpl28 | 5.60285208941422E-40 | 0.857128531911679 | 0.996 | 0.962 | 1.7398536593258E-35 | 2 | Rpl28 |
| mt-Co2 | 6.47857775620827E-40 | 0.670142604630995 | 1 | 0.996 | 2.01179275063535E-35 | 2 | mt-Co2 |
| Ivns1abp | 7.67752062451974E-40 | -1.06242646981053 | 0.548 | 0.829 | 2.38410047953212E-35 | 2 | Ivns1abp |
| Rps15a | 1.94187121079116E-39 | 0.941272043627357 | 1 | 0.98 | 6.03009267086979E-35 | 2 | Rps15a |
| mt-Atp6 | 2.24104034651716E-39 | 0.69265004247674 | 1 | 0.998 | 6.95910258803974E-35 | 2 | mt-Atp6 |
| Pycard | 3.65055581578758E-39 | -0.950699305390988 | 0.702 | 0.859 | 1.13360709747652E-34 | 2 | Pycard |

*(Continued)*

**Table 1.** (Continued)

| | p_val | avg_log2FC | pct.1 | pct.2 | p_val_adj | cluster | gene |
|---|---|---|---|---|---|---|---|
| Pmepa1 | 2.07730008367523E-38 | -0.983359951596355 | 0.697 | 0.945 | 6.45063994983671E-34 | 2 | Pmepa1 |
| Rps3 | 2.75588349361709E-38 | 0.728452549014768 | 1 | 0.978 | 8.55784501272914E-34 | 2 | Rps3 |
| Cd53 | 4.49045896493436E-38 | -0.759006897984039 | 0.899 | 0.955 | 1.39442222238107E-33 | 2 | Cd53 |
| Rpl13 | 6.10702094950391E-38 | 0.819593896024524 | 1 | 0.995 | 1.89641321544945E-33 | 2 | Rpl13 |
| Rpl22 | 1.59647247931969E-37 | 0.790790226705827 | 0.987 | 0.901 | 4.95752599003143E-33 | 2 | Rpl22 |
| Tsc22d3 | 1.75280341918441E-37 | 1.0081073961963 | 0.75 | 0.374 | 5.44298045759336E-33 | 2 | Tsc22d3 |
| Rpl39 | 4.49669520593599E-37 | 0.861742369483312 | 1 | 0.989 | 1.3963587622993E-32 | 2 | Rpl39 |
| Csnk1e | 4.54898314242523E-37 | -1.09265166129746 | 0.504 | 0.757 | 1.41259573521731E-32 | 2 | Csnk1e |
| Zfhx3 | 6.13786298668869E-37 | -0.934197836332535 | 0.689 | 0.913 | 1.90599059325644E-32 | 2 | Zfhx3 |
| Tanc2 | 7.87999510824324E-37 | -1.05982725448567 | 0.43 | 0.779 | 2.44697488096277E-32 | 2 | Tanc2 |
| Ltc4s | 1.41946911233159E-36 | -0.924676039563155 | 0.684 | 0.945 | 4.40787743452328E-32 | 2 | Ltc4s |
| Rps3a1 | 1.57345645959343E-36 | 0.729620660822639 | 1 | 0.987 | 4.88605434397548E-32 | 2 | Rps3a1 |
| Rpl27a.1 | 3.7615177916447E-36 | 0.733875411989987 | 1 | 0.994 | 1.16806411983943E-31 | 2 | Rpl27a |
| Rpl8 | 1.19032041802324E-35 | 0.801573760709803 | 0.991 | 0.95 | 3.69630199408756E-31 | 2 | Rpl8 |
| Ccr5.1 | 1.39313133410864E-35 | -1.02873563003041 | 0.61 | 0.827 | 4.32609073180757E-31 | 2 | Ccr5 |
| Cyth4.1 | 1.61792099199388E-35 | -0.745483595441052 | 0.886 | 0.953 | 5.02413005643861E-31 | 2 | Cyth4 |
| Cd48 | 2.90081136043008E-35 | 0.761626993725923 | 0.794 | 0.371 | 9.00788951754353E-31 | 2 | Cd48 |
| Saraf | 3.24056360820088E-35 | -0.950425840993859 | 0.57 | 0.801 | 1.00629221725462E-30 | 2 | Saraf |
| mt-Co3 | 3.5772538527277E-35 | 0.625568359835822 | 1 | 0.997 | 1.11084463888753E-30 | 2 | mt-Co3 |
| C1qc | 3.75117077978036E-35 | -0.529470818122499 | 0.917 | 0.999 | 1.16485106224519E-30 | 2 | C1qc |
| Mertk | 5.63814159319826E-35 | -1.02146196342641 | 0.447 | 0.8 | 1.75081210893586E-30 | 2 | Mertk |
| Serinc3 | 6.44188403335451E-35 | -0.601390773295421 | 0.974 | 0.999 | 2.00039824887758E-30 | 2 | Serinc3 |
| Fcgrt | 6.54353484334029E-35 | 1.04901210452634 | 0.763 | 0.384 | 2.03196387490246E-30 | 2 | Fcgrt |
| Fam105a | 1.00338882832065E-34 | -0.832422395588597 | 0.754 | 0.891 | 3.11582332858412E-30 | 2 | Fam105a |
| Rps10 | 2.07572719222703E-34 | 0.686997136282966 | 1 | 0.987 | 6.44575565002258E-30 | 2 | Rps10 |
| Nrip1 | 2.11674730336753E-34 | -0.975113945672293 | 0.618 | 0.85 | 6.57313540114719E-30 | 2 | Nrip1 |
| Rps12 | 2.57156588815914E-34 | 0.668437002689803 | 1 | 0.991 | 7.98548355250059E-30 | 2 | Rps12 |
| Rps27a | 2.89960243850952E-34 | 0.748616172118519 | 0.996 | 0.991 | 9.00413545230361E-30 | 2 | Rps27a |
| Rps7 | 3.56506367281024E-34 | 0.847120536430435 | 0.991 | 0.962 | 1.10705922231776E-29 | 2 | Rps7 |
| Rpl10.1 | 8.387418159678E-34 | 0.699711580428214 | 0.996 | 0.986 | 2.60454496112481E-29 | 2 | Rpl10 |
| Rpl17 | 9.563550845489E-34 | 0.87383180455266 | 0.996 | 0.964 | 2.9697694440497E-29 | 2 | Rpl17 |
| Sh3bgrl3 | 9.76158924357623E-34 | 0.795986411196578 | 0.952 | 0.857 | 3.03126630780773E-29 | 2 | Sh3bgrl3 |
| Sft2d1 | 9.85822330190378E-34 | -0.88554150170981 | 0.689 | 0.859 | 3.06127408194018E-29 | 2 | Sft2d1 |
| Pag1 | 1.86811546576051E-33 | -1.06360172440379 | 0.439 | 0.735 | 5.80105895582611E-29 | 2 | Pag1 |
| Ybx1.1 | 1.96528295466773E-33 | 0.741968184007215 | 0.943 | 0.655 | 6.10279315912969E-29 | 2 | Ybx1 |
| Tmem173 | 2.14931437160311E-33 | -1.00883233801719 | 0.535 | 0.778 | 6.67426591813912E-29 | 2 | Tmem173 |
| Rps23 | 2.99005499986355E-33 | 0.682515133189967 | 0.991 | 0.986 | 9.28501779107628E-29 | 2 | Rps23 |
| Capza2 | 3.09653814842897E-33 | -0.704135964682668 | 0.868 | 0.956 | 9.61567991231647E-29 | 2 | Capza2 |
| Rgs2.1 | 1.40863055805506E-32 | -0.903480834111316 | 0.627 | 0.859 | 4.37422047192838E-28 | 2 | Rgs2 |
| Stab1 | 1.56342977170071E-32 | 1.85187288313102 | 0.776 | 0.601 | 4.85491847006221E-28 | 2 | Stab1 |
| Slco2b1 | 1.66100400695275E-32 | -0.848262399588577 | 0.645 | 0.892 | 5.15791574279036E-28 | 2 | Slco2b1 |
| Rrbp1 | 2.72969020582539E-32 | -0.694365179796032 | 0.912 | 0.977 | 8.4765069961496E-28 | 2 | Rrbp1 |
| Rpl7.1 | 2.79852752833827E-32 | 0.740244390373251 | 0.987 | 0.945 | 8.69026753374885E-28 | 2 | Rpl7 |
| Sem1 | 2.85631882126126E-32 | 0.760395327591849 | 0.873 | 0.575 | 8.8697268356626E-28 | 2 | Sem1 |
| Rpl30.1 | 3.65671927697142E-32 | 0.663597450528015 | 0.996 | 0.993 | 1.13552103707793E-27 | 2 | Rpl30 |
| H2afj | 7.25277323500704E-32 | 0.711669514668641 | 0.759 | 0.36 | 2.25220367266674E-27 | 2 | H2afj |
| Calm1.1 | 1.17470735463791E-31 | 0.776829449931705 | 0.952 | 0.734 | 3.64781874835709E-27 | 2 | Calm1 |

(*Continued*)

**Table 1.** (Continued)

| | p_val | avg_log2FC | pct.1 | pct.2 | p_val_adj | cluster | gene |
|---|---|---|---|---|---|---|---|
| **Itm2c.1** | 1.29778111744128E-31 | -0.783257557834447 | 0.776 | 0.885 | 4.02999970399041E-27 | 2 | Itm2c |
| **Cd52.1** | 1.55573908049899E-31 | 1.84343415323955 | 0.785 | 0.575 | 4.83103656667353E-27 | 2 | Cd52 |
| **Rpl18** | 1.66767624865276E-31 | 0.790669945685992 | 0.996 | 0.955 | 5.17863505494141E-27 | 2 | Rpl18 |
| **Rnase4.1** | 4.5759743818341E-31 | -0.781528453072231 | 0.746 | 0.955 | 1.42097732479094E-26 | 2 | Rnase4 |
| **Cyfip1** | 8.17903003125012E-31 | -0.808421153189823 | 0.728 | 0.884 | 2.5398341956041E-26 | 2 | Cyfip1 |
| **Rps15** | 1.11461615447363E-30 | 0.76834726437119 | 0.965 | 0.856 | 3.46121754448695E-26 | 2 | Rps15 |
| **Rps4x.1** | 3.29110068340267E-30 | 0.778118990663748 | 0.996 | 0.984 | 1.02198549521703E-25 | 2 | Rps4x |
| **Rpl24** | 3.5565992195344E-30 | 0.781497756878446 | 0.987 | 0.912 | 1.10443075564202E-25 | 2 | Rpl24 |
| **Elmo1** | 6.62421466554566E-30 | -0.976764742893299 | 0.526 | 0.76 | 2.05701738009189E-25 | 2 | Elmo1 |
| **Gmfg** | 1.01118716645815E-29 | 0.73616623240517 | 0.772 | 0.396 | 3.14003950800251E-25 | 2 | Gmfg |
| **Fam102b** | 1.08754203658136E-29 | -1.04028643720283 | 0.425 | 0.71 | 3.37714428619609E-25 | 2 | Fam102b |
| **Rps9.1** | 1.09511560293501E-29 | 0.704884356242325 | 1 | 0.992 | 3.4006624817941E-25 | 2 | Rps9 |
| **Prdx1.1** | 1.51334436846022E-29 | 0.841059755905203 | 0.882 | 0.562 | 4.69938826737951E-25 | 2 | Prdx1 |
| **Arsb** | 2.34653092631895E-29 | -0.858093440530267 | 0.575 | 0.812 | 7.28668248549822E-25 | 2 | Arsb |
| **St3gal6** | 2.47913778446241E-29 | -0.831997443141878 | 0.575 | 0.81 | 7.69846656209111E-25 | 2 | St3gal6 |
| **Ubc.1** | 4.72295742003184E-29 | -0.596793288073421 | 0.961 | 0.982 | 1.46661996764249E-24 | 2 | Ubc |
| **Scamp2** | 1.7992441589114E-28 | -0.699814393833229 | 0.816 | 0.923 | 5.58719288666757E-24 | 2 | Scamp2 |
| **Hpgds.1** | 1.88230663134722E-28 | -0.782773761312334 | 0.684 | 0.896 | 5.84512678232252E-24 | 2 | Hpgds |
| **Rps17** | 5.19530330078899E-28 | 0.670345331197419 | 0.956 | 0.737 | 1.61329753399401E-23 | 2 | Rps17 |
| **Rpl18a.1** | 7.76043637396675E-28 | 0.793812803597784 | 0.991 | 0.99 | 2.40984830720789E-23 | 2 | Rpl18a |
| **Bsg** | 8.15809532181704E-28 | -0.704895218165514 | 0.838 | 0.908 | 2.53333334028384E-23 | 2 | Bsg |
| **Fcrls.1** | 1.10454665367038E-27 | -0.698843764515964 | 0.807 | 0.984 | 3.42994872364262E-23 | 2 | Fcrls |
| **Fcgr2b** | 1.30796736690552E-27 | 0.923588939518299 | 0.833 | 0.534 | 4.06163106445171E-23 | 2 | Fcgr2b |
| **Rpl6** | 1.3304284799526E-27 | 0.629321133923917 | 0.996 | 0.977 | 4.13137955879682E-23 | 2 | Rpl6 |
| **Rps25.1** | 1.75439516044992E-27 | 0.675317724024114 | 0.987 | 0.915 | 5.44792329174513E-23 | 2 | Rps25 |
| **Mef2c** | 1.8442992871488E-27 | -0.625578907102627 | 0.925 | 0.982 | 5.72710257638315E-23 | 2 | Mef2c |
| **Rpl29** | 2.43402483342969E-27 | 0.712549025409821 | 0.978 | 0.897 | 7.55837731524921E-23 | 2 | Rpl29 |
| **Rpl10a** | 3.02778906567368E-27 | 0.815767368598848 | 0.978 | 0.901 | 9.40219338563647E-23 | 2 | Rpl10a |
| **Comt** | 3.06714731521305E-27 | -0.799587225546452 | 0.632 | 0.804 | 9.5244125579311E-23 | 2 | Comt |
| **Rpl11** | 3.40973355260908E-27 | 0.680598526419851 | 1 | 0.974 | 1.0588245600917E-22 | 2 | Rpl11 |
| **B2m** | 3.44383123112973E-27 | 0.579322228662936 | 0.991 | 0.964 | 1.06941291220271E-22 | 2 | B2m |
| **Tmem256** | 6.1781720998658E-27 | 0.677440811848066 | 0.719 | 0.35 | 1.91850778217133E-22 | 2 | Tmem256 |
| **Glul.1** | 1.21799877868603E-26 | -0.76840490613593 | 0.711 | 0.9 | 3.78225160745373E-22 | 2 | Glul |
| **Tpst2** | 1.50011468190259E-26 | -0.852008747731548 | 0.588 | 0.747 | 4.65830612171211E-22 | 2 | Tpst2 |
| **Mef2a** | 1.61073037904817E-26 | -0.771936677762354 | 0.728 | 0.886 | 5.00180104605828E-22 | 2 | Mef2a |
| **Cox8a** | 3.46709588545212E-26 | 0.639076388156213 | 0.978 | 0.773 | 1.07663728530945E-21 | 2 | Cox8a |
| **Fam49b** | 5.58292474366426E-26 | -0.672354580710964 | 0.789 | 0.885 | 1.73366562065006E-21 | 2 | Fam49b |
| **Rhoh.1** | 6.13284524338528E-26 | -0.879333826138547 | 0.496 | 0.71 | 1.90443243342843E-21 | 2 | Rhoh |
| **Camk1** | 1.48727846897817E-25 | -0.710522147433676 | 0.732 | 0.862 | 4.6184458297179E-21 | 2 | Camk1 |
| **Commd8** | 1.65174766647102E-25 | -0.852890850953875 | 0.579 | 0.741 | 5.12917202869247E-21 | 2 | Commd8 |
| **Ctsz** | 1.83029403078458E-25 | -0.603759026638196 | 0.969 | 0.992 | 5.68361205379535E-21 | 2 | Ctsz |
| **Rack1** | 1.93941295589834E-25 | 0.863300856820349 | 0.956 | 0.874 | 6.02245905195112E-21 | 2 | Rack1 |
| **C1qb.1** | 2.01183469974534E-25 | -0.46318479879185 | 0.921 | 1 | 6.24735029311922E-21 | 2 | C1qb |
| **Lyn** | 2.28953271321653E-25 | -0.702786006518549 | 0.741 | 0.889 | 7.10968593435129E-21 | 2 | Lyn |
| **Calm2** | 3.65052984655318E-25 | -0.614166804344637 | 0.895 | 0.955 | 1.13359903325016E-20 | 2 | Calm2 |
| **Hint1.1** | 3.985705354814E-25 | 0.654508830560937 | 0.899 | 0.573 | 1.23768108383039E-20 | 2 | Hint1 |
| **Rpl12** | 6.43598945712959E-25 | 0.756182890797111 | 0.969 | 0.93 | 1.99856780612245E-20 | 2 | Rpl12 |

(*Continued*)

**Table 1.** (Continued)

| | p_val | avg_log2FC | pct.1 | pct.2 | p_val_adj | cluster | gene |
|---|---|---|---|---|---|---|---|
| **Psmb8** | 7.75352707732209E-25 | 0.735953070738275 | 0.868 | 0.582 | 2.40770276332083E-20 | 2 | Psmb8 |
| **Rps26** | 7.80870237134121E-25 | 0.834676440921305 | 0.991 | 0.894 | 2.42483634737259E-20 | 2 | Rps26 |
| **Inpp5d** | 1.14097929978285E-24 | -0.723984430038733 | 0.715 | 0.862 | 3.54308301961567E-20 | 2 | Inpp5d |
| **Aes** | 2.48151757874713E-24 | 0.583932304064419 | 0.746 | 0.366 | 7.70585653728345E-20 | 2 | Aes |
| **Myl6** | 2.81157060156106E-24 | 0.68334685805671 | 0.908 | 0.668 | 8.73077018902755E-20 | 2 | Myl6 |
| **Tmsb4x** | 5.96640437559762E-24 | 0.351196725090372 | 1 | 1 | 1.85274755075433E-19 | 2 | Tmsb4x |
| **Lrp1** | 6.03510512267145E-24 | -0.830068501201574 | 0.539 | 0.719 | 1.87408119374317E-19 | 2 | Lrp1 |
| **Kctd12** | 9.14781295030041E-24 | -0.76111521251526 | 0.693 | 0.866 | 2.84067035545679E-19 | 2 | Kctd12 |
| **Retreg1** | 1.93719930313547E-23 | -0.832257718252235 | 0.526 | 0.706 | 6.01558499602658E-19 | 2 | Retreg1 |
| **Arhgap45.1** | 2.14521954624873E-23 | -0.71780158952278 | 0.746 | 0.864 | 6.66155025696619E-19 | 2 | Arhgap45 |
| **Cttnbp2nl** | 3.33725450876171E-23 | -0.827507243556233 | 0.553 | 0.74 | 1.03631764260578E-18 | 2 | Cttnbp2nl |
| **Rpl14** | 3.94460636808284E-23 | 0.684022418979118 | 0.965 | 0.848 | 1.22491861548076E-18 | 2 | Rpl14 |
| **Ucp2.1** | 4.45340374812402E-23 | 0.755344589434527 | 0.917 | 0.712 | 1.38291546590495E-18 | 2 | Ucp2 |
| **Pmp22** | 5.63640904019733E-23 | -0.847044435448738 | 0.548 | 0.77 | 1.75027409925248E-18 | 2 | Pmp22 |
| **Tmem59** | 6.00716115893676E-23 | -0.575754757571115 | 0.868 | 0.92 | 1.86540375468463E-18 | 2 | Tmem59 |
| **Rpl27** | 1.15482406206726E-22 | 0.740210454514907 | 0.925 | 0.717 | 3.58607515993747E-18 | 2 | Rpl27 |
| **Rpl23a** | 1.50294500510015E-22 | 0.707093700402501 | 0.925 | 0.742 | 4.6670951243375E-18 | 2 | Rpl23a |
| **Daglb** | 3.66474970894117E-22 | -0.773376118406158 | 0.509 | 0.7 | 1.1380147271175E-17 | 2 | Daglb |
| **Mpc1** | 5.21675078696293E-22 | -0.774423292019402 | 0.588 | 0.731 | 1.6199576218756E-17 | 2 | Mpc1 |
| **Tmem176a** | 5.3530396448917E-22 | 0.972885278837874 | 0.842 | 0.665 | 1.66227940092822E-17 | 2 | Tmem176a |
| **Tomm7** | 6.46684834253959E-22 | 0.580152159964919 | 0.825 | 0.474 | 2.00815041580882E-17 | 2 | Tomm7 |
| **Pfdn5** | 7.24795528442122E-22 | 0.568547018024899 | 0.956 | 0.762 | 2.25070755447132E-17 | 2 | Pfdn5 |
| **Ctsb.1** | 9.70653139672359E-22 | 0.669640432860307 | 0.969 | 0.981 | 3.01416919462458E-17 | 2 | Ctsb |
| **Actr3** | 1.08420298615427E-21 | 0.687592131968084 | 0.851 | 0.605 | 3.36677553290485E-17 | 2 | Actr3 |
| **Rpl3** | 1.16736165147781E-21 | 0.674643660646569 | 0.987 | 0.95 | 3.62500813633405E-17 | 2 | Rpl3 |
| **Tmem86a.** | 1.86614112622408E-21 | -0.692928953459548 | 0.667 | 0.803 | 5.79492803926364E-17 | 2 | Tmem86a |
| **Rpl7a** | 3.90595359722725E-21 | 0.589709881901216 | 0.974 | 0.862 | 1.21291577054698E-16 | 2 | Rpl7a |
| **Bmp2k** | 3.91901003201135E-21 | -0.769547939336852 | 0.526 | 0.703 | 1.21697018524048E-16 | 2 | Bmp2k |
| **Pid1** | 4.28378399015217E-21 | 0.840207692327847 | 0.763 | 0.494 | 1.33024344246195E-16 | 2 | Pid1 |
| **Entpd1.1** | 5.23967263322189E-21 | -0.739274124008819 | 0.596 | 0.776 | 1.62707554279439E-16 | 2 | Entpd1 |
| **Naca** | 5.84668580030608E-21 | 0.561787133525499 | 0.974 | 0.837 | 1.81557134156905E-16 | 2 | Naca |
| **Sec61g** | 7.94864504561606E-21 | 0.607931814588986 | 0.873 | 0.59 | 2.46829274601516E-16 | 2 | Sec61g |
| **Tmed5** | 1.61241682790358E-20 | -0.764937442767754 | 0.522 | 0.702 | 5.00703797568898E-16 | 2 | Tmed5 |
| **Snx3** | 1.62629907124698E-20 | 0.614404847258087 | 0.754 | 0.447 | 5.05014650594323E-16 | 2 | Snx3 |
| **Ppfia4** | 2.22624146172467E-20 | -0.774532140291307 | 0.57 | 0.721 | 6.91314761109361E-16 | 2 | Ppfia4 |
| **H2-T23** | 2.53369663680952E-20 | 0.674358150249728 | 0.763 | 0.466 | 7.8678881662846E-16 | 2 | H2-T23 |
| **Son.1** | 2.60482431248504E-20 | -0.535917096816427 | 0.89 | 0.951 | 8.08876093755979E-16 | 2 | Son |
| **Ehd4** | 3.11003457430786E-20 | 0.6897944233036 | 0.781 | 0.471 | 9.65759036359819E-16 | 2 | Ehd4 |
| **Mt1** | 3.47650849655269E-20 | 0.680895418997291 | 0.728 | 0.399 | 1.07956018343451E-15 | 2 | Mt1 |
| **Rpl36a** | 3.62895326120945E-20 | 0.650968035845183 | 0.952 | 0.841 | 1.12689885620337E-15 | 2 | Rpl36a |
| **Rps27l** | 6.43260459701728E-20 | 0.5421500450493 | 0.768 | 0.437 | 1.99751670551178E-15 | 2 | Rps27l |
| **Spcs2** | 6.86775129445452E-20 | -0.643852255285866 | 0.737 | 0.825 | 2.13264280946696E-15 | 2 | Spcs2 |
| **Tmem176b** | 7.85020462903516E-20 | 0.773257735340698 | 0.904 | 0.836 | 2.43772404345429E-15 | 2 | Tmem176b |
| **Cox6c** | 8.57056456935238E-20 | 0.605245459499571 | 0.846 | 0.581 | 2.661417415721E-15 | 2 | Cox6c |
| **Cd63** | 1.27519916407629E-19 | 0.691534146975824 | 0.789 | 0.64 | 3.9598759642061E-15 | 2 | Cd63 |
| **Tcn2** | 1.4602976459746E-19 | -0.744801159901859 | 0.57 | 0.712 | 4.53466228004493E-15 | 2 | Tcn2 |
| **Atp5l** | 1.89838258220387E-19 | 0.567280719247145 | 0.882 | 0.591 | 5.89504743251766E-15 | 2 | Atp5l |

*(Continued)*

**Table 1.** (Continued)

| | p_val | avg_log2FC | pct.1 | pct.2 | p_val_adj | cluster | gene |
|---|---|---|---|---|---|---|---|
| Rpl15 | 3.07097086627031E-19 | 0.557641703050111 | 0.987 | 0.942 | 9.53628583102918E-15 | 2 | Rpl15 |
| Rnf13 | 3.87666192829017E-19 | -0.688859824809731 | 0.649 | 0.753 | 1.20381982859195E-14 | 2 | Rnf13 |
| Cox7c | 4.52920782720762E-19 | 0.555958155038005 | 0.904 | 0.719 | 1.40645490658278E-14 | 2 | Cox7c |
| Rplp1 | 5.91682305275823E-19 | 0.359073311600416 | 1 | 1 | 1.83735106257301E-14 | 2 | Rplp1 |
| Rpl31 | 7.77773702822651E-19 | 0.623893897532578 | 0.904 | 0.631 | 2.41522067937518E-14 | 2 | Rpl31 |
| C1qa | 7.81610007737381E-19 | -0.409850072089153 | 0.925 | 1 | 2.42713355702689E-14 | 2 | C1qa |
| Ptpn18 | 1.14761038011707E-18 | 0.516656744246749 | 0.934 | 0.806 | 3.56367451337755E-14 | 2 | Ptpn18 |
| Eef1a1.1 | 3.57994467976022E-18 | 0.568783481346547 | 1 | 0.999 | 1.11168022140594E-13 | 2 | Eef1a1 |
| Rps6 | 4.04092882049739E-18 | 0.643250621339852 | 0.982 | 0.886 | 1.25482962662905E-13 | 2 | Rps6 |
| H3f3a.1 | 4.40439225179128E-18 | 0.524539791686793 | 0.982 | 0.908 | 1.36769592594875E-13 | 2 | H3f3a |
| Eif3f | 7.11532154483854E-18 | 0.638658177445678 | 0.825 | 0.562 | 2.20952079931871E-13 | 2 | Eif3f |
| Eef1b2 | 7.75419073252325E-18 | 0.577171799788871 | 0.961 | 0.821 | 2.40790884817044E-13 | 2 | Eef1b2 |
| Arpc3 | 1.04315733764028E-17 | 0.483922068005486 | 0.974 | 0.786 | 3.23931648057436E-13 | 2 | Arpc3 |
| Arglu1 | 2.02097951461616E-17 | -0.682988503365991 | 0.662 | 0.751 | 6.27574768673757E-13 | 2 | Arglu1 |
| Eif3h | 3.41739574262987E-17 | 0.562016403816608 | 0.789 | 0.504 | 1.06120389995885E-12 | 2 | Eif3h |
| Prdx5 | 3.87550996381471E-17 | 0.739891645843199 | 0.846 | 0.588 | 1.20346210906338E-12 | 2 | Prdx5 |
| Pnp | 4.90305156600558E-17 | -0.494625803065984 | 0.816 | 0.872 | 1.52254460279171E-12 | 2 | Pnp |
| Tgfb1 | 6.88233198928374E-17 | -0.586690661816359 | 0.772 | 0.834 | 2.13717055263228E-12 | 2 | Tgfb1 |
| Foxn3 | 7.3705271395484E-17 | -0.667641328948523 | 0.61 | 0.746 | 2.28876979264396E-12 | 2 | Foxn3 |
| Ms4a6d | 8.60366474539993E-17 | 0.61154469289023 | 0.724 | 0.447 | 2.67169601338904E-12 | 2 | Ms4a6d |
| Rplp0 | 1.09056191663698E-16 | 0.702211541507748 | 0.987 | 0.971 | 3.38652191973281E-12 | 2 | Rplp0 |
| Hspa8.1 | 1.56197945904361E-16 | 0.470378971744203 | 0.965 | 0.887 | 4.85041481416811E-12 | 2 | Hspa8 |
| Atp5e | 2.42588741559493E-16 | 0.585325510235871 | 0.904 | 0.701 | 7.53310819164692E-12 | 2 | Atp5e |
| Rpl21 | 3.13796727693619E-16 | 0.422360188942156 | 1 | 0.992 | 9.74432978506994E-12 | 2 | Rpl21 |
| Tnfaip8l2.1 | 3.64328222922003E-16 | -0.604274789119811 | 0.68 | 0.783 | 1.1313484306397E-11 | 2 | Tnfaip8l2 |
| Ptma.1 | 3.81800635973008E-16 | 0.537655722787899 | 0.974 | 0.898 | 1.18560551488698E-11 | 2 | Ptma |
| Irf8.1 | 5.12200357781794E-16 | -0.352892873093243 | 0.75 | 0.883 | 1.59053577101981E-11 | 2 | Irf8 |
| Sec11c | 5.63946739952162E-16 | -0.612254554062919 | 0.632 | 0.732 | 1.75122381157345E-11 | 2 | Sec11c |
| Serp1 | 9.88418587553427E-16 | 0.541444407721974 | 0.715 | 0.416 | 3.06933623992966E-11 | 2 | Serp1 |
| Clec4a2 | 1.53916622018373E-15 | 0.590675749404319 | 0.737 | 0.447 | 4.77957286353653E-11 | 2 | Clec4a2 |
| Serf2 | 1.63799845503095E-15 | 0.453107248407598 | 0.939 | 0.845 | 5.0864766024076E-11 | 2 | Serf2 |
| Cd164.1 | 1.71026950314588E-15 | -0.584039146591802 | 0.588 | 0.786 | 5.31089988811891E-11 | 2 | Cd164 |
| 2010107E0 | 2.22991439473136E-15 | 0.450199688477386 | 0.746 | 0.431 | 6.92455316995929E-11 | 2 | 2010107E0 |
| Smdt1 | 2.74099793053947E-15 | 0.49467478923778 | 0.715 | 0.43 | 8.51162087370421E-11 | 2 | Smdt1 |
| Rpl22l1 | 4.23611667368067E-15 | 0.522468784729402 | 0.754 | 0.479 | 1.31544131067806E-10 | 2 | Rpl22l1 |
| Cd33 | 4.29042563578647E-15 | -0.636769274627958 | 0.632 | 0.742 | 1.33230587268077E-10 | 2 | Cd33 |
| Evi2a | 5.66013255909521E-15 | -0.639937308576002 | 0.693 | 0.758 | 1.75764096357583E-10 | 2 | Evi2a |
| Selenos.1 | 6.53286622059542E-15 | -0.596870120933453 | 0.689 | 0.766 | 2.0286509474815E-10 | 2 | Selenos |
| Adap2.1 | 8.59651579534985E-15 | -0.60070179142482 | 0.614 | 0.75 | 2.66947604992999E-10 | 2 | Adap2 |
| Atp2b1 | 9.88937082897032E-15 | 0.563152562728331 | 0.785 | 0.544 | 3.07094632352015E-10 | 2 | Atp2b1 |
| Itm2b.1 | 1.56483621131164E-14 | -0.34207768307802 | 0.996 | 1 | 4.85928588698605E-10 | 2 | Itm2b |
| Cox4i1 | 1.73829051328893E-14 | 0.465310919735593 | 0.943 | 0.787 | 5.39791353091612E-10 | 2 | Cox4i1 |
| Fam46a | 3.47214424122003E-14 | 0.476616892301095 | 0.746 | 0.485 | 1.07820495122605E-09 | 2 | Fam46a |
| Hsp90b1.1 | 6.18215951978017E-14 | -0.484638532373759 | 0.886 | 0.926 | 1.91974599567733E-09 | 2 | Hsp90b1 |
| Orai1.1 | 6.71121736687443E-14 | -0.619303901072518 | 0.627 | 0.703 | 2.08403432893552E-09 | 2 | Orai1 |
| Adgre1 | 6.99987429333859E-14 | 0.597590887376601 | 0.789 | 0.596 | 2.17367096431043E-09 | 2 | Adgre1 |
| Arpc1b | 7.63029712041265E-14 | 0.440521332744438 | 0.974 | 0.874 | 2.36943616480174E-09 | 2 | Arpc1b |

(*Continued*)

**Table 1.** (Continued)

| | p_val | avg_log2FC | pct.1 | pct.2 | p_val_adj | cluster | gene |
|---|---|---|---|---|---|---|---|
| Rab14 | 9.80977666089721E-14 | -0.464272454452693 | 0.855 | 0.884 | 3.04622994650841E-09 | 2 | Rab14 |
| Unc93b1.1 | 1.08174346111864E-13 | -0.430922987828662 | 0.939 | 0.972 | 3.3591379698117E-09 | 2 | Unc93b1 |
| Cebpb | 1.08548840940362E-13 | 0.809938093235967 | 0.829 | 0.666 | 3.37076715772105E-09 | 2 | Cebpb |
| Calr | 1.27446320696064E-13 | -0.49503901974792 | 0.877 | 0.918 | 3.95759059657486E-09 | 2 | Calr |
| Pdia3 | 1.39633655237055E-13 | -0.515791708181206 | 0.746 | 0.838 | 4.33604389607626E-09 | 2 | Pdia3 |
| mt-Nd5 | 1.68909338778915E-13 | 0.57037688786207 | 0.789 | 0.564 | 5.24514169710163E-09 | 2 | mt-Nd5 |
| Ncf1 | 1.83549288950651E-13 | -0.593821472983735 | 0.645 | 0.718 | 5.69975606978457E-09 | 2 | Ncf1 |
| Hk2 | 1.94290967488945E-13 | -0.615854643090809 | 0.605 | 0.725 | 6.03331741343421E-09 | 2 | Hk2 |
| Akr1a1 | 1.95008687975717E-13 | 0.472807958731164 | 0.882 | 0.609 | 6.05560478770994E-09 | 2 | Akr1a1 |
| Atox1 | 2.14680821129842E-13 | 0.459402577988173 | 0.86 | 0.642 | 6.66648353854499E-09 | 2 | Atox1 |
| Tmcc3.1 | 2.85500668047395E-13 | -0.572372442918023 | 0.64 | 0.778 | 8.86565224487575E-09 | 2 | Tmcc3 |
| Cox6b1 | 2.91510308413542E-13 | 0.432586906239381 | 0.825 | 0.566 | 9.05226960716572E-09 | 2 | Cox6b1 |
| Tmed9 | 4.1261014248474E-13 | -0.446285221443814 | 0.833 | 0.836 | 1.28127827545786E-08 | 2 | Tmed9 |
| Wasf2 | 5.80925546477217E-13 | -0.539868907824566 | 0.715 | 0.768 | 1.8039480994757E-08 | 2 | Wasf2 |
| Pou2f2 | 6.07475280846745E-13 | -0.523732848806352 | 0.759 | 0.855 | 1.8863929896134E-08 | 2 | Pou2f2 |
| Eif3k | 6.32109411786736E-13 | 0.46403757658889 | 0.706 | 0.433 | 1.96288935642135E-08 | 2 | Eif3k |
| Atp5j2.1 | 6.48674554089823E-13 | 0.400823430424063 | 0.868 | 0.613 | 2.01432909281513E-08 | 2 | Atp5j2 |
| Vapa | 7.76325531603396E-13 | -0.527717274224154 | 0.689 | 0.725 | 2.41072367328803E-08 | 2 | Vapa |
| Atp5h | 7.92275180019517E-13 | 0.435181751620122 | 0.829 | 0.597 | 2.46025211651461E-08 | 2 | Atp5h |
| Nsa2 | 1.07668680358859E-12 | 0.495672231672429 | 0.724 | 0.492 | 3.34343553118363E-08 | 2 | Nsa2 |
| Picalm.1 | 1.15093543288937E-12 | -0.551282814916409 | 0.75 | 0.801 | 3.57399979975136E-08 | 2 | Picalm |
| Tmbim6 | 1.18646866729683E-12 | -0.409831108048267 | 0.908 | 0.928 | 3.68434115255686E-08 | 2 | Tmbim6 |
| Arpc5 | 1.35497631737332E-12 | 0.441132733784555 | 0.785 | 0.523 | 4.20760795833937E-08 | 2 | Arpc5 |
| Rpl13a | 1.6774315897726E-12 | 0.465434219821935 | 0.759 | 0.48 | 5.20892831572086E-08 | 2 | Rpl13a |
| Cox7a2.1 | 1.87555283437666E-12 | 0.391213564657996 | 0.785 | 0.5 | 5.82415421658985E-08 | 2 | Cox7a2 |
| Ndufa6 | 3.24436685093557E-12 | 0.452574139901997 | 0.728 | 0.467 | 1.00747323822102E-07 | 2 | Ndufa6 |
| Snrnp70 | 4.13613670415112E-12 | -0.495022876687839 | 0.759 | 0.801 | 1.28439453074005E-07 | 2 | Snrnp70 |
| Scand1 | 4.68957594510304E-12 | 0.433166918575086 | 0.724 | 0.449 | 1.45625401823285E-07 | 2 | Scand1 |
| Npc2.1 | 5.60309454406054E-12 | 0.380155710831326 | 0.961 | 0.906 | 1.73992894876712E-07 | 2 | Npc2 |
| Cox5b | 5.69592409992186E-12 | 0.412029800084562 | 0.728 | 0.477 | 1.76875531074874E-07 | 2 | Cox5b |
| Arl6ip1.1 | 5.83813307652816E-12 | -0.458777819236145 | 0.785 | 0.848 | 1.81291546425429E-07 | 2 | Arl6ip1 |
| Eef1g | 6.47500465450656E-12 | 0.428483441042687 | 0.706 | 0.455 | 2.01068319536392E-07 | 2 | Eef1g |
| Gabarap.1 | 1.01582746029182E-11 | 0.390442387362671 | 0.952 | 0.826 | 3.15444901244419E-07 | 2 | Gabarap |
| Nfkbia | 1.15086054013787E-11 | 0.482027663203991 | 0.719 | 0.447 | 3.57376723529014E-07 | 2 | Nfkbia |
| Erp29 | 1.21534700611123E-11 | -0.39655207901034 | 0.912 | 0.94 | 3.77401705807719E-07 | 2 | Erp29 |
| Fcgr3.1 | 1.2902289123316E-11 | -0.375803714089854 | 0.904 | 0.977 | 4.00654784146332E-07 | 2 | Fcgr3 |
| Git2 | 1.29284053524272E-11 | -0.55988854371515 | 0.627 | 0.725 | 4.01465771408923E-07 | 2 | Git2 |
| Tsc22d4 | 1.48716300925404E-11 | -0.494477224250263 | 0.737 | 0.817 | 4.61808729263658E-07 | 2 | Tsc22d4 |
| Cmtm7 | 1.92498356468171E-11 | -0.510427757890673 | 0.772 | 0.814 | 5.97765146340613E-07 | 2 | Cmtm7 |
| Gng5 | 2.05514432429599E-11 | 0.413648900219296 | 0.943 | 0.791 | 6.38183967023634E-07 | 2 | Gng5 |
| Man2b1 | 2.62346445839065E-11 | -0.415589677224585 | 0.895 | 0.924 | 8.14664418264049E-07 | 2 | Man2b1 |
| Uqcrh | 2.81757942265659E-11 | 0.475159873244601 | 0.846 | 0.615 | 8.74942938117551E-07 | 2 | Uqcrh |
| Atp6v0b | 4.61401899983726E-11 | -0.343793062997714 | 0.952 | 0.974 | 1.43279132001947E-06 | 2 | Atp6v0b |
| Rsrp1.1 | 6.43177903556122E-11 | -0.370787565919689 | 0.956 | 0.967 | 1.99726034391283E-06 | 2 | Rsrp1 |
| Jpt1 | 8.62927444891784E-11 | 0.341009440074251 | 0.75 | 0.489 | 2.67964859462246E-06 | 2 | Jpt1 |
| Fcer1g.1 | 1.08958511112352E-10 | 0.272297431879171 | 0.987 | 0.998 | 3.38348864557187E-06 | 2 | Fcer1g |
| Creg1 | 1.41038051668578E-10 | -0.475781888805909 | 0.724 | 0.78 | 4.37965461846435E-06 | 2 | Creg1 |

*(Continued)*

**Table 1.** (*Continued*)

| | p_val | avg_log2FC | pct.1 | pct.2 | p_val_adj | cluster | gene |
|---|---|---|---|---|---|---|---|
| **Psme1** | 1.47323393780053E-10 | 0.422671529364266 | 0.816 | 0.55 | 4.57483334705199E-06 | 2 | Psme1 |
| **Ubl5** | 1.47450856708473E-10 | 0.392864945793996 | 0.803 | 0.543 | 4.57879145336821E-06 | 2 | Ubl5 |
| **Rpl5** | 1.99864697414574E-10 | 0.476468163493443 | 0.978 | 0.884 | 6.20639844881477E-06 | 2 | Rpl5 |
| **Pld4** | 2.30090399846905E-10 | -0.408974129671601 | 0.886 | 0.931 | 7.14499718644595E-06 | 2 | Pld4 |
| **Vdac2** | 3.919204262526E-10 | -0.442481150148007 | 0.728 | 0.78 | 1.2170304996422E-05 | 2 | Vdac2 |
| **Tmem55b** | 4.56930863337687E-10 | -0.467993373728767 | 0.654 | 0.708 | 1.41890740992252E-05 | 2 | Tmem55b |
| **Mpeg1.1** | 4.71286620147909E-10 | -0.508901748180138 | 0.68 | 0.77 | 1.4634863415453E-05 | 2 | Mpeg1 |
| **Ost4** | 6.35857589584537E-10 | 0.353310442518575 | 0.719 | 0.453 | 1.97452857293686E-05 | 2 | Ost4 |
| **Gpr183** | 7.32353795986329E-10 | -0.499465593977062 | 0.654 | 0.732 | 2.27417824267635E-05 | 2 | Gpr183 |
| **Cltc** | 7.47922259956494E-10 | 0.473139696164029 | 0.825 | 0.645 | 2.3225229938429E-05 | 2 | Cltc |
| **Zfp36l1** | 8.51516131995623E-10 | 0.537085332735128 | 0.807 | 0.648 | 2.64421304468601E-05 | 2 | Zfp36l1 |
| **Cd84** | 8.65355225546132E-10 | -0.522710669275974 | 0.658 | 0.72 | 2.6871875818884E-05 | 2 | Cd84 |
| **Tmed2** | 8.90348849238215E-10 | -0.427134474025206 | 0.746 | 0.752 | 2.76480028153943E-05 | 2 | Tmed2 |
| **Elob** | 1.29786719630003E-09 | 0.385235777012019 | 0.803 | 0.596 | 4.03026700467049E-05 | 2 | Elob |
| **Srsf9** | 1.4331232907252E-09 | -0.460266446373846 | 0.711 | 0.714 | 4.45027775468897E-05 | 2 | Srsf9 |
| **Aif1** | 1.60268597213407E-09 | -0.33429851865648 | 0.89 | 0.947 | 4.97682074926794E-05 | 2 | Aif1 |
| **Pdia6** | 1.94660805115674E-09 | -0.400423903771478 | 0.781 | 0.811 | 6.04480198125702E-05 | 2 | Pdia6 |
| **Chchd2** | 1.97686579783406E-09 | 0.317890286717916 | 0.921 | 0.758 | 6.13876136201412E-05 | 2 | Chchd2 |
| **Gpx1.1** | 2.93448564451033E-09 | 0.636187030617774 | 0.842 | 0.688 | 9.11245827189793E-05 | 2 | Gpx1 |
| **Tnfaip8** | 3.0552979798693E-09 | 0.35818091509245 | 0.776 | 0.5 | 9.48761681688815E-05 | 2 | Tnfaip8 |
| **Syngr2** | 3.3173348646555E-09 | -0.47593829723017 | 0.697 | 0.74 | 0.000103013199552147 | 2 | Syngr2 |
| **Ier5** | 6.21051944992603E-09 | -0.493171392040171 | 0.658 | 0.701 | 0.000192855260478553 | 2 | Ier5 |
| **Celf2** | 7.21452141717146E-09 | -0.442634223025389 | 0.719 | 0.784 | 0.000224032533567425 | 2 | Celf2 |
| **Sec62** | 7.48421264989531E-09 | -0.431900661025612 | 0.772 | 0.779 | 0.000232407255417199 | 2 | Sec62 |
| **Ndufb8** | 8.13975524301523E-09 | 0.305804498089606 | 0.702 | 0.448 | 0.000252763819561352 | 2 | Ndufb8 |
| **Srrm2** | 1.03075971495809E-08 | -0.413640317673794 | 0.838 | 0.836 | 0.000320081814285935 | 2 | Srrm2 |
| **Hsp90ab1.** | 1.07825571632432E-08 | 0.35167219462207 | 0.947 | 0.842 | 0.000334830747590192 | 2 | Hsp90ab1 |
| **Ndufa7** | 1.75951932985345E-08 | 0.276316399359865 | 0.759 | 0.492 | 0.000546383537499391 | 2 | Ndufa7 |
| **Hnrnpk** | 1.93996633697784E-08 | -0.417497792561588 | 0.763 | 0.78 | 0.000602417746621728 | 2 | Hnrnpk |
| **Lrrc25** | 3.70835540880664E-08 | 0.267227337642733 | 0.702 | 0.441 | 0.00115155560509673 | 2 | Lrrc25 |
| **Rpl36al** | 3.93891109134153E-08 | 0.370586462376766 | 0.759 | 0.561 | 0.00122315006119429 | 2 | Rpl36al |
| **Cd300c2.1** | 5.65715851736389E-08 | -0.32719075565138 | 0.842 | 0.902 | 0.00175671743439701 | 2 | Cd300c2 |
| **Bri3** | 6.46423008951682E-08 | 0.377938167940918 | 0.899 | 0.74 | 0.00200733736969766 | 2 | Bri3 |
| **Rpl4** | 6.9124941094832E-08 | 0.426069966657198 | 0.904 | 0.767 | 0.00214653679581782 | 2 | Rpl4 |
| **Atpif1** | 7.22844032746643E-08 | 0.303405735826749 | 0.842 | 0.629 | 0.00224464757488815 | 2 | Atpif1 |
| **Ssr4** | 7.25749685482032E-08 | -0.391343642451748 | 0.776 | 0.829 | 0.00225367049832735 | 2 | Ssr4 |
| **Luc7l2** | 1.29567945512172E-07 | -0.406661315044541 | 0.715 | 0.763 | 0.00402347341198948 | 2 | Luc7l2 |
| **Plekho1** | 1.52226021603442E-07 | -0.38843882657898 | 0.754 | 0.793 | 0.00472707464885167 | 2 | Plekho1 |
| **Sec61b** | 1.71237666370415E-07 | 0.378977311169671 | 0.737 | 0.518 | 0.0053174432538005 | 2 | Sec61b |
| **Npm1.1** | 1.71443561553923E-07 | 0.367223103794778 | 0.846 | 0.676 | 0.00532383691693398 | 2 | Npm1 |
| **mt-Nd1** | 2.46361188125729E-07 | 0.268297311325853 | 0.996 | 0.976 | 0.00765025397486825 | 2 | mt-Nd1 |
| **Grn** | 2.93040379222069E-07 | -0.320905218024805 | 0.912 | 0.967 | 0.0090997828959829 | 2 | Grn |
| **Slc3a2** | 3.3370673524831E-07 | -0.348898592772481 | 0.737 | 0.783 | 0.0103625952496658 | 2 | Slc3a2 |
| **Eef1d** | 3.55973951207723E-07 | 0.269002142533624 | 0.811 | 0.58 | 0.0110540591068534 | 2 | Eef1d |
| **Canx** | 4.41443875948037E-07 | -0.372167163753741 | 0.706 | 0.756 | 0.0137081566798144 | 2 | Canx |
| **Uqcrb** | 5.85956651206878E-07 | 0.298075406343959 | 0.719 | 0.49 | 0.0181957118899272 | 2 | Uqcrb |
| **Timp2.1** | 6.22958875926944E-07 | -0.309246663169858 | 0.754 | 0.829 | 0.0193447419741594 | 2 | Timp2 |

(*Continued*)

**Table 1.** (Continued)

| | p_val | avg_log2FC | pct.1 | pct.2 | p_val_adj | cluster | gene |
|---|---|---|---|---|---|---|---|
| **Ppia.1** | 6.78980834658084E-07 | 0.253461947964809 | 1 | 0.978 | 0.0210843918586375 | 2 | Ppia |
| **Cd68** | 6.89605518846881E-07 | -0.318176816151383 | 0.912 | 0.924 | 0.0214143201767522 | 2 | Cd68 |
| **Pnisr** | 7.93169266411563E-07 | -0.362165507531715 | 0.768 | 0.764 | 0.0246302852298783 | 2 | Pnisr |
| **AC149090.** | 9.49503992252892E-07 | -0.29735851658342 | 0.816 | 0.89 | 0.0294849474714291 | 2 | AC149090. |
| **Tmem50a** | 1.14847766254682E-06 | -0.309422963631777 | 0.86 | 0.867 | 0.0356636768550665 | 2 | Tmem50a |
| **Rbm3.1** | 1.18434860854952E-06 | 0.397520068361228 | 0.772 | 0.583 | 0.0367775773412884 | 2 | Rbm3 |
| **Bst2.1** | 2.01646289492601E-74 | 1.38439663150328 | 0.75 | 0.452 | 6.26172222761374E-70 | 3 | Bst2 |
| **Ly6e.2** | 1.92434322389912E-67 | 0.739293231124573 | 0.987 | 0.95 | 5.97566301317395E-63 | 3 | Ly6e |
| **H2-K1.1** | 4.96994621078869E-59 | 1.27752330969304 | 0.857 | 0.721 | 1.54331739683621E-54 | 3 | H2-K1 |
| **H2-D1.1** | 4.65728297656834E-54 | 1.30019558248527 | 0.86 | 0.73 | 1.44622608271377E-49 | 3 | H2-D1 |
| **B2m.1** | 3.28005039221323E-45 | 0.689996466519465 | 0.987 | 0.964 | 1.01855404829398E-40 | 3 | B2m |
| **Fcgr1** | 9.07626885562554E-33 | 0.605810030608563 | 0.943 | 0.848 | 2.8184537677374E-28 | 3 | Fcgr1 |
| **Ctss.2** | 4.9740995639246E-31 | 0.312989083056316 | 1 | 0.999 | 1.5446071375855E-26 | 3 | Ctss |
| **Cd52.2** | 1.8577059954905E-18 | 0.690089060723365 | 0.708 | 0.575 | 5.76873442779664E-14 | 3 | Cd52 |
| **P2ry12.1** | 3.3125041389265E-17 | -0.270613801836562 | 0.996 | 0.993 | 1.02863191026085E-12 | 3 | P2ry12 |
| **Fth1.1** | 6.59341646245358E-14 | 0.279168891302981 | 0.998 | 0.996 | 2.04745361408571E-09 | 3 | Fth1 |
| **Fcrls.2** | 1.98059140044036E-13 | -0.287053091425957 | 0.958 | 0.982 | 6.15033047578744E-09 | 3 | Fcrls |
| **Pmp22.1** | 1.82547290265218E-11 | -0.353956253292346 | 0.662 | 0.77 | 5.6686410046058E-07 | 3 | Pmp22 |
| **Ctsc.1** | 1.16760387624373E-10 | 0.256879870591931 | 0.86 | 0.793 | 3.62576031689965E-06 | 3 | Ctsc |
| **Ltc4s.1** | 2.00381776458447E-09 | -0.27512615461476 | 0.91 | 0.942 | 6.22245530436416E-05 | 3 | Ltc4s |
| **Scoc.1** | 5.03017141696932E-09 | -0.275312760761899 | 0.809 | 0.876 | 0.000156201913011148 | 3 | Scoc |
| **Ecscr.1** | 1.71398903752471E-08 | -0.26091754977706 | 0.792 | 0.867 | 0.000532245015822548 | 3 | Ecscr |
| **mt-Co1.1** | 4.89170012285646E-50 | 0.911573447295116 | 1 | 0.995 | 1.51901963915062E-45 | 4 | mt-Co1 |
| **mt-Atp6.1** | 4.88980916070267E-44 | 0.879017763203261 | 1 | 0.998 | 1.518432438673E-39 | 4 | mt-Atp6 |
| **mt-Co2.1** | 3.06810772268287E-36 | 0.796225032041781 | 0.996 | 0.996 | 9.52739491124712E-32 | 4 | mt-Co2 |
| **mt-Nd4.1** | 3.7542266892755E-34 | 0.754760456201156 | 0.996 | 0.985 | 1.16580001382072E-29 | 4 | mt-Nd4 |
| **mt-Co3.1** | 8.47897809981095E-33 | 0.767706717834705 | 1 | 0.997 | 2.63297706933429E-28 | 4 | mt-Co3 |
| **mt-Cytb** | 1.17225110026362E-29 | 0.7108238306546 | 1 | 0.995 | 3.64019134164862E-25 | 4 | mt-Cytb |
| **mt-Nd2** | 1.68140822162614E-26 | 0.75607274901141 | 0.996 | 0.984 | 5.22127695061567E-22 | 4 | mt-Nd2 |
| **mt-Nd5.1** | 1.79530917608E-25 | 0.912413599377397 | 0.787 | 0.564 | 5.57497358448124E-21 | 4 | mt-Nd5 |
| **mt-Nd1.1** | 2.63088450330986E-22 | 0.613005696500472 | 0.989 | 0.976 | 8.16968564812812E-18 | 4 | mt-Nd1 |
| **Cox8a.1** | 3.29069008015369E-16 | 0.706411586331548 | 0.886 | 0.773 | 1.02185799059013E-11 | 4 | Cox8a |
| **mt-Nd3** | 2.99026928224297E-12 | 0.502502909497057 | 0.924 | 0.877 | 9.28568320214909E-08 | 4 | mt-Nd3 |
| **Cox6c.1** | 2.5258890333218E-11 | 0.715996887894111 | 0.707 | 0.583 | 7.84364321517419E-07 | 4 | Cox6c |
| **Ly86.1** | 2.08763416869304E-10 | -0.274517307712901 | 0.973 | 0.997 | 6.48273038404251E-06 | 4 | Ly86 |
| **Rplp0.1** | 2.19739072057878E-10 | -0.31178168522705 | 0.966 | 0.971 | 6.82355740461328E-06 | 4 | Rplp0 |
| **Rps4x.2** | 3.39074375974169E-08 | -0.264979678898781 | 0.989 | 0.984 | 0.00105292765971259 | 4 | Rps4x |
| **Rps3.1** | 1.67955234009716E-07 | -0.259162542879798 | 0.97 | 0.978 | 0.0052155138817037 | 4 | Rps3 |
| **Ctsh.1** | 1.97605186356664E-07 | -0.260443089324299 | 0.932 | 0.959 | 0.00613623385193348 | 4 | Ctsh |
| **Calm1.2** | 5.22910084451952E-07 | 0.324001925143003 | 0.833 | 0.735 | 0.0162379268524865 | 4 | Calm1 |
| **Cd68.1** | 1.05331549467576E-06 | -0.291650987813162 | 0.882 | 0.925 | 0.0327086060561664 | 4 | Cd68 |
| **Ndufa4.2** | 1.10795031705453E-06 | 0.534336615943904 | 0.749 | 0.639 | 0.0344051811954944 | 4 | Ndufa4 |
| **Cox7c.1** | 1.13620517299142E-06 | 0.383436152828528 | 0.764 | 0.721 | 0.0352825792369025 | 4 | Cox7c |
| **Rpl12.1** | 1.31378473666861E-06 | -0.272205465122428 | 0.928 | 0.931 | 0.0407969574277704 | 4 | Rpl12 |
| **Ccr1** | 0 | 3.07531506005891 | 1 | 0.022 | 0 | 5 | Ccr1 |
| **Ccr5.2** | 5.14078518699322E-24 | -0.753167180196993 | 0.608 | 0.828 | 1.596368024117E-19 | 5 | Ccr5 |
| **Olfml3.1** | 9.78357254864611E-07 | 0.276063226593715 | 0.981 | 0.977 | 0.0303809278353108 | 5 | Olfml3 |

(*Continued*)

**Table 1.** (Continued)

| | p_val | avg_log2FC | pct.1 | pct.2 | p_val_adj | cluster | gene |
|---|---|---|---|---|---|---|---|
| **Cdkn1a** | 0 | 3.43376426832513 | 0.972 | 0.05 | 0 | 6 | Cdkn1a |
| **Bax** | 1.78413315147855E-50 | 1.12222946231033 | 0.76 | 0.421 | 5.54026867528634E-46 | 6 | Bax |
| **Rps19.1** | 4.02821470596027E-16 | 0.407865038514133 | 0.984 | 0.955 | 1.25088151264184E-11 | 6 | Rps19 |
| **Rpl39.1** | 1.98736775387821E-12 | -0.350866959010792 | 0.972 | 0.989 | 6.171373086118E-08 | 6 | Rpl39 |
| **Ier5.1** | 5.62964216852923E-12 | 0.443912759843309 | 0.823 | 0.698 | 1.74817278259338E-07 | 6 | Ier5 |
| **Rps21.2** | 8.52504182983921E-12 | -0.316852473757971 | 0.98 | 0.993 | 2.64728123941997E-07 | 6 | Rps21 |
| **P2ry6** | 1.18390267393689E-11 | 0.388269615427526 | 0.894 | 0.824 | 3.67637297337621E-07 | 6 | P2ry6 |
| **Rps27.2** | 3.8523551423191E-10 | -0.295547680851792 | 0.984 | 0.99 | 1.19627184234435E-05 | 6 | Rps27 |
| **Rpl27a.2** | 4.56205919601485E-09 | -0.267107089592812 | 0.98 | 0.994 | 0.000141665624213849 | 6 | Rpl27a |
| **Rpl37.2** | 7.16647630177139E-09 | -0.273654908740777 | 0.976 | 0.99 | 0.000222540588598907 | 6 | Rpl37 |
| **Spi1** | 1.9974177378623E-08 | 0.307728327827713 | 0.906 | 0.834 | 0.000620258130138381 | 6 | Spi1 |
| **Rpl34.2** | 5.86662206674482E-08 | -0.27461414839775 | 0.953 | 0.98 | 0.00182176215038627 | 6 | Rpl34 |
| **Rpl35.1** | 5.9931739241275E-08 | -0.351479459708379 | 0.85 | 0.881 | 0.00186106029865931 | 6 | Rpl35 |
| **Serpine2.1** | 7.44329446370467E-08 | 0.360029125698304 | 0.843 | 0.802 | 0.00231136622981421 | 6 | Serpine2 |
| **Rps13.1** | 4.38472064916861E-07 | -0.27805651534537 | 0.957 | 0.973 | 0.0136158730318633 | 6 | Rps13 |
| **Rps28.2** | 1.18174271621467E-06 | -0.283720314635626 | 0.925 | 0.961 | 0.0366966565666141 | 6 | Rps28 |
| **Exosc2** | 0 | 2.62781902812805 | 0.802 | 0.174 | 0 | 7 | Exosc2 |
| **AC160336.** | 1.21301659981191E-66 | 1.26554518644694 | 0.724 | 0.549 | 3.76678044739591E-62 | 7 | AC160336. |
| **Ctsb.2** | 4.9558050594268E-10 | -0.263144543594134 | 0.976 | 0.981 | 1.53892614510381E-05 | 7 | Ctsb |
| **H2-D1.3** | 2.87271958679412E-07 | -0.308652586201546 | 0.674 | 0.736 | 0.00892065613287179 | 7 | H2-D1 |
| **Cd63.1** | 0 | 2.29804463066512 | 0.982 | 0.625 | 0 | 8 | Cd63 |
| **Cd9.2** | 0 | 1.54516801762404 | 1 | 0.969 | 0 | 8 | Cd9 |
| **Ctsb.3** | 0 | 1.4831233398661 | 1 | 0.98 | 0 | 8 | Ctsb |
| **Ctsz.1** | 0 | 1.17071096447867 | 1 | 0.992 | 0 | 8 | Ctsz |
| **Ftl1.1** | 5.65089639532407E-294 | 1.16671616962344 | 1 | 0.99 | 1.75477285763998E-289 | 8 | Ftl1 |
| **Ctsd.1** | 1.79024356661208E-292 | 0.869949891126517 | 1 | 0.999 | 5.5592433474005E-288 | 8 | Ctsd |
| **Cd83** | 2.36809750628E-288 | 1.77625205682349 | 0.739 | 0.242 | 7.35365318625127E-284 | 8 | Cd83 |
| **Selplg.1** | 4.50056348535248E-217 | -0.844289085888205 | 0.991 | 0.995 | 1.3975599791065E-212 | 8 | Selplg |
| **P2ry12.2** | 9.29318214429668E-209 | -0.88717634919471 | 0.985 | 0.993 | 2.88581185126845E-204 | 8 | P2ry12 |
| **Cd68.2** | 1.49821242901075E-161 | 0.872830488647879 | 0.992 | 0.921 | 4.65239905580709E-157 | 8 | Cd68 |
| **C3ar1** | 3.61601153140924E-158 | 1.17628780865338 | 0.806 | 0.434 | 1.12288006084851E-153 | 8 | C3ar1 |
| **Tmem119.** | 7.96821126874067E-151 | -0.764803249988697 | 0.954 | 0.99 | 2.47436864528204E-146 | 8 | Tmem119 |
| **Cadm1** | 2.3262945185331E-149 | 1.12785916619523 | 0.72 | 0.347 | 7.22384236840084E-145 | 8 | Cadm1 |
| **Mt1.1** | 1.02058751643929E-129 | 1.00238842930129 | 0.753 | 0.386 | 3.16923041479893E-125 | 8 | Mt1 |
| **Serinc3.1** | 1.04518546520817E-126 | -0.524012008424553 | 0.996 | 0.999 | 3.24561442511093E-122 | 8 | Serinc3 |
| **Cyba** | 3.82448354443321E-123 | 0.616547774323673 | 0.996 | 0.984 | 1.18761687505285E-118 | 8 | Cyba |
| **Siglech.2** | 2.65875617706483E-100 | -0.586574062026276 | 0.96 | 0.986 | 8.25623555663941E-96 | 8 | Siglech |
| **Ctsa.1** | 9.18401407056005E-93 | 0.621153848629843 | 0.981 | 0.896 | 2.85191188933101E-88 | 8 | Ctsa |
| **Hsp90ab1.** | 2.90712968092798E-89 | 0.726831439873448 | 0.937 | 0.839 | 9.02750979818565E-85 | 8 | Hsp90ab1 |
| **Eif4a1.1** | 1.64176735410592E-88 | 0.850410645825236 | 0.916 | 0.758 | 5.09818016470511E-84 | 8 | Eif4a1 |
| **Calr.1** | 4.96261680677063E-87 | 0.638624914782328 | 0.982 | 0.914 | 1.54104139700648E-82 | 8 | Calr |
| **Pnp.1** | 5.13930432087951E-87 | -0.721953086032058 | 0.744 | 0.877 | 1.59590817076271E-82 | 8 | Pnp |
| **Slc2a5.1** | 3.11323812318665E-84 | -0.8343339218013318 | 0.529 | 0.756 | 9.66753834393152E-80 | 8 | Slc2a5 |
| **Grn.1** | 4.51604419356786E-81 | 0.524743282064129 | 0.991 | 0.965 | 1.40236720342863E-76 | 8 | Grn |
| **Fth1.2** | 4.37030643933904E-80 | 0.582813678919012 | 1 | 0.996 | 1.35711125860795E-75 | 8 | Fth1 |
| **Aldoa** | 2.82589552574676E-79 | 0.755421841772364 | 0.799 | 0.538 | 8.77525337610142E-75 | 8 | Aldoa |
| **Rhob.2** | 2.8780395990474E-79 | -0.544094079948244 | 0.964 | 0.986 | 8.93717636692191E-75 | 8 | Rhob |

*(Continued)*

**Table 1.** (Continued)

| | p_val | avg_log2FC | pct.1 | pct.2 | p_val_adj | cluster | gene |
|---|---|---|---|---|---|---|---|
| Prdx1.2 | 7.61299861361345E-78 | 0.701066346725705 | 0.802 | 0.555 | 2.36406445948539E-73 | 8 | Prdx1 |
| Zfhx3.1 | 1.19082320953699E-76 | -0.655635845403785 | 0.801 | 0.916 | 3.69786331257521E-72 | 8 | Zfhx3 |
| Sdf2l1.1 | 2.79868653556119E-76 | 0.899185252664238 | 0.765 | 0.52 | 8.69076129887817E-72 | 8 | Sdf2l1 |
| Dnase2a | 5.24110074378046E-75 | 0.725973414614582 | 0.745 | 0.469 | 1.62751901396615E-70 | 8 | Dnase2a |
| Hspa5.1 | 9.69014544666345E-75 | 0.717897713426487 | 0.947 | 0.845 | 3.0090808655524E-70 | 8 | Hspa5 |
| Npc2.2 | 1.14135772665376E-74 | 0.505345295003798 | 0.973 | 0.904 | 3.54425814857792E-70 | 8 | Npc2 |
| Ivns1abp.1 | 1.14067577595922E-73 | -0.696114089343054 | 0.703 | 0.831 | 3.54214048708617E-69 | 8 | Ivns1abp |
| Gapdh.1 | 2.09532399991922E-73 | 0.655580269017518 | 0.942 | 0.826 | 6.50660961694916E-69 | 8 | Gapdh |
| Adrb2 | 2.37770893871912E-72 | -0.776856535810651 | 0.476 | 0.707 | 7.38349956740447E-68 | 8 | Adrb2 |
| Trem2.2 | 8.4390685350099E-72 | 0.361999080628309 | 1 | 0.996 | 2.62058395217662E-67 | 8 | Trem2 |
| Creg1.1 | 5.53050457214583E-69 | 0.562616411197819 | 0.919 | 0.773 | 1.71738758478845E-64 | 8 | Creg1 |
| Slc25a5 | 9.17219396097847E-69 | 0.615716490058619 | 0.902 | 0.75 | 2.84824139070265E-64 | 8 | Slc25a5 |
| P2ry13.1 | 3.28378602971041E-68 | -0.560659783020534 | 0.895 | 0.946 | 1.01971407580597E-63 | 8 | P2ry13 |
| Commd8.1 | 1.97143990728E-64 | -0.689170976350403 | 0.567 | 0.747 | 6.12191234407659E-60 | 8 | Commd8 |
| Rps2.1 | 1.06072351372311E-63 | 0.463212915612133 | 0.975 | 0.929 | 3.29386472716438E-59 | 8 | Rps2 |
| Vsir.2 | 1.44019860510305E-63 | -0.455538521385318 | 0.95 | 0.976 | 4.47224872842651E-59 | 8 | Vsir |
| Lamp1 | 1.44048180581282E-63 | 0.432711845443972 | 0.989 | 0.968 | 4.47312815159054E-59 | 8 | Lamp1 |
| Susd3.1 | 2.49406935986811E-62 | -0.643544215633243 | 0.644 | 0.796 | 7.74483358319846E-58 | 8 | Susd3 |
| Csf1r.1 | 5.68150480837528E-62 | -0.302051251303159 | 1 | 0.999 | 1.76427768814478E-57 | 8 | Csf1r |
| Ifngr1.1 | 1.39615877046287E-59 | -0.423827282757582 | 0.968 | 0.985 | 4.33549182991836E-55 | 8 | Ifngr1 |
| Syngr1.1 | 1.40977942076103E-59 | 0.5333823855998 | 0.932 | 0.805 | 4.37778803528922E-55 | 8 | Syngr1 |
| Ltc4s.2 | 7.58691910422862E-59 | -0.564182294135025 | 0.89 | 0.944 | 2.35596598943611E-54 | 8 | Ltc4s |
| Txnip.2 | 8.85849601099603E-58 | -0.59230117569223 | 0.759 | 0.863 | 2.7508287662946E-53 | 8 | Txnip |
| Pdia6.1 | 7.79778103288753E-57 | 0.535705404261183 | 0.932 | 0.805 | 2.42144494414257E-52 | 8 | Pdia6 |
| Cx3cr1.1 | 9.25539750103571E-57 | -0.38126083411941 | 0.994 | 0.995 | 2.87407858599662E-52 | 8 | Cx3cr1 |
| Ssh2.1 | 2.86369276444416E-56 | -0.616931158263941 | 0.674 | 0.814 | 8.89262514142845E-52 | 8 | Ssh2 |
| Srgap2.1 | 4.01773302646846E-56 | -0.547167987541366 | 0.796 | 0.894 | 1.24762663670925E-51 | 8 | Srgap2 |
| Lpcat2.1 | 8.25845776565788E-56 | -0.415881376807884 | 0.967 | 0.981 | 2.56449888996974E-51 | 8 | Lpcat2 |
| Elmo1.1 | 1.58885845794808E-55 | -0.655133992864196 | 0.62 | 0.764 | 4.93388216946619E-51 | 8 | Elmo1 |
| Cotl1 | 2.11411084462814E-55 | 0.531945879892493 | 0.897 | 0.752 | 6.56494840582377E-51 | 8 | Cotl1 |
| Ctsl.2 | 1.27263286238336E-54 | 0.42806463878946 | 0.998 | 0.985 | 3.95190682755905E-50 | 8 | Ctsl |
| Asph | 1.79790327758073E-54 | 0.579853438524778 | 0.83 | 0.662 | 5.58302904787143E-50 | 8 | Asph |
| Ptgs1.1 | 3.69722153716382E-54 | -0.502842936711786 | 0.84 | 0.916 | 1.14809820393548E-49 | 8 | Ptgs1 |
| Apoe.1 | 6.47282283145754E-53 | 0.554447251148479 | 0.754 | 0.53 | 2.01000567385251E-48 | 8 | Apoe |
| Tyrobp.1 | 3.10159317614079E-52 | 0.305938905874624 | 1 | 0.998 | 9.63137728987E-48 | 8 | Tyrobp |
| Cmtm7.1 | 1.5400700048622E-51 | -0.548754393893354 | 0.691 | 0.819 | 4.78237938609859E-47 | 8 | Cmtm7 |
| Rgs10.2 | 1.66585897471301E-51 | -0.358646616428772 | 0.981 | 0.991 | 5.17299187417631E-47 | 8 | Rgs10 |
| Maf.1 | 4.08340130401126E-51 | -0.531198348565383 | 0.869 | 0.924 | 1.26801860693462E-46 | 8 | Maf |
| Ssr4.1 | 7.79359985860166E-51 | 0.438635378351858 | 0.939 | 0.823 | 2.42014656409157E-46 | 8 | Ssr4 |
| Cmtm6.2 | 8.343662334429E-49 | -0.516184684673656 | 0.753 | 0.847 | 2.59095746471024E-44 | 8 | Cmtm6 |
| Bcl2a1b | 2.29711597099703E-48 | 0.661957595851463 | 0.785 | 0.601 | 7.13323422473708E-44 | 8 | Bcl2a1b |
| Gusb | 5.78942773791784E-48 | 0.535575860894955 | 0.836 | 0.636 | 1.79779099545563E-43 | 8 | Gusb |
| Calm2.1 | 1.05031706839045E-47 | -0.418555464175271 | 0.93 | 0.956 | 3.26154959247287E-43 | 8 | Calm2 |
| Nrip1.1 | 1.18069866325771E-47 | -0.538909753398394 | 0.744 | 0.852 | 3.66642355901417E-43 | 8 | Nrip1 |
| mt-Nd1.2 | 1.9537330253942E-47 | 0.476994712017097 | 0.979 | 0.976 | 6.06692716375661E-43 | 8 | mt-Nd1 |
| Rnase4.2 | 1.36624359127097E-46 | -0.445893747711319 | 0.904 | 0.954 | 4.24259622397375E-42 | 8 | Rnase4 |
| Frmd4a.1 | 1.45309391865056E-46 | -0.484704054430051 | 0.827 | 0.893 | 4.51229254558557E-42 | 8 | Frmd4a |

*(Continued)*

**Table 1.** (Continued)

| | p_val | avg_log2FC | pct.1 | pct.2 | p_val_adj | cluster | gene |
|---|---|---|---|---|---|---|---|
| Psap.1 | 1.65466999957967E-46 | 0.377436847358631 | 1 | 0.992 | 5.13824674969474E-42 | 8 | Psap |
| Hsp90b1.2 | 3.34452027332648E-46 | 0.444385462872208 | 0.977 | 0.923 | 1.03857388047607E-41 | 8 | Hsp90b1 |
| Hspa8.2 | 4.64694407314333E-46 | 0.471526992668645 | 0.951 | 0.885 | 1.4430155430332E-41 | 8 | Hspa8 |
| Cd164.2 | 9.76234682818389E-46 | -0.573393832812395 | 0.664 | 0.79 | 3.03150156055594E-41 | 8 | Cd164 |
| St3gal6.1 | 2.63526344708144E-45 | -0.513259352936535 | 0.697 | 0.813 | 8.18328358222198E-41 | 8 | St3gal6 |
| Glul.2 | 3.10760019011743E-45 | -0.486655830058533 | 0.855 | 0.9 | 9.65003087037166E-41 | 8 | Glul |
| Manf.1 | 8.06641154769464E-45 | 0.562405815075004 | 0.867 | 0.688 | 2.50486277790562E-40 | 8 | Manf |
| Arhgap5.1 | 8.26471238944432E-45 | -0.496287053145072 | 0.827 | 0.897 | 2.56644113829414E-40 | 8 | Arhgap5 |
| Adgrg1.1 | 1.76814132654111E-44 | -0.563366193930639 | 0.662 | 0.763 | 5.4906092613081E-40 | 8 | Adgrg1 |
| mt-Atp6.2 | 2.26461394958704E-44 | 0.379968099492182 | 0.998 | 0.998 | 7.03230569765264E-40 | 8 | mt-Atp6 |
| Ssr2.1 | 3.5620009979267E-43 | 0.480146262859831 | 0.72 | 0.489 | 1.10610816988618E-38 | 8 | Ssr2 |
| Hpgd.1 | 1.06149621335982E-42 | -0.506817022093743 | 0.756 | 0.845 | 3.29626419134624E-38 | 8 | Hpgd |
| Pmepa1.1 | 1.12993168925879E-42 | -0.485243666497403 | 0.892 | 0.944 | 3.50877687465531E-38 | 8 | Pmepa1 |
| Timp2.2 | 2.91883780301469E-42 | 0.418838401706748 | 0.93 | 0.823 | 9.06386702970152E-38 | 8 | Timp2 |
| Eif5a.2 | 1.12471932250698E-41 | 0.506050269259257 | 0.814 | 0.627 | 3.49259091218094E-37 | 8 | Eif5a |
| F11r.1 | 1.14244009534915E-40 | -0.395677475901032 | 0.918 | 0.949 | 3.54761922808772E-36 | 8 | F11r |
| Atp6ap2 | 1.12674631950163E-39 | 0.459997633941989 | 0.864 | 0.688 | 3.49888534594842E-35 | 8 | Atp6ap2 |
| Ctsc.2 | 8.92662014852459E-39 | 0.424248652824609 | 0.91 | 0.789 | 2.77198335472134E-34 | 8 | Ctsc |
| Rpsa.2 | 1.07208100139989E-38 | 0.348408228978954 | 0.967 | 0.926 | 3.32913313364707E-34 | 8 | Rpsa |
| Srgn.1 | 1.25531540440601E-38 | 0.544902528289219 | 0.737 | 0.55 | 3.89813092530198E-34 | 8 | Srgn |
| Mbnl1 | 3.11958903347287E-38 | -0.466372473172114 | 0.821 | 0.868 | 9.6872598256433E-34 | 8 | Mbnl1 |
| Ncl.1 | 5.39752535990159E-38 | 0.573025026718125 | 0.725 | 0.532 | 1.67609355001024E-33 | 8 | Ncl |
| Arhgap45.2 | 2.11705059672011E-37 | -0.436209353845278 | 0.779 | 0.867 | 6.57407721799497E-33 | 8 | Arhgap45 |
| Mef2a.1 | 1.92288220047842E-36 | -0.427905366142296 | 0.795 | 0.888 | 5.97112609714565E-32 | 8 | Mef2a |
| Alox5ap | 2.15109418600895E-35 | -0.46890723391594 | 0.715 | 0.808 | 6.67979277581361E-31 | 8 | Alox5ap |
| Fam102b.1 | 3.03171785061314E-35 | -0.520044804258609 | 0.584 | 0.713 | 9.41439344150899E-31 | 8 | Fam102b |
| Tmem173. | 2.26393552520826E-34 | -0.459229994702944 | 0.68 | 0.78 | 7.0301989864292E-30 | 8 | Tmem173 |
| Ckb.1 | 3.04536386991144E-34 | -0.389467832746993 | 0.924 | 0.945 | 9.45676842523599E-30 | 8 | Ckb |
| Rsrp1.2 | 3.42027370654977E-34 | -0.353632733601948 | 0.942 | 0.968 | 1.0620975940949E-29 | 8 | Rsrp1 |
| Serpine2.2 | 5.79996807417822E-34 | 0.410739977569245 | 0.916 | 0.797 | 1.80106408607456E-29 | 8 | Serpine2 |
| Pdia3.1 | 6.68232734120519E-34 | 0.395636647450952 | 0.938 | 0.831 | 2.07506310926445E-29 | 8 | Pdia3 |
| Glmp | 1.1221728070886E-33 | 0.433231412248728 | 0.716 | 0.493 | 3.48468321785222E-29 | 8 | Glmp |
| Ptma.2 | 1.34349120677872E-33 | 0.348988784339256 | 0.973 | 0.895 | 4.17194324440995E-29 | 8 | Ptma |
| Crybb1.3 | 2.16678752994459E-33 | -0.547399860214337 | 0.65 | 0.751 | 6.72852531673695E-29 | 8 | Crybb1 |
| mt-Co2.2 | 2.18792909249224E-33 | 0.323932311570829 | 0.996 | 0.996 | 6.79417621091615E-29 | 8 | mt-Co2 |
| Slc3a2.1 | 4.81550555062895E-33 | 0.472699984515899 | 0.894 | 0.777 | 1.49535893863681E-28 | 8 | Slc3a2 |
| mt-Co3.2 | 7.78332976398412E-33 | 0.330243744764552 | 0.995 | 0.997 | 2.41695739160999E-28 | 8 | mt-Co3 |
| Tmem176b | 9.25337770620232E-33 | -0.502776632558105 | 0.767 | 0.841 | 2.87345137910701E-28 | 8 | Tmem176b |
| Slco2b1.1 | 4.07562227134426E-32 | -0.390407821211703 | 0.854 | 0.891 | 1.26560298392053E-27 | 8 | Slco2b1 |
| mt-Nd2.1 | 5.54387532988732E-32 | 0.378265178004132 | 0.995 | 0.984 | 1.72153960618991E-27 | 8 | mt-Nd2 |
| Tnfaip8l2.2 | 6.43387657448121E-32 | -0.42878838441006 | 0.689 | 0.786 | 1.99791169267365E-27 | 8 | Tnfaip8l2 |
| Ybx1.3 | 8.31649704517378E-32 | 0.412576831129288 | 0.828 | 0.651 | 2.58252182743781E-27 | 8 | Ybx1 |
| Tsc22d4.1 | 2.84679965680923E-31 | -0.40979953878853 | 0.743 | 0.82 | 8.8401669742897E-27 | 8 | Tsc22d4 |
| Ccr5.3 | 4.79441639813565E-31 | -0.475058709648057 | 0.758 | 0.828 | 1.48881012411306E-26 | 8 | Ccr5 |
| H2-K1.3 | 5.94604051329431E-31 | 0.352520375604056 | 0.881 | 0.716 | 1.84642396059328E-26 | 8 | H2-K1 |
| Cfl1 | 6.84169804627813E-31 | 0.316152382715626 | 0.964 | 0.928 | 2.12455249431075E-26 | 8 | Cfl1 |
| Fam49b.1 | 7.91358930980715E-31 | -0.369350308421919 | 0.842 | 0.886 | 2.45740688837442E-26 | 8 | Fam49b |

*(Continued)*

**Table 1.** (Continued)

| | p_val | avg_log2FC | pct.1 | pct.2 | p_val_adj | cluster | gene |
|---|---|---|---|---|---|---|---|
| Abhd12.1 | 1.44236057381535E-30 | 0.323710406196794 | 0.986 | 0.953 | 4.47896228986881E-26 | 8 | Abhd12 |
| Pmp22.2 | 1.46922281469256E-30 | 0.601295803227464 | 0.862 | 0.762 | 4.56237760646481E-26 | 8 | Pmp22 |
| Pld4.1 | 8.01365025949018E-30 | -0.365263790811843 | 0.895 | 0.932 | 2.48847881507949E-25 | 8 | Pld4 |
| Zfp36l2.1 | 1.05118544658868E-29 | -0.469341907918229 | 0.703 | 0.779 | 3.26424616729183E-25 | 8 | Zfp36l2 |
| Kctd12.1 | 1.06632303539353E-29 | -0.411060117067163 | 0.807 | 0.867 | 3.31125292180752E-25 | 8 | Kctd12 |
| Spcs2.1 | 1.51718078376072E-29 | 0.353380627016068 | 0.91 | 0.82 | 4.71130148781217E-25 | 8 | Spcs2 |
| Gnai2 | 1.81604525501735E-29 | -0.31294722920072 | 0.944 | 0.961 | 5.63936533040539E-25 | 8 | Gnai2 |
| Ppia.2 | 1.30843519307151E-28 | 0.255180529847415 | 0.987 | 0.978 | 4.06308380504497E-24 | 8 | Ppia |
| Canx.1 | 1.82964959839069E-27 | 0.347911268266192 | 0.874 | 0.749 | 5.68161089788261E-23 | 8 | Canx |
| Ppfia4.1 | 3.57358274003528E-27 | -0.408605345701883 | 0.621 | 0.724 | 1.10970464826316E-22 | 8 | Ppfia4 |
| Tanc2.1 | 4.43293618389633E-27 | -0.430787794690276 | 0.702 | 0.778 | 1.37655967318533E-22 | 8 | Tanc2 |
| Tmem86a. | 8.03801106610128E-27 | 0.387263850643032 | 0.904 | 0.797 | 2.49604357635643E-22 | 8 | Tmem86a |
| Ptpn18.1 | 1.23027426761042E-26 | -0.387948804438304 | 0.736 | 0.811 | 3.82037068321065E-22 | 8 | Ptpn18 |
| Dad1 | 1.25051437701368E-26 | 0.335097544921475 | 0.853 | 0.675 | 3.88322229494057E-22 | 8 | Dad1 |
| Git2.1 | 2.29123654626136E-26 | -0.437054881693923 | 0.643 | 0.728 | 7.11497684710542E-22 | 8 | Git2 |
| Ecscr.2 | 3.76014584164265E-26 | -0.368424805985026 | 0.814 | 0.867 | 1.16763808820529E-21 | 8 | Ecscr |
| B2m.2 | 4.18012132656751E-26 | 0.264894224934046 | 0.987 | 0.963 | 1.29805307553901E-21 | 8 | B2m |
| Rpl10a.2 | 4.18717109364947E-26 | 0.267339486076959 | 0.972 | 0.898 | 1.30024223971097E-21 | 8 | Rpl10a |
| Gnas.1 | 9.62506960416419E-26 | 0.336030750707358 | 0.906 | 0.8 | 2.98887286418111E-21 | 8 | Gnas |
| Npm1.2 | 1.16624609343738E-25 | 0.424022212543326 | 0.807 | 0.671 | 3.6215439939511E-21 | 8 | Npm1 |
| Atox1.1 | 1.94017907467132E-25 | 0.35244207601904 | 0.802 | 0.636 | 6.02483808057685E-21 | 8 | Atox1 |
| Krtcap2.1 | 2.09207368012998E-25 | 0.376558207810135 | 0.716 | 0.553 | 6.49651639890764E-21 | 8 | Krtcap2 |
| Arpc2.1 | 2.89371402697861E-25 | 0.296343146782166 | 0.952 | 0.893 | 8.98585016797667E-21 | 8 | Arpc2 |
| Pag1.1 | 4.08295281051541E-25 | -0.404001156752083 | 0.639 | 0.735 | 1.26787933624935E-20 | 8 | Pag1 |
| Bin2.1 | 1.15333965206071E-24 | -0.363982603522883 | 0.715 | 0.789 | 3.58146562154413E-20 | 8 | Bin2 |
| Rps9.2 | 1.35530652184561E-24 | -0.265819223946562 | 0.996 | 0.992 | 4.20863334228717E-20 | 8 | Rps9 |
| Rgs2.2 | 2.58628292793391E-24 | -0.35919293434617 | 0.802 | 0.859 | 8.03118437611316E-20 | 8 | Rgs2 |
| Ophn1 | 6.06412348593023E-24 | -0.417371770115936 | 0.621 | 0.7 | 1.88309226608591E-19 | 8 | Ophn1 |
| H2-D1.4 | 7.03474391698801E-24 | 0.293824863571368 | 0.869 | 0.727 | 2.18449902854229E-19 | 8 | H2-D1 |
| Erp29.1 | 1.34760315333823E-23 | 0.275774488579251 | 0.979 | 0.938 | 4.1847120720612E-19 | 8 | Erp29 |
| mt-Nd4.2 | 6.18280053492699E-23 | 0.279258098452354 | 0.986 | 0.986 | 1.91994505011088E-18 | 8 | mt-Nd4 |
| Tpp1 | 7.02469912920861E-23 | 0.33132805368182 | 0.798 | 0.638 | 2.18137982059315E-18 | 8 | Tpp1 |
| Fam105a.1 | 2.79713073247418E-22 | -0.317825770228975 | 0.858 | 0.891 | 8.68593006355208E-18 | 8 | Fam105a |
| Epb41l2.1 | 2.87129950716644E-22 | -0.331600967731752 | 0.909 | 0.916 | 8.91624635960396E-18 | 8 | Epb41l2 |
| Cd84.1 | 1.17397648087063E-21 | 0.338634989142378 | 0.848 | 0.713 | 3.64554916604757E-17 | 8 | Cd84 |
| Tm6sf1 | 1.30846335940964E-21 | -0.375294604858226 | 0.64 | 0.729 | 4.06317126997476E-17 | 8 | Tm6sf1 |
| Wasf2.1 | 1.97627089527403E-21 | -0.352331321243301 | 0.71 | 0.77 | 6.13691401109445E-17 | 8 | Wasf2 |
| Fcgr1.1 | 2.0859944076169E-21 | -0.335460342069659 | 0.804 | 0.852 | 6.47763843397275E-17 | 8 | Fcgr1 |
| Cd52.3 | 2.25198730560763E-21 | 0.405529453782898 | 0.719 | 0.571 | 6.99309618010336E-17 | 8 | Cd52 |
| Hexa | 2.87557606155249E-21 | 0.280108253479925 | 0.979 | 0.94 | 8.92952634393895E-17 | 8 | Hexa |
| Tmed9.1 | 3.41793826931766E-21 | 0.264575197438471 | 0.913 | 0.832 | 1.06137237077121E-16 | 8 | Tmed9 |
| Ywhah.1 | 5.06032945066087E-21 | -0.310117375924705 | 0.861 | 0.889 | 1.57138410431372E-16 | 8 | Ywhah |
| Gabarap.2 | 1.00777190280094E-20 | 0.269071795962169 | 0.925 | 0.823 | 3.12943408976777E-16 | 8 | Gabarap |
| Tpst2.1 | 1.98143311789268E-20 | -0.369364197870582 | 0.663 | 0.749 | 6.15294426099213E-16 | 8 | Tpst2 |
| Celf2.1 | 3.13963279358223E-20 | -0.359427416034551 | 0.727 | 0.786 | 9.7495017139109E-16 | 8 | Celf2 |
| Ddost | 3.53733616391243E-20 | 0.338065951325238 | 0.753 | 0.588 | 1.09844899897973E-15 | 8 | Ddost |
| Rbm3.2 | 5.67522503067435E-20 | 0.319863230590694 | 0.725 | 0.579 | 1.7623276287753E-15 | 8 | Rbm3 |

(*Continued*)

**Table 1.** (Continued)

| | p_val | avg_log2FC | pct.1 | pct.2 | p_val_adj | cluster | gene |
|---|---|---|---|---|---|---|---|
| Bmp2k.1 | 5.67839050545066E-20 | -0.378820543412697 | 0.639 | 0.704 | 1.76331060365759E-15 | 8 | Bmp2k |
| Fkbp2.1 | 6.06425874900776E-20 | 0.341080581066399 | 0.733 | 0.578 | 1.88313426932938E-15 | 8 | Fkbp2 |
| Ppib | 1.17990901279462E-19 | 0.276677381713492 | 0.933 | 0.86 | 3.66397145743113E-15 | 8 | Ppib |
| mt-Nd3.1 | 1.80257283677481E-19 | 0.322989800375962 | 0.924 | 0.876 | 5.59752943003682E-15 | 8 | mt-Nd3 |
| Rps27.3 | 1.96179690767096E-19 | -0.263129105572359 | 0.982 | 0.991 | 6.09196793739063E-15 | 8 | Rps27 |
| Qk.1 | 4.08618647358674E-19 | -0.259427144592222 | 0.933 | 0.948 | 1.26888348564289E-14 | 8 | Qk |
| Itgam.1 | 1.51726815832078E-18 | -0.348368882884458 | 0.759 | 0.797 | 4.71157281203351E-14 | 8 | Itgam |
| Atp6ap1 | 2.4505176011153E-18 | 0.294891853050671 | 0.711 | 0.546 | 7.60959230674334E-14 | 8 | Atp6ap1 |
| Picalm.2 | 2.76314953025217E-18 | -0.337154400684982 | 0.767 | 0.802 | 8.58040823629207E-14 | 8 | Picalm |
| Arsb.1 | 6.15550705637644E-18 | -0.321435432479596 | 0.796 | 0.81 | 1.91146960621657E-13 | 8 | Arsb |
| Tmed3.1 | 6.31768087674275E-18 | 0.309292476936743 | 0.723 | 0.581 | 1.96182944265492E-13 | 8 | Tmed3 |
| Actr3.1 | 9.48092766075481E-18 | 0.318496050711825 | 0.752 | 0.6 | 2.94411246649419E-13 | 8 | Actr3 |
| Akr1a1.1 | 1.58133735972227E-17 | 0.286654861403575 | 0.757 | 0.605 | 4.91052690314556E-13 | 8 | Akr1a1 |
| Foxn3.1 | 5.67970975850568E-17 | -0.328812781350265 | 0.708 | 0.746 | 1.76372027130877E-12 | 8 | Foxn3 |
| Atp6v1g1 | 6.75467183434774E-17 | 0.291408843619294 | 0.828 | 0.716 | 2.09752824472E-12 | 8 | Atp6v1g1 |
| Ucp2.2 | 8.16577942177794E-17 | 0.280251886059107 | 0.829 | 0.708 | 2.5357194838447E-12 | 8 | Ucp2 |
| Inpp5d.1 | 1.32954737965335E-16 | -0.281699562948805 | 0.828 | 0.862 | 4.12864347803755E-12 | 8 | Inpp5d |
| Atp5b | 1.5718254786845E-16 | 0.311681993281979 | 0.726 | 0.584 | 4.88098965895898E-12 | 8 | Atp5b |
| Cd33.1 | 3.75338961888481E-16 | -0.342354740625025 | 0.706 | 0.742 | 1.1655400783523E-11 | 8 | Cd33 |
| Adap2.2 | 8.18950162837603E-16 | -0.332638521542607 | 0.706 | 0.75 | 2.54308594065961E-11 | 8 | Adap2 |
| Myl6.1 | 8.46500370570202E-16 | 0.254229029958553 | 0.8 | 0.664 | 2.62863760073165E-11 | 8 | Myl6 |
| Srsf2.1 | 1.10768934574481E-15 | 0.370438246052577 | 0.772 | 0.682 | 3.43970772534135E-11 | 8 | Srsf2 |
| Golm1.1 | 1.18716005101352E-15 | -0.28314526747922 | 0.849 | 0.865 | 3.68648810641229E-11 | 8 | Golm1 |
| Lrp1.1 | 1.8336777954223E-15 | 0.271717828057505 | 0.818 | 0.711 | 5.69411965812486E-11 | 8 | Lrp1 |
| Evi2a.1 | 2.33175862562294E-15 | 0.316409582377507 | 0.849 | 0.752 | 7.24081006014691E-11 | 8 | Evi2a |
| Tmem55b. | 2.85165465558581E-15 | -0.292515967519428 | 0.647 | 0.71 | 8.85524320199062E-11 | 8 | Tmem55b |
| Cfh | 3.04296031432077E-15 | -0.28333795605719 | 0.788 | 0.813 | 9.44930466406028E-11 | 8 | Cfh |
| Ncf1.1 | 3.19760750604483E-15 | -0.314301919825442 | 0.66 | 0.72 | 9.92953058852102E-11 | 8 | Ncf1 |
| Srrm2.1 | 4.4528764536497E-15 | -0.283997988818107 | 0.834 | 0.837 | 1.38275172515184E-10 | 8 | Srrm2 |
| Limd2.1 | 4.95659390611145E-15 | -0.315276879649246 | 0.653 | 0.71 | 1.53917110566479E-10 | 8 | Limd2 |
| St13 | 9.73498773734598E-15 | 0.255243002437833 | 0.717 | 0.565 | 3.02300574207805E-10 | 8 | St13 |
| Pou2f2.1 | 2.79475461122531E-14 | -0.275270019843823 | 0.821 | 0.855 | 8.67855149423796E-10 | 8 | Pou2f2 |
| Cd86 | 3.2962670110266E-14 | 0.296821624433216 | 0.767 | 0.64 | 1.02358979493409E-09 | 8 | Cd86 |
| Hnrnpf.1 | 8.94709697622951E-14 | 0.258923973138191 | 0.843 | 0.753 | 2.77834202402855E-09 | 8 | Hnrnpf |
| Cttnbp2nl. | 2.89459752897827E-13 | -0.279656275882088 | 0.695 | 0.74 | 8.98859370673621E-09 | 8 | Cttnbp2nl |
| Pnisr.1 | 7.34931013600605E-13 | -0.278718936771654 | 0.726 | 0.765 | 2.28218127653396E-08 | 8 | Pnisr |
| Bri3.1 | 1.81400256566867E-12 | 0.276134042849792 | 0.811 | 0.739 | 5.63302216717092E-08 | 8 | Bri3 |
| Orai1.2 | 2.78215538141171E-12 | -0.257773458393519 | 0.664 | 0.704 | 8.63942710589779E-08 | 8 | Orai1 |
| Arglu1.1 | 5.38956476588582E-12 | -0.262554894028675 | 0.738 | 0.751 | 1.67362154675052E-07 | 8 | Arglu1 |
| Abi3.1 | 1.07306713457451E-11 | -0.259439288002924 | 0.792 | 0.81 | 3.33219537299423E-07 | 8 | Abi3 |
| Btg1.1 | 1.19074769254943E-11 | -0.274321022050926 | 0.76 | 0.785 | 3.69762880967376E-07 | 8 | Btg1 |
| Nfe2l2 | 3.69370483128489E-09 | 0.266359149816771 | 0.701 | 0.595 | 0.00011470061612589 | 8 | Nfe2l2 |
| Ier5.2 | 2.42193698956475E-07 | -0.252914084851068 | 0.662 | 0.702 | 0.00752084093369542 | 8 | Ier5 |
| Klf2 | 0 | 2.82025264998077 | 0.814 | 0.081 | 0 | 9 | Klf2 |
| Junb | 1.16836760080433E-102 | 1.70248126058051 | 0.848 | 0.587 | 3.62813191077768E-98 | 9 | Junb |
| Fcrls.3 | 2.57574494889736E-81 | -0.790511735720266 | 0.878 | 0.985 | 7.99846078981096E-77 | 9 | Fcrls |
| Jun | 2.05156090179544E-80 | 1.89047764374518 | 0.759 | 0.509 | 6.37071206834537E-76 | 9 | Jun |

*(Continued)*

**Table 1.** (Continued)

| | p_val | avg_log2FC | pct.1 | pct.2 | p_val_adj | cluster | gene |
|---|---|---|---|---|---|---|---|
| Btg2 | 8.09839845879443E-80 | 1.94089546005116 | 0.814 | 0.651 | 2.51479567340943E-75 | 9 | Btg2 |
| P2ry12.3 | 1.65252076314068E-43 | -0.412757412373791 | 0.996 | 0.993 | 5.13157272578075E-39 | 9 | P2ry12 |
| Ctss.3 | 5.85387757393451E-42 | 0.390604059286042 | 1 | 0.999 | 1.81780460303388E-37 | 9 | Ctss |
| Jund | 1.34193207002807E-40 | 1.01394096915674 | 0.797 | 0.676 | 4.16710165705818E-36 | 9 | Jund |
| Ier5.3 | 2.58222047894291E-29 | 0.803120437595928 | 0.784 | 0.698 | 8.01856925326143E-25 | 9 | Ier5 |
| Rhob.3 | 2.79178080868891E-26 | 0.415679219283436 | 0.992 | 0.985 | 8.66931694522169E-22 | 9 | Rhob |
| Ly86.2 | 4.99302980331277E-25 | 0.328105505001004 | 1 | 0.997 | 1.55048554482271E-20 | 9 | Ly86 |
| Eef1a1.2 | 2.47929669701862E-24 | 0.29245482371578 | 1 | 0.999 | 7.69896003325193E-20 | 9 | Eef1a1 |
| Rps8.1 | 5.52648186936241E-23 | 0.297128075596735 | 0.992 | 0.989 | 1.71613841489311E-18 | 9 | Rps8 |
| Cd164.3 | 8.76905685619847E-23 | -0.469565999765011 | 0.624 | 0.789 | 2.72305522555531E-18 | 9 | Cd164 |
| Rsrp1.3 | 3.18560931266469E-21 | -0.355396896629338 | 0.932 | 0.968 | 9.89227259861767E-17 | 9 | Rsrp1 |
| Tmem176a | 1.98439828077123E-20 | 0.539880018119519 | 0.776 | 0.664 | 6.16215198127891E-16 | 9 | Tmem176a |
| Gpr34.2 | 2.12843218975933E-20 | -0.277039921694065 | 0.992 | 0.994 | 6.60942047885964E-16 | 9 | Gpr34 |
| Fth1.3 | 2.19199760566805E-19 | 0.330507664778572 | 0.998 | 0.996 | 6.80681016488101E-15 | 9 | Fth1 |
| H3f3b.1 | 2.67146802107176E-19 | 0.585715297948255 | 0.976 | 0.98 | 8.29570964583413E-15 | 9 | H3f3b |
| P2ry13.2 | 5.15700588928796E-19 | -0.359955828571151 | 0.893 | 0.945 | 1.60140503880059E-14 | 9 | P2ry13 |
| Rps4x.3 | 1.98263069520838E-18 | 0.294791024882537 | 0.992 | 0.984 | 6.15666309783057E-14 | 9 | Rps4x |
| Srsf5 | 2.15993702535101E-18 | 0.492190333489135 | 0.808 | 0.737 | 6.70725244482249E-14 | 9 | Srsf5 |
| Rpl10a.3 | 3.39450595974726E-18 | 0.320687445665616 | 0.938 | 0.901 | 1.05409593568032E-13 | 9 | Rpl10a |
| Ecscr.3 | 6.10041658134621E-18 | -0.377100273492835 | 0.767 | 0.868 | 1.89436236100544E-13 | 9 | Ecscr |
| Mcl1 | 8.28685557133532E-18 | 0.456863050903963 | 0.771 | 0.665 | 2.57331726056676E-13 | 9 | Mcl1 |
| Rps11.2 | 1.06223182907011E-17 | 0.269803998487436 | 0.991 | 0.991 | 3.29854849881142E-13 | 9 | Rps11 |
| Gm42418 | 1.10865672015124E-17 | -0.416569643884278 | 0.962 | 0.981 | 3.44271171308565E-13 | 9 | Gm42418 |
| Rps25.2 | 4.8011995384876E-17 | 0.317410567520731 | 0.945 | 0.915 | 1.49091649268655E-12 | 9 | Rps25 |
| Rpl39.2 | 8.09618189794176E-17 | 0.273674024793626 | 0.994 | 0.989 | 2.51410736476785E-12 | 9 | Rpl39 |
| Cebpb.1 | 1.09902198141848E-16 | 0.467272430937489 | 0.759 | 0.665 | 3.4127929588988E-12 | 9 | Cebpb |
| Rpl23.2 | 2.51874634811659E-16 | 0.269322720454198 | 0.996 | 0.987 | 7.82146303480645E-12 | 9 | Rpl23 |
| H2-D1.5 | 3.46757952696842E-16 | 0.447767623551638 | 0.825 | 0.731 | 1.0767874705095E-11 | 9 | H2-D1 |
| Rpl32.1 | 1.24659169879033E-15 | 0.29385711386566 | 0.989 | 0.976 | 3.8710412022536E-11 | 9 | Rpl32 |
| Tmem176b | 9.60171917033495E-15 | 0.365325780539131 | 0.895 | 0.836 | 2.98162185396411E-10 | 9 | Tmem176b |
| Rps26.1 | 9.74070498327059E-15 | 0.313264172190597 | 0.932 | 0.895 | 3.02478111845502E-10 | 9 | Rps26 |
| Ddx5 | 2.84147964342529E-14 | 0.305351617075877 | 0.97 | 0.969 | 8.82364673672857E-10 | 9 | Ddx5 |
| Gpx1.2 | 5.32081330322768E-14 | 0.393179791738437 | 0.748 | 0.688 | 1.65227215505129E-09 | 9 | Gpx1 |
| Rpsa.3 | 9.0822634976825E-14 | 0.296969070154704 | 0.953 | 0.927 | 2.82031528393535E-09 | 9 | Rpsa |
| Psap.2 | 9.66474024142945E-14 | 0.263543211954765 | 0.996 | 0.993 | 3.00119178717109E-09 | 9 | Psap |
| Scoc.2 | 1.49716050811527E-13 | -0.291990643074513 | 0.795 | 0.877 | 4.64913252585034E-09 | 9 | Scoc |
| Ctsh.2 | 1.81797505212348E-13 | 0.262350594762351 | 0.981 | 0.958 | 5.64535792935906E-09 | 9 | Ctsh |
| Crybb1.4 | 3.46794919477157E-13 | -0.439946095341386 | 0.662 | 0.749 | 1.07690226345242E-08 | 9 | Crybb1 |
| Rps5.1 | 3.90683227039514E-13 | 0.267510985717204 | 0.957 | 0.935 | 1.2131886249258E-08 | 9 | Rps5 |
| Kctd12.2 | 2.36396580907382E-11 | 0.395449704186906 | 0.898 | 0.863 | 7.34082302691694E-07 | 9 | Kctd12 |
| Rack1.1 | 3.42340862099143E-11 | 0.292098147095996 | 0.908 | 0.874 | 1.06307107907647E-06 | 9 | Rack1 |
| Ctsc.3 | 3.78717790789658E-11 | 0.295448959591156 | 0.821 | 0.794 | 1.17603235573912E-06 | 9 | Ctsc |
| Slc2a5.2 | 6.18298733346207E-11 | -0.327957207138233 | 0.641 | 0.748 | 1.92000305665998E-06 | 9 | Slc2a5 |
| Rpl12.2 | 6.86227555623138E-11 | 0.250831212315769 | 0.953 | 0.93 | 2.13094242847653E-06 | 9 | Rpl12 |
| Maf.2 | 9.22665231438888E-11 | -0.267527733792997 | 0.882 | 0.923 | 2.86515234318718E-06 | 9 | Maf |
| Hnrnpa2b1 | 1.21897330120586E-10 | 0.393305228609403 | 0.784 | 0.735 | 3.78527779223455E-06 | 9 | Hnrnpa2b1 |
| Tmcc3.2 | 1.2934751937977E-10 | 0.270706395051237 | 0.831 | 0.774 | 4.01662851930001E-06 | 9 | Tmcc3 |

(*Continued*)

**Table 1.** (*Continued*)

| | p_val | avg_log2FC | pct.1 | pct.2 | p_val_adj | cluster | gene |
|---|---|---|---|---|---|---|---|
| **Rhoa** | 2.80652806231772E-10 | 0.289825411182493 | 0.893 | 0.855 | 8.71511159191521E-06 | 9 | Rhoa |
| **Canx.2** | 1.84188104039672E-09 | -0.279387458745 | 0.664 | 0.758 | 5.71959319474392E-05 | 9 | Canx |
| **Btg1.2** | 4.79174153976171E-09 | 0.457718074690793 | 0.81 | 0.783 | 0.000148797950034221 | 9 | Btg1 |
| **Srsf2.2** | 4.25975194152108E-08 | 0.280104036585748 | 0.733 | 0.685 | 0.00132278077040054 | 9 | Srsf2 |
| **Tgfb1.1** | 4.50515354617021E-08 | -0.255774461657109 | 0.763 | 0.835 | 0.00139898533069224 | 9 | Tgfb1 |
| **Pabpc1** | 2.96376028677717E-07 | 0.292098202929933 | 0.712 | 0.649 | 0.00920336481852915 | 9 | Pabpc1 |
| **Tmem86a.** | 4.9540598341497E-07 | -0.256723117227967 | 0.711 | 0.804 | 0.0153838420029851 | 9 | Tmem86a |
| **Apoe.2** | 0 | 2.44003497858863 | 1 | 0.505 | 0 | 10 | Apoe |
| **Lyz2.1** | 5.32864964531461E-213 | 0.901723909166408 | 0.707 | 0.343 | 1.65470557435955E-208 | 10 | Lyz2 |
| **Fau.2** | 5.62092126642786E-137 | 0.436383701380954 | 1 | 0.998 | 1.74546468086384E-132 | 10 | Fau |
| **Ctss.4** | 3.41684295195587E-126 | 0.350108610489573 | 1 | 0.999 | 1.06103224187086E-121 | 10 | Ctss |
| **Rps12.1** | 1.1191163682531E-123 | 0.479481668759049 | 0.999 | 0.99 | 3.47519205833634E-119 | 10 | Rps12 |
| **Rpl23.3** | 5.8071642958955E-120 | 0.480767623427662 | 0.999 | 0.986 | 1.80329872880443E-115 | 10 | Rpl23 |
| **Eef1a1.3** | 2.29028694521351E-117 | 0.409590399680411 | 0.999 | 0.999 | 7.1120280509715E-113 | 10 | Eef1a1 |
| **Rpl32.2** | 1.78386135837906E-114 | 0.493211893354908 | 0.99 | 0.975 | 5.5394246761745E-110 | 10 | Rpl32 |
| **Rps24.1** | 5.32205197027601E-111 | 0.415744395776548 | 0.999 | 0.994 | 1.65265679832981E-106 | 10 | Rps24 |
| **Rpl30.2** | 1.59887548287395E-109 | 0.409811170735646 | 1 | 0.992 | 4.96498803696846E-105 | 10 | Rpl30 |
| **Cd63.2** | 7.0900321665476E-105 | 0.641150455113905 | 0.834 | 0.627 | 2.20166768867803E-100 | 10 | Cd63 |
| **Rpl27a.3** | 3.30325138611377E-104 | 0.39845492132839 | 1 | 0.994 | 1.02575865292991E-99 | 10 | Rpl27a |
| **Rpl13.1** | 5.15265706995985E-100 | 0.394215696867408 | 0.999 | 0.994 | 1.60005459993463E-95 | 10 | Rpl13 |
| **Rpl39.3** | 1.57861668635376E-97 | 0.434118140092837 | 0.995 | 0.988 | 4.90207839613432E-93 | 10 | Rpl39 |
| **Rpl21.1** | 1.97898335458091E-96 | 0.404715348040053 | 0.998 | 0.991 | 6.1453370109801E-92 | 10 | Rpl21 |
| **Rps21.3** | 4.72160902988962E-96 | 0.407707564080355 | 1 | 0.992 | 1.46620125205162E-91 | 10 | Rps21 |
| **Rpl41.1** | 8.82870490594265E-95 | 0.407272687093557 | 0.997 | 0.989 | 2.74157773444237E-90 | 10 | Rpl41 |
| **Rps29.1** | 1.05505067599865E-94 | 0.343252389904265 | 1 | 1 | 3.27624886417862E-90 | 10 | Rps29 |
| **Rps9.3** | 2.75762275113222E-93 | 0.387808719111042 | 1 | 0.992 | 8.56324592909087E-89 | 10 | Rps9 |
| **Rpl37a.1** | 9.73524209661492E-93 | 0.402148492265271 | 0.998 | 0.99 | 3.02308472826183E-88 | 10 | Rpl37a |
| **Rpl35a.1** | 2.13476017455015E-91 | 0.369946186473899 | 0.997 | 0.996 | 6.62907077003057E-87 | 10 | Rpl35a |
| **Rps27a.1** | 3.10538076507345E-89 | 0.400949509386775 | 0.998 | 0.99 | 9.64313888978257E-85 | 10 | Rps27a |
| **Rps15a.1** | 1.53736502862585E-88 | 0.40665532148171 | 0.995 | 0.979 | 4.77397962339184E-84 | 10 | Rps15a |
| **Rplp1.1** | 1.61616939041949E-88 | 0.31168045604988 | 1 | 1 | 5.01869080806963E-84 | 10 | Rplp1 |
| **Rpl37.3** | 2.08255099489522E-88 | 0.389553544395368 | 1 | 0.989 | 6.46694560444811E-84 | 10 | Rpl37 |
| **Rps10.1** | 2.97167881274376E-88 | 0.376619587342426 | 0.996 | 0.986 | 9.22795421721319E-84 | 10 | Rps10 |
| **Rpl18a.2** | 4.92684236482868E-88 | 0.386705855339309 | 0.997 | 0.99 | 1.52993235955025E-83 | 10 | Rpl18a |
| **Rps27.4** | 5.46287559460672E-88 | 0.398201608108706 | 0.998 | 0.99 | 1.69638675839323E-83 | 10 | Rps27 |
| **Rps16.1** | 2.75199196332616E-86 | 0.406622627856963 | 0.994 | 0.975 | 8.54576064371673E-82 | 10 | Rps16 |
| **Rps19.2** | 1.72685937025609E-83 | 0.460442404822361 | 0.986 | 0.953 | 5.36241640245624E-79 | 10 | Rps19 |
| **Rps11.3** | 1.97493949173616E-83 | 0.386684165914418 | 0.998 | 0.99 | 6.13277960368829E-79 | 10 | Rps11 |
| **Rplp2.1** | 1.03107469587375E-81 | 0.400882682574048 | 0.992 | 0.969 | 3.20179625309676E-77 | 10 | Rplp2 |
| **Rps4x.4** | 1.20514410998781E-81 | 0.392169912629153 | 0.998 | 0.983 | 3.74233400474514E-77 | 10 | Rps4x |
| **Rps5.2** | 1.36023725659512E-81 | 0.441972016353041 | 0.976 | 0.932 | 4.22394475290484E-77 | 10 | Rps5 |
| **Rps7.1** | 2.74340030282821E-81 | 0.412372551929382 | 0.984 | 0.961 | 8.51908096037243E-77 | 10 | Rps7 |
| **Rpl34.3** | 9.39477652591638E-81 | 0.386903788610049 | 0.993 | 0.979 | 2.91735995459282E-76 | 10 | Rpl34 |
| **Rpl19.1** | 2.79663453448762E-80 | 0.372463630566697 | 0.993 | 0.979 | 8.68438921994441E-76 | 10 | Rpl19 |
| **Rps23.1** | 1.91938750090788E-79 | 0.36023126763653 | 0.996 | 0.985 | 5.96027400656923E-75 | 10 | Rps23 |
| **Rps28.3** | 1.94794449129769E-77 | 0.431116843282908 | 0.984 | 0.958 | 6.04895202882671E-73 | 10 | Rps28 |
| **Rps18.1** | 2.42056743269659E-77 | 0.421331677491286 | 0.986 | 0.945 | 7.51658804875273E-73 | 10 | Rps18 |

(*Continued*)

**Table 1.** (Continued)

| | p_val | avg_log2FC | pct.1 | pct.2 | p_val_adj | cluster | gene |
|---|---|---|---|---|---|---|---|
| Rps14.2 | 2.77086009426189E-75 | 0.384581456823455 | 0.996 | 0.97 | 8.60435185071144E-71 | 10 | Rps14 |
| Rpl10a.4 | 7.4482631842686E-75 | 0.476135843439428 | 0.969 | 0.897 | 2.31290916661093E-70 | 10 | Rpl10a |
| Rpl9.1 | 1.97032386124247E-74 | 0.384219739440568 | 0.992 | 0.97 | 6.11844668631625E-70 | 10 | Rpl9 |
| Rplp0.2 | 2.84660380197437E-74 | 0.388083167460363 | 0.987 | 0.97 | 8.83955878627102E-70 | 10 | Rplp0 |
| Rpl10.2 | 3.01341092943011E-73 | 0.344537608979151 | 0.993 | 0.985 | 9.35754495915933E-69 | 10 | Rpl10 |
| Tpt1.1 | 7.83724032078323E-73 | 0.320611437626757 | 1 | 0.999 | 2.43369823681282E-68 | 10 | Tpt1 |
| Rps20.1 | 1.24503165961084E-71 | 0.38551875844823 | 0.99 | 0.974 | 3.86619681258953E-67 | 10 | Rps20 |
| Fth1.4 | 2.30597221173909E-70 | 0.364652912601929 | 0.997 | 0.996 | 7.1607355091134E-66 | 10 | Fth1 |
| Rps13.2 | 5.36387795949655E-69 | 0.365783471422239 | 0.991 | 0.972 | 1.66564502276246E-64 | 10 | Rps13 |
| Rpsa.4 | 1.63487468179917E-68 | 0.410456724629459 | 0.97 | 0.924 | 5.07677634939098E-64 | 10 | Rpsa |
| Rpl3.1 | 5.97723052301204E-68 | 0.387927199526091 | 0.981 | 0.948 | 1.85610939431093E-63 | 10 | Rpl3 |
| Rpl36.1 | 7.04870864668595E-68 | 0.404048425273195 | 0.982 | 0.937 | 2.18883549605539E-63 | 10 | Rpl36 |
| Rps8.2 | 1.16696488474549E-67 | 0.346950650496585 | 0.996 | 0.988 | 3.62377605660016E-63 | 10 | Rps8 |
| Rps3a1.1 | 1.93801590074705E-67 | 0.345791660404939 | 0.995 | 0.986 | 6.01812077658983E-63 | 10 | Rps3a1 |
| Rpl26.1 | 2.37993036253762E-67 | 0.359023090019497 | 0.996 | 0.976 | 7.39039775478806E-63 | 10 | Rpl26 |
| Rpl17.1 | 1.08608517346208E-65 | 0.379172928689477 | 0.983 | 0.963 | 3.37262028915178E-61 | 10 | Rpl17 |
| Rps25.3 | 4.25329726385286E-65 | 0.408598988381816 | 0.972 | 0.911 | 1.32077639934423E-60 | 10 | Rps25 |
| Ctsb.4 | 5.30318151315412E-65 | 0.384643994854843 | 0.993 | 0.98 | 1.64679695527975E-60 | 10 | Ctsb |
| Rps3.2 | 1.95664128551142E-61 | 0.330556146332798 | 0.991 | 0.977 | 6.07595818389862E-57 | 10 | Rps3 |
| Rpl28.1 | 1.31963476522321E-60 | 0.358901980683292 | 0.982 | 0.96 | 4.09786183644764E-56 | 10 | Rpl28 |
| Rpl36a.2 | 2.62634783211756E-58 | 0.431164599701323 | 0.918 | 0.836 | 8.15559792307465E-54 | 10 | Rpl36a |
| Rpl11.1 | 5.62240790628403E-58 | 0.328057923109089 | 0.993 | 0.973 | 1.74592632713838E-53 | 10 | Rpl11 |
| Rpl38.2 | 1.93951840703342E-57 | 0.36332541060221 | 0.972 | 0.947 | 6.02278650936088E-53 | 10 | Rpl38 |
| P2ry12.4 | 8.00248196633807E-57 | -0.297726944702868 | 0.996 | 0.993 | 2.48501072500696E-52 | 10 | P2ry12 |
| Rpl12.3 | 6.78621413778579E-55 | 0.379181959201782 | 0.973 | 0.927 | 2.10732307620662E-50 | 10 | Rpl12 |
| Rpl6.1 | 1.8192087184904E-54 | 0.332113960434845 | 0.989 | 0.976 | 5.64918883352823E-50 | 10 | Rpl6 |
| Rps2.2 | 5.47282412061971E-54 | 0.369715494690236 | 0.965 | 0.928 | 1.69947607417604E-49 | 10 | Rps2 |
| Rps26.2 | 1.23494681583076E-53 | 0.405454598280664 | 0.949 | 0.891 | 3.83488034719926E-49 | 10 | Rps26 |
| Lag3 | 2.92486249964473E-51 | 0.509404109510691 | 0.78 | 0.626 | 9.08257552014678E-47 | 10 | Lag3 |
| Npc2.3 | 5.79152611367355E-51 | 0.346329895070549 | 0.957 | 0.903 | 1.79844260407905E-46 | 10 | Npc2 |
| Rack1.2 | 2.47586316279089E-50 | 0.369052063003554 | 0.936 | 0.87 | 7.68829787941454E-46 | 10 | Rack1 |
| Rpl22.1 | 5.71801362044721E-46 | 0.334626348160738 | 0.951 | 0.898 | 1.77561476955747E-41 | 10 | Rpl22 |
| Rpl35.2 | 2.01505787678937E-44 | 0.376770581519025 | 0.937 | 0.876 | 6.25735922479402E-40 | 10 | Rpl35 |
| Cd52.4 | 6.1872743017145E-44 | 0.448903136652846 | 0.721 | 0.567 | 1.9213342889114E-39 | 10 | Cd52 |
| Ctsl.3 | 2.72123491388623E-43 | 0.284190613683329 | 0.996 | 0.984 | 8.45025077809092E-39 | 10 | Ctsl |
| Rpl18.1 | 2.29040308462909E-42 | 0.300520279548212 | 0.981 | 0.954 | 7.11238869869871E-38 | 10 | Rpl18 |
| Rpl7.2 | 5.14463693650129E-41 | 0.297961810999176 | 0.969 | 0.943 | 1.59756410789175E-36 | 10 | Rpl7 |
| Psap.3 | 1.56936840978199E-40 | 0.28378041916547 | 0.995 | 0.992 | 4.873359722896E-36 | 10 | Psap |
| Tmem119. | 2.16754322664113E-40 | -0.260587757532319 | 0.992 | 0.988 | 6.73087198168871E-36 | 10 | Tmem119 |
| mt-Cytb.1 | 6.45678746505167E-40 | 0.272066084543308 | 0.999 | 0.995 | 2.0050262115225E-35 | 10 | mt-Cytb |
| Rps6.1 | 1.04810303037004E-39 | 0.328657002876134 | 0.94 | 0.883 | 3.2546743402081E-35 | 10 | Rps6 |
| H2-D1.6 | 1.87515117571797E-38 | 0.394205833110366 | 0.836 | 0.725 | 5.82290694595701E-34 | 10 | H2-D1 |
| Rpl15.1 | 4.28089666817188E-37 | 0.293058911889982 | 0.98 | 0.939 | 1.32934684236741E-32 | 10 | Rpl15 |
| Timp2.3 | 6.34282886957515E-37 | 0.332791434751504 | 0.909 | 0.822 | 1.96963864886917E-32 | 10 | Timp2 |
| mt-Atp6.3 | 1.52300244145514E-36 | 0.252990905825849 | 0.997 | 0.998 | 4.72937948145064E-32 | 10 | mt-Atp6 |
| P2ry13.3 | 1.00301657116551E-35 | -0.319634017473755 | 0.906 | 0.946 | 3.11466735844026E-31 | 10 | P2ry13 |
| Slc2a5.3 | 2.8646323532995E-35 | -0.409248653600007 | 0.64 | 0.754 | 8.89554284670095E-31 | 10 | Slc2a5 |

(*Continued*)

**Table 1.** (Continued)

| | p_val | avg_log2FC | pct.1 | pct.2 | p_val_adj | cluster | gene |
|---|---|---|---|---|---|---|---|
| Eef1b2.1 | 4.85093439092742E-35 | 0.327637074869775 | 0.887 | 0.817 | 1.50636065641469E-30 | 10 | Eef1b2 |
| mt-Co3.3 | 2.55484774620138E-34 | 0.262370728644558 | 0.999 | 0.997 | 7.93356870627915E-30 | 10 | mt-Co3 |
| Rpl24.1 | 1.9421406607979E-33 | 0.289836884812964 | 0.955 | 0.909 | 6.03092939397572E-29 | 10 | Rpl24 |
| Rpl29.1 | 3.46201996148353E-33 | 0.310869566924759 | 0.942 | 0.894 | 1.07506105863948E-28 | 10 | Rpl29 |
| Rpl14.1 | 4.11065687970827E-33 | 0.297427868549755 | 0.906 | 0.845 | 1.27648228085581E-28 | 10 | Rpl14 |
| Rpl8.1 | 5.34324561215846E-33 | 0.265504269500336 | 0.977 | 0.949 | 1.65923805994357E-28 | 10 | Rpl8 |
| Rhob.4 | 9.18523876076832E-33 | -0.262286673786257 | 0.975 | 0.986 | 2.85229219238139E-28 | 10 | Rhob |
| Rpl7a.2 | 1.43093670649655E-32 | 0.305299785249621 | 0.928 | 0.858 | 4.44348775468373E-28 | 10 | Rpl7a |
| Cd164.4 | 1.734308858873E-32 | -0.372118114507751 | 0.701 | 0.79 | 5.38554929945832E-28 | 10 | Cd164 |
| Rpl5.1 | 2.70937165640831E-32 | 0.292399526422159 | 0.926 | 0.882 | 8.41341180464472E-28 | 10 | Rpl5 |
| Selenop.2 | 2.08396899359105E-31 | 0.259331014138179 | 0.996 | 0.983 | 6.47134891579827E-27 | 10 | Selenop |
| Maf.3 | 3.0337486787823E-30 | -0.309198484593745 | 0.894 | 0.924 | 9.42069977222267E-26 | 10 | Maf |
| Rps15.1 | 1.13040603362523E-28 | 0.277606219890133 | 0.912 | 0.853 | 3.51024985621643E-24 | 10 | Rps15 |
| Fcrls.4 | 5.91734346350332E-28 | -0.251735939660225 | 0.964 | 0.983 | 1.83751266572169E-23 | 10 | Fcrls |
| Txnip.3 | 1.07404125338772E-27 | -0.320025913854964 | 0.796 | 0.863 | 3.33522030414488E-23 | 10 | Txnip |
| Hspa5.2 | 1.12762003732415E-26 | -0.330468576798668 | 0.8 | 0.854 | 3.50159850190269E-22 | 10 | Hspa5 |
| Pnp.2 | 8.47659834155748E-26 | -0.30822938493096 | 0.821 | 0.875 | 2.63223808300384E-21 | 10 | Pnp |
| Qk.2 | 2.29737035424808E-25 | -0.255632507695065 | 0.931 | 0.949 | 7.13402416104656E-21 | 10 | Qk |
| Olfml3.2 | 4.51994681811757E-24 | -0.269320234359044 | 0.977 | 0.977 | 1.40357908543005E-19 | 10 | Olfml3 |
| Ecscr.4 | 1.34264082090337E-21 | -0.279329181152027 | 0.828 | 0.868 | 4.16930254115122E-17 | 10 | Ecscr |
| Sgk1.1 | 1.44186267417416E-21 | -0.350283048414311 | 0.769 | 0.821 | 4.47741616211303E-17 | 10 | Sgk1 |
| Rpl23a.2 | 4.86343099846509E-19 | 0.25512405688848 | 0.82 | 0.738 | 1.51024122795336E-14 | 10 | Rpl23a |
| St3gal6.2 | 3.86338671578639E-18 | -0.25832486025904 | 0.778 | 0.81 | 1.19969747685315E-13 | 10 | St3gal6 |
| Adrb2.1 | 8.65493769283677E-16 | -0.268154918999491 | 0.633 | 0.701 | 2.6876178017566E-11 | 10 | Adrb2 |
| Tmem176a | 6.7798943506942E-13 | 0.257733376709122 | 0.732 | 0.662 | 2.10536059272107E-08 | 10 | Tmem176a |
| Fau.3 | 1.00914476477373E-232 | 0.381999423516817 | 1 | 0.998 | 3.13369723805188E-228 | 11 | Fau |
| Rps29.2 | 9.43918258808252E-224 | 0.38214608646211 | 1 | 1 | 2.93114936907726E-219 | 11 | Rps29 |
| Tpt1.2 | 6.71159108957138E-214 | 0.354767200406928 | 1 | 0.999 | 2.0841503810446E-209 | 11 | Tpt1 |
| Rps24.2 | 2.09495723409771E-190 | 0.37257557177707 | 1 | 0.994 | 6.50547069904363E-186 | 11 | Rps24 |
| Rps27.5 | 3.28508428123223E-182 | 0.401307940626227 | 0.999 | 0.989 | 1.02011722185104E-177 | 11 | Rps27 |
| Rpl35a.2 | 6.53988713176546E-181 | 0.359563973889564 | 1 | 0.995 | 2.03083115102713E-176 | 11 | Rpl35a |
| Rplp1.2 | 1.60244849602338E-178 | 0.303506987704643 | 1 | 0.999 | 4.97608331470139E-174 | 11 | Rplp1 |
| Rpl13.2 | 1.41602548197314E-171 | 0.352014520317668 | 0.998 | 0.994 | 4.39718392917118E-167 | 11 | Rpl13 |
| Rpl18a.3 | 3.12666419403505E-169 | 0.375090781319993 | 0.999 | 0.989 | 9.70923032173704E-165 | 11 | Rpl18a |
| Rps21.4 | 4.65892503892541E-167 | 0.367580377191951 | 0.998 | 0.992 | 1.44673599233751E-162 | 11 | Rps21 |
| Eef1a1.4 | 2.36315207780235E-166 | 0.321560710358294 | 1 | 0.999 | 7.33829614719964E-162 | 11 | Eef1a1 |
| Rpl27a.4 | 9.94077990134489E-166 | 0.351562712985399 | 0.998 | 0.993 | 3.08691038276463E-161 | 11 | Rpl27a |
| Rpl30.3 | 4.42758546847812E-164 | 0.360361593397331 | 0.999 | 0.992 | 1.37489811552651E-159 | 11 | Rpl30 |
| Rpl37.4 | 2.91162145312323E-160 | 0.381895592625492 | 0.998 | 0.988 | 9.04145809838356E-156 | 11 | Rpl37 |
| Rps4x.5 | 6.5612485964065E-160 | 0.383473565703717 | 0.996 | 0.983 | 2.03746452664211E-155 | 11 | Rps4x |
| Rpl39.4 | 9.21922912384732E-158 | 0.373089627349552 | 0.996 | 0.988 | 2.86284721982831E-153 | 11 | Rpl39 |
| Rpl23.4 | 6.49515229467765E-157 | 0.361843681811832 | 0.998 | 0.986 | 2.01693964206625E-152 | 11 | Rpl23 |
| Rps11.4 | 9.08134146129344E-155 | 0.352002225092217 | 0.998 | 0.99 | 2.82002896397545E-150 | 11 | Rps11 |
| Rps12.2 | 4.08413651522183E-152 | 0.35773662754193 | 0.999 | 0.99 | 1.26824691207183E-147 | 11 | Rps12 |
| Rps3a1.2 | 2.5381184623679E-150 | 0.368090150208279 | 0.998 | 0.985 | 7.88161926119103E-146 | 11 | Rps3a1 |
| Rps14.3 | 4.56068389300495E-147 | 0.37416042567773 | 0.996 | 0.969 | 1.41622916929483E-142 | 11 | Rps14 |
| Rps23.2 | 1.18296497115089E-145 | 0.353254698094795 | 0.999 | 0.984 | 3.67346112491485E-141 | 11 | Rps23 |

(*Continued*)

**Table 1.** (Continued)

| | p_val | avg_log2FC | pct.1 | pct.2 | p_val_adj | cluster | gene |
|---|---|---|---|---|---|---|---|
| **Rpl32.3** | 8.27201479142128E-145 | 0.363332215199131 | 0.993 | 0.974 | 2.56870875318005E-140 | 11 | Rpl32 |
| **Rps9.4** | 3.63828761099171E-144 | 0.335241390641472 | 0.999 | 0.991 | 1.12979745184126E-139 | 11 | Rps9 |
| **Rpl34.4** | 7.48676616892516E-142 | 0.364636271007404 | 0.995 | 0.978 | 2.32486549843633E-137 | 11 | Rpl34 |
| **Rps10.2** | 1.55928194586442E-139 | 0.347071749677379 | 0.998 | 0.985 | 4.84203822649277E-135 | 11 | Rps10 |
| **Rpl37a.2** | 5.53164883999499E-139 | 0.347883044725671 | 0.998 | 0.989 | 1.71774291428364E-134 | 11 | Rpl37a |
| **Rps28.4** | 3.09118346777759E-138 | 0.391989780553777 | 0.988 | 0.956 | 9.59905202248976E-134 | 11 | Rps28 |
| **Rps27a.2** | 7.54793958262459E-134 | 0.32725449420847 | 0.998 | 0.989 | 2.34386167859241E-129 | 11 | Rps27a |
| **Rps15a.2** | 9.79798161519934E-133 | 0.345163038528053 | 0.995 | 0.978 | 3.04256723096785E-128 | 11 | Rps15a |
| **Rpl10.3** | 6.07551720648778E-132 | 0.338773052296286 | 0.996 | 0.984 | 1.88663035813065E-127 | 11 | Rpl10 |
| **Rps3.3** | 2.6712323087565E-126 | 0.345375495809468 | 0.992 | 0.976 | 8.29497768838156E-122 | 11 | Rps3 |
| **Rpl21.2** | 6.37364036898494E-126 | 0.322197493857587 | 0.998 | 0.991 | 1.97920654378089E-121 | 11 | Rpl21 |
| **Rps16.2** | 6.59716723865906E-111 | 0.325465074810795 | 0.994 | 0.974 | 2.0486183426208E-106 | 11 | Rps16 |
| **Rps7.2** | 2.19006157515073E-109 | 0.338042230691532 | 0.99 | 0.959 | 6.80079820931557E-105 | 11 | Rps7 |
| **Rps8.3** | 8.64234547986014E-109 | 0.298039019433599 | 0.998 | 0.987 | 2.68370754186097E-104 | 11 | Rps8 |
| **Rpl9.2** | 1.61761781340538E-108 | 0.325624632190307 | 0.986 | 0.969 | 5.02318859596774E-104 | 11 | Rpl9 |
| **Rpl26.2** | 1.95725094165712E-108 | 0.312442268029107 | 0.993 | 0.976 | 6.07785134912785E-104 | 11 | Rpl26 |
| **Rpl17.2** | 5.80985842206008E-107 | 0.33049098392024 | 0.99 | 0.961 | 1.80413533580232E-102 | 11 | Rpl17 |
| **Rps20.2** | 4.73480693974359E-104 | 0.318091813664898 | 0.991 | 0.972 | 1.47029959899858E-99 | 11 | Rps20 |
| **Rps13.3** | 1.21905880387685E-102 | 0.323812065551955 | 0.99 | 0.971 | 3.78554330367878E-98 | 11 | Rps13 |
| **Rplp0.3** | 4.73238316161159E-101 | 0.305816802408739 | 0.993 | 0.968 | 1.46954694317525E-96 | 11 | Rplp0 |
| **Rps5.3** | 6.38893685139109E-98 | 0.346412740033853 | 0.977 | 0.929 | 1.98395656046248E-93 | 11 | Rps5 |
| **Rpl19.2** | 8.97050772983177E-96 | 0.289876055080998 | 0.997 | 0.977 | 2.78561176534466E-91 | 11 | Rpl19 |
| **Rplp2.2** | 2.65154903912819E-94 | 0.309600479990419 | 0.992 | 0.967 | 8.23385523120477E-90 | 11 | Rplp2 |
| **Rpl6.2** | 3.82755198171549E-94 | 0.299438453505658 | 0.993 | 0.975 | 1.18856971688211E-89 | 11 | Rpl6 |
| **Rpl38.3** | 8.03963332160593E-94 | 0.310446880242178 | 0.984 | 0.943 | 2.49654733535829E-89 | 11 | Rpl38 |
| **Rpl41.2** | 1.00792066986345E-91 | 0.264868248455206 | 0.998 | 0.989 | 3.12989605612699E-87 | 11 | Rpl41 |
| **Rpl3.2** | 5.81365803873062E-91 | 0.329307412654396 | 0.983 | 0.946 | 1.80531523076702E-86 | 11 | Rpl3 |
| **Rpl7.3** | 2.51383515046433E-89 | 0.319576218282623 | 0.974 | 0.941 | 7.80621229273688E-85 | 11 | Rpl7 |
| **Rpl28.2** | 2.65000638549962E-84 | 0.304941248955486 | 0.983 | 0.959 | 8.22906482889196E-80 | 11 | Rpl28 |
| **Rpl11.2** | 3.49864512547977E-84 | 0.279991023874725 | 0.988 | 0.972 | 1.08643427081523E-79 | 11 | Rpl11 |
| **Rps19.3** | 9.84093819519042E-83 | 0.298265270404637 | 0.985 | 0.951 | 3.05590653775248E-78 | 11 | Rps19 |
| **Rpl36.2** | 9.18438382503191E-81 | 0.305223102981423 | 0.971 | 0.936 | 2.85202670918716E-76 | 11 | Rpl36 |
| **Rpl22.2** | 1.90522207659401E-80 | 0.330100650419271 | 0.964 | 0.893 | 5.91628611444738E-76 | 11 | Rpl22 |
| **Rpl35.3** | 1.68473527225578E-76 | 0.350261700737557 | 0.932 | 0.873 | 5.23160844093587E-72 | 11 | Rpl35 |
| **Rps25.4** | 1.24177001344861E-73 | 0.315818509530406 | 0.96 | 0.909 | 3.85606842276196E-69 | 11 | Rps25 |
| **Rpsa.5** | 1.01807660796439E-70 | 0.291177408056235 | 0.971 | 0.921 | 3.16143329071183E-66 | 11 | Rpsa |
| **Rpl36a.3** | 1.06678674559224E-70 | 0.33583204067196 | 0.921 | 0.83 | 3.31269288108757E-66 | 11 | Rpl36a |
| **Rps18.2** | 3.32921781962459E-70 | 0.277240190734208 | 0.979 | 0.943 | 1.03382200952802E-65 | 11 | Rps18 |
| **Rpl12.4** | 2.19353700289621E-69 | 0.290606781007832 | 0.969 | 0.925 | 6.81159045509361E-65 | 11 | Rpl12 |
| **Rpl8.2** | 2.36471572132581E-66 | 0.275503203926762 | 0.98 | 0.946 | 7.34315172943303E-62 | 11 | Rpl8 |
| **Rpl24.2** | 3.494280603238E-64 | 0.295092117044214 | 0.955 | 0.906 | 1.0850789557235E-59 | 11 | Rpl24 |
| **Rack1.3** | 2.49565553948227E-61 | 0.304077322097207 | 0.939 | 0.865 | 7.7497591467543E-57 | 11 | Rack1 |
| **Rps6.2** | 5.00150686649441E-61 | 0.298369380250195 | 0.938 | 0.88 | 1.55311792725251E-56 | 11 | Rps6 |
| **Rps26.3** | 1.26966434268137E-60 | 0.297643234062698 | 0.94 | 0.889 | 3.94268868332845E-56 | 11 | Rps26 |
| **Rpl29.2** | 3.16301339934733E-56 | 0.272722548359096 | 0.941 | 0.891 | 9.82210550899325E-52 | 11 | Rpl29 |
| **Rpl5.2** | 1.36563026567911E-55 | 0.289839622000832 | 0.931 | 0.879 | 4.24069166401335E-51 | 11 | Rpl5 |
| **Rps15.2** | 1.3639847310554E-50 | 0.280519633695111 | 0.919 | 0.848 | 4.23558178534633E-46 | 11 | Rps15 |

*(Continued)*

**Table 1.** (Continued)

| | p_val | avg_log2FC | pct.1 | pct.2 | p_val_adj | cluster | gene |
|---|---|---|---|---|---|---|---|
| **Eef1b2.2** | 9.30812220480122E-43 | 0.274538812066377 | 0.89 | 0.812 | 2.89045118825692E-38 | 11 | Eef1b2 |
| **Rpl23a.3** | 2.46982059543098E-36 | 0.262569513023566 | 0.823 | 0.732 | 7.66953389499183E-32 | 11 | Rpl23a |
| **Cox7a2l.1** | 5.54353874942074E-32 | 0.258018288196526 | 0.739 | 0.65 | 1.72143508785762E-27 | 11 | Cox7a2l |
| **Sgk1.2** | 5.23988650792421E-29 | -0.30962880324034 | 0.767 | 0.825 | 1.6271419573057E-24 | 11 | Sgk1 |
| **Hspa5.3** | 2.53981627427702E-26 | -0.263058929255922 | 0.806 | 0.856 | 7.88689147651242E-22 | 11 | Hspa5 |
| **Rps24.3** | 5.998497889333E-256 | -0.394633682522278 | 0.992 | 0.996 | 1.86271354957458E-251 | 12 | Rps24 |
| **Fau.4** | 9.76651237936016E-254 | -0.363755424813346 | 0.998 | 0.999 | 3.03279508916271E-249 | 12 | Fau |
| **Rplp1.3** | 4.05274804226819E-252 | -0.303853877590739 | 0.999 | 1 | 1.25849984956554E-247 | 12 | Rplp1 |
| **Rps29.3** | 1.02783850290498E-244 | -0.3478531891415 | 1 | 1 | 3.19174690307084E-240 | 12 | Rps29 |
| **Rpl32.4** | 7.36146606438397E-244 | -0.449145532814297 | 0.963 | 0.982 | 2.28595605697315E-239 | 12 | Rpl32 |
| **Rpl13.3** | 4.72415500770956E-233 | -0.369887383196817 | 0.992 | 0.996 | 1.46699185454405E-228 | 12 | Rpl13 |
| **Eef1a1.5** | 6.86996179280032E-233 | -0.336337302721933 | 0.998 | 0.999 | 2.13332923551828E-228 | 12 | Eef1a1 |
| **Rpl30.4** | 6.35993481676531E-230 | -0.36908453872846 | 0.987 | 0.995 | 1.97495055865013E-225 | 12 | Rpl30 |
| **Tpt1.3** | 3.6035813545058E-227 | -0.332500402993238 | 0.998 | 0.999 | 1.11902011801468E-222 | 12 | Tpt1 |
| **Rpl23.5** | 1.93266493794577E-223 | -0.40265984053389 | 0.98 | 0.991 | 6.00150443180301E-219 | 12 | Rpl23 |
| **Rps12.3** | 1.89927131848971E-222 | -0.384929209045021 | 0.987 | 0.993 | 5.89780722530608E-218 | 12 | Rps12 |
| **Rpl27a.5** | 6.67858021661473E-207 | -0.34821989057002 | 0.991 | 0.996 | 2.07389951466537E-202 | 12 | Rpl27a |
| **Rpl35a.3** | 7.08543577549241E-201 | -0.335045257881166 | 0.995 | 0.996 | 2.20024037136366E-196 | 12 | Rpl35a |
| **Rps15a.3** | 7.95132309895143E-200 | -0.379339545341713 | 0.971 | 0.984 | 2.46912436191739E-195 | 12 | Rps15a |
| **Rps4x.6** | 4.6211320100241E-193 | -0.377586458260358 | 0.975 | 0.989 | 1.43500012307278E-188 | 12 | Rps4x |
| **Rps23.3** | 2.34472949716352E-189 | -0.346853130664285 | 0.978 | 0.989 | 7.28108850754189E-185 | 12 | Rps23 |
| **Rpl39.5** | 6.25394959624099E-189 | -0.375730006217627 | 0.986 | 0.99 | 1.94203896812071E-184 | 12 | Rpl39 |
| **Rps27a.3** | 1.29381882390593E-188 | -0.342395023515312 | 0.987 | 0.992 | 4.01769559387507E-184 | 12 | Rps27a |
| **Rps5.4** | 1.57135258894067E-182 | -0.434082927439527 | 0.904 | 0.95 | 4.87952119443746E-178 | 12 | Rps5 |
| **Rps10.3** | 1.25335640909659E-181 | -0.336601909385188 | 0.978 | 0.991 | 3.89204765716765E-177 | 12 | Rps10 |
| **Rps21.5** | 4.94076568327923E-181 | -0.337282923967583 | 0.989 | 0.994 | 1.5342559676287E-176 | 12 | Rps21 |
| **Rps19.4** | 8.52476128990415E-181 | -0.41525534389086 | 0.932 | 0.966 | 2.64719412335393E-176 | 12 | Rps19 |
| **Rplp0.4** | 1.00783904092009E-176 | -0.367291936966278 | 0.957 | 0.978 | 3.12964257376914E-172 | 12 | Rplp0 |
| **Rps11.5** | 2.76412675132683E-176 | -0.342949794204847 | 0.986 | 0.993 | 8.5834428008952E-172 | 12 | Rps11 |
| **Fth1.5** | 5.21330909873393E-175 | -0.397345754084254 | 0.994 | 0.997 | 1.61888887442985E-170 | 12 | Fth1 |
| **Rps27.6** | 7.53801247519455E-175 | -0.356280739673766 | 0.986 | 0.992 | 2.34077901392217E-170 | 12 | Rps27 |
| **Rpl41.3** | 6.04889103227734E-174 | -0.344843624686359 | 0.985 | 0.992 | 1.87836213225308E-169 | 12 | Rpl41 |
| **Rpsa.6** | 1.05790406705618E-170 | -0.432708526614709 | 0.895 | 0.943 | 3.28510949942956E-166 | 12 | Rpsa |
| **Rps14.4** | 1.80112590003184E-169 | -0.368835901755145 | 0.957 | 0.979 | 5.59303625736888E-165 | 12 | Rps14 |
| **Rps20.3** | 3.96431676055841E-169 | -0.357999180417248 | 0.963 | 0.98 | 1.2310392836562E-164 | 12 | Rps20 |
| **Rplp2.3** | 4.42229075933991E-168 | -0.349224307008848 | 0.954 | 0.978 | 1.37325394949782E-163 | 12 | Rplp2 |
| **Rpl21.3** | 2.81581138828779E-167 | -0.315616129596249 | 0.987 | 0.994 | 8.74393910405007E-163 | 12 | Rpl21 |
| **Rpl9.3** | 1.27075603753764E-166 | -0.351294713038499 | 0.958 | 0.977 | 3.94607872336564E-162 | 12 | Rpl9 |
| **Rps16.3** | 1.82837205686959E-166 | -0.364080728464849 | 0.965 | 0.981 | 5.67764374819714E-162 | 12 | Rps16 |
| **Rpl37a.3** | 3.69790109019501E-164 | -0.336910181842119 | 0.986 | 0.992 | 1.14830922553826E-159 | 12 | Rpl37a |
| **Rps8.4** | 3.73433120826548E-164 | -0.324564582726674 | 0.983 | 0.991 | 1.15962187010268E-159 | 12 | Rps8 |
| **Rps9.5** | 6.15482563128004E-162 | -0.315387430991811 | 0.987 | 0.995 | 1.91125800328139E-157 | 12 | Rps9 |
| **Rps28.5** | 7.27605386219421E-162 | -0.405258706831361 | 0.944 | 0.967 | 2.25943300582717E-157 | 12 | Rps28 |
| **Rps18.3** | 1.67553575870695E-160 | -0.38181297434791 | 0.922 | 0.96 | 5.20304119151269E-156 | 12 | Rps18 |
| **Rpl18a.4** | 2.21614169561422E-159 | -0.327274803948634 | 0.985 | 0.993 | 6.88178480739085E-155 | 12 | Rpl18a |
| **Rpl19.3** | 2.86365460716991E-159 | -0.334999459091495 | 0.97 | 0.984 | 8.89250665164473E-155 | 12 | Rpl19 |
| **Rpl37.5** | 1.14640363064214E-157 | -0.330275149065809 | 0.985 | 0.992 | 3.55992719423302E-153 | 12 | Rpl37 |

*(Continued)*

**Table 1.** (Continued)

| | p_val | avg_log2FC | pct.1 | pct.2 | p_val_adj | cluster | gene |
|---|---|---|---|---|---|---|---|
| **Rpl12.5** | 2.51687814090459E-157 | -0.385278273948043 | 0.9 | 0.945 | 7.81566169095103E-153 | 12 | Rpl12 |
| **Rps3a1.3** | 5.90360652282855E-157 | -0.325667325739164 | 0.979 | 0.991 | 1.83324693353395E-152 | 12 | Rps3a1 |
| **Rpl3.3** | 2.08190974563607E-152 | -0.365191454118041 | 0.922 | 0.964 | 6.46495433312367E-148 | 12 | Rpl3 |
| **Rpl34.5** | 1.62582426919469E-150 | -0.32491207001262 | 0.971 | 0.984 | 5.04867210313026E-146 | 12 | Rpl34 |
| **Rps7.3** | 1.76873748120025E-149 | -0.355279365881805 | 0.942 | 0.972 | 5.49246050037114E-145 | 12 | Rps7 |
| **Rps2.3** | 4.16723898776737E-142 | -0.369740104528157 | 0.909 | 0.941 | 1.2940527228714E-137 | 12 | Rps2 |
| **Rps13.4** | 1.44419860567106E-137 | -0.325902154289488 | 0.963 | 0.978 | 4.48466990019035E-133 | 12 | Rps13 |
| **Rpl38.4** | 4.11375574979499E-137 | -0.364147579138721 | 0.929 | 0.958 | 1.27744457298384E-132 | 12 | Rpl38 |
| **Rpl17.3** | 4.78141555868286E-137 | -0.339193322166393 | 0.948 | 0.973 | 1.48477297343779E-132 | 12 | Rpl17 |
| **Rpl26.3** | 7.36095779827572E-135 | -0.309841088700522 | 0.97 | 0.982 | 2.28579822509856E-130 | 12 | Rpl26 |
| **Rpl36.3** | 2.82739964761746E-134 | -0.357061997447862 | 0.918 | 0.951 | 8.77992412574649E-130 | 12 | Rpl36 |
| **Rpl11.3** | 1.15645617274359E-128 | -0.299634189101647 | 0.962 | 0.98 | 3.59114335322068E-124 | 12 | Rpl11 |
| **Rpl10a.5** | 9.35831399712361E-127 | -0.372085188043589 | 0.868 | 0.918 | 2.90603724552679E-122 | 12 | Rpl10a |
| **Rpl10.4** | 1.94579027655934E-126 | -0.285769900995899 | 0.981 | 0.988 | 6.04226254579973E-122 | 12 | Rpl10 |
| **Rps25.5** | 2.04610605598998E-126 | -0.353084534513678 | 0.881 | 0.932 | 6.35377313566568E-122 | 12 | Rps25 |
| **Rpl6.3** | 1.4342376550081E-118 | -0.291404873015504 | 0.966 | 0.982 | 4.45373819009667E-114 | 12 | Rpl6 |
| **Rps26.4** | 1.0411885363562E-114 | -0.364583591613055 | 0.858 | 0.913 | 3.23320276194691E-110 | 12 | Rps26 |
| **Rps3.4** | 1.39913011300749E-114 | -0.285066365736839 | 0.971 | 0.981 | 4.34471873992215E-110 | 12 | Rps3 |
| **Rpl28.3** | 3.0926701472285E-112 | -0.302314255755959 | 0.952 | 0.967 | 9.60366860818867E-108 | 12 | Rpl28 |
| **Rpl22.3** | 4.33368878606153E-106 | -0.321218565685524 | 0.866 | 0.919 | 1.34574037873569E-101 | 12 | Rpl22 |
| **Rhob.5** | 2.21122722150697E-104 | 0.252182566302041 | 0.994 | 0.98 | 6.86652389094558E-100 | 12 | Rhob |
| **Rpl36a.4** | 2.70741622684613E-104 | -0.358370601150229 | 0.788 | 0.867 | 8.40733960922529E-100 | 12 | Rpl36a |
| **Ftl1.2** | 6.84809776551012E-100 | -0.342876047619353 | 0.987 | 0.992 | 2.12653979912386E-95 | 12 | Ftl1 |
| **Rpl18.2** | 5.65203863603851E-99 | -0.282870723419333 | 0.935 | 0.965 | 1.75512755764904E-94 | 12 | Rpl18 |
| **Rack1.4** | 3.11856957286769E-98 | -0.341617954725745 | 0.836 | 0.893 | 9.68409409462603E-94 | 12 | Rack1 |
| **Rpl8.3** | 5.01189483862522E-98 | -0.276297040404362 | 0.934 | 0.959 | 1.55634370423829E-93 | 12 | Rpl8 |
| **Rpl24.3** | 7.71138595738696E-97 | -0.310609875188744 | 0.886 | 0.925 | 2.39461668134737E-92 | 12 | Rpl24 |
| **Rpl7.4** | 1.28182443222934E-94 | -0.280498806413339 | 0.926 | 0.954 | 3.98044940940175E-90 | 12 | Rpl7 |
| **Rpl35.4** | 1.51352432837602E-91 | -0.343336833023415 | 0.846 | 0.897 | 4.69994709690606E-87 | 12 | Rpl35 |
| **Rpl29.3** | 5.32935827018169E-83 | -0.291393811270776 | 0.861 | 0.914 | 1.65492562363952E-78 | 12 | Rpl29 |
| **Rpl15.2** | 3.58838345990969E-81 | -0.256817914774629 | 0.923 | 0.951 | 1.11430071580575E-76 | 12 | Rpl15 |
| **Maf.4** | 5.95651685372799E-81 | 0.254707396642929 | 0.944 | 0.911 | 1.84967717858815E-76 | 12 | Maf |
| **Rps6.3** | 7.52314826845287E-81 | -0.286056724047544 | 0.853 | 0.903 | 2.33616323180267E-76 | 12 | Rps6 |
| **Rpl5.3** | 1.48269489999568E-73 | -0.279652101424222 | 0.849 | 0.902 | 4.60421247295657E-69 | 12 | Rpl5 |
| **Rpl14.2** | 3.37756180943416E-67 | -0.262588510408555 | 0.812 | 0.867 | 1.04883426868359E-62 | 12 | Rpl14 |
| **Ctsb.5** | 6.61924338402115E-65 | -0.346605447257156 | 0.976 | 0.983 | 2.05547364804009E-60 | 12 | Ctsb |
| **Cd164.5** | 5.17020258536369E-64 | 0.2647341953667 | 0.827 | 0.764 | 1.60550300883299E-59 | 12 | Cd164 |
| **Rps15.3** | 1.43352638386833E-63 | -0.263614745836866 | 0.821 | 0.874 | 4.45152947982634E-59 | 12 | Rps15 |
| **Nrip1.2** | 8.24849263444341E-62 | 0.254328721625188 | 0.879 | 0.832 | 2.56140441777371E-57 | 12 | Nrip1 |
| **Sgk1.3** | 4.6793254259069E-60 | 0.288705226928264 | 0.854 | 0.8 | 1.45307092450687E-55 | 12 | Sgk1 |
| **Eef1b2.3** | 5.97330209763794E-57 | -0.261423029127211 | 0.782 | 0.841 | 1.85488950037951E-52 | 12 | Eef1b2 |
| **H2-D1.7** | 1.49715182429795E-48 | -0.425000702810103 | 0.692 | 0.753 | 4.64910555999243E-44 | 12 | H2-D1 |
| **H2-K1.4** | 9.36544322381961E-25 | -0.27956804594711 | 0.685 | 0.742 | 2.9082510842927E-20 | 12 | H2-K1 |
| **Xist** | 0 | 2.24173168755572 | 0.998 | 0.307 | 0 | 13 | Xist |
| **Hspa8.3** | 6.06325114543686E-80 | -0.31450326549286 | 0.841 | 0.905 | 1.88282137819251E-75 | 13 | Hspa8 |
| **Hsp90ab1.** | 5.25446982066137E-56 | -0.296097155215301 | 0.797 | 0.861 | 1.63167051340997E-51 | 13 | Hsp90ab1 |
| **H2-D1.8** | 1.15028084631823E-40 | -0.388842643535434 | 0.68 | 0.753 | 3.571967112072E-36 | 13 | H2-D1 |

express microglia marker genes and no border macrophage genes such as *Mrc1*, *Ms4a7*, and *Pf4* were detected. Cluster 6 expresses high levels of *Cdkn1a* and *Bax* suggesting that these cells may be undergoing a4poptosis. Lastly, Cluster 7 shows enrichment of *Pmepa1* which has been identified in TGF-ß signaling [37]. Given that these clusters do not have many differentially expressed genes, they are unlikely to be distinct populations.

The remaining clusters express genes associated with either distinct functions or were characteristic of non-microglial populations. The presence of non-microglia macrophages has been reported in numerous studies [11, 12, 35, 38, 39]. Cluster 2 appears to be a non-microglia macrophage population, based on, expression of genes associated with antigen presentation including *H2-Aa*, *H2-Eb1*, and *Cd74* in over half of the cells. These genes have been shown by previous findings to define adult choroid plexus macrophages [14]. The remaining cells in cluster 2 appear to have the same transcription profile associated with border macrophages or CNS-associated macrophages marked with expression of marker genes such as *Mrc1*, *Pf4*, and *Ms4a7 [40]*. These two groups thus likely comprise non-microglia myeloid lineage cells in the hippocampus.

In the healthy brain, there is some degree of microglial turnover. Approximately one percent of cells in the total myeloid population in the hippocampus express genes undergoing cell cycle transition. These cells, enriched in Cluster 1, are expressing *Mki67*, *Top2a*, *H2afz*, known to be found in proliferating microglia [14]. Cluster 3 expresses genes known to be implicated in interferon response. Many of these genes, such as *Rtp4*, *Ifit3*, *Ifit2*, *Ifitm3*, *Oasl2*, have also been implicated in the aging transcriptome [15]. By contrast, cluster 9 comprises cells that exhibit an activation profile, potentially as an artifact of the isolation protocol, as evidenced by the upregulation of immediate early genes such as *Fos*, *Egr1*, and *Jun*; these genes have been shown to correlate specifically with activation due to homogenization and other isolation-associated experimental steps [14]. Cluster 10 exhibits increased expression of genes associated with encoding ribosomal protein subunits (genes with *Rps* and *Rpl* as their prefix) and also contains the highest *ApoE* expression. This group of cells undergoing high metabolic activity has been identified by other groups and may reflect cells with more metabolic needs which are typically enriched with genes associated with immune reactivity [41–43]. The expansion of cells falling in this cluster may be a sign of reactivity in certain contexts. Cluster 8 exhibits expression of genes associated with the Disease-associated microglial (DAM) phenotype, namely *Cd9*, *Cd63*, *Cst7 [8]* and more detailed analysis localizes this novel cluster to the dentate gyrus subgranular zone (Figs 3 and 4).

## Myeloid Cd68 expression localizes to the subgranular zone of the dentate gyrus

In order to test whether microglia associated with the neurogenic niche are represented as a distinct transcriptomic cluster, we used our double reporter mice (S1 Fig) to visualize both neural progenitor cells and myeloid lineage cells. The neural progenitor pool clearly demarcates the SGZ from the rest of the dentate gyrus/hippocampus (Fig 2A). Somas from these cells line the interior (medial area) of the dentate gyrus. Processes of these cells protrude through the granule cell layer in the dentate gyrus to the molecular layer. Microglia in this region break the tile pattern they normally have in the adult cortex where processes of adjacent microglia seldomly coincide (Fig 2B and 2E). Such high density, or clustering, of microglia is often associated with increased microglial activity, particularly in clearing apoptotic cells [44, 45]. Additionally, neural stem cell processes are highly wrapped around microglial processes. Microglia are observed in very close apposition to neural progenitors in this region (Fig 2D). Cd68, (macrosialin) is found on the surface of lysosomes and increased Cd68 expression is detected

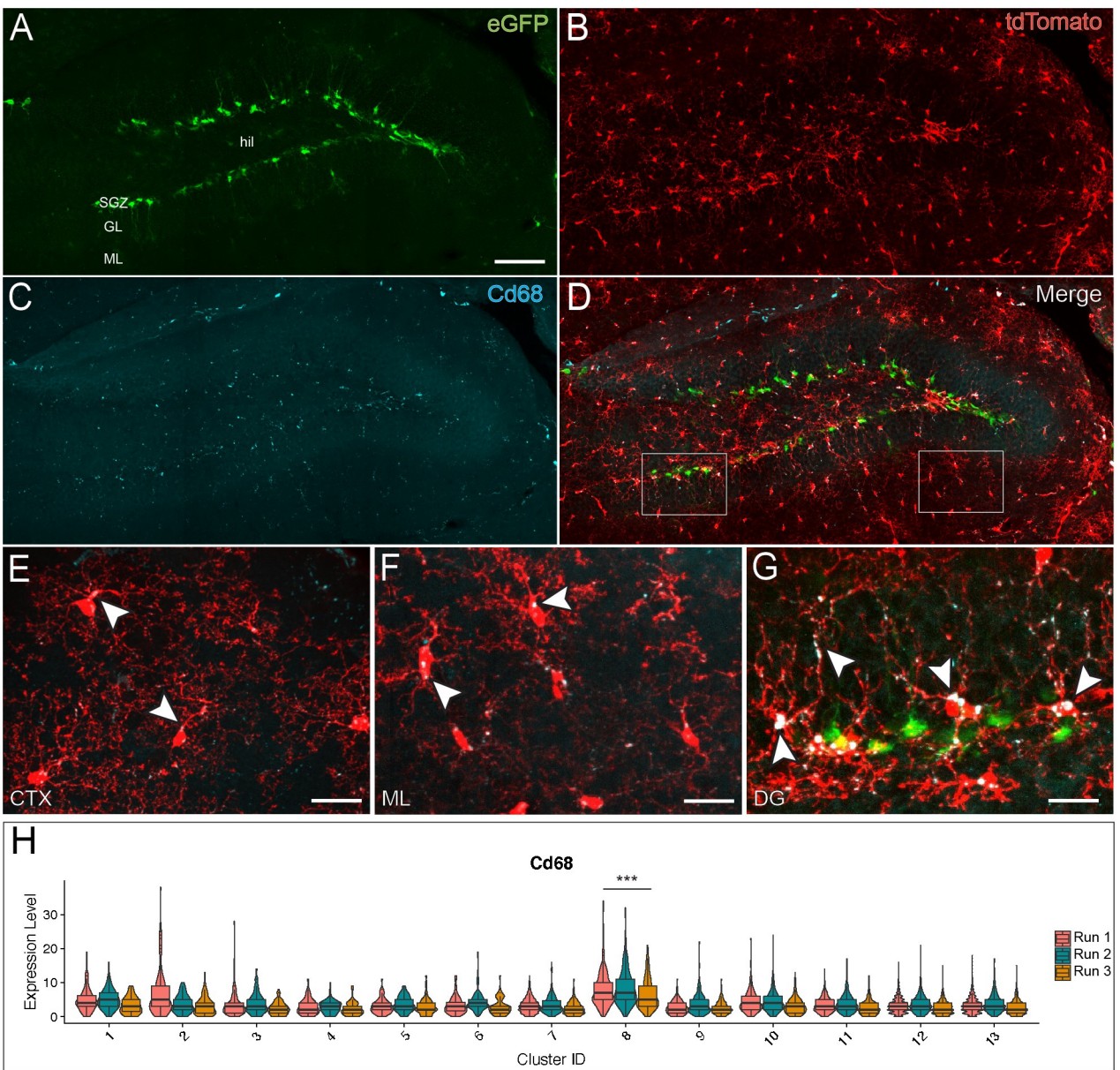

**Fig 2. Increased Cd68 is localized to subgranular zone of the dentate gyrus and transcriptomic cluster 8.** (A) Distribution of Nestin-eGFP expressing neural progenitors located in the SGZ. hil = hilus, GL = granule cell layer, ML = molecular layer. (B) Distribution of myeloid lineage cells expressing tdTomato in the dentate gyrus. (C) CD68+ lysosomal content staining in the dentate gyrus. (D) Merge image at showing colocalization of Cd68+ lysosomes in myeloid cells in apposition to Nestin-GFP cells. CD68+ lysosomal puncta in CTX (E) and ML (F) vs SGZ (G) with arrow heads to highlight CD68/tdTomato colocalization. Scale bars A&B = 40 μm; D-G: 100 μm; E = 25 μm F& G =: 20 μm. (H) Violin Plot with superimposed boxplots to show *Cd68* transcript counts and median values across clusters and runs.

in regions with neuronal death such as the cerebellum and olfactory bulb [18, 45, 46]. Cd68 puncta show higher colocalization in cells within the SGZ as well as processes from these cells (Fig 2C, 2D and 2G) as compared to microglia in the cortex or elsewhere in the dentate gyrus (Fig 2E and 2F). We analyzed the hippocampal clusters to see whether any transcriptomic cluster expresses elevated levels of Cd68 and observe that cluster 8 has significantly higher expression of Cd68 when compared to all other clusters (log2FC = 0.87; p-val <0.001) (Fig 2H).

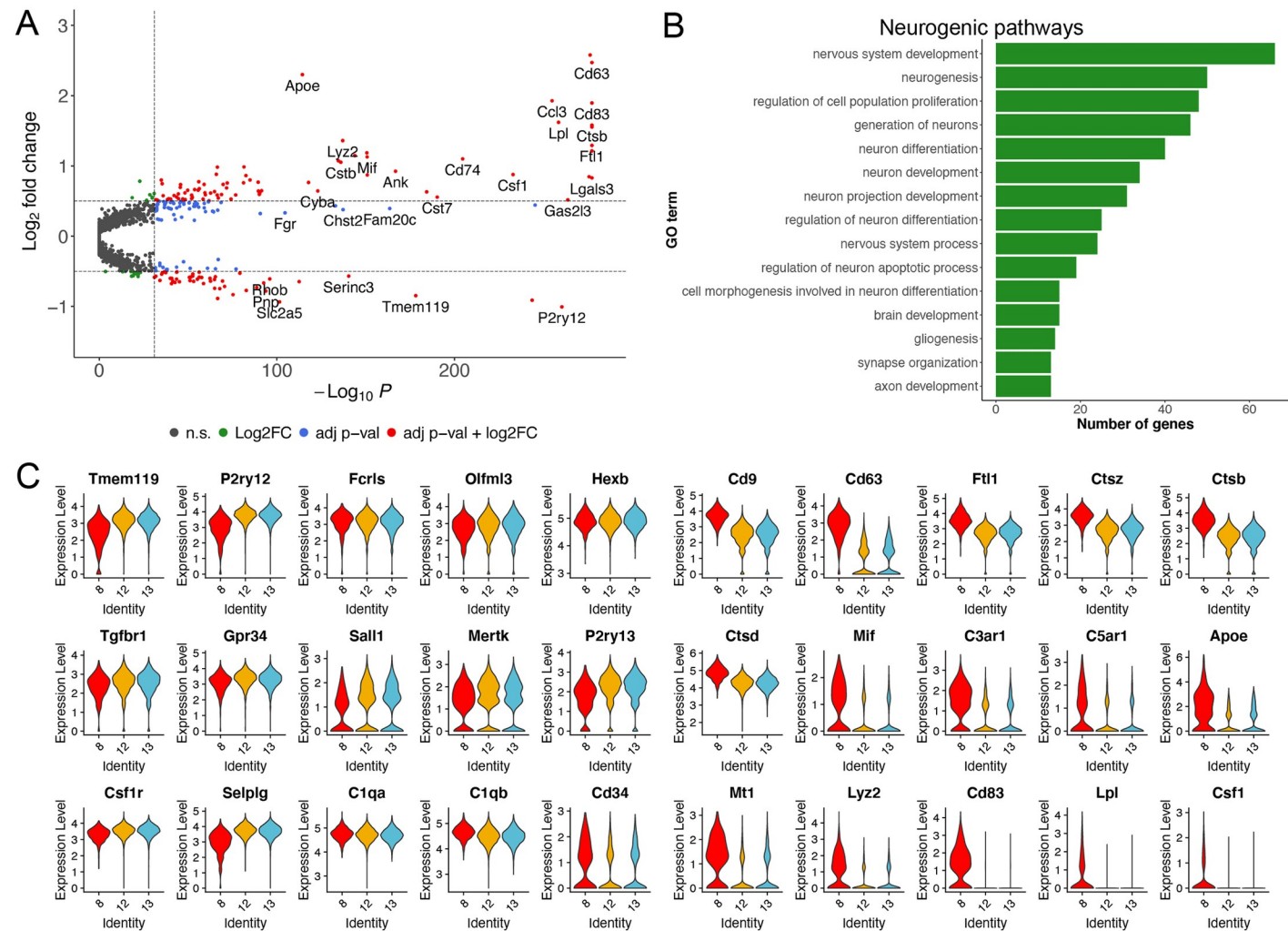

**Fig 3. Transcriptomic profile of cluster 8.** (A) Volcano plot showing differentially expressed genes in Cluster 8 microglia compared to other hippocampal microglia. Statistically significant genes (up or down- regulated) are represented by red dots (LFC > 0.25 and p-val < 10e-32). (B) Gene set enrichment analysis of upregulated genes in cluster 8 showing select ontology related to neurogenesis. (C) Violin Plots showing comparison of expression profiles of key genes in Cluster 8 vs homeostatic clusters (13 and 14).

## Transcriptomic analysis of the putative subgranular zone cluster demonstrates unique expression profile

After identifying Cluster 8 as the putative SGZ cluster, we tested for other genes differentially expressed by this cluster. We cross-referenced the Allen Institute *In Situ* Hybridization atlas to confirm spatial patterning of genes enriched or downregulated in this cluster (S6 Fig) [47]. Some previous studies have shown many microglia-specific marker genes to be downregulated in the context of immune activation [8]. Similarly, the SGZ microglia also exhibit decreased expression of microglia marker genes such as *Tmem119*, *P2ry12*, *Selplg,* and *Siglech* (Fig 3A).

Using the cluster-8 specific gene list, we next examined the gene set enrichment analysis applying the Kolmogorov-Smirnov test through the topGO package for annotation of terms. We set a stringent false discovery rate (adjusted p-value less than 0.05) and found several gene ontology (GO) terms related to nervous system development, which we have highlighted in

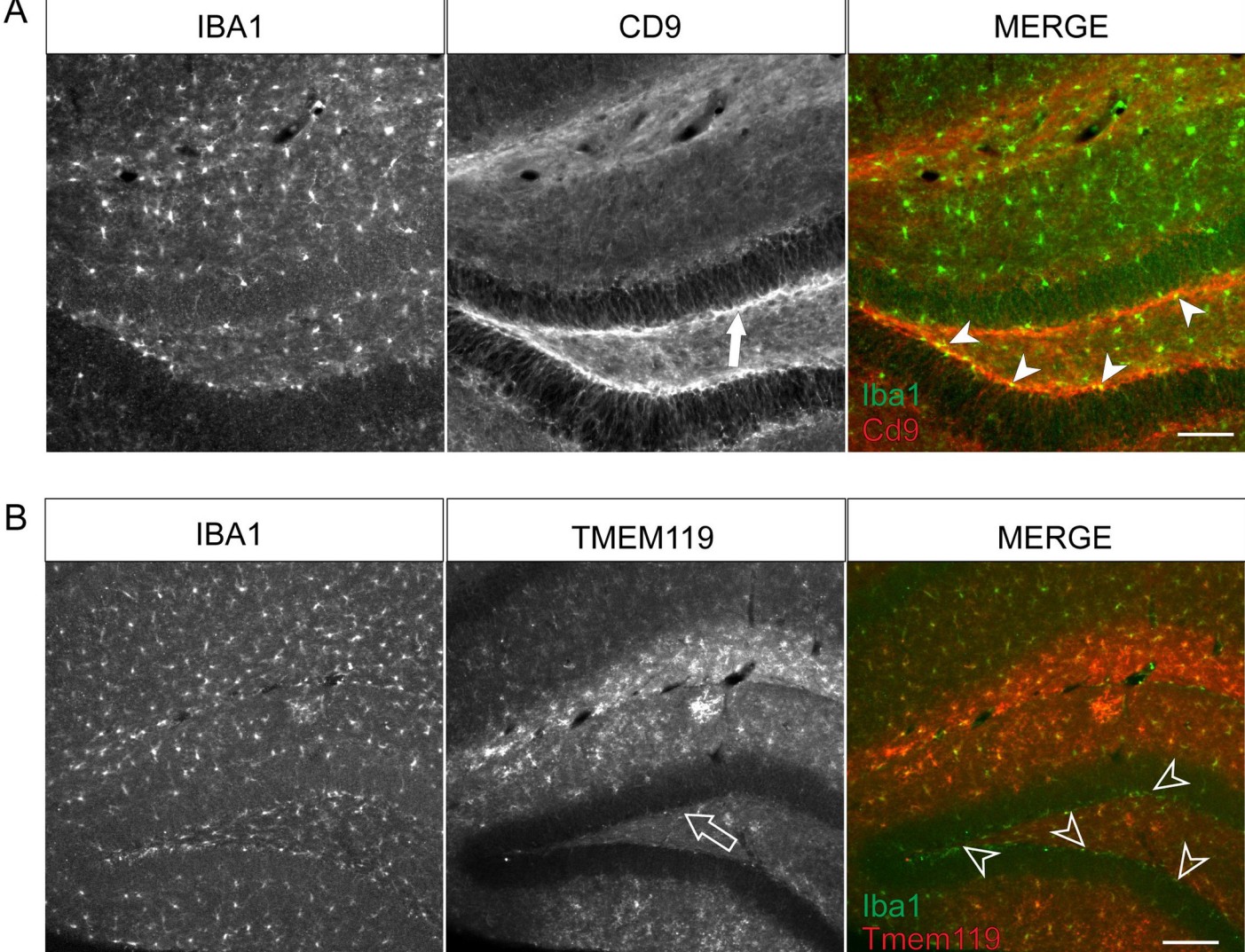

**Fig 4. Immunoreactivity of marker genes for cluster 8 in the dentate gyrus.** (A) Immunohistochemistry staining for Cd9 colocalized with Iba1 in the dentate gyrus. Decreased immunoreactivity to Cd9 as indicated with an arrow in molecular layer (middle) with increased Iba1 colocalization in SGZ as indicated by arrowheads (right panel). (B) Immunohistochemistry staining for Tmem119 colocalized with Iba1 in the dentate gyrus. Decreased Tmem119 immunoreactivity in SGZ (arrow in middle panel) and sparse colocalization with Iba1 in SGZ indicated by arrowheads (right panel). Scale for A and B = 80 μm.

Fig 3B (see full list in S2 Table). This further suggests that this cluster of cells correlates to microglia spatially aligned to the neurogenic niche, the SGZ of the dentate gyrus in the hippocampus. Interestingly, other microglia-specific genes such as *Hexb*, *Fclrs*, *Olfml3* retain stable expression in this cluster (Fig 3C).

Cluster 8 shows an upregulation of genes associated with lysosomal function such as *Ctsz*, *Ctsb*, and *Ctsd* (Fig 3C). It is well established that complement receptors *C3ar1* and *C5ar1*, which are typically not found in resident microglia are expressed in the SGZ, and that complement cascade pathways are necessary for normal neuronal development and synaptic pruning [48–50]. Therefore, it is unsurprising that we see them upregulated in cluster 8.

## Altered immunoreactivity in the SGZ correlates with cluster 8 gene expression

To validate whether the transcriptomic differences between cluster 8 and homeostatic clusters align with protein levels, we stained for candidate marker genes. We observed increased immunoreactivity to Cd9 in the SGZ (Fig 4A and S7 Fig). This is, however, not just localized to microglia and also expressed by neural progenitor/precursor cells [51]. Conversely, *Tmem119* immunoreactivity is decreased in the SGZ (Fig 4B). We also observed decreased immunoreactivity of Iba1+ cell processes in the granular layer of the dentate gyrus, similar to reports in the subventricular zone where this other brain neurogenic region also demonstrates lower Iba1 expression [18].

We next utilized confocal imaging to test whether neural stem/progenitor cells express Cd9 using our double reporter mice. We found little to no colocalization of GFP with Cd9 staining (S7 Fig). We note increased staining around vessels, suggesting that Cd9 is enriched in vascularized regions such as the neurogenic niche also often referred to as the neurovascular niche. Lastly, we referred to the Allen Brain Atlas in situ hybridization database to confirm spatial localization of genes for which we could not find suitable antibodies. We note high specificity of *Cd63* in the SGZ, further suggesting that this transcriptomic cluster might be specific to the neurogenic niche of the hippocampus (S6 Fig). These observations suggest that microglial cells deviating from conventionally defined homeostatic signatures may require alternative, combinatorial marker genes for identification and isolation of these cells.

## SGZ microglia display morphology and gene expression profiles that deviate from a more homeostatic phenotype

Microglial morphology and distribution are well established methods to compare activation states of immune cells [18, 44, 45]. We first directly compared cells specifically in the sub granular zone with cells in the cortex. As the cortex is the largest region of the murine brain, microglia from this region represent the highest number of any one region. While the distribution of microglia in the resting brain is characterized as tiled with microglial branches from adjacent cells maintaining distinct, non-overlapping territories, some regions of the brain do not have this patterning, particularly in regions with high neuronal densities [44, 52]. We noted deviation in SGZ microglia from this tiled distribution compared to cortical microglia (Fig 5A and 5B) in the homeostatic brain. To characterize morphometric traits, we utilized Sholl analysis to compare ramification of myeloid cells derived from the cortex versus the sub granular zone of the hippocampus. We find that cells with their cell bodies located in the sub granular zone are less ramified than cells in the cortex (Fig 5C), which is consistent with a phagocytic phenotype [53, 54]. These results are in accordance with transcriptome analyses that suggest that subsets of microglia within the hippocampus display an alternative phenotype [28].

We next asked whether microglia in this population express genes correlating with immune reactivity and activation. We again filtered the list of gene ontology terms from S2 Table, but specifically for terms that show increased metabolic activity and immune processes (Fig 5D). These are typically associated with microglial reactivity. We closely examined whether genes upregulated in the cluster 8 overlapped with known activation profiles. We compared its transcriptome profile to that of Disease Associated Microglia (DAM) described by Keren-Shaul and others first by plotting genes shown to be differentially expressed in cells from diseased brains [8]. When plotting for these genes in our homeostatic clusters 12 and 13 along with our putative SGZ cluster 8, we find that these genes seem to follow a similar pattern of expression as DAM (S8A Fig). From the Keren-Shaul dataset, we integrated transcriptome data from microglia, excluding any potential non-microglial cells and clusters 8, 12, and 13 of our

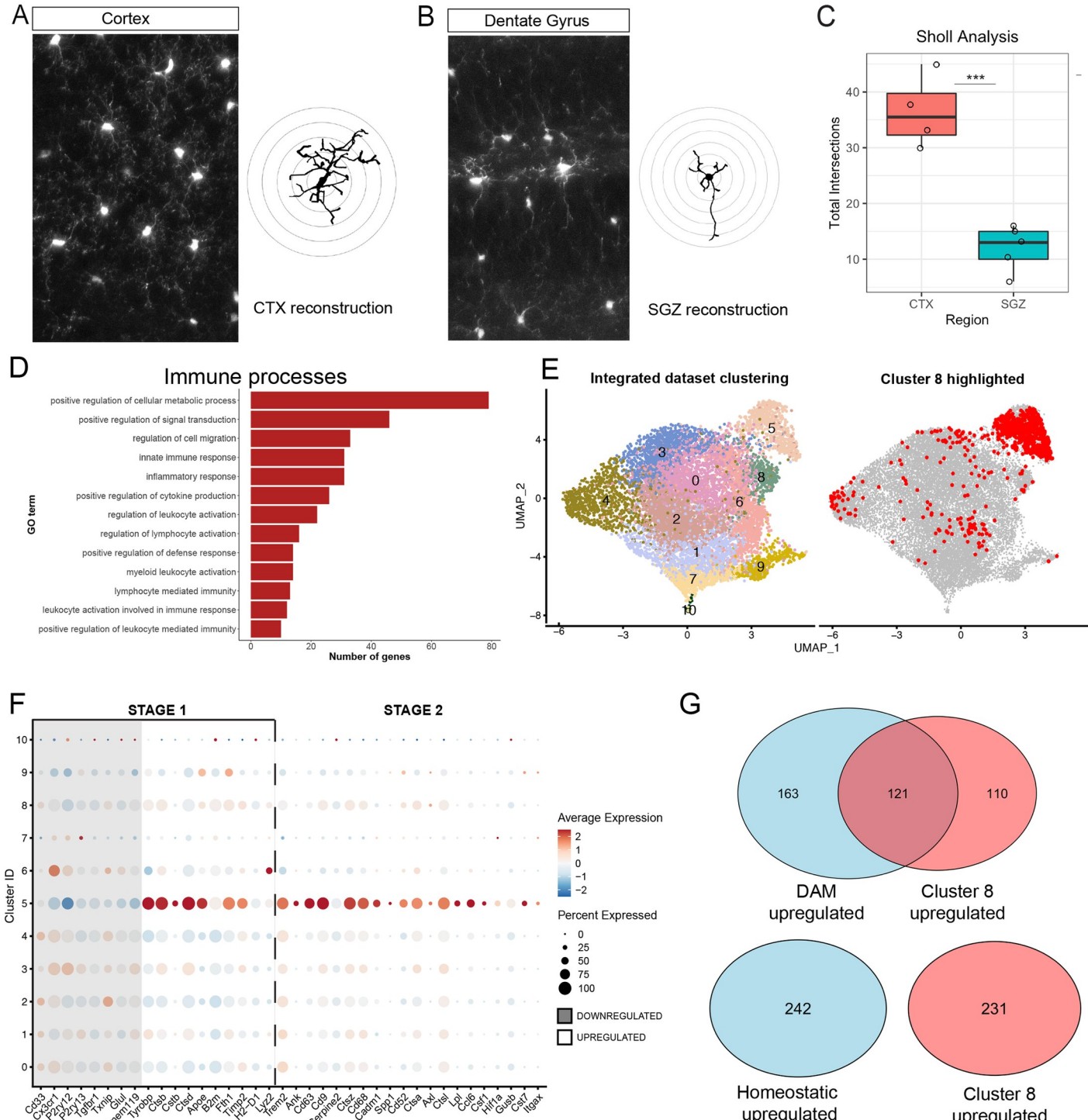

**Fig 5. Subgranular zone (SGZ) microglia display a reactive phenotype.** Representative projections of 3D z-stacked images of myeloid lineage cells labelled with tamoxifen induced tdTomato ($Cx3cr1^{CreErt2/+}$; $Rosa26^{loxp-tdTomato/+}$) in the cortex, CTX (A) and dentate gyrus (DG), with representative manual tracing of processes (right in A and B, respectively). (C) Sholl analysis to display morphometric differences between cell ramification of cells from cortex vs SGZ. p-val = 0.000227. (D) Gene set enrichment analysis of upregulated genes in cluster 4 showing select ontology related to immune activation. (E) UMAP plots showing integration of hippocampal myeloid dataset from this study (left) [8] with cells from cluster 8 in the hippocampal myeloid cells highlighted in red (right). (F) Dot Plot of key genes in integrated dataset (see E) known to be down regulated or upregulated in Stage-1 TREM2-independent activation versus Step-2 Trem2-dependent activation in DAM microglia. Shading corresponds to genes known to be downregulated in DAM profile (G) Venn diagram illustrating overlap of upregulated genes in Cluster 8 with Disease Associated Microglia (DAM) microglia (top) and overlap of upregulated genes in Cluster 8 with homeostatic microglia from [8] (bottom).

dataset. Cells from cluster 8 predominantly fell in cluster 5 of this integrated dataset (Fig 5E and S8 Fig) [8]. While the cluster identities for individual cells from the Keren-Shaul dataset were not included in the publicly available dataframes, we identified cells showing the DAM state by plotting key genes previously identified in their study (Fig 5F). We found the pattern of downregulated homeostatic genes and upregulated genes associated with reactive microglia in cluster 5 of the integrated dataset.

Additionally, we compared the list of all upregulated genes between cells labelled as DAM in the Keren-Shaul dataset compared to homeostatic cells in their dataset with the list of top upregulated genes by log fold change in cluster 8 relative to homeostatic clusters 12 and 13 (Table 2) in our study and found that over half the genes enriched in cluster 8 are also upregulated in DAM (Fig 5G). By contrast, we found no overlap between genes enriched in homeostatic cells from the Keren-Shaul dataset with those upregulated in cluster 8 (Fig 5G). As reactive microglia are not only found in disease, but also in development, we compared the whole list of upregulated genes in Table 2 with genes reported to be found in early postnatal microglia by Hammond and others (labeled cluster 3 in their dataset). We observed over a hundred genes enriched in both Cluster 8 and with many of these genes overlapping the DAM upregulated genes (S8 Fig) [15]. We investigated the expression patterns on the integrated dataset of genes overlapping in all three datasets and plotted three which were more specific to the reactive cluster 5 in the integrated dataset (S8 Fig). Therefore, cluster 8's gene expression profiles are consistent with other phenotypic states found in development and disease and appear to be distinct from most microglia in the adult brain which are typically not presented with a phagocytic challenge.

## Discussion

As the involvement of immune cells in CNS development and health are examined in greater depth, the role of innate immune cells reveals diverse function depending on cell type and context [20, 39, 55]. Different cell types are tuned to recognize and respond to specific cues [11, 56, 57]. Some populations present in the brain express different sets of genes which may explain mechanisms involving aberrant immune activation and inflammation leading to neurodegenerative diseases coupled with cognitive decline [8, 58–60]. Immune cells are undoubtedly an important layer of defense for the central nervous system, while dysregulation of the neuroimmune axis can disrupt healthy neural and cognitive functions [61–64]. This makes characterizing the different players in the immune system key to uncovering pathological development and recovery in neurological diseases [21, 65, 66]. By sampling a large number of hippocampal cells, we provide evidence of heterogeneity in the adult hippocampus, which may help explain specialized roles for microglia in mitigating disease within discrete brain areas such as the hippocampal neurogenic niche.

We have uncovered a novel context in which immune cell heterogeneity potentially explains a specialized function. The subgranular zone of the hippocampus is a select site in the adult mammalian brain where neuronal development persists through adulthood. Thus, we have identified a subset of cells in the hippocampus with a distinct transcriptomic signature correlating to function within this niche. This population of cells comprises less than five percent of total immune cells in the hippocampus, highlighting why it may have been missed in studies that examine a smaller number of cells in the hippocampus.

Previous studies suggest that immune dynamics alter the neurogenic niche and immune input in this region appears highly specialized [16, 17, 28]. In the SGZ, cells labelled with tdTomato (resulting from fractalkine Cre recombinase reporter) show stark morphological differences when compared to cells from non-neurogenic regions. Similar to microglia in the adult

**Table 2. List of upregulated genes.**

| | p_val | avg_log2FC | pct.1 | pct.2 | p_val_adj |
|---|---|---|---|---|---|
| **Ctsz** | 0 | 1.21316411097896 | 1 | 0.991 | 0 |
| **Cd9** | 0 | 1.5543421129946 | 1 | 0.974 | 0 |
| **Ftl1** | 0 | 1.29233968681864 | 1 | 0.988 | 0 |
| **Cd63** | 0 | 2.4706052666907 | 0.982 | 0.585 | 0 |
| **Lgals3** | 0 | 0.829277774804711 | 0.154 | 0.001 | 0 |
| **Ctsb** | 0 | 1.58007383569573 | 1 | 0.977 | 0 |
| **Cd83** | 0 | 1.89489310308902 | 0.739 | 0.216 | 0 |
| **Ccl4** | 1.34568412142678E-281 | 2.57797468165877 | 0.264 | 0.02 | 4.17875290226658E-277 |
| **Ctsd** | 4.97895240219732E-281 | 0.845592630366848 | 1 | 1 | 1.54611408945433E-276 |
| **Gas2l3** | 4.54139318224689E-269 | 0.516476416191356 | 0.158 | 0.004 | 1.41023882488313E-264 |
| **P2ry12** | 1.10192603799569E-265 | -1.00623764254092 | 0.985 | 1 | 3.42181092578801E-261 |
| **Lpl** | 7.76064173468084E-264 | 1.61971828850796 | 0.389 | 0.057 | 2.40991207787044E-259 |
| **Ccl3** | 3.65487488733058E-260 | 1.92695749766185 | 0.275 | 0.025 | 1.13494829876276E-255 |
| **Itgax** | 1.32483942819687E-250 | 0.442361492759136 | 0.149 | 0.004 | 4.11402387637974E-246 |
| **Selplg** | 6.06512079659385E-249 | -0.911138664302458 | 0.991 | 1 | 1.88340196096629E-244 |
| **Csf1** | 3.2412266656412E-238 | 0.877158522936439 | 0.274 | 0.028 | 1.00649811648156E-233 |
| **Cd74** | 7.90449700843159E-210 | 1.0997831043934 | 0.31 | 0.043 | 2.45458345602826E-205 |
| **Cst7** | 1.68324179253974E-195 | 0.555691686979492 | 0.185 | 0.014 | 5.22697073837367E-191 |
| **Plaur** | 1.42548459170699E-189 | 0.630800536987574 | 0.214 | 0.021 | 4.42655730262773E-185 |
| **Tmem119** | 1.99055499282937E-183 | -0.847447500045601 | 0.954 | 0.999 | 6.18127041923303E-179 |
| **Ank** | 4.76231172722668E-172 | 0.923590018242182 | 0.349 | 0.068 | 1.4788406606557E-167 |
| **Fam20c** | 9.09780361712421E-169 | 0.393238869351446 | 0.115 | 0.004 | 2.82514095722558E-164 |
| **Cd68** | 4.11575467235566E-156 | 0.871154160257722 | 0.992 | 0.918 | 1.2780652984066E-151 |
| **Mif** | 4.91689937546175E-156 | 1.12750523875927 | 0.653 | 0.268 | 1.52684476306214E-151 |
| **C3ar1** | 7.28773442939688E-156 | 1.18564814151043 | 0.806 | 0.422 | 2.26306017236061E-151 |
| **Cadm1** | 4.3280606929276E-149 | 1.14776033150113 | 0.72 | 0.336 | 1.34399268697481E-144 |
| **Serinc3** | 1.1313269549074E-145 | -0.568979451404833 | 0.996 | 1 | 3.51310959307396E-141 |
| **Chst2** | 1.76392884399378E-142 | 0.377086774950856 | 0.103 | 0.005 | 5.47752823925388E-138 |
| **Lyz2** | 2.80210208313845E-142 | 1.35961382873573 | 0.654 | 0.279 | 8.70136759876981E-138 |
| **Cstb** | 3.07814858153347E-141 | 1.05254537321382 | 0.557 | 0.198 | 9.55857479023588E-137 |
| **Mt1** | 8.77481426751793E-140 | 1.07344863386028 | 0.753 | 0.365 | 2.72484307449234E-135 |
| **Nes** | 3.42213777542466E-138 | 0.431832942380394 | 0.11 | 0.006 | 1.06267644340262E-133 |
| **Cyba** | 3.06812357639266E-128 | 0.643865546819513 | 0.996 | 0.983 | 9.52744414177214E-124 |
| **Fth1** | 4.90300782311598E-123 | 0.764439179435085 | 1 | 0.994 | 1.52253101931221E-118 |
| **Apoe** | 1.30313503913993E-119 | 2.29920561902052 | 0.754 | 0.449 | 4.04662523704122E-115 |
| **Siglech** | 8.83413706086301E-118 | -0.646255195550367 | 0.96 | 0.993 | 2.74326458150979E-113 |
| **Fgr** | 8.28797040183072E-110 | 0.332736968583023 | 0.115 | 0.01 | 2.57366344888049E-105 |
| **Slc2a5** | 9.40910967879824E-107 | -0.93581431699063 | 0.529 | 0.791 | 2.92181082855722E-102 |
| **Rhob** | 3.39067862081181E-101 | -0.608529496680619 | 0.964 | 0.992 | 1.05290743212069E-96 |
| **Pnp** | 5.68251320897039E-99 | -0.78037994727031 | 0.744 | 0.884 | 1.76459082678157E-94 |
| **P2ry13** | 8.86445785140815E-98 | -0.665371924270032 | 0.895 | 0.964 | 2.75268009659777E-93 |
| **Mfsd12** | 1.25107066793727E-96 | 0.647176621003028 | 0.297 | 0.079 | 3.88494974514561E-92 |
| **Sulf2** | 7.36585947817111E-96 | 0.322789125657732 | 0.149 | 0.021 | 2.28732034375648E-91 |
| **Rps2** | 1.65536181775218E-95 | 0.613476248060951 | 0.975 | 0.917 | 5.14039505266586E-91 |
| **Npc2** | 2.41479374849408E-95 | 0.595713731507814 | 0.973 | 0.891 | 7.49865902719867E-91 |
| **Ctsa** | 7.90944747065718E-95 | 0.642309559039543 | 0.981 | 0.892 | 2.45612072306318E-90 |
| **Hsp90ab1** | 1.10526330234185E-94 | 0.768389306887071 | 0.937 | 0.825 | 3.43217413276214E-90 |

*(Continued)*

**Table 2.** (Continued)

| | p_val | avg_log2FC | pct.1 | pct.2 | p_val_adj |
|---|---|---|---|---|---|
| Zfhx3 | 6.90955450599031E-94 | -0.730581763236332 | 0.801 | 0.929 | 2.14562396074517E-89 |
| Pld3 | 4.25612980120331E-90 | 0.799466292950042 | 0.609 | 0.296 | 1.32165598716766E-85 |
| Ivns1abp | 5.42104342841098E-88 | -0.77171959722175 | 0.703 | 0.846 | 1.68339661582446E-83 |
| Eif4a1 | 8.31472515642672E-88 | 0.862075934098564 | 0.916 | 0.752 | 2.58197160282519E-83 |
| Atf3 | 1.09177659056209E-86 | 0.985888098965569 | 0.225 | 0.053 | 3.39029384667244E-82 |
| Prdx1 | 1.92081956743974E-86 | 0.759903519869938 | 0.802 | 0.53 | 5.96472100277062E-82 |
| Vsir | 2.18414905903913E-84 | -0.527805954030406 | 0.95 | 0.983 | 6.78243807303421E-80 |
| Cx3cr1 | 2.98478897474066E-82 | -0.468087065801117 | 0.994 | 0.998 | 9.26866520326218E-78 |
| Nab2 | 4.59372768530848E-82 | 0.586682880198139 | 0.278 | 0.078 | 1.42649025811884E-77 |
| Adrb2 | 1.48935004990287E-80 | -0.832061987359219 | 0.476 | 0.719 | 4.6248787099634E-76 |
| Grn | 2.09040166041137E-79 | 0.527480248748557 | 0.991 | 0.964 | 6.49132427607543E-75 |
| Aldoa | 2.78271375563998E-77 | 0.757008365792895 | 0.799 | 0.529 | 8.64116102538882E-73 |
| Gapdh | 6.30362869296444E-77 | 0.687114693582737 | 0.942 | 0.818 | 1.95746581802625E-72 |
| Susd3 | 8.92059004040641E-76 | -0.715643358436196 | 0.644 | 0.814 | 2.7701108252474E-71 |
| Dnase2a | 5.9995018301071E-75 | 0.736974353683346 | 0.745 | 0.46 | 1.86302530330316E-70 |
| Ltc4s | 1.38542069972164E-74 | -0.638259235072411 | 0.89 | 0.955 | 4.30214689884561E-70 |
| Calr | 5.07915517047771E-74 | 0.592962075661038 | 0.982 | 0.92 | 1.57723005508844E-69 |
| Lamp1 | 8.80046227934479E-74 | 0.479896362375576 | 0.989 | 0.965 | 2.73280755160494E-69 |
| Commd8 | 4.74577308215989E-73 | -0.746453712174256 | 0.567 | 0.757 | 1.47370491520311E-68 |
| Rpsa | 3.15030216262623E-72 | 0.542170257938339 | 0.967 | 0.906 | 9.78263330560322E-68 |
| Csf1r | 3.31940341793199E-72 | -0.332214413260663 | 1 | 1 | 1.03077434337042E-67 |
| Tyrobp | 6.6543263906078E-72 | 0.36972771346155 | 1 | 0.999 | 2.06636797407544E-67 |
| Cebpd | 7.01898109246718E-72 | -0.886805201414117 | 0.416 | 0.662 | 2.17960419864383E-67 |
| C5ar1 | 9.83244862997868E-72 | 0.982565348891724 | 0.558 | 0.288 | 3.05327027306728E-67 |
| Creg1 | 1.19134604088356E-71 | 0.585336993870734 | 0.919 | 0.763 | 3.69948686075571E-67 |
| Sdf2l1 | 3.86056173202341E-71 | 0.881063114436453 | 0.765 | 0.522 | 1.19882023464523E-66 |
| Ifngr1 | 1.20140591499939E-70 | -0.465331691139475 | 0.968 | 0.986 | 3.7307257878476E-66 |
| Ptgs1 | 1.41407948365729E-70 | -0.577933036754695 | 0.84 | 0.933 | 4.391141020601E-66 |
| Elmo1 | 3.56775241952291E-70 | -0.742191907788343 | 0.62 | 0.782 | 1.10789415883445E-65 |
| Slc25a5 | 4.23383739494567E-70 | 0.634582031922411 | 0.902 | 0.745 | 1.31473352625248E-65 |
| Srgap2 | 4.34188939906534E-70 | -0.616983535997769 | 0.796 | 0.906 | 1.34828691509176E-65 |
| Fxyd5 | 1.2867385583887E-69 | 0.346519496333807 | 0.132 | 0.023 | 3.99570924536443E-65 |
| Trem2 | 5.33445604855432E-69 | 0.355647681278468 | 1 | 0.999 | 1.65650863675757E-64 |
| Hmox1 | 6.44133916650025E-68 | 0.614063258717888 | 0.302 | 0.102 | 2.00022905137332E-63 |
| Txnip | 6.95951761726012E-68 | -0.638966702848192 | 0.759 | 0.872 | 2.16113900568778E-63 |
| Maf | 1.43241040877241E-67 | -0.599860174585344 | 0.869 | 0.937 | 4.44806404236096E-63 |
| Got1 | 4.88581578537014E-67 | 0.50288421121787 | 0.346 | 0.125 | 1.51719237583099E-62 |
| S1pr1 | 1.11291697245524E-66 | 0.488531208232475 | 0.264 | 0.082 | 3.45594107456526E-62 |
| Lpcat2 | 3.46520723112635E-66 | -0.457011622282256 | 0.967 | 0.986 | 1.07605080148166E-61 |
| Gpr65 | 4.92654181042412E-65 | 0.349856028660476 | 0.136 | 0.026 | 1.529839028391E-60 |
| Apbb2 | 5.50700881264941E-65 | 0.58933610246463 | 0.387 | 0.156 | 1.71009144659202E-60 |
| Nme1 | 1.03319222280149E-64 | 0.617959107016174 | 0.499 | 0.236 | 3.20837180946548E-60 |
| Cacna1a | 2.35393917026855E-64 | 0.466613332157352 | 0.204 | 0.054 | 7.30968730543494E-60 |
| Glipr1 | 2.47564748560677E-64 | 0.402925315282 | 0.225 | 0.063 | 7.6876281370547E-60 |
| Pkm | 2.84282022129259E-64 | 0.737191820182225 | 0.691 | 0.429 | 8.82780963317989E-60 |
| Cd164 | 6.58332821620466E-64 | -0.674086540911718 | 0.664 | 0.816 | 2.04432091097803E-59 |
| Sdc4 | 1.16134169168885E-63 | 0.527084310727749 | 0.209 | 0.058 | 3.6063143552014E-59 |

(*Continued*)

**Table 2.** (Continued)

| | p_val | avg_log2FC | pct.1 | pct.2 | p_val_adj |
|---|---|---|---|---|---|
| **Cmtm6** | 7.25774044112845E-63 | -0.588213829776584 | 0.753 | 0.861 | 2.25374613918362E-58 |
| **Aplp2** | 1.54024166309499E-61 | 0.676463426508205 | 0.46 | 0.211 | 4.78291243640886E-57 |
| **Arhgap5** | 1.6330383964775E-61 | -0.58341651663721 | 0.827 | 0.91 | 5.07107413258157E-57 |
| **Rnase4** | 1.7342655506065E-61 | -0.512249790390645 | 0.904 | 0.963 | 5.38541481429837E-57 |
| **Pdgfa** | 2.93804991339461E-61 | 0.775302793475312 | 0.365 | 0.149 | 9.12352639606428E-57 |
| **Plin2** | 3.39733878887266E-61 | 0.413205438391448 | 0.251 | 0.078 | 1.05497561410863E-56 |
| **Hspa5** | 6.60104334289085E-61 | 0.65459591021307 | 0.947 | 0.858 | 2.0498219892679E-56 |
| **Nrip1** | 1.74792200519455E-60 | -0.612028423098356 | 0.744 | 0.869 | 5.42782220273063E-56 |
| **Ctsl** | 2.10848833230955E-60 | 0.460529293443523 | 0.998 | 0.986 | 6.54748881832084E-56 |
| **Psap** | 5.29767740928728E-60 | 0.445380962728947 | 1 | 0.993 | 1.64508776590598E-55 |
| **Scd2** | 1.81891231399637E-59 | 0.482773140626862 | 0.336 | 0.127 | 5.64826840865291E-55 |
| **Nceh1** | 4.23303886520062E-59 | 0.424251858399195 | 0.181 | 0.047 | 1.31448555881075E-54 |
| **Tpi1** | 6.69427759448965E-59 | 0.663206734801895 | 0.487 | 0.239 | 2.07877402141687E-54 |
| **mt-Atp6** | 8.28514597497733E-59 | 0.460948833826779 | 0.998 | 0.998 | 2.57278637960971E-54 |
| **Dpp7** | 1.98350277475901E-58 | 0.424468311114338 | 0.229 | 0.07 | 6.15937116645916E-54 |
| **Ssh2** | 2.48741175253527E-58 | -0.64276415770462 | 0.674 | 0.816 | 7.72415971514778E-54 |
| **Frmd4a** | 7.00243162399543E-58 | -0.538165079495026 | 0.827 | 0.906 | 2.1744650921993E-53 |
| **Pmepa1** | 8.07063542729683E-58 | -0.567143331483457 | 0.892 | 0.956 | 2.50617441923849E-53 |
| **Cotl1** | 1.40026649014461E-57 | 0.553502045934712 | 0.897 | 0.741 | 4.34824753184605E-53 |
| **Capg** | 3.01218792921459E-57 | 0.53396852890683 | 0.29 | 0.105 | 9.35374717659008E-53 |
| **Rhoc** | 3.31928202253687E-57 | 0.58494057735236 | 0.374 | 0.157 | 1.03073664645837E-52 |
| **St3gal6** | 9.06384537425771E-57 | -0.581099272195043 | 0.697 | 0.827 | 2.81459590406825E-52 |
| **Calm2** | 1.98642331542267E-56 | -0.457691301788096 | 0.93 | 0.96 | 6.16844032138201E-52 |
| **Adgrg1** | 2.54011540340229E-56 | -0.631518305206502 | 0.662 | 0.782 | 7.88782036218513E-52 |
| **Anxa5** | 4.79872108327651E-56 | 0.389616253498145 | 0.201 | 0.058 | 1.49014685798986E-51 |
| **Cmtm7** | 4.98883593427602E-56 | -0.586094685274199 | 0.691 | 0.822 | 1.54918322267073E-51 |
| **Uap1l1** | 7.24881444698784E-56 | 0.474409326235679 | 0.32 | 0.122 | 2.25097435022314E-51 |
| **Hif1a** | 5.03204379833433E-55 | 0.653191597752922 | 0.512 | 0.265 | 1.56260056069676E-50 |
| **mt-Nd1** | 1.27608557526488E-53 | 0.520390796544433 | 0.979 | 0.974 | 3.96262853687004E-49 |
| **Gnl3** | 1.82717015796429E-53 | 0.616290373611735 | 0.426 | 0.2 | 5.6739114915265E-49 |
| **Ssr4** | 7.53897622443318E-53 | 0.45670983091204 | 0.939 | 0.814 | 2.34107828697324E-48 |
| **Gaa** | 1.17172865138326E-52 | 0.481994161871853 | 0.427 | 0.19 | 3.63856898114044E-48 |
| **Rgs10** | 1.1744417795469E-52 | -0.363248010659844 | 0.981 | 0.992 | 3.64699405802699E-48 |
| **Ctsc** | 3.14688415535714E-52 | 0.528662496677226 | 0.91 | 0.768 | 9.77201936763054E-48 |
| **Hspa8** | 3.80828837400191E-52 | 0.514928302750921 | 0.951 | 0.872 | 1.18258778877881E-47 |
| **Cox6a2** | 1.31166642786174E-51 | 0.270088241869821 | 0.116 | 0.023 | 4.07311775843906E-47 |
| **Syngr1** | 4.07815857552635E-51 | 0.495894306710773 | 0.932 | 0.819 | 1.2663905824582E-46 |
| **Id2** | 5.13004735203389E-51 | 0.73459631434953 | 0.273 | 0.105 | 1.59303360422708E-46 |
| **Bcl2a1b** | 7.34863786997375E-51 | 0.694498438592724 | 0.785 | 0.589 | 2.28197251776295E-46 |
| **Timp2** | 1.30496723397884E-50 | 0.474988225831741 | 0.93 | 0.811 | 4.0523147516745E-46 |
| **Rpl10a** | 1.69210900399288E-50 | 0.424073020099445 | 0.972 | 0.877 | 5.2545060900991E-46 |
| **Dock10** | 2.32679252778372E-50 | -0.6479276023627 | 0.544 | 0.695 | 7.22538883652677E-46 |
| **Renbp** | 7.94640048410583E-50 | 0.532906728251976 | 0.454 | 0.223 | 2.46759574232938E-45 |
| **Cxcl16** | 9.43230042376968E-50 | 0.602180017993279 | 0.373 | 0.168 | 2.9290122505932E-45 |
| **Mef2a** | 5.20458524572297E-49 | -0.508319662100362 | 0.795 | 0.9 | 1.61617985635435E-44 |
| **Slamf8** | 7.7520986511249E-49 | 0.401094433918377 | 0.186 | 0.056 | 2.40725919413382E-44 |
| **Glul** | 9.0328119522076E-49 | -0.513087417236526 | 0.855 | 0.903 | 2.80495909551903E-44 |

*(Continued)*

**Table 2.** (Continued)

| | p_val | avg_log2FC | pct.1 | pct.2 | p_val_adj |
|---|---|---|---|---|---|
| F11r | 9.23182938042976E-49 | -0.441225560157002 | 0.918 | 0.955 | 2.86675997750485E-44 |
| Asph | 9.56049840849747E-49 | 0.557115143524005 | 0.83 | 0.667 | 2.96882157079072E-44 |
| Il10ra | 9.8315954159412E-49 | -0.611479440775794 | 0.529 | 0.699 | 3.05300532451222E-44 |
| Nav3 | 7.56176135331455E-48 | -0.659496669639548 | 0.516 | 0.687 | 2.34815375304477E-43 |
| H2-D1 | 2.56348688205036E-47 | 0.578557305024221 | 0.869 | 0.687 | 7.96039581483099E-43 |
| Adssl1 | 3.53597991589356E-47 | 0.306382037827989 | 0.179 | 0.053 | 1.09802784328243E-42 |
| Il4i1 | 4.20175651273496E-47 | 0.466290645043656 | 0.192 | 0.061 | 1.30477144989959E-42 |
| Pdia6 | 1.39516487264208E-46 | 0.489157209247046 | 0.932 | 0.813 | 4.33240547901544E-42 |
| Lrba | 1.41998096714974E-46 | -0.639531221241718 | 0.48 | 0.652 | 4.4094668972901E-42 |
| Fam102b | 4.98193707083843E-46 | -0.600723477825392 | 0.584 | 0.735 | 1.54704091860746E-41 |
| Ms4a6b | 6.74592017119362E-46 | -0.675205168844468 | 0.353 | 0.556 | 2.09481059076076E-41 |
| B2m | 1.12607164109117E-45 | 0.387611579094594 | 0.987 | 0.957 | 3.49679026708041E-41 |
| Gusb | 2.20831554805777E-45 | 0.52775162670477 | 0.836 | 0.635 | 6.85748227138379E-41 |
| Pgk1 | 2.60157539088621E-45 | 0.481031829728836 | 0.381 | 0.177 | 8.07867206131895E-41 |
| Pde3b | 3.35206535493397E-45 | -0.600037422922937 | 0.599 | 0.721 | 1.04091685466764E-40 |
| mt-Co2 | 1.24274205865524E-44 | 0.396079210932013 | 0.996 | 0.996 | 3.85908691474211E-40 |
| Eif5a | 2.4170403298965E-44 | 0.531295411154843 | 0.814 | 0.608 | 7.50563533642761E-40 |
| mt-Co3 | 4.8481056297552E-44 | 0.403682409006687 | 0.995 | 0.997 | 1.50548224120788E-39 |
| H2-K1 | 1.34360142659825E-43 | 0.521306515292163 | 0.881 | 0.691 | 4.17228551001555E-39 |
| Ecscr | 1.77748247088031E-43 | -0.472796485355958 | 0.814 | 0.896 | 5.51961631682464E-39 |
| Arhgap45 | 4.7752216343876E-43 | -0.473616024242607 | 0.779 | 0.871 | 1.48284957412638E-38 |
| Mbnl1 | 7.56394416450135E-43 | -0.497500101930406 | 0.821 | 0.87 | 2.3488315814026E-38 |
| Abcg1 | 7.92852332395055E-43 | 0.482203326305026 | 0.51 | 0.275 | 2.46204434778636E-38 |
| Tmem173 | 1.15399896757026E-42 | -0.519497631944832 | 0.68 | 0.793 | 3.58351299399594E-38 |
| Scpep1 | 2.78590022785178E-42 | 0.456699152950326 | 0.368 | 0.172 | 8.65105597754812E-38 |
| Dtnbp1 | 4.9079114166475E-42 | 0.446761420062665 | 0.415 | 0.203 | 1.52405373221155E-37 |
| Plekhm2 | 6.58051470289304E-42 | 0.407477513791209 | 0.218 | 0.079 | 2.04344723068938E-37 |
| Hpgd | 8.47797549378708E-42 | -0.508153796550951 | 0.756 | 0.843 | 2.6326657300857E-37 |
| Il11ra1 | 1.52883061508586E-41 | 0.490177078291142 | 0.425 | 0.213 | 4.74747770902612E-37 |
| Srgn | 3.44371530112524E-41 | 0.575642140578165 | 0.737 | 0.534 | 1.06937691245842E-36 |
| Slco2b1 | 4.34640691226016E-41 | -0.444594167814038 | 0.854 | 0.902 | 1.34968973846415E-36 |
| Sall1 | 1.18451927478039E-40 | -0.526085879035089 | 0.592 | 0.714 | 3.67828770397553E-36 |
| Slc15a3 | 1.22714213657559E-40 | 0.746930747140177 | 0.554 | 0.341 | 3.81064447670818E-36 |
| Ccr5 | 2.13181090841299E-40 | -0.54155723509141 | 0.758 | 0.841 | 6.61991241389486E-36 |
| Ccl6 | 3.65725060136654E-40 | 0.798272506123976 | 0.637 | 0.434 | 1.13568602924235E-35 |
| Rapgef5 | 2.07759564357415E-39 | -0.62956656004199 | 0.34 | 0.526 | 6.45155775199082E-35 |
| Ckb | 2.42312692389765E-39 | -0.417576140079039 | 0.924 | 0.95 | 7.52453603677936E-35 |
| Rsrp1 | 2.42851834206525E-39 | -0.384093406316538 | 0.942 | 0.97 | 7.54127800761523E-35 |
| C1qbp | 4.49682276672034E-39 | 0.496477885564906 | 0.517 | 0.296 | 1.39639837374967E-34 |
| Ssr2 | 3.34190183013306E-38 | 0.454747008083742 | 0.72 | 0.49 | 1.03776077531122E-33 |
| Selenow | 4.34727701825501E-38 | 0.465379260277558 | 0.408 | 0.21 | 1.34995993247873E-33 |
| Lipa | 5.06489828428849E-38 | 0.511643152162503 | 0.602 | 0.373 | 1.5728028642201E-33 |
| Atp5g1 | 8.32463194321884E-38 | 0.514547877550272 | 0.655 | 0.437 | 2.58504795732775E-33 |
| Ptma | 1.03202222180045E-37 | 0.385685821422643 | 0.973 | 0.889 | 3.20473860535695E-33 |
| Arl11 | 1.1263253988904E-37 | 0.413895991508187 | 0.448 | 0.235 | 3.49757826117436E-33 |
| Hsd17b12 | 1.15723930293936E-37 | 0.44315662785853 | 0.464 | 0.248 | 3.59357520741758E-33 |
| Mettl1 | 1.27448087741679E-37 | 0.367524736031922 | 0.285 | 0.122 | 3.95764546864237E-33 |

(*Continued*)

**Table 2.** (Continued)

| | p_val | avg_log2FC | pct.1 | pct.2 | p_val_adj |
|---|---|---|---|---|---|
| **Tnfrsf12a** | 1.49404595039677E-37 | 0.391129481708753 | 0.106 | 0.026 | 4.63946088976708E-33 |
| **Manf** | 4.03636107863532E-37 | 0.51833924559785 | 0.867 | 0.697 | 1.25341120574863E-32 |
| **Alox5ap** | 4.56098489137684E-37 | -0.489946018174042 | 0.715 | 0.808 | 1.41632263831925E-32 |
| **Crybb1** | 4.65994772545172E-37 | -0.570595980047922 | 0.65 | 0.761 | 1.44705356718452E-32 |
| **Tsc22d4** | 7.61387646197082E-37 | -0.449178642918815 | 0.743 | 0.827 | 2.3643370577358E-32 |
| **Cfl1** | 8.64782056780436E-37 | 0.357689979068086 | 0.964 | 0.92 | 2.68540772092029E-32 |
| **Rpl12** | 1.43027169743147E-36 | 0.360732032917497 | 0.972 | 0.911 | 4.44142270203393E-32 |
| **Nrp2** | 1.86121375230495E-36 | 0.381401397875395 | 0.305 | 0.137 | 5.77962706503256E-32 |
| **Fchsd2** | 2.94245025300602E-36 | -0.585132766236506 | 0.55 | 0.671 | 9.13719077065958E-32 |
| **Cd52** | 4.10648684413692E-36 | 0.607683096004775 | 0.719 | 0.529 | 1.27518735970984E-31 |
| **Hsp90b1** | 5.21796424925618E-36 | 0.394919780147213 | 0.977 | 0.926 | 1.62033443832152E-31 |
| **Ybx1** | 1.10078801772538E-35 | 0.449645010630102 | 0.828 | 0.637 | 3.41827703144262E-31 |
| **Siglecf** | 1.85626910896376E-35 | 0.365474034182325 | 0.225 | 0.089 | 5.76427246406515E-31 |
| **Ppia** | 1.86196145422586E-35 | 0.298070488274917 | 0.987 | 0.975 | 5.78194890380757E-31 |
| **Tnfaip8l2** | 4.34056175515685E-35 | -0.455328357507087 | 0.689 | 0.791 | 1.34787464803946E-30 |
| **Hsp90aa1** | 4.95683494488193E-35 | 0.542542963378589 | 0.665 | 0.465 | 1.53924595543419E-30 |
| **Ncl** | 5.34720815572171E-35 | 0.555769526277748 | 0.725 | 0.529 | 1.66046854859626E-30 |
| **mt-Nd2** | 6.37796598093349E-35 | 0.409317822483824 | 0.995 | 0.983 | 1.98054977605928E-30 |
| **Tanc2** | 1.20846051211416E-34 | -0.491893007656888 | 0.702 | 0.793 | 3.75263242826812E-30 |
| **Kctd12** | 1.61289159875745E-34 | -0.442687859566028 | 0.807 | 0.873 | 5.00851228162151E-30 |
| **Cpd** | 2.47259870435313E-34 | 0.361393105831768 | 0.239 | 0.099 | 7.67816075662778E-30 |
| **Gga2** | 2.53536388068479E-34 | 0.252459505003485 | 0.143 | 0.045 | 7.87306545869048E-30 |
| **Fam49b** | 3.19507640188712E-34 | -0.396681310114844 | 0.842 | 0.887 | 9.92167075078009E-30 |
| **Npnt** | 3.55508175207553E-34 | 0.380220024707806 | 0.31 | 0.145 | 1.10395953647202E-29 |
| **Tpd52** | 3.57782112428065E-34 | 0.413351733171847 | 0.471 | 0.262 | 1.11102079372287E-29 |
| **Atp6ap2** | 4.45107453378929E-34 | 0.430346229411463 | 0.864 | 0.69 | 1.38219217497759E-29 |
| **Neat1** | 5.83218213790618E-34 | 0.418926472909667 | 0.265 | 0.118 | 1.81106751928401E-29 |
| **Qk** | 9.04642106126178E-34 | -0.351134672140777 | 0.933 | 0.959 | 2.80918513215362E-29 |
| **Gpr34** | 9.70164297950284E-34 | -0.295508692418895 | 0.994 | 0.999 | 3.01265119442502E-29 |
| **Ldha** | 1.09981846490267E-33 | 0.589677058542156 | 0.589 | 0.39 | 3.41526627906225E-29 |
| **Epb41l2** | 2.68551553006558E-33 | -0.409889683349906 | 0.909 | 0.927 | 8.33933137551264E-29 |
| **Degs1** | 2.70083169875143E-33 | 0.362194420927888 | 0.42 | 0.219 | 8.38689267413281E-29 |
| **Tmem256** | 2.83506461863131E-33 | 0.433735219835239 | 0.539 | 0.327 | 8.80372616023581E-29 |
| **Ranbp1** | 3.71815025560179E-33 | 0.434065288948232 | 0.558 | 0.334 | 1.15459719887202E-28 |
| **Numb** | 4.09206681981799E-33 | -0.492154464348105 | 0.535 | 0.674 | 1.27070950955808E-28 |
| **Glmp** | 4.76389604189922E-33 | 0.436446723942437 | 0.716 | 0.487 | 1.47933263789097E-28 |
| **Rpl41** | 7.20150170299407E-33 | 0.322278590479056 | 0.992 | 0.986 | 2.23628232383075E-28 |
| **Rplp0** | 1.80232812919758E-32 | 0.304482437819109 | 0.979 | 0.964 | 5.59676953959725E-28 |
| **Npm1** | 2.93613237905256E-32 | 0.482536597318115 | 0.807 | 0.648 | 9.11757187667191E-28 |
| **Plek** | 7.12401711659187E-32 | 0.509301833047534 | 0.614 | 0.401 | 2.21222103521527E-27 |
| **Pag1** | 2.00632874067246E-31 | -0.457569141620418 | 0.639 | 0.749 | 6.23025263841018E-27 |
| **Pgam1** | 3.05771802647204E-31 | 0.445280801826828 | 0.591 | 0.378 | 9.49513178760362E-27 |
| **Sparc** | 3.16669775512935E-31 | -0.270627876052526 | 0.999 | 1 | 9.83354653900318E-27 |
| **Ppa1** | 5.52914172778572E-31 | 0.259306316096504 | 0.164 | 0.059 | 1.7169643807293E-26 |
| **Zfp36l2** | 8.08830189219789E-31 | -0.48735866589604 | 0.703 | 0.779 | 2.51166038658421E-26 |
| **Hspd1** | 8.54323149950983E-31 | 0.464596738031814 | 0.469 | 0.276 | 2.65292967754279E-26 |
| **Oxct1** | 9.65720787703265E-31 | 0.427226950903729 | 0.67 | 0.469 | 2.99885276205495E-26 |

(*Continued*)

**Table 2.** (Continued)

| | p_val | avg_log2FC | pct.1 | pct.2 | p_val_adj |
|---|---|---|---|---|---|
| Speg | 2.02576350776271E-30 | 0.284247564160621 | 0.185 | 0.072 | 6.29060342065555E-26 |
| Git2 | 2.45457106397613E-30 | -0.477957080409749 | 0.643 | 0.734 | 7.62217952496507E-26 |
| Bin2 | 2.63336426847757E-30 | -0.411471218127436 | 0.715 | 0.799 | 8.1773860629034E-26 |
| mt-Co1 | 2.9815285100886E-30 | 0.314362725096399 | 0.998 | 0.995 | 9.25854048237814E-26 |
| Tceal9 | 3.10988034443876E-30 | 0.315215003506278 | 0.243 | 0.107 | 9.65711143358569E-26 |
| Atp6v1a | 3.58475644284532E-30 | 0.383248952984662 | 0.288 | 0.139 | 1.11317441819676E-25 |
| Rgs2 | 3.75744085442515E-30 | -0.402337867752876 | 0.802 | 0.868 | 1.16679810852464E-25 |
| Phgdh | 3.77743806387575E-30 | 0.444600846374229 | 0.588 | 0.381 | 1.17300784197534E-25 |
| Pld4 | 9.74867268662707E-30 | -0.370617885375037 | 0.895 | 0.931 | 3.0272553293783E-25 |
| Gnas | 1.18071110411391E-29 | 0.377988569498271 | 0.906 | 0.787 | 3.66646219160493E-25 |
| Wasf2 | 3.60333338977188E-29 | -0.420581383513958 | 0.71 | 0.784 | 1.11894311752586E-24 |
| Serpine2 | 6.66360639380897E-29 | 0.383617023599912 | 0.916 | 0.805 | 2.0692496934695E-24 |
| Amdhd2 | 9.50418401866414E-29 | 0.283721022231996 | 0.24 | 0.106 | 2.95133426331578E-24 |
| Cln5 | 1.25163031951808E-28 | 0.385480891055797 | 0.441 | 0.254 | 3.88668763119949E-24 |
| Sco2 | 1.2688681370891E-28 | 0.323027067247246 | 0.264 | 0.123 | 3.94021622610277E-24 |
| Ctc1 | 1.96077032129541E-28 | -0.515265712329096 | 0.406 | 0.54 | 6.08878007871863E-24 |
| Ankrd44 | 1.99210757079243E-28 | -0.481944304536418 | 0.55 | 0.655 | 6.18609163958174E-24 |
| mt-Nd4 | 2.12925484249E-28 | 0.324380571816951 | 0.986 | 0.985 | 6.6119750623842E-24 |
| Dock4 | 2.41473238350482E-28 | -0.491027036550414 | 0.577 | 0.673 | 7.49846847049752E-24 |
| Ophn1 | 3.18027531495293E-28 | -0.459503522424536 | 0.621 | 0.709 | 9.87570893552335E-24 |
| Atox1 | 3.21306135826157E-28 | 0.382989950584401 | 0.802 | 0.626 | 9.97751943580966E-24 |
| Bst2 | 3.90015542102088E-28 | 0.473950904762105 | 0.635 | 0.42 | 1.21111526288961E-23 |
| Rps19 | 4.84974100350518E-28 | 0.303320235999598 | 0.978 | 0.94 | 1.50599007381846E-23 |
| Ccl9 | 4.85264746596657E-28 | 0.781795381828554 | 0.646 | 0.49 | 1.5068926176066E-23 |
| Itgam | 4.86545775763175E-28 | -0.425248716600044 | 0.759 | 0.817 | 1.51087059747739E-23 |
| Snrpf | 5.71359125903477E-28 | 0.38402897475651 | 0.509 | 0.313 | 1.77424149366807E-23 |
| Dad1 | 6.59339415932594E-28 | 0.349207348922655 | 0.853 | 0.663 | 2.04744668829548E-23 |
| Pdia3 | 1.70792539383179E-27 | 0.360406539745383 | 0.938 | 0.837 | 5.30362072546585E-23 |
| Rpl14 | 1.86754165159592E-27 | 0.327840117468946 | 0.917 | 0.822 | 5.79927709070081E-23 |
| Cyth4 | 2.39025681170063E-27 | -0.314149804904163 | 0.94 | 0.959 | 7.42246447737397E-23 |
| Plekha1 | 2.85601278093158E-27 | 0.396901309178602 | 0.391 | 0.218 | 8.86877648862684E-23 |
| Tnfrsf13b | 2.87678049562323E-27 | -0.534342219557837 | 0.312 | 0.456 | 8.9332664730588E-23 |
| Eef1g | 3.34249016135725E-27 | 0.402534009396938 | 0.628 | 0.427 | 1.03794346980627E-22 |
| Cysltr1 | 3.8713375903377E-27 | -0.549305982203782 | 0.309 | 0.452 | 1.20216646192756E-22 |
| Nav2 | 3.97079235172207E-27 | -0.462955746775159 | 0.602 | 0.708 | 1.23305014898025E-22 |
| Fam110a | 4.23106492500756E-27 | -0.573835045335538 | 0.207 | 0.368 | 1.3138725911626E-22 |
| Abca1 | 4.26215470350909E-27 | 0.425279299180042 | 0.654 | 0.454 | 1.32352690008068E-22 |
| Slc3a2 | 4.69825171128444E-27 | 0.435281850487179 | 0.894 | 0.785 | 1.45894810390516E-22 |
| Rplp2 | 6.82383987623299E-27 | 0.259607179512908 | 0.988 | 0.96 | 2.11900699676663E-22 |
| Ppfia4 | 6.94789351207204E-27 | -0.414871911456713 | 0.621 | 0.72 | 2.15752937230373E-22 |
| Bax | 7.70025700162577E-27 | 0.402427060709431 | 0.618 | 0.402 | 2.39116080671485E-22 |
| Golm1 | 9.87441790978386E-27 | -0.36860114428039 | 0.849 | 0.883 | 3.06630299352518E-22 |
| Tpst2 | 1.01725104788008E-26 | -0.429870060579117 | 0.663 | 0.762 | 3.15886967898202E-22 |
| mt-Cytb | 1.03181450622033E-26 | 0.30454419833674 | 0.996 | 0.995 | 3.20409358616599E-22 |
| Fam212a | 1.13149238992867E-26 | -0.495590373060695 | 0.416 | 0.554 | 3.51362331844549E-22 |
| Fam105a | 1.40097553025469E-26 | -0.349219741897841 | 0.858 | 0.895 | 4.35044931409989E-22 |
| Emp3 | 1.44511045424651E-26 | 0.392396547417138 | 0.295 | 0.15 | 4.48750149357168E-22 |

(*Continued*)

**Table 2.** (Continued)

| | p_val | avg_log2FC | pct.1 | pct.2 | p_val_adj |
|---|---|---|---|---|---|
| Hspe1 | 1.78813839928288E-26 | 0.392766359842521 | 0.688 | 0.497 | 5.55270617129311E-22 |
| Zfp710 | 2.44348399038032E-26 | -0.539164156650091 | 0.373 | 0.503 | 7.58775083532802E-22 |
| Bmp2k | 2.62996673621556E-26 | -0.443222658658454 | 0.639 | 0.718 | 8.16683570597017E-22 |
| Gna15 | 2.72773098304417E-26 | -0.416083889741095 | 0.587 | 0.685 | 8.47042302164705E-22 |
| Gnai2 | 3.05770691698163E-26 | -0.298679297813492 | 0.944 | 0.957 | 9.49509728930305E-22 |
| Ypel3 | 3.4471788769902E-26 | -0.437403754258679 | 0.537 | 0.654 | 1.07045245667177E-21 |
| Cndp2 | 3.61671663232009E-26 | 0.35504280335393 | 0.481 | 0.288 | 1.12309901583436E-21 |
| Krtcap2 | 7.4034101962161E-26 | 0.386307527376267 | 0.716 | 0.545 | 2.29898096823099E-21 |
| Vegfb | 1.04203975127724E-25 | 0.34051092407815 | 0.371 | 0.204 | 3.23584603964122E-21 |
| Stmn1 | 1.19106567546169E-25 | 0.292210714096892 | 0.247 | 0.117 | 3.69861624201118E-21 |
| Slc35f6 | 1.63740645745082E-25 | 0.374017824549108 | 0.354 | 0.195 | 5.08463827232204E-21 |
| Ywhah | 2.14595764889782E-25 | -0.344055033282341 | 0.861 | 0.893 | 6.66384228712242E-21 |
| Eif1a | 2.82744485653522E-25 | 0.362899744606182 | 0.331 | 0.179 | 8.78006451299882E-21 |
| Eya4 | 3.39899818163469E-25 | 0.297263128577378 | 0.255 | 0.124 | 1.05549090534302E-20 |
| Cd48 | 4.18505561516487E-25 | 0.439721785823438 | 0.52 | 0.345 | 1.29958532017715E-20 |
| Gabarap | 4.56519320585279E-25 | 0.306175787038689 | 0.925 | 0.811 | 1.41762944621347E-20 |
| Hnrnpa1 | 4.6228163739757E-25 | 0.384980462086857 | 0.586 | 0.395 | 1.43552316861067E-20 |
| Eef2k | 6.42429967611457E-25 | -0.521510095504093 | 0.301 | 0.444 | 1.99493777842386E-20 |
| Foxn3 | 6.86271308219647E-25 | -0.40110624227457 | 0.708 | 0.763 | 2.13107829341447E-20 |
| Abca9 | 8.18064836253267E-25 | -0.517741276729229 | 0.329 | 0.461 | 2.54033673601727E-20 |
| Plekho2 | 1.11946463496997E-24 | 0.301166850724053 | 0.153 | 0.061 | 3.47627353097224E-20 |
| A830008E2 | 1.90463000073343E-24 | -0.556834154765782 | 0.226 | 0.38 | 5.91444754127752E-20 |
| Btg2 | 2.3474921785515E-24 | -0.563013359238349 | 0.535 | 0.662 | 7.28966746205596E-20 |
| Hexa | 2.53828913455285E-24 | 0.306464484576321 | 0.979 | 0.936 | 7.88214924952697E-20 |
| Arpc2 | 2.56276037140693E-24 | 0.295028510273021 | 0.952 | 0.888 | 7.95813978132994E-20 |
| Picalm | 2.83930104582616E-24 | -0.391340020597646 | 0.767 | 0.811 | 8.81688153760398E-20 |
| Csmd3 | 3.5594532771125E-24 | -0.499633798501414 | 0.475 | 0.597 | 1.10531702614174E-19 |
| Atp6v1c1 | 4.6926561120587E-24 | 0.351002030323345 | 0.475 | 0.288 | 1.45721050247759E-19 |
| Spcs2 | 4.88697108308929E-24 | 0.319237328977595 | 0.91 | 0.824 | 1.51755113043172E-19 |
| Tmem176b | 7.17504520039447E-24 | -0.426396027903934 | 0.767 | 0.826 | 2.22806678607849E-19 |
| Tgfbr1 | 8.39155054450364E-24 | -0.263542279712594 | 0.987 | 0.987 | 2.60582819058472E-19 |
| Pmp22 | 9.11727216729362E-24 | 0.547298658890186 | 0.862 | 0.774 | 2.83118652610969E-19 |
| Col27a1 | 1.03616588823734E-23 | -0.497799037849485 | 0.368 | 0.497 | 3.21760593274341E-19 |
| Pten | 1.0643907370072E-23 | -0.476323460985719 | 0.457 | 0.558 | 3.30525255562847E-19 |
| Cox4i1 | 1.47827322268289E-23 | 0.2975999369363 | 0.882 | 0.756 | 4.59048183839717E-19 |
| Tsc22d3 | 1.90309966487181E-23 | -0.570077354715977 | 0.219 | 0.367 | 5.90969538932642E-19 |
| Bmyc | 2.18203247721525E-23 | -0.412985900576483 | 0.535 | 0.633 | 6.77586545149652E-19 |
| Chd9 | 2.22837449509304E-23 | -0.479345180230731 | 0.448 | 0.556 | 6.91977131961243E-19 |
| Rps26 | 2.96012696947443E-23 | 0.294476307294439 | 0.946 | 0.872 | 9.19208227830894E-19 |
| Snn | 3.76279383322177E-23 | -0.486285080087485 | 0.37 | 0.486 | 1.16846036903036E-18 |
| Prkca | 3.99551096596017E-23 | -0.50301747851366 | 0.2 | 0.344 | 1.24072602025961E-18 |
| Mrps18b | 4.54047365570421E-23 | 0.268032111545704 | 0.303 | 0.159 | 1.40995328430583E-18 |
| Dip2b | 7.16520090346016E-23 | -0.457176208802776 | 0.43 | 0.547 | 2.22500983655148E-18 |
| Cct3 | 7.18133986139772E-23 | 0.347759107295508 | 0.49 | 0.309 | 2.23002146715984E-18 |
| Clk1 | 9.59466578713302E-23 | -0.424915532010236 | 0.571 | 0.65 | 2.97943156687842E-18 |
| Mef2c | 1.19426506975782E-22 | -0.264656789705501 | 0.968 | 0.986 | 3.70855132111897E-18 |
| Celf2 | 1.22740575214403E-22 | -0.386544231064657 | 0.727 | 0.788 | 3.81146308213287E-18 |

(*Continued*)

**Table 2.** (Continued)

| | p_val | avg_log2FC | pct.1 | pct.2 | p_val_adj |
|---|---|---|---|---|---|
| **Abhd12** | 1.25350096498218E-22 | 0.279705391607691 | 0.986 | 0.956 | 3.89249654655916E-18 |
| **Pebp1** | 1.28759235511961E-22 | 0.346322736777086 | 0.687 | 0.503 | 3.99836054035293E-18 |
| **Pkig** | 1.52134364274231E-22 | -0.437935702103922 | 0.463 | 0.563 | 4.7242284138077E-18 |
| **mt-Nd3** | 1.52389567932705E-22 | 0.360814831157491 | 0.924 | 0.868 | 4.73215325301429E-18 |
| **Slc16a3** | 1.65809139698158E-22 | 0.361830850603775 | 0.316 | 0.176 | 5.14887121504691E-18 |
| **G3bp1** | 1.91696409836498E-22 | 0.380174457719334 | 0.612 | 0.424 | 5.95274861465277E-18 |
| **Tm6sf1** | 2.22506791082021E-22 | -0.392208837137046 | 0.64 | 0.725 | 6.90950338346998E-18 |
| **Fcgr1** | 2.40226518345125E-22 | -0.344414161211829 | 0.804 | 0.851 | 7.45975407417118E-18 |
| **Ptpn18** | 2.94117241161809E-22 | -0.359820107776922 | 0.736 | 0.8 | 9.13322268979767E-18 |
| **mt-Nd4l** | 3.00643416440556E-22 | 0.437525676271991 | 0.689 | 0.534 | 9.33588001072857E-18 |
| **Hpf1** | 3.2908545829426E-22 | 0.325405936242127 | 0.454 | 0.278 | 1.02190907364117E-17 |
| **Lpar6** | 3.68266539289908E-22 | -0.441485669117467 | 0.579 | 0.652 | 1.14357808445695E-17 |
| **Pea15a** | 3.76050014348994E-22 | 0.297686993077065 | 0.373 | 0.214 | 1.16774810955793E-17 |
| **Mrto4** | 4.00659697083673E-22 | 0.297995208709106 | 0.277 | 0.145 | 1.24416855735393E-17 |
| **Srrm2** | 4.22106077435486E-22 | -0.348102490685013 | 0.834 | 0.847 | 1.31076600226041E-17 |
| **Ran** | 4.45176694180117E-22 | 0.368010010709252 | 0.661 | 0.474 | 1.38240718843752E-17 |
| **Tmem160** | 6.69363595911118E-22 | 0.331029688957387 | 0.473 | 0.297 | 2.07857477438279E-17 |
| **Acadl** | 7.84562625307841E-22 | 0.283939709052962 | 0.292 | 0.156 | 2.43630232036844E-17 |
| **Myl6** | 8.60720705007188E-22 | 0.324543917095488 | 0.8 | 0.643 | 2.67279600525882E-17 |
| **Ntpcr** | 1.27075013162379E-21 | -0.377354672588958 | 0.575 | 0.659 | 3.94606038373135E-17 |
| **Slc35b1** | 1.28227950523366E-21 | 0.324301233020395 | 0.409 | 0.244 | 3.98186254760207E-17 |
| **Cd33** | 1.37174895458831E-21 | -0.397263680216192 | 0.706 | 0.755 | 4.25969202868309E-17 |
| **Atp1b3** | 1.39030721546009E-21 | 0.36416752758522 | 0.639 | 0.447 | 4.31732099616823E-17 |
| **Mydgf** | 1.84362317078189E-21 | 0.35166869377745 | 0.629 | 0.447 | 5.72500303222902E-17 |
| **Tns1** | 1.94664366990437E-21 | 0.278223492035178 | 0.29 | 0.155 | 6.04491258815404E-17 |
| **Cct8** | 2.14096922120587E-21 | 0.322171857740395 | 0.572 | 0.377 | 6.64835172261059E-17 |
| **Erp29** | 2.2439166123147E-21 | 0.265004396150513 | 0.979 | 0.935 | 6.96803425622084E-17 |
| **Akr1a1** | 2.28774416967525E-21 | 0.327941413048129 | 0.757 | 0.585 | 7.10413197009256E-17 |
| **Dnajb11** | 2.84771560588368E-21 | 0.352570790141271 | 0.684 | 0.495 | 8.8430112709506E-17 |
| **Canx** | 2.98045704610599E-21 | 0.304679454238274 | 0.874 | 0.757 | 9.25521326527292E-17 |
| **Ddx5** | 3.07696546690176E-21 | -0.262798198324156 | 0.967 | 0.973 | 9.55490086437002E-17 |
| **Hmgn2** | 3.68501113234346E-21 | 0.286467187348032 | 0.399 | 0.238 | 1.14430650692661E-16 |
| **Bola2** | 4.21623716577286E-21 | 0.31379305469731 | 0.432 | 0.263 | 1.30926812708745E-16 |
| **Arsb** | 4.39635672434192E-21 | -0.355232794149626 | 0.796 | 0.815 | 1.3652006536099E-16 |
| **Slc6a6** | 4.52593770082542E-21 | 0.335763114024004 | 0.602 | 0.406 | 1.40543943423732E-16 |
| **Npl** | 5.08850186991542E-21 | 0.352882100817083 | 0.419 | 0.258 | 1.58013248566483E-16 |
| **Ogfrl1** | 5.17065398789115E-21 | -0.455393350202006 | 0.475 | 0.567 | 1.60564318285984E-16 |
| **Cd34** | 5.79105523560748E-21 | 0.441877995625058 | 0.633 | 0.466 | 1.79829638231319E-16 |
| **Sft2d1** | 5.83530366903873E-21 | -0.316307144515182 | 0.842 | 0.871 | 1.8120368483466E-16 |
| **Inpp5d** | 8.95122246639135E-21 | -0.320474111370355 | 0.828 | 0.869 | 2.77962311248851E-16 |
| **Il6ra** | 9.21620560202215E-21 | -0.394310742173464 | 0.637 | 0.699 | 2.86190832559594E-16 |
| **Tpm4** | 1.05342595620172E-20 | 0.271181645835348 | 0.204 | 0.098 | 3.27120362179321E-16 |
| **Rbm3** | 1.06529018324443E-20 | 0.335309695611084 | 0.725 | 0.572 | 3.30804560602894E-16 |
| **Tcf4** | 1.07209154748229E-20 | -0.43010896893165 | 0.609 | 0.686 | 3.32916588239675E-16 |
| **Eng** | 1.07559986534675E-20 | -0.474809410948009 | 0.196 | 0.328 | 3.34006026186126E-16 |
| **AU020206** | 1.11562881595433E-20 | 0.255641200769782 | 0.246 | 0.125 | 3.46436216218297E-16 |
| **Neu1** | 1.14480087762701E-20 | 0.313046714571744 | 0.377 | 0.222 | 3.55495016529515E-16 |

(*Continued*)

**Table 2.** (Continued)

| | p_val | avg_log2FC | pct.1 | pct.2 | p_val_adj |
|---|---|---|---|---|---|
| Vps29 | 1.20794604224294E-20 | 0.343472020092991 | 0.506 | 0.33 | 3.751034844977E-16 |
| Pld1 | 1.29195425587035E-20 | -0.47277470803636 | 0.22 | 0.35 | 4.01190555075421E-16 |
| Cttnbp2nl | 1.30963284097332E-20 | -0.357387214797646 | 0.695 | 0.758 | 4.06680286107447E-16 |
| Itga6 | 1.62273809533733E-20 | -0.422708778607816 | 0.563 | 0.651 | 5.039088607451E-16 |
| Dnajb14 | 1.68113973048152E-20 | 0.336379748806099 | 0.439 | 0.277 | 5.22044320506425E-16 |
| Clec4a2 | 1.68649949552416E-20 | -0.460954076140435 | 0.344 | 0.456 | 5.23708688345117E-16 |
| Rragc | 1.79207168155747E-20 | 0.344496311440997 | 0.484 | 0.313 | 5.56492019274041E-16 |
| Tpp1 | 2.05433713626289E-20 | 0.317589286024118 | 0.798 | 0.639 | 6.37933310923716E-16 |
| Atp1a1 | 2.10163841726529E-20 | 0.342408747823385 | 0.519 | 0.342 | 6.52621777713392E-16 |
| Adap2 | 2.12062053585025E-20 | -0.378084340796438 | 0.706 | 0.759 | 6.58516294997577E-16 |
| Kcnk12 | 2.20335453563795E-20 | -0.440561300457441 | 0.119 | 0.252 | 6.84207683951653E-16 |
| Rps18 | 2.49754121950701E-20 | 0.251631169651514 | 0.97 | 0.932 | 7.75561474893513E-16 |
| Slc12a9 | 2.50575506470782E-20 | -0.440747503217784 | 0.365 | 0.485 | 7.7811212024372E-16 |
| Commd4 | 3.37034329662654E-20 | 0.281126624875675 | 0.367 | 0.217 | 1.04659270390144E-15 |
| Mapkapk2 | 3.49853753855046E-20 | 0.26652535556196 | 0.338 | 0.189 | 1.08640086184607E-15 |
| Mtus1 | 3.68722416066434E-20 | -0.424309218665912 | 0.416 | 0.521 | 1.1449937186111E-15 |
| Slc12a2 | 3.75485594725543E-20 | 0.332422140712737 | 0.468 | 0.302 | 1.16599541730123E-15 |
| Uqcc2 | 4.31666990210354E-20 | 0.280946972087408 | 0.432 | 0.265 | 1.34045550470021E-15 |
| Cd84 | 4.37521068579692E-20 | 0.333477594641535 | 0.848 | 0.713 | 1.35863417426052E-15 |
| Lyl1 | 5.0726471734059E-20 | -0.446324931864446 | 0.462 | 0.555 | 1.57520912675774E-15 |
| Khdrbs3 | 5.88435176878513E-20 | -0.49942136021998 | 0.173 | 0.302 | 1.82726775476085E-15 |
| Atp6v0a2 | 6.83695758092147E-20 | -0.4540460991717 | 0.273 | 0.398 | 2.12308043760355E-15 |
| Bank1 | 8.72998276938563E-20 | -0.463581420878649 | 0.185 | 0.314 | 2.71092154937732E-15 |
| Grina | 1.00314510712373E-19 | 0.319982884987649 | 0.598 | 0.414 | 3.11506650115132E-15 |
| Ucp2 | 1.01744008111698E-19 | 0.317308618728166 | 0.829 | 0.694 | 3.15945668389257E-15 |
| Mfsd11 | 1.04970316882176E-19 | 0.375373232182121 | 0.429 | 0.271 | 3.25964325014222E-15 |
| Sgpl1 | 1.18012181943746E-19 | 0.326938686688929 | 0.444 | 0.28 | 3.66463228589914E-15 |
| Tmem86a | 1.22210473148769E-19 | 0.33259963190436 | 0.904 | 0.807 | 3.79500182268872E-15 |
| Mfsd1 | 1.22972343918018E-19 | 0.279819868572921 | 0.345 | 0.201 | 3.81866019568621E-15 |
| Yif1b | 1.28729438539454E-19 | 0.317829682125488 | 0.545 | 0.363 | 3.99743525496567E-15 |
| Cln8 | 1.69828330352658E-19 | 0.32443991266458 | 0.458 | 0.288 | 5.2736791424411E-15 |
| Rps6ka1 | 1.73385019340581E-19 | -0.429553091562141 | 0.47 | 0.553 | 5.38412500558305E-15 |
| Rps27l | 2.26903164414125E-19 | 0.335913800147433 | 0.587 | 0.411 | 7.04602396455182E-15 |
| Tmem55b | 2.36488507243722E-19 | -0.344191530534683 | 0.647 | 0.719 | 7.34367761543929E-15 |
| Klf7 | 2.8325196055687E-19 | -0.495953343198119 | 0.256 | 0.379 | 8.79582313117247E-15 |
| Cycs | 3.9005245142487E-19 | 0.277145651818025 | 0.437 | 0.269 | 1.21122987740965E-14 |
| Mtss1 | 4.21296766458036E-19 | -0.442989148338404 | 0.154 | 0.283 | 1.30825284888214E-14 |
| Sipa1 | 4.63539390497423E-19 | -0.421767107683874 | 0.482 | 0.564 | 1.43942886931165E-14 |
| Bcl2l1 | 5.59025175349705E-19 | 0.284054852072817 | 0.353 | 0.206 | 1.73594087701344E-14 |
| Arglu1 | 8.11661808808381E-19 | -0.329966313916827 | 0.738 | 0.768 | 2.52045341489267E-14 |
| Ccng2 | 8.31122021168012E-19 | -0.451676492248546 | 0.296 | 0.413 | 2.58088321233303E-14 |
| Lgals3bp | 8.79950920058822E-19 | 0.321461377148218 | 0.407 | 0.254 | 2.73251159205866E-14 |
| Cox7a2l | 8.87621598780657E-19 | -0.379052985231203 | 0.582 | 0.636 | 2.75633135069357E-14 |
| Rrp1 | 9.29740077857506E-19 | 0.293405835357007 | 0.478 | 0.311 | 2.88712186377091E-14 |
| Atp6v1g1 | 9.44350247136255E-19 | 0.311739476909132 | 0.828 | 0.701 | 2.93249082243221E-14 |
| Grap | 9.77795686660405E-19 | -0.464006627353637 | 0.281 | 0.398 | 3.03634894578655E-14 |
| Atp5b | 1.10492476068254E-18 | 0.33749467525466 | 0.726 | 0.572 | 3.43112285934748E-14 |

*(Continued)*

**Table 2.** (Continued)

| | p_val | avg_log2FC | pct.1 | pct.2 | p_val_adj |
|---|---|---|---|---|---|
| **Tkt** | 1.14127583794376E-18 | 0.297288021736379 | 0.518 | 0.344 | 3.54400385956677E-14 |
| **Cd47** | 1.17798198386232E-18 | -0.373747055718565 | 0.623 | 0.682 | 3.65798745448765E-14 |
| **Wdr83os** | 1.20976344231692E-18 | 0.303102574968087 | 0.564 | 0.389 | 3.75667841742673E-14 |
| **Rcsd1** | 1.23815382135333E-18 | -0.431168500218207 | 0.416 | 0.524 | 3.8448390614485E-14 |
| **Pnisr** | 1.48670064105816E-18 | -0.338582899027254 | 0.726 | 0.779 | 4.61665150067789E-14 |
| **Ncf1** | 1.64229070392813E-18 | -0.352372726457918 | 0.66 | 0.727 | 5.09980532290802E-14 |
| **Slc16a6** | 1.77838692411149E-18 | -0.413486740549916 | 0.496 | 0.578 | 5.52242491544342E-14 |
| **Lat2** | 2.08306280396461E-18 | 0.350389003864938 | 0.549 | 0.383 | 6.46853492515129E-14 |
| **Plbd2** | 2.14664275958688E-18 | 0.285799369878017 | 0.468 | 0.302 | 6.66596976134514E-14 |
| **Gtf2h2** | 2.30374475078653E-18 | -0.396965609894835 | 0.517 | 0.589 | 7.15381857461741E-14 |
| **Rbm39** | 2.33775999241838E-18 | -0.261919826975625 | 0.929 | 0.926 | 7.25944610445679E-14 |
| **Upk1b** | 2.35006043009281E-18 | -0.505774509767325 | 0.154 | 0.28 | 7.29764265356721E-14 |
| **Atp6v1b2** | 2.46514928877113E-18 | 0.285442312166701 | 0.378 | 0.23 | 7.655028086421E-14 |
| **Dusp7** | 2.70754994789811E-18 | -0.424791809376384 | 0.359 | 0.466 | 8.407754853208E-14 |
| **Spg21** | 3.05252671370719E-18 | 0.29090779850723 | 0.363 | 0.221 | 9.47901120407495E-14 |
| **Pou2f2** | 3.24795451227099E-18 | -0.316212820021629 | 0.821 | 0.863 | 1.00858731469551E-13 |
| **Hspa9** | 4.02544128885067E-18 | 0.259641227682286 | 0.352 | 0.209 | 1.2500202834268E-13 |
| **Creld2** | 4.5616502224761E-18 | 0.274122935939713 | 0.325 | 0.19 | 1.4165292435855E-13 |
| **Dapp1** | 4.92002507710908E-18 | -0.425391449544058 | 0.333 | 0.443 | 1.52781538719468E-13 |
| **Sla** | 5.86274068312574E-18 | -0.424364632129824 | 0.33 | 0.439 | 1.82055686433104E-13 |
| **Atp6ap1** | 6.05389879250639E-18 | 0.293302404392222 | 0.711 | 0.539 | 1.87991719203701E-13 |
| **Colgalt1** | 6.39124477930819E-18 | 0.308035376851334 | 0.508 | 0.342 | 1.98467324131857E-13 |
| **Scamp2** | 6.91689931949417E-18 | -0.262392884928979 | 0.901 | 0.928 | 2.14790474568252E-13 |
| **Rsl1d1** | 7.35766121463398E-18 | 0.306352642651962 | 0.551 | 0.366 | 2.28477453698029E-13 |
| **Fkbp2** | 7.77905904855289E-18 | 0.325204521853408 | 0.733 | 0.578 | 2.41563120634713E-13 |
| **Wdr12** | 8.14270766388104E-18 | 0.254913645908266 | 0.253 | 0.138 | 2.52855501086498E-13 |
| **Ppib** | 8.41745387464041E-18 | 0.266468284640856 | 0.933 | 0.858 | 2.61387195169209E-13 |
| **Il6st** | 8.7027564716849E-18 | -0.42636185457079 | 0.387 | 0.486 | 2.70246696715231E-13 |
| **Ndufa12** | 9.39030207484908E-18 | 0.253174833372587 | 0.43 | 0.268 | 2.91597050330289E-13 |
| **Phb** | 1.03525488065212E-17 | 0.254267723508659 | 0.304 | 0.174 | 3.21477698088901E-13 |
| **Tjp1** | 1.08297826060262E-17 | -0.476632069318155 | 0.278 | 0.388 | 3.36297239264931E-13 |
| **Mgat1** | 1.29179754881149E-17 | -0.432793495041039 | 0.416 | 0.511 | 4.01141892832433E-13 |
| **Cd37** | 1.40137257848825E-17 | -0.278288085096139 | 0.904 | 0.916 | 4.35168226797957E-13 |
| **Ggt5** | 1.51270632008367E-17 | -0.426077524703089 | 0.124 | 0.24 | 4.69740693575582E-13 |
| **Alas1** | 1.6623425615383E-17 | 0.274592162943472 | 0.358 | 0.215 | 5.16207235634487E-13 |
| **Il7r** | 1.81245134942228E-17 | -0.469648368612809 | 0.209 | 0.328 | 5.62820517536102E-13 |
| **Abi3** | 1.95577304055468E-17 | -0.316197954427222 | 0.792 | 0.824 | 6.07326202283444E-13 |
| **Sdhb** | 1.98657046045973E-17 | 0.269799655856111 | 0.505 | 0.331 | 6.16889725086559E-13 |
| **Cct2** | 2.05614424939656E-17 | 0.290584052373808 | 0.571 | 0.398 | 6.38494473765113E-13 |
| **Ddost** | 2.17632032943184E-17 | 0.316976146401421 | 0.753 | 0.59 | 6.7581275189847E-13 |
| **Dusp3** | 2.2018042889431E-17 | 0.257253282389059 | 0.284 | 0.162 | 6.83726285845502E-13 |
| **Tomm20** | 2.47177707857286E-17 | 0.351493561067022 | 0.696 | 0.552 | 7.6756093620923E-13 |
| **Ptpre** | 2.86465690103381E-17 | -0.359509103708775 | 0.564 | 0.643 | 8.89561907478028E-13 |
| **Actr3** | 2.97296712302265E-17 | 0.321048632465373 | 0.752 | 0.593 | 9.23195480712224E-13 |
| **Pwwp2a** | 4.3929564092752E-17 | -0.386760492013616 | 0.473 | 0.564 | 1.36414475377223E-12 |
| **Tnfaip8** | 4.49399400016815E-17 | -0.420043677174188 | 0.419 | 0.509 | 1.39551995687222E-12 |
| **Eno1** | 5.00583791198731E-17 | 0.305968390382963 | 0.601 | 0.423 | 1.55446284680942E-12 |

(*Continued*)

**Table 2.** (Continued)

| | p_val | avg_log2FC | pct.1 | pct.2 | p_val_adj |
|---|---|---|---|---|---|
| **Prmt1** | 5.09174335238438E-17 | 0.276208284607023 | 0.414 | 0.263 | 1.58113906321592E-12 |
| **Gbp7** | 5.25700199766624E-17 | -0.454576157339616 | 0.136 | 0.254 | 1.6324568303353E-12 |
| **Ang** | 5.41259363576779E-17 | -0.378054544736609 | 0.564 | 0.622 | 1.68077270171497E-12 |
| **Ptp4a3** | 6.08362417358844E-17 | -0.38464088088297 | 0.458 | 0.54 | 1.88914781462442E-12 |
| **Slamf9** | 6.8877059725291E-17 | 0.318172797931955 | 0.352 | 0.22 | 2.13883933564946E-12 |
| **Cox7b** | 6.93971162875382E-17 | 0.280689487662147 | 0.539 | 0.372 | 2.15498865207692E-12 |
| **Tnrc6b** | 8.26852170579065E-17 | -0.394115289814842 | 0.432 | 0.519 | 2.56762404529917E-12 |
| **Cdkn1a** | 8.29339614202412E-17 | 0.289760707899465 | 0.105 | 0.042 | 2.57534830398275E-12 |
| **Tmem59** | 9.73302281885459E-17 | -0.254573834050175 | 0.908 | 0.92 | 3.02239557593892E-12 |
| **Limd2** | 1.06040622241658E-16 | -0.340310974759087 | 0.653 | 0.711 | 3.2928794424702E-12 |
| **Tmed3** | 1.06282554058161E-16 | 0.300893066937747 | 0.723 | 0.578 | 3.30039215116808E-12 |
| **Ddit4** | 1.18285230665959E-16 | -0.489861585652333 | 0.215 | 0.34 | 3.67311126787003E-12 |
| **Mdh2** | 1.18585765217309E-16 | 0.261007102367462 | 0.446 | 0.288 | 3.68244376729309E-12 |
| **Slfn2** | 1.2509809273484E-16 | 0.268369261927096 | 0.284 | 0.166 | 3.88467107369499E-12 |
| **Ikzf1** | 1.31203969810136E-16 | -0.348030404846223 | 0.564 | 0.635 | 4.07427687451416E-12 |
| **Gpr155** | 1.36916830808104E-16 | -0.425079896635055 | 0.249 | 0.362 | 4.25167834708406E-12 |
| **Cct5** | 1.46571124427011E-16 | 0.262634154920281 | 0.494 | 0.331 | 4.55147312683197E-12 |
| **Gng5** | 1.59347456835924E-16 | 0.266547340884202 | 0.875 | 0.769 | 4.94821657712595E-12 |
| **Irf2** | 1.84647701387984E-16 | -0.385496522741026 | 0.525 | 0.591 | 5.73386507120106E-12 |
| **Rbbp7** | 2.1900299100064E-16 | 0.261251212985066 | 0.344 | 0.21 | 6.80069987954286E-12 |
| **Nop56** | 2.26227028451707E-16 | 0.314039321004599 | 0.4 | 0.26 | 7.02502791451087E-12 |
| **Cfh** | 2.56692165599814E-16 | -0.296556197481258 | 0.788 | 0.812 | 7.97106181837101E-12 |
| **Zfp36l1** | 4.45065883066545E-16 | -0.397051626582696 | 0.577 | 0.65 | 1.38206308668654E-11 |
| **Map4k4** | 4.55184204184691E-16 | -0.374420110721998 | 0.525 | 0.596 | 1.41348350925472E-11 |
| **Casp8** | 5.71473612734207E-16 | -0.386243920167746 | 0.474 | 0.542 | 1.77459700962353E-11 |
| **Orai1** | 5.73103854736507E-16 | -0.304532255045019 | 0.664 | 0.712 | 1.77965940011328E-11 |
| **Psmc5** | 6.27341260741128E-16 | 0.27134189169766 | 0.511 | 0.348 | 1.94808281697943E-11 |
| **Pdia4** | 6.76759851687717E-16 | 0.320574295099112 | 0.573 | 0.411 | 2.10154236744587E-11 |
| **Ogt** | 7.23541633333939E-16 | -0.334368936474543 | 0.613 | 0.669 | 2.24681383399188E-11 |
| **Pip4k2a** | 7.42971727070723E-16 | -0.391253442586827 | 0.434 | 0.513 | 2.30715010407271E-11 |
| **Rock2** | 8.27947521304489E-16 | -0.388773605097724 | 0.471 | 0.547 | 2.57102543790683E-11 |
| **Snx18** | 8.80388343138845E-16 | -0.343873013127123 | 0.614 | 0.668 | 2.73386992194905E-11 |
| **B4galt4** | 9.93181678801525E-16 | -0.39356310119824 | 0.298 | 0.406 | 3.08412706718238E-11 |
| **Fgd2** | 1.00856634483731E-15 | -0.311353869726041 | 0.628 | 0.687 | 3.13190107062331E-11 |
| **Whrn** | 1.3123436583685E-15 | -0.409618387092258 | 0.207 | 0.32 | 4.0752207623317E-11 |
| **Pid1** | 1.33648074370879E-15 | -0.388046487551891 | 0.407 | 0.497 | 4.15017365343891E-11 |
| **Sec61g** | 1.54658056248189E-15 | 0.30026104445314 | 0.724 | 0.566 | 4.80259662067502E-11 |
| **Klf3** | 1.96103615461628E-15 | -0.338396161815291 | 0.546 | 0.614 | 6.08960557092993E-11 |
| **Fam46c** | 2.90453498475351E-15 | 0.290022214628318 | 0.382 | 0.246 | 9.01945248815508E-11 |
| **Morf4l2** | 3.36171005253021E-15 | 0.250687920314127 | 0.385 | 0.245 | 1.04391182261221E-10 |
| **9930111J2** | 3.41627266234598E-15 | -0.378694567978912 | 0.519 | 0.581 | 1.0608551498383E-10 |
| **Sirt2** | 3.42492387105264E-15 | 0.251595821772263 | 0.465 | 0.307 | 1.06354160967798E-10 |
| **Ssr1** | 3.45844774266661E-15 | 0.293344707254471 | 0.635 | 0.481 | 1.07395177753026E-10 |
| **St13** | 3.47496970953192E-15 | 0.263534755232402 | 0.717 | 0.554 | 1.07908234390095E-10 |
| **Snrnp70** | 3.51852115485312E-15 | -0.282435800880467 | 0.795 | 0.814 | 1.09260637421654E-10 |
| **Csk** | 3.61144217646443E-15 | -0.379268699075924 | 0.45 | 0.52 | 1.1214611390575E-10 |
| **Zfp652** | 3.76640816468012E-15 | -0.417103528605731 | 0.36 | 0.443 | 1.16958272737812E-10 |

(*Continued*)

**Table 2.** (Continued)

| | p_val | avg_log2FC | pct.1 | pct.2 | p_val_adj |
|---|---|---|---|---|---|
| **Ptms** | 3.84540517210156E-15 | 0.265344629410965 | 0.861 | 0.723 | 1.1941136680927E-10 |
| **Arrb2** | 3.92458471902766E-15 | -0.329048071067822 | 0.572 | 0.628 | 1.21870129279966E-10 |
| **Bri3** | 4.36035451393967E-15 | 0.313088008367401 | 0.811 | 0.725 | 1.35402088721369E-10 |
| **Sec61b** | 4.73805636530629E-15 | 0.294593502863423 | 0.641 | 0.489 | 1.47130864311856E-10 |
| **Mycbp2** | 4.76926246483581E-15 | -0.36555030851172 | 0.54 | 0.595 | 1.48099907320546E-10 |
| **Pnrc1** | 5.01864205033632E-15 | -0.331902178871134 | 0.611 | 0.67 | 1.55843891589094E-10 |
| **Kdelr2** | 5.57713059386983E-15 | 0.298411420677201 | 0.553 | 0.396 | 1.7318663633144E-10 |
| **Ccni** | 6.87418285705523E-15 | -0.35538547749027 | 0.48 | 0.544 | 2.13464000260136E-10 |
| **Dennd4a** | 7.04290606506537E-15 | -0.353570907328834 | 0.539 | 0.616 | 2.18703362038475E-10 |
| **Stard3** | 7.78873700535385E-15 | -0.360608336442906 | 0.436 | 0.523 | 2.41863650227253E-10 |
| **Fscn1** | 8.3098713983335E-15 | -0.348168455031682 | 0.654 | 0.704 | 2.5804643653245E-10 |
| **Sigmar1** | 8.71901425163909E-15 | 0.256204682064242 | 0.349 | 0.217 | 2.70751549556149E-10 |
| **Gm26917** | 9.17217973786097E-15 | 0.353357847802252 | 0.31 | 0.194 | 2.84823697399797E-10 |
| **Slc11a1** | 9.68141793340423E-15 | 0.254679678964083 | 0.546 | 0.382 | 3.00637071086002E-10 |
| **Il16** | 1.00962597780429E-14 | -0.398779919496901 | 0.281 | 0.377 | 3.13519154887567E-10 |
| **Scoc** | 1.08829198136043E-14 | -0.270121450773693 | 0.871 | 0.888 | 3.37947308971855E-10 |
| **2010107E0** | 1.39661226384444E-14 | 0.266506234406527 | 0.563 | 0.404 | 4.33690006291615E-10 |
| **Ighm** | 1.44216628134759E-14 | -0.347798026166093 | 0.552 | 0.632 | 4.47835895346868E-10 |
| **0610040J0** | 1.49847013934833E-14 | -0.376701643247374 | 0.354 | 0.446 | 4.65319932371836E-10 |
| **Grk2** | 1.71727864009447E-14 | -0.393591078538817 | 0.336 | 0.418 | 5.33266536108537E-10 |
| **Atp6v1d** | 1.77745472129953E-14 | 0.308335909507825 | 0.412 | 0.279 | 5.51953014605144E-10 |
| **Tubgcp5** | 2.12057341485409E-14 | -0.378971940890316 | 0.428 | 0.506 | 6.58501662514642E-10 |
| **Olfml3** | 2.37307789960423E-14 | -0.260508077012623 | 0.989 | 0.986 | 7.36911880164101E-10 |
| **Gpsm3** | 2.68428956663939E-14 | -0.361411519079472 | 0.437 | 0.519 | 8.33552439128531E-10 |
| **Elovl1** | 3.13314908989516E-14 | 0.300216385321233 | 0.501 | 0.345 | 9.72936786885143E-10 |
| **Tmem206** | 3.20897139888298E-14 | 0.329988439763472 | 0.374 | 0.246 | 9.96481888495131E-10 |
| **Maml3** | 3.38781952630243E-14 | -0.410146105599206 | 0.215 | 0.317 | 1.05201959750269E-09 |
| **Srsf2** | 3.68608948629561E-14 | 0.362475194084846 | 0.772 | 0.684 | 1.14464136817937E-09 |
| **Tra2a** | 4.14488149802628E-14 | -0.319424393723889 | 0.632 | 0.675 | 1.2871100515821E-09 |
| **Tomm40** | 4.18849464803916E-14 | 0.253398728298092 | 0.395 | 0.258 | 1.3006532430556E-09 |
| **Sesn1** | 5.02027647396987E-14 | -0.406158865338817 | 0.243 | 0.337 | 1.55894645346186E-09 |
| **Mycl** | 5.44136164760015E-14 | -0.30138139032991 | 0.048 | 0.137 | 1.68970603242928E-09 |
| **Rps17** | 5.5525317128653E-14 | 0.255205254558279 | 0.839 | 0.71 | 1.72422767279606E-09 |
| **Slc23a2** | 5.57977543566448E-14 | 0.267830203122948 | 0.269 | 0.161 | 1.73268766603689E-09 |
| **Rnf167** | 6.52070948388705E-14 | -0.361955540184498 | 0.283 | 0.38 | 2.02487591603144E-09 |
| **Naglu** | 6.82941436458614E-14 | 0.26297945763687 | 0.457 | 0.315 | 2.12073804263493E-09 |
| **Ndufa4** | 7.43317359933091E-14 | 0.267063341070183 | 0.757 | 0.619 | 2.30822339780023E-09 |
| **Atp6v0d1** | 8.1015251662391E-14 | 0.256841723680055 | 0.501 | 0.342 | 2.51576660987223E-09 |
| **March1** | 8.27581105347156E-14 | -0.365552406584209 | 0.391 | 0.479 | 2.56988760643452E-09 |
| **Cox5b** | 8.46558491892441E-14 | 0.277483137143121 | 0.611 | 0.447 | 2.6288180848736E-09 |
| **Cox5a** | 8.78197043571543E-14 | 0.255778217383667 | 0.586 | 0.43 | 2.72706527940271E-09 |
| **Rpl31** | 9.12716164053496E-14 | 0.265747847984656 | 0.747 | 0.595 | 2.83425750423532E-09 |
| **Fry** | 9.1984410020598E-14 | -0.399064182929683 | 0.149 | 0.251 | 2.85639188436963E-09 |
| **Arl8a** | 9.30636242290384E-14 | 0.25150680734014 | 0.409 | 0.27 | 2.88990472318433E-09 |
| **Arid1a** | 9.53790640384842E-14 | -0.354670538769954 | 0.476 | 0.543 | 2.96180607558705E-09 |
| **Nos1ap** | 1.03070669666513E-13 | -0.362519811848667 | 0.115 | 0.214 | 3.20065350515423E-09 |
| **Hnrnpf** | 1.13836137974372E-13 | 0.260906886875476 | 0.843 | 0.746 | 3.53495359251818E-09 |

(*Continued*)

**Table 2.** (Continued)

| | p_val | avg_log2FC | pct.1 | pct.2 | p_val_adj |
|---|---|---|---|---|---|
| Nfia | 1.17748017852189E-13 | -0.408056369440299 | 0.326 | 0.418 | 3.65642919836403E-09 |
| Phf14 | 1.21898825112006E-13 | -0.315583931610452 | 0.532 | 0.597 | 3.78532421620312E-09 |
| Vmp1 | 1.21944254666567E-13 | 0.280717104537476 | 0.504 | 0.358 | 3.78673494016091E-09 |
| Atxn10 | 1.24829647065569E-13 | 0.258684811015794 | 0.458 | 0.314 | 3.87633503032712E-09 |
| Lrpap1 | 1.25749456382314E-13 | 0.257589361516729 | 0.582 | 0.426 | 3.90489786903998E-09 |
| Crlf3 | 1.29849080390834E-13 | -0.315694208421083 | 0.613 | 0.646 | 4.03220349337657E-09 |
| Thrsp | 1.31436544055218E-13 | -0.361849996680576 | 0.264 | 0.369 | 4.08149900254669E-09 |
| Mfng | 1.8871840372422E-13 | -0.375818280980217 | 0.309 | 0.399 | 5.86027259084819E-09 |
| Dock8 | 2.00564292462928E-13 | -0.386343276761699 | 0.38 | 0.449 | 6.22812297385132E-09 |
| Arhgef40 | 2.08235610692363E-13 | -0.385700119646028 | 0.303 | 0.388 | 6.46634041882996E-09 |
| Ankrd12 | 2.21658259126232E-13 | 0.395924993993436 | 0.474 | 0.347 | 6.88315392064687E-09 |
| Cep68 | 3.58505905729271E-13 | -0.358244815516592 | 0.144 | 0.244 | 1.11326838906111E-08 |
| Rcan1 | 3.79462504112328E-13 | 0.31594502120239 | 0.284 | 0.176 | 1.17834491402001E-08 |
| Psmb6 | 5.04650659733684E-13 | 0.259279992102095 | 0.599 | 0.45 | 1.56709169367101E-08 |
| Chst7 | 5.11801440964736E-13 | -0.363209906220227 | 0.276 | 0.371 | 1.5892970146278E-08 |
| Frmd4b | 5.29142728141766E-13 | -0.278060792014388 | 0.694 | 0.712 | 1.64314691369863E-08 |
| Soga1 | 5.40450090652629E-13 | -0.353874503760134 | 0.254 | 0.351 | 1.67825966650361E-08 |
| Dpysl2 | 5.78068558930173E-13 | -0.356324351006645 | 0.423 | 0.493 | 1.79507629604587E-08 |
| Mertk | 6.35957529969865E-13 | -0.273873208453055 | 0.804 | 0.813 | 1.97483891781542E-08 |
| Armc3 | 6.46876835584872E-13 | -0.330035880316145 | 0.074 | 0.161 | 2.0087466375417E-08 |
| Psmd7 | 6.88748250932346E-13 | 0.255606129628242 | 0.585 | 0.423 | 2.13876994362021E-08 |
| Mgat4a | 7.33268642955137E-13 | -0.331794596018824 | 0.463 | 0.531 | 2.27701911696859E-08 |
| Akirin2 | 7.43793709357552E-13 | -0.317072005917028 | 0.535 | 0.591 | 2.30970260566801E-08 |
| mt-Atp8 | 8.87174971959925E-13 | 0.252874091257314 | 0.338 | 0.222 | 2.75494444042715E-08 |
| Sult1a1 | 9.43543911598618E-13 | -0.3832758948998 | 0.248 | 0.343 | 2.92998690868719E-08 |
| Snx24 | 1.02220021930258E-12 | 0.259419801881412 | 0.352 | 0.233 | 3.1742383410003E-08 |
| Uqcr11 | 1.12702879465377E-12 | 0.254647488403128 | 0.654 | 0.496 | 3.49976251603834E-08 |
| 1700017B0 | 1.2071249021999E-12 | -0.292974047958738 | 0.579 | 0.627 | 3.74848495880134E-08 |
| Btg1 | 1.32061728999243E-12 | -0.282260309018144 | 0.76 | 0.786 | 4.10091287061351E-08 |
| Soat1 | 1.3485726701717E-12 | 0.276635833403159 | 0.374 | 0.25 | 4.18772271268418E-08 |
| Adora3 | 1.71806282743118E-12 | -0.357770048652599 | 0.347 | 0.425 | 5.33510049802206E-08 |
| Anp32b | 1.8115591645384E-12 | 0.28238330071732 | 0.389 | 0.267 | 5.62543467364109E-08 |
| Siglece | 2.32433062938259E-12 | -0.371961023988845 | 0.187 | 0.281 | 7.21774390342177E-08 |
| Gm32036 | 2.61605955363153E-12 | -0.330051478209053 | 0.122 | 0.213 | 8.12364973189198E-08 |
| Fcrl1 | 2.93706032511877E-12 | -0.312221475664913 | 0.087 | 0.175 | 9.12045342759132E-08 |
| Sft2d2 | 3.03657087822497E-12 | -0.386455262085356 | 0.227 | 0.314 | 9.42946354815201E-08 |
| Snx3 | 3.29244022761008E-12 | 0.250095106444104 | 0.572 | 0.429 | 1.02240146387976E-07 |
| Mfap3 | 3.3969359238781E-12 | -0.362783174606334 | 0.378 | 0.441 | 1.05485051244187E-07 |
| Hfe | 3.5368718118132E-12 | -0.336339887554912 | 0.4 | 0.474 | 1.09830480372235E-07 |
| Evi2a | 4.23508467591846E-12 | 0.284046077537508 | 0.849 | 0.756 | 1.31512084441296E-07 |
| Fam91a1 | 4.2917903016291E-12 | -0.362783322793053 | 0.26 | 0.344 | 1.33272964236489E-07 |
| Myo1b | 4.8114936476746E-12 | -0.349878576759932 | 0.132 | 0.224 | 1.49411312241239E-07 |
| Rapgef6 | 5.65288965439106E-12 | -0.368775099753797 | 0.302 | 0.384 | 1.75539182437805E-07 |
| BC017643 | 5.84631313762014E-12 | -0.328821511762247 | 0.409 | 0.478 | 1.81545561862518E-07 |
| Ubash3b | 5.99421038024171E-12 | -0.326270499807639 | 0.496 | 0.545 | 1.86138214937646E-07 |
| Pik3r1 | 6.33350479393534E-12 | -0.308800828559661 | 0.578 | 0.614 | 1.96674324366074E-07 |
| Tmem44 | 6.5195586825707E-12 | -0.326300562511423 | 0.076 | 0.158 | 2.02451855769868E-07 |

(*Continued*)

**Table 2.** (Continued)

| | p_val | avg_log2FC | pct.1 | pct.2 | p_val_adj |
|---|---|---|---|---|---|
| Tmem100 | 6.66674616907444E-12 | -0.340335796741541 | 0.331 | 0.424 | 2.07022468788269E-07 |
| Pold4 | 7.42188548763467E-12 | -0.323339410081192 | 0.407 | 0.477 | 2.30471810047519E-07 |
| Klhdc8b | 7.74592357565568E-12 | -0.370925438228524 | 0.228 | 0.316 | 2.40534164794836E-07 |
| Cryl1 | 1.06624846978481E-11 | -0.354023711590468 | 0.295 | 0.373 | 3.31102137322278E-07 |
| Zbtb20 | 1.14264813480778E-11 | -0.425925194116129 | 0.338 | 0.423 | 3.54826525301861E-07 |
| Rtn4rl1 | 1.15908859032059E-11 | -0.313795702018367 | 0.458 | 0.523 | 3.59931779952254E-07 |
| Ckap4 | 1.16014306543555E-11 | 0.270591369637953 | 0.326 | 0.215 | 3.60259226109702E-07 |
| Irf2bp2 | 1.23672785296879E-11 | -0.30365330557541 | 0.499 | 0.558 | 3.840411001824E-07 |
| Plcl2 | 1.32653526615747E-11 | -0.358652414706463 | 0.315 | 0.391 | 4.11928996199878E-07 |
| Etv1 | 2.50252176614799E-11 | -0.307576895201074 | 0.091 | 0.175 | 7.77108084041935E-07 |
| Fermt3 | 2.68431837138697E-11 | -0.291950529313464 | 0.591 | 0.618 | 8.33561383866796E-07 |
| Lag3 | 3.38996223187597E-11 | 0.255518545051852 | 0.749 | 0.619 | 1.05268497186444E-06 |
| Agmo | 3.65982105274457E-11 | -0.326326386160307 | 0.191 | 0.278 | 1.13648423150877E-06 |
| Tuba1a | 3.66470061363511E-11 | -0.313085156619056 | 0.482 | 0.538 | 1.13799948155211E-06 |
| Kmt2e | 3.8398873523139E-11 | -0.347292190578359 | 0.458 | 0.507 | 1.19240021951404E-06 |
| Ppp1r18 | 3.840214953127E-11 | -0.300392745715527 | 0.51 | 0.557 | 1.19250194939453E-06 |
| Filip1l | 4.24320362984864E-11 | -0.35325481569522 | 0.198 | 0.288 | 1.3176420231769E-06 |
| Adap2os | 4.32867317369491E-11 | -0.30225746268662 | 0.47 | 0.538 | 1.34418288062748E-06 |
| Arhgap31 | 4.35743708673276E-11 | -0.343321492673757 | 0.433 | 0.49 | 1.35311493854312E-06 |
| Plxna4 | 4.83864633828803E-11 | -0.341720946807565 | 0.145 | 0.233 | 1.50254484742858E-06 |
| Rhoh | 6.28701975795654E-11 | -0.269646045304644 | 0.687 | 0.72 | 1.95230824543824E-06 |
| Arhgap27 | 6.49351636229224E-11 | -0.339360939968431 | 0.184 | 0.266 | 2.01643163598261E-06 |
| Prkab1 | 6.80240112541581E-11 | -0.332032795223066 | 0.233 | 0.313 | 2.11234962147537E-06 |
| Kif21b | 6.83783527535236E-11 | -0.329249464737552 | 0.295 | 0.378 | 2.12335298805517E-06 |
| Cd86 | 7.19896970323852E-11 | 0.257365440764221 | 0.767 | 0.647 | 2.23549606194666E-06 |
| Zfp467 | 8.25737288369693E-11 | -0.338324996994412 | 0.166 | 0.249 | 2.56416200157441E-06 |
| Tlr3 | 1.03403632108671E-10 | -0.344453307996895 | 0.199 | 0.283 | 3.21099298787057E-06 |
| Gm26740 | 1.07053986318923E-10 | -0.357099866645611 | 0.132 | 0.22 | 3.32434743716153E-06 |
| Capn3 | 1.20445186466862E-10 | -0.339335918525183 | 0.204 | 0.286 | 3.74018437535547E-06 |
| Gm31243 | 1.29906956046967E-10 | -0.309440708917158 | 0.113 | 0.194 | 4.03400070612645E-06 |
| Ctsf | 1.49161496733332E-10 | -0.274051039753143 | 0.598 | 0.635 | 4.63191195806014E-06 |
| Abhd6 | 1.49599371263832E-10 | -0.304553701403873 | 0.407 | 0.467 | 4.64550927585578E-06 |
| Ccdc50 | 1.49625739943834E-10 | -0.33601221438744 | 0.382 | 0.443 | 4.64632810247587E-06 |
| Cmtm8 | 1.57069891881241E-10 | -0.329801400710897 | 0.132 | 0.213 | 4.87749135258819E-06 |
| Camk2d | 1.67572780664287E-10 | -0.295375334217713 | 0.566 | 0.605 | 5.2036375579681E-06 |
| Plxnb2 | 1.76677322086986E-10 | -0.271180330234748 | 0.547 | 0.582 | 5.48636088276717E-06 |
| Tlr2 | 2.0070829394644E-10 | 0.365148627193511 | 0.366 | 0.258 | 6.23259465191881E-06 |
| Sik2 | 2.55658357362055E-10 | -0.320084583993081 | 0.168 | 0.251 | 7.93895897116391E-06 |
| Rnase6 | 2.57097995515787E-10 | -0.275402371844687 | 0.051 | 0.121 | 7.98366405475173E-06 |
| Tab2 | 2.76787220785046E-10 | -0.322977711989916 | 0.357 | 0.425 | 8.59507356703805E-06 |
| Serpinf1 | 2.9252313403133E-10 | -0.328831849826937 | 0.271 | 0.351 | 9.08372088107489E-06 |
| St3gal5 | 2.95376030277258E-10 | -0.291506150397422 | 0.593 | 0.623 | 9.1723118681997E-06 |
| Pnn | 2.97221239750054E-10 | -0.28965587780706 | 0.613 | 0.626 | 9.22961115795842E-06 |
| Garnl3 | 3.05774102324112E-10 | -0.314695046891293 | 0.164 | 0.246 | 9.49520319947067E-06 |
| Wdfy2 | 3.53155042047174E-10 | -0.317315160849578 | 0.132 | 0.212 | 1.09665235206909E-05 |
| G3bp2 | 4.07566377273629E-10 | -0.324232407352645 | 0.413 | 0.473 | 1.2656158713478E-05 |
| Stab1 | 5.00075085126394E-10 | -0.269190258596502 | 0.596 | 0.62 | 1.55288316184299E-05 |

(*Continued*)

**Table 2.** (Continued)

| | p_val | avg_log2FC | pct.1 | pct.2 | p_val_adj |
|---|---|---|---|---|---|
| 1810011H1 | 5.4741893572433E-10 | -0.345300704795401 | 0.267 | 0.336 | 1.69990002110476E-05 |
| Rgs19 | 6.54843424080239E-10 | -0.298723080466954 | 0.356 | 0.426 | 2.03348528479637E-05 |
| Rassf2 | 6.66620186518655E-10 | -0.307689557620538 | 0.449 | 0.489 | 2.07005566519638E-05 |
| Ralgps1 | 6.88570059642538E-10 | -0.319846054456458 | 0.132 | 0.208 | 2.13821660620797E-05 |
| Sall3 | 7.2100684301764E-10 | -0.332077730039377 | 0.285 | 0.362 | 2.23894254962268E-05 |
| Rab3il1 | 8.25441703034906E-10 | -0.255321356899655 | 0.683 | 0.703 | 2.56324412043429E-05 |
| Sema4b | 8.77724462414094E-10 | -0.306150580593703 | 0.129 | 0.207 | 2.72559777313449E-05 |
| mt-Nd5 | 8.92775887643216E-10 | 0.25421372022666 | 0.668 | 0.547 | 2.77233696389848E-05 |
| Gm6277 | 9.27738412679472E-10 | -0.317879039385527 | 0.233 | 0.316 | 2.88090609289356E-05 |
| Pik3cg | 9.36388254754199E-10 | -0.301240301246207 | 0.313 | 0.393 | 2.90776644748821E-05 |
| Yipf4 | 9.5970918577271E-10 | -0.301900284198654 | 0.453 | 0.486 | 2.98018493458E-05 |
| Mat2a | 1.12238953929417E-09 | 0.297990199837501 | 0.647 | 0.534 | 3.48535623637018E-05 |
| Nrm | 1.2003915130591E-09 | -0.332245759858035 | 0.165 | 0.24 | 3.72757576550241E-05 |
| Akap13 | 1.28140817253234E-09 | -0.282248131869738 | 0.627 | 0.64 | 3.97915679816468E-05 |
| Ggta1 | 1.3740174683028E-09 | -0.312541637867199 | 0.159 | 0.237 | 4.26673644432067E-05 |
| Eif5 | 1.50053864322529E-09 | 0.254011790557805 | 0.642 | 0.532 | 4.65962264880749E-05 |
| Arid4a | 1.578265936477E-09 | -0.345925775127187 | 0.27 | 0.343 | 4.90098921254204E-05 |
| Tcf7l2 | 1.58995724011868E-09 | -0.349097782965097 | 0.247 | 0.316 | 4.93729421774054E-05 |
| Ier5 | 1.70322316536736E-09 | -0.275338381393287 | 0.662 | 0.712 | 5.28901889541527E-05 |
| Fez2 | 1.76595936974633E-09 | -0.273407185222967 | 0.509 | 0.556 | 5.48383363087329E-05 |
| Hist1h1c | 1.95942692953849E-09 | -0.372566077479789 | 0.184 | 0.263 | 6.08460844429587E-05 |
| Jmjd1c | 2.5547115678097E-09 | -0.274847023103296 | 0.565 | 0.594 | 7.93314583151946E-05 |
| 3222401L1 | 2.67947939304363E-09 | -0.321430285930938 | 0.302 | 0.369 | 8.3205873592184E-05 |
| Lst1 | 2.70642616983557E-09 | -0.321026851479065 | 0.456 | 0.512 | 8.4042651851904E-05 |
| Snta1 | 3.04046972609653E-09 | -0.30668878769722 | 0.311 | 0.373 | 9.44157064044754E-05 |
| Gp9 | 3.14779804987945E-09 | -0.329360237354145 | 0.202 | 0.275 | 9.77485728429065E-05 |
| Dok3 | 3.56706647679233E-09 | -0.327151945694815 | 0.142 | 0.213 | 0.000110768115303832 |
| Dusp6 | 3.85249902300743E-09 | -0.263184647659365 | 0.606 | 0.651 | 0.00011963165216145 |
| Rtn1 | 4.07994756406942E-09 | -0.283531289630688 | 0.278 | 0.354 | 0.000126694611707048 |
| Thap3 | 4.17306608039058E-09 | -0.305824209631277 | 0.246 | 0.312 | 0.000129586220994369 |
| Slc25a37 | 4.26781637603559E-09 | -0.314852295544989 | 0.171 | 0.244 | 0.000132528501925033 |
| Ulk2 | 4.29804749750075E-09 | -0.305790960168544 | 0.303 | 0.37 | 0.000133467268939891 |
| Rbm5 | 4.99409704957006E-09 | -0.288818911346762 | 0.547 | 0.577 | 0.000155081695680299 |
| Lifr | 5.69080121467615E-09 | -0.326907778629069 | 0.153 | 0.225 | 0.000176716450119338 |
| Gpr84 | 5.69361806580752E-09 | 0.305274892722692 | 0.464 | 0.355 | 0.000176803921797521 |
| Trim12a | 5.98290448136089E-09 | -0.30733731278951 | 0.264 | 0.335 | 0.0001857871328597 |
| Kat6a | 8.69973674686093E-09 | -0.324517732144981 | 0.237 | 0.309 | 0.000270152925200273 |
| Cdk5r1 | 8.98624730331975E-09 | -0.316809051500348 | 0.214 | 0.287 | 0.000279049937509988 |
| Jun | 9.72891974385823E-09 | -0.502742942226946 | 0.465 | 0.527 | 0.00030211214480603 |
| Slc29a3 | 1.14777218628046E-08 | -0.2547779958623 | 0.64 | 0.659 | 0.000356417697005672 |
| Clasp2 | 1.31805649323952E-08 | -0.301406456275779 | 0.276 | 0.34 | 0.00040929608284567 |
| Hmox2 | 1.36708217816625E-08 | -0.275015978552862 | 0.448 | 0.488 | 0.000424520028785966 |
| Wdr44 | 1.37284708056473E-08 | -0.32166310016436 | 0.175 | 0.246 | 0.000426310203927765 |
| Prkcd | 1.72084609914569E-08 | -0.262432396844864 | 0.56 | 0.572 | 0.00053437433916771 |
| Ninj1 | 1.73510354497654E-08 | 0.330546243819882 | 0.513 | 0.403 | 0.000538801703821566 |
| Helz | 1.76319713827638E-08 | -0.334449057181911 | 0.246 | 0.309 | 0.000547525607348963 |
| Vgll4 | 1.85711993447595E-08 | -0.292733044204024 | 0.413 | 0.454 | 0.000576691453252817 |

(*Continued*)

**Table 2.** (Continued)

|  | p_val | avg_log2FC | pct.1 | pct.2 | p_val_adj |
|---|---|---|---|---|---|
| Cmklr1 | 1.93408850339667E-08 | -0.294728037402751 | 0.263 | 0.336 | 0.000600592502959769 |
| Slc25a45 | 1.93438409323988E-08 | -0.290863520218869 | 0.295 | 0.356 | 0.000600684292473782 |
| 4933406I18 | 2.18394371348499E-08 | -0.275261516049534 | 0.118 | 0.188 | 0.000678180041348493 |
| Asb2 | 2.18671373638281E-08 | -0.299346157089326 | 0.23 | 0.305 | 0.000679040216558955 |
| Trio | 2.22753220855024E-08 | -0.323941419928939 | 0.221 | 0.286 | 0.000691715576721107 |
| Mapk14 | 2.49750065185523E-08 | -0.296487919667262 | 0.225 | 0.292 | 0.000775548877420605 |
| Irf2bpl | 2.56931716590577E-08 | -0.305391879595496 | 0.336 | 0.397 | 0.000797850059528718 |
| Fcgr2b | 2.91806527644632E-08 | -0.292263177657661 | 0.504 | 0.524 | 0.000906146810294875 |
| Nfe2l2 | 3.08390951647958E-08 | 0.255846904988063 | 0.701 | 0.592 | 0.000957646422152404 |
| Gm32849 | 3.17602353262688E-08 | -0.258035440493209 | 0.109 | 0.177 | 0.000986250587586626 |
| Cdk19 | 3.8481379302782E-08 | -0.289019152418222 | 0.135 | 0.201 | 0.00119496227148929 |
| Kdm2b | 4.03006532429599E-08 | -0.290002965755192 | 0.269 | 0.337 | 0.00125145618515363 |
| Csad | 4.49526931525609E-08 | -0.310745062991901 | 0.323 | 0.383 | 0.00139591598046648 |
| Ppcdc | 5.81807564676372E-08 | -0.265522257315796 | 0.563 | 0.584 | 0.00180668703058954 |
| Zfp691 | 6.05144940667064E-08 | -0.287794053694264 | 0.222 | 0.293 | 0.00187915658425343 |
| Pdk1 | 6.610897270047E-08 | -0.301507254693624 | 0.28 | 0.337 | 0.00205288192926769 |
| Abl1 | 6.6157359156016E-08 | -0.281200641230685 | 0.232 | 0.297 | 0.00205438447387176 |
| Arhgap12 | 6.91442150002903E-08 | -0.299920602583469 | 0.268 | 0.328 | 0.00214713530840402 |
| Inpp4b | 6.99898231231546E-08 | -0.293236091592834 | 0.212 | 0.281 | 0.00217339397744332 |
| Ets1 | 8.12291339213466E-08 | -0.258129291601876 | 0.265 | 0.331 | 0.00252240829565958 |
| Dbnl | 8.15076638964446E-08 | -0.306365293418463 | 0.365 | 0.411 | 0.00253105748697629 |
| Ddx6 | 8.63287689751023E-08 | -0.286138422873663 | 0.382 | 0.43 | 0.00268076726298385 |
| Fcho2 | 8.99912031948466E-08 | -0.268905607061916 | 0.422 | 0.466 | 0.00279449683280957 |
| Cd2ap | 1.16686048984007E-07 | -0.27126250567033 | 0.412 | 0.454 | 0.00362345187910038 |
| Gmfg | 1.17298689892795E-07 | -0.281594406648446 | 0.327 | 0.381 | 0.00364247621724097 |
| Gabarapl2 | 1.23776220455145E-07 | -0.284462602750837 | 0.509 | 0.524 | 0.00384362297379361 |
| Etv5 | 1.27843959723451E-07 | -0.295418034097322 | 0.213 | 0.273 | 0.00396993848129234 |
| Lcp2 | 1.31690040928617E-07 | -0.2723629018795 | 0.318 | 0.376 | 0.00408937084095635 |
| Crebrf | 1.34588425133811E-07 | -0.276517819739463 | 0.172 | 0.24 | 0.00417937436568022 |
| Tbc1d9 | 1.38145060290491E-07 | -0.266388430315232 | 0.157 | 0.221 | 0.00428981855720062 |
| I830077J02 | 1.4637366069222E-07 | -0.30208590489188 | 0.228 | 0.291 | 0.00454534128547552 |
| Sh2b3 | 1.49622296908862E-07 | -0.280971926905668 | 0.306 | 0.36 | 0.00464622118591089 |
| Lpin2 | 1.55521845162501E-07 | -0.291460760232745 | 0.336 | 0.385 | 0.00482941985783114 |
| Cirbp | 1.55771296935472E-07 | -0.283304293742543 | 0.414 | 0.446 | 0.0048371660837372 |
| Cbl | 1.61780389166837E-07 | -0.264558963084066 | 0.394 | 0.44 | 0.00502376642479778 |
| Gm37494 | 1.63058760011155E-07 | -0.287468795740807 | 0.204 | 0.27 | 0.00506346367462639 |
| Gimap1 | 1.69961687886768E-07 | -0.272064366048699 | 0.136 | 0.2 | 0.00527782029394781 |
| Gng2 | 1.96746260200635E-07 | -0.29522924849199 | 0.316 | 0.365 | 0.0061095616180103 |
| Nipa2 | 1.97417047674926E-07 | -0.282142299807652 | 0.405 | 0.449 | 0.00613039158144947 |
| Nr3c1 | 2.00158840065216E-07 | -0.256541721340712 | 0.435 | 0.485 | 0.00621553246054517 |
| Xist | 2.17007356334849E-07 | -0.359478333487127 | 0.509 | 0.557 | 0.00673872943626608 |
| Arl10 | 2.17556299101516E-07 | -0.263628757101459 | 0.391 | 0.433 | 0.00675577575599939 |
| Cd79b | 2.23938168374618E-07 | -0.290858792235169 | 0.161 | 0.224 | 0.00695395194253703 |
| Zcchc11 | 2.2625448077169E-07 | -0.298796094943654 | 0.257 | 0.318 | 0.00702588039140329 |
| Hps3 | 2.34715786241391E-07 | -0.304186513367248 | 0.392 | 0.424 | 0.00728862931015391 |
| Fbxl20 | 2.37536294859142E-07 | -0.265026077491498 | 0.133 | 0.196 | 0.00737621456426093 |
| Tmem63a | 2.77714222325414E-07 | -0.268973762116124 | 0.333 | 0.385 | 0.00862385974587108 |

(*Continued*)

**Table 2.** (Continued)

| | p_val | avg_log2FC | pct.1 | pct.2 | p_val_adj |
|---|---|---|---|---|---|
| Prex1 | 2.83525925765431E-07 | -0.260451960982122 | 0.513 | 0.53 | 0.00880433057279394 |
| Eva1b | 2.87920381952709E-07 | -0.254077167852617 | 0.167 | 0.232 | 0.00894079162077747 |
| Fbrsl1 | 2.97270291232568E-07 | -0.274304507436561 | 0.302 | 0.355 | 0.00923113435364493 |
| Acads | 3.09890074406194E-07 | -0.257886565657563 | 0.407 | 0.448 | 0.00962301648053553 |
| Sort1 | 3.47223923378202E-07 | -0.289851456974882 | 0.287 | 0.336 | 0.0107823444926633 |
| Lrrc3 | 3.57488321265882E-07 | -0.280278370734734 | 0.302 | 0.358 | 0.0111010848402694 |
| Tbc1d23 | 3.63201324115574E-07 | -0.280473449415491 | 0.271 | 0.331 | 0.0112784907177609 |
| Per3 | 3.75119928883498E-07 | -0.251639323204655 | 0.096 | 0.156 | 0.0116485991516193 |
| Cdkn1b | 4.24874230795943E-07 | -0.252477946346461 | 0.138 | 0.2 | 0.0131936194889064 |
| AI467606 | 4.6053456777113E-07 | -0.259271591383804 | 0.137 | 0.199 | 0.0143009799329969 |
| Ppm1l | 5.01054988908223E-07 | -0.25323177671416 | 0.095 | 0.152 | 0.015559260570567 |
| Gmip | 5.36304216907414E-07 | -0.267194910392705 | 0.289 | 0.344 | 0.0166538548476259 |
| Pan3 | 5.48806520867436E-07 | -0.275472289304947 | 0.336 | 0.382 | 0.0170420888924965 |
| Ep300 | 6.06397927648205E-07 | -0.282757640694748 | 0.333 | 0.38 | 0.0188304748472597 |
| Trim26 | 6.37527759210646E-07 | -0.269228386584243 | 0.299 | 0.349 | 0.0197971495067682 |
| Mkrn1 | 7.33767432973374E-07 | -0.261907936073313 | 0.327 | 0.377 | 0.0227856800961222 |
| Borcs6 | 8.49438171344948E-07 | -0.282131008977835 | 0.204 | 0.258 | 0.0263776035347747 |
| Mkln1 | 8.91067448766098E-07 | -0.25150557483357 | 0.491 | 0.512 | 0.0276703174865336 |
| Ralgps2 | 9.81312479117344E-07 | -0.28420563063336 | 0.191 | 0.247 | 0.0304726964140309 |
| Camk2n1 | 9.81381163537971E-07 | -0.271861652410143 | 0.425 | 0.467 | 0.0304748292713446 |
| Arap3 | 9.92555302065236E-07 | -0.268218229797015 | 0.188 | 0.246 | 0.0308218197950318 |
| Gm13889 | 1.01792145397341E-06 | 0.38643275581675 | 0.111 | 0.068 | 0.0316095149102364 |
| Hps4 | 1.08518438324778E-06 | -0.257917137610395 | 0.467 | 0.478 | 0.0336982306529932 |
| Usf1 | 1.09585610937582E-06 | -0.279859027515085 | 0.26 | 0.309 | 0.0340296197644473 |
| Spred1 | 1.11749334199504E-06 | -0.278377273266421 | 0.249 | 0.303 | 0.0347015207489719 |
| Abcb4 | 1.23266801682812E-06 | -0.266543229285136 | 0.137 | 0.195 | 0.0382780399265637 |
| Iffo1 | 1.4803462029996E-06 | -0.257780126263516 | 0.351 | 0.399 | 0.0459691906417465 |
| Mgll | 1.53873254098203E-06 | -0.274564544984353 | 0.387 | 0.421 | 0.0477822615951149 |
| Chsy1 | 1.55153963505454E-06 | -0.264049531225795 | 0.153 | 0.21 | 0.0481799602873486 |

subventricular zone, these cells exhibit altered immunoreactivity to some common markers such as Tmem119 and Iba1 [18]. These cells, do however, express Cd68+ puncta, which are generally found only in low levels by microglia that are ramified or surveilling. It has been long known that microglia show spatial patterning in morphology and density across the brain [44]. These morphological differences often correlate with differential phagocytic activity, proliferative potential, and immunoreactivity as different regions of the brain have different needs.

In this study, we characterize the transcriptomic profile of these cells relative to other myeloid cells in the hippocampus at the level of single cell resolution. We identify a greater number of differentially expressed genes by DAM-like microglia, and separate confounding results from CNS-associated macrophages. This approach allows us to computationally separate different subsets of myeloid cells with great precision.

We first noted downregulation of several microglia-specific marker genes in the healthy, adult subgranular zone; strikingly similar to cases of disease and injury [67, 68]. Due to the lower levels of expression of these marker genes, these cells may not be captured by conventional flow cytometry panels, requiring judicious use of marker genes depending on spatial, temporal, and associated disease context. We classify myeloid cells in the SGZ as microglia as

they still retain expression, albeit to a lesser degree, of some microglia-specific genes while maintaining robust expression of other marker genes such as *Hexb* and *Olfm3*. This further corroborates findings which suggest not all microglia-specific markers retain consistent expression and comprehensively label microglia in the brain [67].

Next, we compared genes upregulated in the SGZ to known microglial phenotypes. Studies have reported that microglia described as alternatively activated M2 microglia in the macrophage polarization scheme promote neurogenesis [69–71]. Microglia in the dentate gyrus have also been reported to express a handful of genes in M2 microglia. Our results indicate that this profile does not accurately capture the SGZ transcriptome. Instead, we observe great overlap of genes enriched in our putative SGZ cluster with those upregulated in disease-associated microglia (DAM), which are involved in plaque clearance [8, 72]. This DAM signature is translationally relevant and potentially highly conserved, as it has been observed in postmortem human tissue from patients with AD and MS [9, 10].

Many of these DAM genes are also found in previously described transcriptome profiles or cell states such as proliferation-associated macrophages (PAM) found in developing white matter and early postnatal microglia [8, 15]. These are primarily genes which are associated with lysosomal function. In PAM these genes reflect phagocytosis of oligodendrocyte progenitors, important in myelination, and are no longer present in the mature, adult brain. The presence of these genes in the adult hippocampus may result from engulfment of excess neural progenitors that are undergoing apoptosis [19, 73]. This points to a broader role of DAM genes extending beyond disease, particularly since phagocytic microglia have been shown to support neuronal development in the adult hippocampus [19].

During early postnatal development, developing neurons are pruned through a CD11B-DAP12-dependent mechanism [74]. Interestingly, cluster 8 also shows upregulation of *Dap12* (also known as *Tyrobp*), which is activated downstream of the DAM-specific receptor TREM2. Our data suggest that processes similar to those found in embryonic development of the nervous system as well as in PAMs during postnatal development persist throughout adulthood in the hippocampus. This population specifically may be targeted in diseases in which neurogenesis is perturbed or needs to be altered [21, 27]. Notably, gene networks associated with phagocytic microglia are involved in neurogenic function *in vivo* within the neurogenic niche [19].

We also ruled out whether transcriptomic differences in SGZ microglia could explain how sex-specific differences may contribute to altered risk for neurodevelopment diseases, particularly those affecting hippocampal function and neurogenesis [75–78]. In the healthy adult murine hippocampus, there appear to be no sex-related differences in the transcription of immune -related genes Additionally, no significant differences in genes related to immune function are found specifically in SGZ microglia from male and female mice. Thus, differences in immune function that occur may come into play later in life or are due to post-transcriptional regulation or at the level of protein interactions.

Adult hippocampal neurogenesis is sensitive to various external or environmental factors. As such, the immune cells in this niche are tuned to varying inputs and may be responsible for relaying the state of the outside world to progenitors. Important questions that remain to be answered are whether this specialized microglial phenotype is indicative of a distinct ontogeny, or whether it arises as a response to the environmental cues provided by the niche. Furthermore, whether this signature describes an activated population or a reactive state is not known and of great interest [79]. This is important to uncover because it can elucidate if alterations in adult neurogenesis in pathological contexts result from differential properties of immune cells in the SGZ. Given that these cells express many genes found in the DAM gene signature, the role of these genes specifically in the context of adult neurogenesis need to be examined.

While some genes expressed by microglia have been characterized in various developmental stages and disease models, many of the genes enriched by subgranular zone microglia have unknown functions. Our findings point to a need to separate the SGZ population and examine it separately in disease models. Extracting SGZ-specific differences is a necessary component of understanding hippocampal physiology, particularly from an immune perspective.

Further understanding the mechanisms behind how this population progresses and the trajectory it takes will be fundamentally important, particularly in the context of aging and disease [77, 80] While we highlight one specific subset of cells in this paper, the potential contributions of other populations cannot be excluded from regulation of the neurogenic niche or in disease development and progression. Our data provide a reference point for comparing immune alterations in the hippocampus and its neurogenic niche in the context of disease or other pathology.

## Supporting information

**S1 Fig.** A. Generation of double reporter mouse line. Schematic displaying breeding schemes and selection of breeders to generate mice expressing eGFP in Nestin+ neural progenitor cells and tdTomato in cells from fractalkine (Cx3Cr1) expressing cells and their daughter cells. B. Percentage of cells showing dual expression of Tmem119 and tdTomato, Tmem119 only, TdTomato only, or neither. Results are shown for samples derived from transgenic mouse or wildtype mouse (C57/6Bl). "Count" column contains number of cells per sample recorded. C. Percentage of TdTomato+ cells with CD11B and CD45 expression. Unstained samples (WT unstained, TdTomato unstained) are negative controls for antibody staining. D. Isotype control stainings to show nonspecific background binding for FITC and APC conjugated antibodies.
(PDF)

**S2 Fig. Quality control (QC) metrics pre-processing.** (A) Violin Plots displaying transcript reads, number of unique genes, percentage of reads corresponding to mitochondrial genes, and percentage of reads corresponding to ribosomal genes in each sequencing sample. Dashed line corresponds to lower limit for filtering cells, solid lines represent maximum values for filtering. (B) UMAP plot displaying cell cycle scoring for cells in dataset. (C) UMAP plot displaying predicted doublets based on DoubletFinder. Predicted doublets are highlighted in magenta.
(PDF)

**S3 Fig.** A. Breakdown of subpopulation frequencies within clusters. B. QC violin plots showing number unique genes (nFeature_RNA) and number of reads (nCount_RNA) in each cluster. C. Elbow plot showing variance represented by principal components. D. Dot Plot of myeloid lineage marker genes.
(PDF)

**S4 Fig. Heat Map displaying scaled expression levels of top five or fewer differentially expressed genes based on set thresholds (P$_{adj}$ <0.001 and expressed in at least 70% of cells in cluster).** Thirty cells were sampled for each cluster. Each vertical line represents scaled z-score value of gene expression across row.
(PDF)

**S5 Fig. Sex-specific differences in hippocampal myeloid cells.** (A) Feature plot displaying scaled expression of *Xist* projected onto UMAP. (B) Distribution of cells *Xist*+ female cells versus *Xist*—male cells across clusters. (C) Violin Plots displaying genes differentially expressed

between bulk female (F) and Male (M) cells. (D) Violin plots displaying genes differentially expressed between female and male cells in SGZ cluster 8.
(PDF)

**S6 Fig. Distribution of differentially expressed genes in cluster 8 in the dentate gyrus.** *In situ hybridization* images from the Allen Institute of genes enriched in cluster 8 (top three) and marker genes (*Hexb*) or genes associated with homeostatic microgglia (*Selplg*, *P2ry12*). For enriched genes, hemicortices (left) are shown to display level of specificity of these genes in the SGZ with more zoomed in fields of view alongside them (right).
(PDF)

**S7 Fig.** Cd9 immunoreactivity in dual reporter mice in individual channels (top) show expression of Cd9, tdTomato+ myeloid cells and eGFP+ neural progenitor cells. Merge images (middle) show colocalization of Cd9 in both tdTomato and GFP positive cells (left middle). Diffuse immunoreactivity in cortex (bottom) Scale = 10 μm.
(PDF)

**S8 Fig.** A. DAM marker gene expression levels in clusters 8, 12, and 13 of hippocampal myeloid cells. B. Correlation matrix showing split of Cross-dataset comparisons between cluster 8 microglia and reactive microglia. Plot showing relation between clusters 8,12,13 from this study and integrated dataset from Keren-Shaul et. al. 2017. Scale represents fraction of original cluster (from hippocampal myeloid dataset) represented in each integrated cluster from Fig 5E. C. Overlap of upregulated cluster 8, DAM, and early postnatal microglia (Hammond et. al. Cluster 3). D. Feature plots showing localization of *Ccl3*, *Igf1*, *and Lgals3* as examples of genes enriched in reactive microglia from the intersection of genes featured in the venn diagram (part C of this figure).
(PDF)

**S1 Table. Quality control (QC) metrics pre and post- processing.** Table displaying cells, mean transcript reads, mean number of unique genes, and the mean percentage of reads corresponding to mitochondrial genes in each sequencing sample.
(DOCX)

**S2 Table. Gene ontology analysis of genes enriched in SGZ vs homeostatic clusters.**
(DOCX)

## Author Contributions

**Conceptualization:** Elizabeth M. Bradshaw, Vilas Menon, Steven G. Kernie.

**Data curation:** Sana Chintamen, Pallavi Gaur, Vilas Menon.

**Formal analysis:** Sana Chintamen, Pallavi Gaur, Vilas Menon.

**Funding acquisition:** Steven G. Kernie.

**Investigation:** Sana Chintamen.

**Methodology:** Nicole Vo, Steven G. Kernie.

**Supervision:** Elizabeth M. Bradshaw, Vilas Menon, Steven G. Kernie.

**Writing – original draft:** Sana Chintamen.

**Writing – review & editing:** Sana Chintamen, Pallavi Gaur, Steven G. Kernie.

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
