## [Author Response · Author response to previous submission]

10 Jun 2023

to Reviewers_PLOS_6_2023.docx
---

## [Decision Letter · Decision Letter 0]

11 Jul 2023

PONE-D-23-16545Unique Microglial
Transcriptomic Signature within the Hippocampal Neurogenic
NichePLOS ONE

Dear Dr. Kernie,

Thank you for submitting your manuscript to PLOS ONE. After careful consideration, we
feel that it has merit but does not fully meet PLOS ONE’s publication criteria as it
currently stands. Therefore, we invite you to submit a revised version of the
manuscript that addresses the points raised during the review process.

Please submit your revised manuscript by Aug 25 2023 11:59PM. If you will need more
time than this to complete your revisions, please reply to this message or contact
the journal office at plosone@plos.org. When
you're ready to submit your revision, log on to https://www.editorialmanager.com/pone/ and select the 'Submissions
Needing Revision' folder to locate your manuscript file.

Please include the following items when submitting your revised
manuscript:A rebuttal letter that responds to each point raised by the academic
editor and reviewer(s). You should upload this letter as a separate file
labeled 'Response to Reviewers'.A marked-up copy of your manuscript that highlights changes made to the
original version. You should upload this as a separate file labeled
'Revised Manuscript with Track Changes'.An unmarked version of your revised paper without tracked changes. You
should upload this as a separate file labeled 'Manuscript'.

If you would like to make changes to your financial disclosure, please include your
updated statement in your cover letter. Guidelines for resubmitting your figure
files are available below the reviewer comments at the end of this letter.

We look forward to receiving your revised manuscript.

Kind regards,

Giuseppe Biagini, MD

Academic Editor

PLOS ONE

Journal Requirements:

 "This research was supported by National Institutes of Health/National Institute of
Neurological Disorders and Stroke Grants R01-NS-095803 (SGK) and the Paul Allen
Foundation (SGK), and National Institute of Aging Grant R01-AG-066831 (VM). This
research was funded in part through the National Institute of Health and National
Cancer Institute (NIH/NCI) Cancer Center Support Grant P30CA013696 and used the
Genomics and High Throughput Screening Shared Resource. "

"This research was supported by National Institutes of Health/National Institute of
Neurological Disorders and Stroke Grants R01-NS-095803 (SGK) and the Paul Allen
Foundation (SGK), and National Institute of Aging Grant R01-AG-066831 (VM). This
research was funded in part through the National Institute of Health and National
Cancer Institute (NIH/NCI) Cancer Center Support Grant P30CA013696 and used the
Genomics and High Throughput Screening Shared Resource. Research reported in this
publication was performed in the CCTI Flow Cytometry Core, supported in part by the
Office of the Director, National Institutes of Health under awards S10OD020056. The
content is solely the responsibility of the authors and does not necessarily
represent the official views of the National Institutes of Health. Images were
collected and/or image processing and analysis for this work was performed in the
Confocal and Specialized Microscopy Shared Resource of the Herbert Irving
Comprehensive Cancer Center at Columbia University, supported by NIH grant #P30
CA013696 (National Cancer Institute)."

"This research was supported by National Institutes of Health/National Institute of
Neurological Disorders and Stroke Grants R01-NS-095803 (SGK) and the Paul Allen
Foundation (SGK), and National Institute of Aging Grant R01-AG-066831 (VM). This
research was funded in part through the National Institute of Health and National
Cancer Institute (NIH/NCI) Cancer Center Support Grant P30CA013696 and used the
Genomics and High Throughput Screening Shared Resource. "

Reviewers' comments:

Reviewer's Responses to Questions

**Comments to the Author**

1. Is the manuscript technically sound, and do the data support the conclusions?

Reviewer #1: Partly

Reviewer #2: Partly

2. Has the statistical analysis been performed
appropriately and rigorously? 

Reviewer #1: I Don't Know

Reviewer #2: Yes

3. Have the authors made all data underlying the
findings in their manuscript fully available?

Reviewer #1: Yes

Reviewer #2: Yes

4. Is the manuscript presented in an intelligible
fashion and written in standard English?

Reviewer #1: Yes

Reviewer #2: Yes

5. Review Comments to the Author

Reviewer #1: Chintamen et al., explore the diversity of the myeloid cells in the
hippocampus, which are known to be extremely heterogenous both in morphology and
gene expression profile. They enrich for myeloid cells utilizing a tdTomato
construct, applying single-cell RNA sequencing on 12 mouse hippocampi, and revealing
a rare microglia subpopulation with high Cd68 expression and, possibly, specific
Cd63 expression (cluster 8). Indirect evidence of immunohistochemistry of cluster 8
markers, suggests this subpopulation is localized in dentate gyrus subgranular zone.
Cluster markers are a mix of mostly DAM genes and some genes suggesting neurogenesis
and synaptic pruning. The last figure aims to prove that the morphology of the SGZ
microglia is distinct from homeostatic microglia and that cluster 8 is
transcriptionally similar to DAMs. This study reports a highly relevant findings for
any scientist working on the hippocampus and raises an extremely interesting point
of SGZ specialized microglia. The efforts that are made to connect a transcriptional
single-cell cluster to a biological function in the SGZ is appreciated. However,
especially for figure 3 and 5, which are conceptionally the most interesting, I have
major concerns to address:

Figure 5A-C: The argument made in figure 5A is that microglia morphology in the
dentate gyrus are different from microglia in the cortex. This is quite unsurprising
and does not support the following claim, which conains a comparison between
hippocampal microglia:

‘These results are in accordance with transcriptome analyses that suggest microglia
display an alternative phenotype in the neurogenic niche’

In short, the claim is about differences within the hippocampus, the plots are
comparing cortex and dentate gyrus. Figure 5 is also not convincingly proving a
phagocytic function of SGZ microglia. In my opinion, Fig 5A-C have no relevance
regarding the message about hippocampal diversity in myeloid cells and would be more
suitable to place in a supplement.

It would be highly interesting and valuable to instead analyze morphology in figure 2
images (tdTomato) for the different hippocampal zones (SGZ, GL, GM). Direct evidence
for altered morphology in the SGZ compared to the GL and GM would be a convincing
argument that supports the claim.

Figure 5E: The results from integrated keren-shaul and hippocampal microglia are a
valuable addition to this paper adding a relevant perspective on what a “DAM”
signature actually represents. However, I was struggling to understand the
integrated UMAP presented in figure 5E and relate it back to figure 1D. A plot is
missing that shows the distribution of cluster 8 microglia (potential SGZ microglia)
in the integrated data. The only message figure 5E is currently conveying, is that
the abundance of cluster 7, 9 and 10 is altered between the two datasets. There are
no abundance differences visible in cluster 5.

In short, the claim in line 468: “found that microglia exclusively from cluster 8
with DAM” is not supported.

Cluster 5 seems to be the DAM phenotype, based on the marker expression in figure 5F,
however, it is not shown how integrated cluster 5 (DAM) is linked to original
cluster 8 (hippocampal SGZ microglia). My suggestion is that the authors could show
the UMAP with all integrated data but color the cells as non-cluster 8 (grey) and
cluster 8 (red) to indicate the hippocampal SGZ microglia. It is essential to
visualize the abundance and distribution of the cluster 8 “SGZ” microglia in the
integrated data.

Figure 3: The volcano plot in figure 3A clearly illustrates the point that cluster 8
is expressing DAM genes. The top GO terms should be microglia-related, such as
innate immune response , antigen presentation, and cytokine production. The fact
that only a selection of GO terms related to neurons is shown, while omitting the
top GO terms that are most significant, is misleading.

I would strongly suggest to explain when and why a selection of GO terms is
displayed. It could be improved by showing both the most significant GO term
enrichment and the ‘cherry-picked’ GO terms to make it clear to the reader. However,
the finding does raise an interesting scientific point.. which genes are part of GO
term “neurogenesis”, “neuron development”, “synapse organization” ( I did not find
this in S Table 2??) and are these genes specific to neurons? Where are they in the
volcano plot? Is it possible these genes are not expressed by microglia at all; this
would suggest that microglia phagocytose apoptotic and immature neurons in the SGZ
and that neuron-derived RNA is detected in your microglia transcriptome. Reports
using RiboTag suggest this could be an explanation (PMID: 29777220) and could be
used to extend the argument made in line 417-422. Some neuronal genes could be added
to figure 3C.

Optional, out of interest, neuronal RNA in the differential gene list could be
further investigated using a simple webtool https://www.syngoportal.org/ .

Figure 1C: Please do NOT show fluoresence intensity, as it is very hard to compare
the different antibody results. Please DO show the percentage of cells that is
positive for Tmem119 (and other markers) and tdTomato in comparison to a control.
How many cells are triple positive for Tmem119+CD45+CD11b+ and how many cells are
quadruple positive Tmem119+CD45+CD11b+tdTomato+?

Negative controls using unstained cells, single staining controls and fluorescent
minus one (FMO) controls are not shown. These are important for setting parameters
of true positive and negative cell populations. Otherwise, if lacking, please show
ungated populations that serve as “negative” cells (e.g. non-microglia cells should
be negative for CD45).

Minor comments:

Line 60-61 versus line 71-76: please be consistent on terms “microglia”, “myeloid
lineage cells” and “immune cells”. Could the authors make an effort to distinguish
microglia, CNS-associated macrophages, and infiltrating immune cells in the
introduction? For example, in line 71 it appears to me microglia are grouped under
immune cells whereas in line 60 microglia are specifically not immune cells.

Line 226: methods: Why was Harmony used for the data integration with the Keren-Shaul
dataset, whereas previously (line 189) Seurat’s IntegrateData with CCA was used to
integrate samples and correct technical effects? What is the rationale for this
different methodology? Harmony strongly alters the gene expression information and
clustering results might be even more overlapping if the CCA approach is used for
the Keren-Shaul study.

Line 339: Clusters 4,5,6, 7 and 11 failed to show enrichment of at least ten nuclear
genes. This raises the concern this single-cell dataset is a bit overclustered.
However, I understand it cannot be changed at this point in the project. Cluster 4
expresses mitochondrial-derived markers, suggesting this is a stressed, less viable
cluster of cells that should be removed. Cluster 5 and 6 are appropriately addressed
in the text. In my opinion cluster 7 and 11 could be merged to one cluster. This is
sometimes done in single-cell analysis to revert the overclustering (It can be done
by simply giving them the same label in cluster annotation using
Seurat::RenameIdents()).

Line 408: incorrect reference, should reference supplemental figure 7

Supplemental figure 4: the heatmap is very blurred.

Figura 1A: this is outside the scope of the current study, but relevant for follow-up
work using the same reporter. Did you check tdTomato signal in mice without a TAM
injection? Unfortunately, Rosa26-floxP-tdTomato constructs can be leaky due to the
short stop codon, please refer to PMID: 32125704 “STOP floxing around: specificity
and leakiness of inducible Cre/loxP systems” to verify the validity of your strain.
For the main message and findings of this paper I do not believe it matters.

Please reconsider the manuscript title “Unique Microglial Transcriptomic Signature
within the Hippocampal Neurogenic Niche”, since the single-cell data does not prove
the microglia cluster 8 is unique to the SGZ nor is there sufficient evidence from
immunohistochemistry to localize it to the neurogenic niche. A more suitable
alternative could be “Distinct microglial transcriptomic signatures within the
hippocampus”

Reviewer #2: Chintamen et al., used single-cell RNAseq to explore the transcriptional
profiles of myeloid lineage cells isolated from the hippocampus. The authors noted
microglial heterogeneity and highlighted an unique population (cluster 8) of
microglia that demonstrate higher level of phagocytosis-related genes. Based on
histological analysis of selected markers, the authors have attributed this
population of microglia to a neurogenic niche in the subgranular zone of hippocampus
dentate gyrus.

Major concern

1. While the authors have cited other single-cell RNAseq studies on microglia, they
have not rigorously compared their results to these studies. I appreciate that they
have compared it with Keren-Shaul et al. (2017). However, from what I can gather
from their differential gene expression tables, their cluster 8 is also very similar
to an early postnatal microglia population described by Hammond et al. 2019 (cluster
3) .

2. Similarly, in line 374 and 375, the authors mentioned that a ribosomal enriched
cluster that has been reported but not characterised, but the observation has not
only been reported by multiple studies, it has been reviewed (https://www.ncbi.nlm.nih.gov/pmc/articles/PMC8225243/). These are
just some of the existing literature that I was able to identify with relatively few
searches. Along with the lack of mentions of bulk-RNAseq work that highlighted
microglial heterogeneity across the brain, my main concern is whether other
information and links to existing work were also missed and that the authors have
overstated their findings due to insufficient comparison to existing work.

3. If the authors want to keep clusters with upregulation on mitochondrial genes and
not regress out cell cycle genes, they should really include more quality control
figures (i.e. plots showing nCount_RNA, nFeature_RNA, % ribosomal RNA, %
mitochondrial RNA, cell cycle scoring (you can get that from using CellCycleScoring
in Seurat). It is difficult to review and assess their quality control.

4. While I do not believe Cluster 8 is doublet and the authors have included a figure
showing the lack of clusters demonstrating doublets in their response to other
reviewers, I have two issues with the manuscript in its current form. 1) Even if
there isn’t a significant pattern in doublets, the figure shown to the reviewer
should be included in the supplementary. It should also be demonstrated that the
removal of doublets identified by Doubletfinder does not significantly improve the
cluster separation. 2) Doubletfinder works by averaging the transcriptional profiles
of randomly selected cell pairs. If there were to be doublet in this dataset, part
of the doublet pair could be neuronal. However, as the cells were FACs sorted, the
majority of the neuronal cells would not be present in the dataset, therefore
Doubletfinder would not be able to generate a good representation of doublet
transcriptional profiles. In order to more accurately remove doublets in this
dataset, you would need integrate a single cell RNAseq dataset with non-FACS sorted,
then carry out doublet detection, remove all doublet and then keep the singlet cells
from this study. I suspect doublets might be the source of the small fraction of
expression of other cell type markers indicated in line 309-311. While I don’t
expect this level of doublet removal, I do not think the response to Reviewer 2’s
comment 4 was sufficient and at least 1) should be incorporated in the
manuscript.

5. It would be useful to see Allen Brain atlas showing the Cluster 8 upregulated gene
expression in other brain regions to see if they are specific to the hippocampus
region.

6. The rational of selecting 20 PCs is unclear – is a Jackson straw plot or elbow
plot made to determine this number?

Minor concerns

1. It is not clear to me what clusters Figure 5E are referring to and how they relate
to the main clustering in this study. The comparison to Keren-Shaul et al’s work is
generally not well described and readers cannot easily follow the rationale.

2. Line 530 – citation style change

3. Line 441 – did you mean Supp Figure 7?

4. Figure 1E and 5F – the macrophage clusters should be removed from these figures
(and placed in supplementary) to allow the colour scale to reflect a wider range of
gene expression.

6. PLOS authors have the option to publish the peer
review history of their article (what does this mean?). If published, this will
include your full peer review and any attached files.

If you choose “no”, your identity will remain anonymous but your review may still be
made public.

**Do you want your identity to be public for this peer review?** For
information about this choice, including consent withdrawal, please see our
Privacy Policy.

Reviewer #1: No

Reviewer #2: No

---

## [Author Response · Author response to Decision Letter 0]

5 Sep 2023

PONE-D-23-16545

Unique Microglial Transcriptomic Signature within the Hippocampal Neurogenic
Niche

PLOS ONE

Reviewers' comments:

Reviewer's Responses to Questions

Comments to the Author

1. Is the manuscript technically sound, and do the data support the conclusions?

Reviewer #1: Partly

Reviewer #2: Partly

2. Has the statistical analysis been performed appropriately and rigorously?

Reviewer #1: I Don't Know

Reviewer #2: Yes

3. Have the authors made all data underlying the findings in their manuscript fully
available?

Reviewer #1: Yes

Reviewer #2: Yes

4. Is the manuscript presented in an intelligible fashion and written in standard
English?

Reviewer #1: Yes

Reviewer #2: Yes

5. Review Comments to the Author

Reviewer #1: Chintamen et al., explore the diversity of the myeloid cells in the
hippocampus, which are known to be extremely heterogenous both in morphology and
gene expression profile. They enrich for myeloid cells utilizing a tdTomato
construct, applying single-cell RNA sequencing on 12 mouse hippocampi, and revealing
a rare microglia subpopulation with high Cd68 expression and, possibly, specific
Cd63 expression (cluster 8). Indirect evidence of immunohistochemistry of cluster 8
markers, suggests this subpopulation is localized in dentate gyrus subgranular zone.
Cluster markers are a mix of mostly DAM genes and some genes suggesting neurogenesis
and synaptic pruning. The last figure aims to prove that the morphology of the SGZ
microglia is distinct from homeostatic microglia and that cluster 8 is
transcriptionally similar to DAMs. This study reports a highly relevant findings for
any scientist working on the hippocampus and raises an extremely interesting point
of SGZ specialized microglia. The efforts that are made to connect a transcriptional
single-cell cluster to a biological function in the SGZ is appreciated. However,
especially for figure 3 and 5, which are conceptionally the most interesting, I have
major concerns to address:

Figure 5A-C: The argument made in figure 5A is that microglia morphology in the
dentate gyrus are different from microglia in the cortex. This is quite unsurprising
and does not support the following claim, which contains a comparison between
hippocampal microglia:

‘These results are in accordance with transcriptome analyses that suggest microglia
display an alternative phenotype in the neurogenic niche. 

We thank the reviewer for the feedback. While the majority of our manuscript focuses
on differences between transcriptionally defined clusters within the hippocampus,
our intention with Figure 5 is to highlight the differences between SGZ microglia
and well characterized microglia. As cortical microglia have been most
characterized, we sought to specifically compare the traits of our cluster 8
microglia with these more well characterized microglia. The genesis for this study
was from the observation that SGZ microglia which are located in a clear spatially
confined niche have very different properties from cells traditionally thought to be
homeostatic microglia. The fact that they have in a resting state properties and
marker gene expression similar to those found in disease is important to point out
because this reflects regional heterogeneity, which is not well understood in the
adult brain. For this reason, we took a molecular approach to characterizing the
cells within the hippocampus. Since cortical microglia are highly represented within
whole brain homogenates and have been examined by others such as Hammond et. al.
2019, Li et. al. 2019, we excluded them from the RNA seq experiments for this study. 

In short, the claim is about differences within the hippocampus, the plots are
comparing cortex and dentate gyrus. Figure 5 is also not convincingly proving a
phagocytic function of SGZ microglia. In my opinion, Fig 5A-C have no relevance
regarding the message about hippocampal diversity in myeloid cells and would be more
suitable to place in a supplement.

It would be highly interesting and valuable to instead analyze morphology in figure 2
images (tdTomato) for the different hippocampal zones (SGZ, GL, GM). Direct evidence
for altered morphology in the SGZ compared to the GL and GM would be a convincing
argument that supports the claim.

Figure 5E: The results from integrated keren-shaul and hippocampal microglia are a
valuable addition to this paper adding a relevant perspective on what a “DAM”
signature actually represents. However, I was struggling to understand the
integrated UMAP presented in figure 5E and relate it back to figure 1D. A plot is
missing that shows the distribution of cluster 8 microglia (potential SGZ microglia)
in the integrated data. The only message figure 5E is currently conveying, is that
the abundance of cluster 7, 9 and 10 is altered between the two datasets. There are
no abundance differences visible in cluster 5.

In short, the claim in line 468: “found that microglia exclusively from cluster 8
with DAM” is not supported.

Cluster 5 seems to be the DAM phenotype, based on the marker expression in figure 5F,
however, it is not shown how integrated cluster 5 (DAM) is linked to original
cluster 8 (hippocampal SGZ microglia). My suggestion is that the authors could show
the UMAP with all integrated data but color the cells as non-cluster 8 (grey) and
cluster 8 (red) to indicate the hippocampal SGZ microglia. It is essential to
visualize the abundance and distribution of the cluster 8 “SGZ” microglia in the
integrated data.

We appreciate these insights and agree that it is important to visualize the
abundance and distribution of these microglia. We have added description in the text
to better explain the motivation for the morphological characterizations.
Qualitatively, one can observe consistent and prominent differences in the SGZ. The
presence of a subset of cells that appeared to have an alternative morphology
compared to cortical microglia is what motivated us to directly compare results to a
study that specifically examine a distinct phenotypic state. Since cortical
microglia are in high abundance and likely dominate the results for characteristic
traits in homeostatic microglia, we chose to compare SGZ microglia with cortical
microglia for morphological characterization. In order to clarify the rationale for
these comparisons to cortical microglia, we have included additional text in lines
576-581 (see below).

“As the cortex is the largest region of the murine brain, microglia from this region
represent the highest number of any one region. While the distribution of microglia
in the resting cortex is characterized as tiled with microglial branches from
adjacent cells maintaining distinct, non-overlapping territories, some regions of
the brain do not have this patterning, particularly in regions with high neuronal
densities [1, 2]”

We have also now provided a correlation matrix in supp fig 8 to show that the cluster
5 in the integrated dataset corresponds to cluster 8 from figure 1D. The UMAP in
figure 5E is a representation of the integrated dataset split by Hippocampal
microglia and Keren-Shaul data to show the DAM signature via the cluster 5.
Additionally, we have now included the plot with cells from cluster 8 highlighted
which the reviewer has asked for in Figure 5D. 

Figure 3: The volcano plot in figure 3A clearly illustrates the point that cluster 8
is expressing DAM genes. The top GO terms should be microglia-related, such as
innate immune response , antigen presentation, and cytokine production. The fact
that only a selection of GO terms related to neurons is shown, while omitting the
top GO terms that are most significant, is misleading.

I would strongly suggest to explain when and why a selection of GO terms is
displayed. It could be improved by showing both the most significant GO term
enrichment and the ‘cherry-picked’ GO terms to make it clear to the reader. However,
the finding does raise an interesting scientific point.. which genes are part of GO
term “neurogenesis”, “neuron development”, “synapse organization” ( I did not find
this in S Table 2??) and are these genes specific to neurons? Where are they in the
volcano plot? Is it possible these genes are not expressed by microglia at all; this
would suggest that microglia phagocytose apoptotic and immature neurons in the SGZ
and that neuron-derived RNA is detected in your microglia transcriptome. Reports
using RiboTag suggest this could be an explanation (PMID: 29777220) and could be
used to extend the argument made in line 417-422. Some neuronal genes could be added
to figure 3C.

Optional, out of interest, neuronal RNA in the differential gene list could be
further investigated using a simple webtool https://www.syngoportal.org/ .

We appreciate this feedback. To ensure we add more clarity for readers, we have
explicitly stated that we have highlighted processes relating to neurogenesis and
immune responses in Figure 3B (lines 495-501) and Figure 5D (570-574), respectively.
In addition, we also explicitly point readers to the full list of GO terms which is
included as supplemental table 2, where they can also refer to the information for
specific terms such as “neurogenesis” Neurogenesis (GO:0022008), “neuron
development” – (GO:0048666), “synapse organization” (GO:0050808), “axon development”
(GO:0061564), “generation of neurons” (GO:0048699), “neuron differentiation”
(GO:0030182), “neuron projection development” (GO:0031175), and etc. 

We also thank the reviewer for pointing us to the “syngo portal” as a useful
resource. We tried to use this tool on our most differentially expressed genes.
However, these were not in the database as the reviewer pointed out this database is
for neuronal gene enrichment. The top differentially expressed genes are exclusively
expressed in microglia. Their annotation as genes enriched in the context of
neurogenesis likely is due to the upregulation of these genes during processes
associated with embryonic and early postnatal development of the nervous system. 

Figure 1C: Please do NOT show fluoresence intensity, as it is very hard to compare
the different antibody results. Please DO show the percentage of cells that is
positive for Tmem119 (and other markers) and tdTomato in comparison to a control.
How many cells are triple positive for Tmem119+CD45+CD11b+ and how many cells are
quadruple positive Tmem119+CD45+CD11b+tdTomato+?

Negative controls using unstained cells, single staining controls and fluorescent
minus one (FMO) controls are not shown. These are important for setting parameters
of true positive and negative cell populations. Otherwise, if lacking, please show
ungated populations that serve as “negative” cells (e.g. non-microglia cells should
be negative for CD45).

Thank you for these suggestions. As suggested, we now represent the information in
supplemental figure S1B-S1C regarding the overlap between TdTomato cells and the
cell surface markers (Tmem119, Cd11b/Cd45). We have also now included data from
unstained samples, single stained samples, and samples with isotype controls. These
include percentages of cells with dual expression or triple expression after gating
on fluorescence levels.

Due to limitations of how many channels we had available, we were not able to stain
for Tmem119 and Cd11b in the same flow samples. However, the flow samples originated
from the same homogenates so percentages of cells expressing these respective marker
genes can be directly compared.

Minor comments:

Line 60-61 versus line 71-76: please be consistent on terms “microglia”, “myeloid
lineage cells” and “immune cells”. Could the authors make an effort to distinguish
microglia, CNS-associated macrophages, and infiltrating immune cells in the
introduction? For example, in line 71 it appears to me microglia are grouped under
immune cells whereas in line 60 microglia are specifically not immune cells.

We recognize that without an explanation of these terms it is difficult to understand
the distinction between these cell populations. Thus, we have changed the text in
line 60 and added the text listed below (lines 66-75) to describe that macrophages
come from microglia in the introduction.

“Under resting conditions, the immune compartment of the central nervous system (CNS)
is comprised of myeloid lineage cells, microglia and other macrophages which contain
distinct transcriptomic and phenotypic properties [11, 12]. The latter of the two
are typically found in the meninges, perivascular regions, and choroid plexus [13].
Collectively, these non-microglial macrophages are termed CNS-associated macrophages
(CAMs) or Border Associated Macrophages (BAMs) [11, 12]. However, as microglia are
the primary macrophage in the brain parenchyma and found to be actively interacting
with neurons, they are more widely studied and characterized compared to macrophages
in non-parenchymal tissue, particularly in the context of neurodevelopmental
processes.”

Line 226: methods: Why was Harmony used for the data integration with the Keren-Shaul
dataset, whereas previously (line 189) Seurat’s IntegrateData with CCA was used to
integrate samples and correct technical effects? What is the rationale for this
different methodology? Harmony strongly alters the gene expression information and
clustering results might be even more overlapping if the CCA approach is used for
the Keren-Shaul study. 

The reason for choosing Harmony to integrate our dataset with Keren-Shaul’s dataset
is the ability of Harmony to correct for multiple experimental/technical factors.
Harmony projects cells into a shared embedding in which cells group by their cell
types rather than dataset-specific conditions and it corrects for multiple
experimental factors which was the case for our integration step since both datasets
have different technical factors and tissue sources. Harmony (https://www.ncbi.nlm.nih.gov/pmc/articles/PMC6884693/) specifically
mentions that it does not alter the expression values of individual genes to account
for dataset-specific differences.

Line 339: Clusters 4,5,6, 7 and 11 failed to show enrichment of at least ten nuclear
genes. This raises the concern this single-cell dataset is a bit overclustered.
However, I understand it cannot be changed at this point in the project. Cluster 4
expresses mitochondrial-derived markers, suggesting this is a stressed, less viable
cluster of cells that should be removed. Cluster 5 and 6 are appropriately addressed
in the text. In my opinion cluster 7 and 11 could be merged to one cluster. This is
sometimes done in single-cell analysis to revert the overclustering (It can be done
by simply giving them the same label in cluster annotation using
Seurat::RenameIdents()).

We thank the reviewer for these suggestions. All cells that are present in the
initial clustering (Fig 1D) have passed the QC filters and therefore these cells
express an acceptable number of mitochondrial genes. We intentionally chose to
retain a higher number of clusters so as not to obscure or confound any cell states
or subpopulations that may be identified in future studies. However, we recognize
that these populations may also potentially represent cells that are transitioning
towards states where their quality may be potentially compromised. Hence, we focused
our downstream analyses in Figures 3 and 5 on cells derived from cluster 8 versus
clusters from homeostatic clusters (now labeled 12 and 13). We also appreciate the
reviewer’s feedback and suggestions on how to merge clusters. Upon receiving this
feedback, we agree that clusters 7 and 11 should be merged. We have thus updated all
main text figures and supplemental figures to reflect these changes. Note, we have
also changed the supplemental tables including the differential gene expression
analysis between clusters to reflect this.

Line 408: incorrect reference, should reference supplemental figure 7

We had rearranged supplemental figures 6 and 7 in response to previous reviewers’
feedback on the order of these figures. However, the supplementary figures may not
have been updated upon our resubmission. We thank the reviewer for their vigilance
in noting this discrepancy. 

Supplemental figure 4: the heatmap is very blurred.

We apologize for the lack of resolution in this figure. We have reattached and
updated version incorporating changes in clustering (see above for merging of
clusters 7 and 11).

Figure 1A: this is outside the scope of the current study, but relevant for follow-up
work using the same reporter. Did you check tdTomato signal in mice without a TAM
injection? Unfortunately, Rosa26-floxP-tdTomato constructs can be leaky due to the
short stop codon, please refer to PMID: 32125704 “STOP floxing around: specificity
and leakiness of inducible Cre/loxP systems” to verify the validity of your strain.
For the main message and findings of this paper I do not believe it matters.

We checked tdTomato expression with Tamoxifen injection compared to vehicle
injections. We noted tdTomato expression in some cells. These appeared to all have
microglial morphology, thus leading us to believe there is some low-level leaky
tdTomato expression.

Please reconsider the manuscript title “Unique Microglial Transcriptomic Signature
within the Hippocampal Neurogenic Niche”, since the single-cell data does not prove
the microglia cluster 8 is unique to the SGZ nor is there sufficient evidence from
immunohistochemistry to localize it to the neurogenic niche. A more suitable
alternative could be “Distinct microglial transcriptomic signatures within the
hippocampus”

We appreciate the reviewer’s views on how best to capture the results of our
manuscript. We have changed the title to reflect a more accurate and specific
summary of our finding and have changed the title of our manuscript to the
reviewer’s suggested title. 

Reviewer #2: Chintamen et al., used single-cell RNAseq to explore the transcriptional
profiles of myeloid lineage cells isolated from the hippocampus. The authors noted
microglial heterogeneity and highlighted a unique population (cluster 8) of
microglia that demonstrate higher level of phagocytosis-related genes. Based on
histological analysis of selected markers, the authors have attributed this
population of microglia to a neurogenic niche in the subgranular zone of hippocampus
dentate gyrus.

Major concern

1. While the authors have cited other single-cell RNAseq studies on microglia, they
have not rigorously compared their results to these studies. I appreciate that they
have compared it with Keren-Shaul et al. (2017). However, from what I can gather
from their differential gene expression tables, their cluster 8 is also very similar
to an early postnatal microglia population described by Hammond et al. 2019 (cluster
3).

We thank the reviewer for the keen observation that our cluster 8 bears resemblance
to cluster 3 in Hammond et. al. We created a Venn diagram to illustrate the overlap
of the genes enriched between cluster 8 in our manuscript with the early postnatal
microglia. This can now be found as Supplemental Figure S8B. We investigated some
specific genes that are present in both lists and highlighted their presence by our
cluster 5 from in the integrated dataset which we have included in the manuscript as
Supplemental Figure S8D.

Additionally, we have also included a Venn diagram (Supplemental Figure S8C) showing
overlap between enriched genes in our Cluster 8 and the genes upregulated in cluster
3 from the Hammond et. al. dataset and incorporated language to describe
similarities and differences between these enriched genes in these datasets.

2. Similarly, in line 374 and 375, the authors mentioned that a ribosomal enriched
cluster that has been reported but not characterized, but the observation has not
only been reported by multiple studies, it has been reviewed (https://www.ncbi.nlm.nih.gov/pmc/articles/PMC8225243/). These are
just some of the existing literature that I was able to identify with relatively few
searches. Along with the lack of mentions of bulk-RNAseq work that highlighted
microglial heterogeneity across the brain, my main concern is whether other
information and links to existing work were also missed and that the authors have
overstated their findings due to insufficient comparison to existing work.

We thank the reviewer for pointing out the literature that has emerged for ribosomal
gene enriched clusters which were not published when we first conducted these
experiments. We have included the reference to the review mentioned as well as a
couple of other papers and included the description bellow in lines 468-475 of the
main text.

“Cluster 10 exhibits increased expression of genes associated with encoding ribosomal
protein subunits (genes with Rps and Rpl as their prefix) and also contains the
highest ApoE expression. This group of cells undergoing high metabolic activity has
been identified by other groups and may reflect cells with more metabolic needs
which are typically enriched with immune reactivity [3-5].”

We have referenced Kreisel et al 2019 which was a bulk RNAseq data set comparing
regional differences between microglia from the dentate gyrus (DG) vs CA1. In
addition, we have also alluded to the work by Artegiani et al, Hammond et al, and
Li. et al which do use single cell rnaseq data but do not enrich for microglia (in
the case of Artegiani et al) or do not enrich for microglia in specific regions. We
have also referenced Ayata et al, which looks at regional heterogeneity in their
main text with an emphasis on cerebellar microglia.

3. If the authors want to keep clusters with upregulation on mitochondrial genes and
not regress out cell cycle genes, they should really include more quality control
figures (i.e. plots showing nCount_RNA, nFeature_RNA, % ribosomal RNA, %
mitochondrial RNA, cell cycle scoring (you can get that from using CellCycleScoring
in Seurat). It is difficult to review and assess their quality control.

nCount_RNA, nFeature_RNA, and % mitochondrial RNA are presented as “Transcript
reads”, “unique genes”, “%mito genes” respectively. In addition, as per the
reviewer’s suggestion we have added % ribo genes as a violin plot along the
previously described QC plots in Supplemental Figure S2A. We have also added a UMAP
plot showing the cell cycle classification of each of the cells in Supplemental
Figure S2B. We noted that this analysis has been developed with human genes in mind.
We converted the list of human genes used in the “Cell Cycle Scoring vignette”
(https://satijalab.org/seurat/articles/cell_cycle_vignette.html) and
converted these genes into their orthologous murine equivalent. For this, we used
BiomaRt as suggested on some seurat forums (https://github.com/satijalab/seurat/issues/2493). We do find that
not every gene has an ortholog in the mouse genomic reference which may potentially
skew the scoring results. 

4. While I do not believe Cluster 8 is doublet and the authors have included a figure
showing the lack of clusters demonstrating doublets in their response to other
reviewers, I have two issues with the manuscript in its current form. 1) Even if
there isn’t a significant pattern in doublets, the figure shown to the reviewer
should be included in the supplementary. It should also be demonstrated that the
removal of doublets identified by Doubletfinder does not significantly improve the
cluster separation. 2) Doubletfinder works by averaging the transcriptional profiles
of randomly selected cell pairs. If there were to be doublet in this dataset, part
of the doublet pair could be neuronal. However, as the cells were FACs sorted, the
majority of the neuronal cells would not be present in the dataset, therefore
Doubletfinder would not be able to generate a good representation of doublet
transcriptional profiles. In order to more accurately remove doublets in this
dataset, you would need integrate a single cell RNAseq dataset with non-FACS sorted,
then carry out doublet detection, remove all doublet and then keep the singlet cells
from this study. I suspect doublets might be the source of the small fraction of
expression of other cell type markers indicated in line 309-311. While I don’t
expect this level of doublet removal, I do not think the response to Reviewer 2’s
comment 4 was sufficient and at least 1) should be incorporated in the
manuscript.

Thank you for your acknowledgement regarding cluster 8. To answer the point raised
here, we investigated the doublets in our data by using DoubletFinder (2.0.3). A
total of 1108 cells (~6% of total cells) were predicted to be doublets based on
DoubletFinder. We highlighted these cells on a UMAP and it seems like the chances of
clusters to be rearranged/improved is highly unlikely even if we remove these cells
given these cells are not confined to any specific cluster as well as are low in
number. We have incorporated this figure as Supplemental Figure S2C in the
manuscript as suggested and included an explanation in lines 374-385 of the
manuscript . 

5. It would be useful to see Allen Brain atlas showing the Cluster 8 upregulated gene
expression in other brain regions to see if they are specific to the hippocampus
region.

We have added images from the Allen Brain atlas showing coronal sections of these
genes. 

6. The rational of selecting 20 PCs is unclear – is a Jackson straw plot or elbow
plot made to determine this number?

We determined this based on an elbow plot which we have now included as a
supplementary figure S3C and referenced in lines 239-240. 

Minor concerns

1. It is not clear to me what clusters Figure 5E are referring to and how they relate
to the main clustering in this study. The comparison to Keren-Shaul et al’s work is
generally not well described and readers cannot easily follow the rationale.

Given the differences in transcriptomic signature between cluster 8 and homeostatic
microglia characterized by several groups, we sought to understand the functional
role these cells may potentially have. When examining the list of genes found to be
enriched in our cluster, we noted that many of them were also similar to that
published by Keren-Shaul et. al. We found this observation interesting given that
these are cells in a healthy brain. In order to classify whether these cells are
actually DAM-like beyond some genes, we performed integration analysis which showed
us that these cells are more similar to DAM than they are to homeostatic microglia.
We have expanded our explanation and rationale of these analyses and data in lines
594-694.

2. Line 530 – citation style change

Thank you for pointing out this discrepancy, we have fixed the citation notation. 

3. Line 441 – did you mean Supp Figure 7?

The ordering of the supplemental text was rearranged after the first round of
revision. We had swapped the confocal images and images from the Allen brain atlas
and have kept the reference to Supplemental Figure 6. However, a previous version of
our supplemental file was inadvertently attached to our manuscript submission. Thank
you for your attention to this.

4. Figure 1E and 5F – the macrophage clusters should be removed from these figures
(and placed in supplementary) to allow the colour scale to reflect a wider range of
gene expression.

For Figure 1E we have intentionally kept all clusters in the figure in order to
identify cell identity across various clusters. Further analyses in this study focus
on expression levels between microglial subpopulations cluster 8 and homeostatic
clusters 12 and 13 with cell expression displayed as violin plots which do capture
the full range of expression of homeostatic and upregulated genes (see figure
3C).

Figure 5F on the other hand does not consist of macrophages. The cells we included
for this integration analysis were clusters 12 and 13 as homeostatic microglial
cells along with cells from cluster 8. Cells that were integrated from the
Keren-Shaul dataset were specifically cells expressing microglial marker genes only
and hence excluded any non-microglial macrophages. In order to highlight the scale
for DAM genes within the non-integrated cells in homeostatic microglial clusters 12
and 13 alongside cluster8, we have selectively plotted gene expression of these
genes in Supplemental Figure 8A. 

1. Parkhurst CN, Gan WB. Microglia dynamics and function in the CNS. Curr Opin
Neurobiol. 2010;20(5):595-600. Epub 2010/08/14. doi: 10.1016/j.conb.2010.07.002.
PubMed PMID: 20705452; PubMed Central PMCID: PMCPMC3708473.

2. Lawson LJ, Perry VH, Dri P, Gordon S. Heterogeneity in the distribution and
morphology of microglia in the normal adult mouse brain. Neuroscience.
1990;39(1):151-70. Epub 1990/01/01. doi: 10.1016/0306-4522(90)90229-w. PubMed PMID:
2089275.

3. Zhan L, Fan L, Kodama L, Sohn PD, Wong MY, Mousa GA, et al. A MAC2-positive
progenitor-like microglial population is resistant to CSF1R inhibition in adult
mouse brain. Elife. 2020;9. Epub 2020/10/16. doi: 10.7554/eLife.51796. PubMed PMID:
33054973; PubMed Central PMCID: PMCPMC7591254.

4. Tansley S, Uttam S, Urena Guzman A, Yaqubi M, Pacis A, Parisien M, et al.
Single-cell RNA sequencing reveals time- and sex-specific responses of mouse spinal
cord microglia to peripheral nerve injury and links ApoE to chronic pain. Nat
Commun. 2022;13(1):843. Epub 2022/02/13. doi: 10.1038/s41467-022-28473-8. PubMed
PMID: 35149686; PubMed Central PMCID: PMCPMC8837774.

5. Mendes MS, Majewska AK. An overview of microglia ontogeny and maturation in the
homeostatic and pathological brain. Eur J Neurosci. 2021;53(11):3525-47. Epub
2021/04/10. doi: 10.1111/ejn.15225. PubMed PMID: 33835613; PubMed Central PMCID:
PMCPMC8225243.

to reviewers_August_sk.docx
---

## [Decision Letter · Decision Letter 1]

16 Nov 2023

PONE-D-23-16545R1Distinct Microglial
Transcriptomic Signatures within the HippocampusPLOS
ONE

Dear Dr. Kernie,

Thank you for submitting your manuscript to PLOS ONE. After careful consideration, we
feel that it has merit but does not fully meet PLOS ONE’s publication criteria as it
currently stands. Therefore, we invite you to submit a revised version of the
manuscript that addresses the points raised during the review process.

Please submit your revised manuscript by Dec 31 2023 11:59PM. If you will need more
time than this to complete your revisions, please reply to this message or contact
the journal office at plosone@plos.org. When
you're ready to submit your revision, log on to https://www.editorialmanager.com/pone/ and select the 'Submissions
Needing Revision' folder to locate your manuscript file.

Please include the following items when submitting your revised
manuscript:A rebuttal letter that responds to each point raised by the academic
editor and reviewer(s). You should upload this letter as a separate file
labeled 'Response to Reviewers'.A marked-up copy of your manuscript that highlights changes made to the
original version. You should upload this as a separate file labeled
'Revised Manuscript with Track Changes'.An unmarked version of your revised paper without tracked changes. You
should upload this as a separate file labeled 'Manuscript'.If you would like to make changes to your financial disclosure,
please include your updated statement in your cover letter. Guidelines for
resubmitting your figure files are available below the reviewer comments at the end
of this letter.

We look forward to receiving your revised manuscript.

Kind regards,

Giuseppe Biagini, MD

Academic Editor

PLOS ONE

Journal Requirements:

Reviewers' comments:

Reviewer's Responses to Questions

**Comments to the Author**

1. If the authors have adequately addressed your comments raised in a previous round
of review and you feel that this manuscript is now acceptable for publication, you
may indicate that here to bypass the “Comments to the Author” section, enter your
conflict of interest statement in the “Confidential to Editor” section, and submit
your "Accept" recommendation.

Reviewer #1: All comments have been addressed

2. Is the manuscript technically sound, and do the data
support the conclusions?

Reviewer #1: Yes

3. Has the statistical analysis been performed
appropriately and rigorously? 

Reviewer #1: Yes

4. Have the authors made all data underlying the
findings in their manuscript fully available?

Reviewer #1: Yes

5. Is the manuscript presented in an intelligible
fashion and written in standard English?

Reviewer #1: Yes

6. Review Comments to the Author

Reviewer #1: Thank you for submitting your revised manuscript. I have reviewed it
with great interest and observed several improvements. I am happy to accept this
paper. Before finalizing the article, I would like to propose three minor
modifications to the authors.

Minor details:

Line 499" your heading of the paragraph is missing a word in track changes: "SGZ
microglia display morphology and gene expression profiles "that" deviate from a more
homeostatic phenotype."

Supplemental figure 4: the legend can be adjusted to “scaled expression” (expression
of -2 is not possible so I assume this is a z-score , scale = TRUE setting
somewhere).

The phrase “resting brain” and “resting cortex” is used in multiple instances in the
manuscript. It sounds a bit awkward, you might want to consider “healthy brain”
“under control conditions” or otherwise.

7. PLOS authors have the option to publish the peer
review history of their article (what does this mean?). If published, this will
include your full peer review and any attached files.

If you choose “no”, your identity will remain anonymous but your review may still be
made public.

**Do you want your identity to be public for this peer review?** For
information about this choice, including consent withdrawal, please see our
Privacy Policy.

Reviewer #1: No

you for submitting your revised manuscript.docx
---

## [Author Response · Author response to Decision Letter 1]

7 Dec 2023

Response to Reviewer 1:

Thank you for submitting your revised manuscript. I have reviewed it with great
interest and observed several improvements. I am happy to accept this paper. Before
finalizing the article, I would like to propose three minor modifications to the
authors.

Minor details:

Line: 499 your heading of the paragraph is missing a word in track changes: "SGZ
microglia display morphology and gene expression profiles that deviate from a more
homeostatic phenotype."

Thank you for pointing out this typo, it has been corrected.

Supplemental figure 4: the legend can be adjusted to “scaled expression” (expression
of -2 is not possible so I assume this is a z-score , scale = TRUE setting
somewhere).

This has been corrected on Supplemental Figure 4.

The phrase “resting brain” and “resting cortex” is used in multiple instances in the
manuscript. It sounds a bit awkward, you might want to consider “healthy brain”
“under control conditions” or otherwise.

The two instances where “resting brain” and “resting cortex” were used have been each
changed to “healthy brain”.

to reviewers_12_2023.docx
---

## [Editor Report · Decision Letter 2]

10 Dec 2023

Distinct Microglial Transcriptomic Signatures within the Hippocampus

PONE-D-23-16545R2

Dear Dr. Kernie,

We’re pleased to inform you that your manuscript has been judged scientifically
suitable for publication and will be formally accepted for publication once it meets
all outstanding technical requirements.

Kind regards,

Giuseppe Biagini, MD

Academic Editor

PLOS ONE
---

## [Editor Report · Acceptance letter]

21 Dec 2023

PONE-D-23-16545R2 

PLOS ONE

Dear Dr. Kernie, 

I'm pleased to inform you that your manuscript has been deemed suitable for
publication in PLOS ONE. Congratulations! Your manuscript is now being handed over
to our production team.

Kind regards, 

on behalf of

Dr. Giuseppe Biagini 

Academic Editor

PLOS ONE